# Many-Body Open Quantum Systems

Rosario Fazio[1,2] , Jonathan Keeling[3] , Leonardo Mazza[4] , Marco Schirò[5] ,

**1** The *Abdus Salam* International Center for Theoretical Physics (ICTP), I-34151 Trieste, Italy

**2** Dipartimento di Fisica "E. Pancini", Università di Napoli Federico II, Monte S. Angelo, I-80126 Napoli, Italy

**3** SUPA, School of Physics and Astronomy, University of St Andrews, KY16 9SS St Andrews, United Kingdom

**4** Université Paris-Saclay, CNRS, LPTMS, 91405, Orsay, France

**5** JEIP, UAR 3573 CNRS, Collège de France, PSL Research University, 11 Place Marcelin Berthelot, 75321 Paris Cedex 05, France

October 15, 2024

## Abstract

These Lecture Notes discuss the recent theoretical advances in the understanding of open quantum many-body physics in platforms where both dissipative and coherent processes can be tuned and controlled to a high degree. We start by reviewing the theoretical frameworks and methods used to describe and tackle open quantum many-body systems. We then discuss the use of dissipative processes to engineer many-body stationary states with desired properties and the emergence of dissipative phase transitions arising out of the competition between coherent evolution and dissipation. We review the dynamics of open quantum many body systems in the presence of correlated many-body dissipative processes, such as heating and many-body losses. Finally we provide a different perspective on open quantum many-body systems by looking at stochastic quantum trajectories, relevant for the case in which the environment represents a monitoring device, and the associated measurement-induced phase transitions.

# 1 Introduction

## 1.1 Scope and structure of the Lecture Notes

Understanding the effect of an external environment on the dynamics of quantum system has been a problem of paramount importance since the early days of quantum mechanics [1] because it touches questions that are at the heart of its fundamental principles [2,3]. Much more recently, with the birth of quantum information processing, the properties of open quantum systems have become very relevant, as decoherence and damping are the ultimate limiting factors for the development of quantum technologies [4, 5]. Quantum systems in contact with an external environment are referred to as *open quantum systems* and

are described by a quantum state that is, generically, mixed. For this purpose, several techniques, both in the design and in the operating mode of qubits and gates, have been implemented with the scope of reducing unavoidable environmental effects [6, 7].

If on one side an external environment typically tends to hinder quantum coherence effects, on the other side several new phenomena emerge from the competition between unitary evolution and dissipative effects. This phenomenology gets considerably richer in the case of many-body systems. Exotic phases of matter that are not allowed at equilibrium, novel dynamical regimes in the evolution of complex quantum systems or critical phenomena controlled by dissipation are only few of the most exciting examples. Furthermore, and contrary to what is generally believed, dissipation is not always detrimental for manipulating quantum systems: in the same way as optical pumping is a powerful tool for tailoring the properties of quantum systems at the single-atom level, it is also possible to control the dynamics of many-body quantum systems by suitably engineered dissipation. In fact, with the help of only dissipative dynamics, one can perform a universal quantum computation in a way that is completely equivalent to the standard formulation in terms of unitary quantum circuits [8, 9].

This set of Lecture Notes aims to give a pedagogical introduction to the physics of *many-body open quantum systems*. Obviously this is a broad research field, that essentially includes every aspect of the equilibrium and nonequilibrium statistical mechanics of quantum systems coupled to baths. One could even go further by noting that, for ergodic systems, the system itself can play the role of the bath for one of its subparts and conclude that the recent studies on the unitary dynamics of closed quantum systems and the problem of thermalisation belong to this subject. Given the potential breadth of such a field, these Lecture Notes will necessarily have a more narrow focus, as we outline next. First of all, our interest will be on *Synthetic Quantum Matter*—artificial quantum complex systems which can be used to realise tunable and controllable phases of matter. In such platforms, one naturally faces the presence of a bath, but one could also engineer the bath itself and its coupling with the setup, and mix its action with an external drive, i.e. *driven-dissipative systems*. Monitoring the system by performing measurements is also a way to perturb the unitary dynamics that we will consider and that is gaining increasing experimental interest. Secondly, in these Lecture Notes, we will be studying those situations in which the external bath *modifies* the collective behaviour of synthetic quantum systems, for instance by inducing new phases, provoking novel kinds of critical properties, or transforming the conditions under which various states of matter can be found. Thirdly, we will consider specifically Markovian systems, whose dynamics could be described by a time-local Lindblad master equation, with a term that accounts for the unitary evolution due to the Hamiltonian and a contribution that describes the effect of the external bath, responsible for the relaxation to a steady state and for the loss of quantum coherence.

Synthetic open quantum systems also allow one to explore phenomena that are otherwise rare (or non-existent) in nature, even though they are possible according to the laws of quantum physics. They can also be seen, following Feynman's vision [10], as opensystem quantum simulators. In the last decades there has been a considerable experimental progress in realising platforms where many-body effects in the presence of dissipation can be explored, ranging from optical lattices and trapped ions to Bose-Einstein condensates in cavities, from cavity arrays to quantum circuits subject to measurements. In essentially all the cases mentioned above, the dynamics of the density matrix of the system can be accurately described by a Lindblad equation. The unitary part typically represents a many-body Hamiltonian (for example the exchange Hamiltonian for interacting spinsystems, or the Bose–Hubbard Hamiltonian for atoms in optical lattices) or a sequence of quantum gates; on its own, it will induce correlations and entanglement between local

degrees of freedom. The presence of the environment or of measurements will result in a local incoherent dynamics that may compete against such ordering: this interplay will strongly affect both the steady state and the transient dynamics. Our goal is to give a broad and pedagogical introduction to these phenomena.

Understanding the influence of decoherence and dissipative effects in many-body quantum systems touches many questions that are beyond the scope of these Lecture Notes. Here we note some of these topics, and provide references to reviews that discuss these points. We will concentrate on systems whose dynamics is time-local (Markovian) and specifically of Lindblad form, thus we will not consider non-Markovian open systems [11]. We will further focus our attention on those phenomena where dissipation is an essential element, rather than a problem to be avoided. This means that we will not cover the problems related to the reliability of quantum simulators [12,13] in the presence of imperfections and noise [5–7]. Focusing on many-body systems we will also put to one side the rich topic of spin-boson dynamics and associated models [14,15] (other than brief remarks in the next section), and similarly we will ignore questions about the role of dissipation and decoherence in the emergence of classical behaviour [2,3]. While we will discuss some aspects of exciton-polariton physics—in particular, methods first developed in the context of modelling polariton condensates—we will not aim to provide a comprehensive overview of this large field. Reviews of this field can be found in a number of places, e.g. Refs. [16–19].

Likewise, we will not discuss those situations where few qubits are coupled to two or more reservoirs (as in quantum thermal machines [20,21] or transport setups [22,23]), nor will we explore the whole field of quantum thermodynamics [24,25]. We also refer to more specialised reviews for the exciting problem of developing efficient numerical tools for the simulation of open quantum systems [26]. Open quantum systems are often described in terms of effective non-Hermitian models for pure states: we will not enter into this subject and stick to the formalism based on density matrices and Lindblad master equations, referring the interested reader to existing reviews [27]. Although very interesting and very important for quantum technological applications, these topics will not be considered further below, except when of relevance to the many-body realm.

These Lecture Notes are organised as follows. The remainder of this introduction contains a brief historical perspective of the field of open-system dynamics in synthetic quantum matter, and a discussion of the conceptual differences between considering equilibrium vs. driven-dissipative systems. In Sec. 2 we present a brief overview of the experimental platforms used to realise synthetic quantum systems. In Sec. 3 we introduce the general frameworks that are currently employed to study the open-system dynamics of these setups: the Lindblad dynamics for the density matrix, the Schwinger–Keldysh path integral, and quantum trajectories. Section 4 summarises the key theoretical approaches to determine the dynamics in these various frameworks.

We then proceed by discussing the key properties of open many-body systems. In Sec. 5 we review the applications of dissipative state preparation in the many-body context, where a tailored design of dissipative processes and many-body jump operators can stabilise states with non-trivial properties. As one system parameter is tuned, these states can undergo sharp, non-analytic changes in their observable properties, leading to dissipative phase transitions, that we discuss in Sec. 6. We review their general classification, their relation with classical and quantum phase transitions and symmetry breaking and provide several examples of recent theoretical and experimental interest. Dissipative phase transitions can be also associated to the absence of a steady-state and to time-translation symmetry breaking, such as in boundary or dissipative time crystals. The focus of Sec. 7

shifts from the steady state to the dynamical transient approach to it. This is particularly relevant for many-body dephasing problems leading to heating dynamics or many-body losses associated to the emergence of the quantum Zeno effect. Further insight on open quantum many-body systems can be obtained by following the dynamics along quantum trajectories and the unravelling of the Lindblad master equation. The case of monitored dynamics unveils new phenomenology that is invisible to the average state, i.e. the density matrix. In Sec. 8 we will consider the dynamics under different unravelling corresponding to continuous weak monitoring and review the emergence of measurement-induced entanglement transitions. Finally, in Sec. 9 we will draw our conclusions. For completeness, Appendix A presents a self-contained derivation of Lindblad master equations.

Finally, these Lecture Notes should not be intended as a comprehensive review, but rather as a pedagogical introduction to the field. Many of the topics discussed in these lectures notes were also covered in the past in several, interesting reviews. The reader is encourage to read Refs. [28–33] to acquire a more comprehensive picture of the physics of many-body open systems.

## 1.2 Historical Perspective

It is useful to frame the field of open quantum systems in the broader perspective of dissipative quantum systems that have been investigated for a long time. We focus our discussion here on Josephson junction arrays (see e.g. the review [34]), which were one of the first successful attempts to observe dissipative phase transitions in synthetic (i.e. artificially realised) systems. There are also a number of other topics where there was early work that could be understood in this context. This includes work on the analogy between lasers and phase transitions [35–37]—a topic we discuss further in Sec. 2.6 in relation to polaritons and photons. In addition a great deal of work has been done on phase transitions of two-level systems coupled to external baths, as reviewed in Refs. [14, 38].

Turning to Josephson junction arrays, early works initially prompted by proposals of P. W. Anderson [39] and A. J. Leggett [40] identified small Josephson junctions as very promising candidates to observe macroscopic quantum phenomena. The original motivation was mainly driven by the possibility to explore macroscopic quantum dynamics and the classical-to-quantum boundary in meso- or nano-structures. Parallel to these studies, Josephson junction arrays (regular lattices of small superconducting islands connected by tunnel junctions) were identified as promising platforms to study classical and quantum statistical mechanics models. The first experimental evidence of Berezinskii–Kosterlitz–Thouless transitions was observed [41], and in the late 80s a superconductor-insulator quantum phase transition was seen in quantum arrays using devices with small enough geometrical capacitance in order to have charging energies comparable to Josephson couplings [42]. Furthermore, several theoretical works pointed out that by including an external environment *à la* Caldeira–Leggett the zero- and finite-temperature phase diagram would be considerably enriched. In particular, on increasing the strength of dissipation the whole array would undergo a transition to a phase of global coherence solely controlled by dissipation, as in [43]. For further details, see the review [34] and references cited there.

The experimental verification of a transition induced by dissipation came much later from the independent work of two research groups. The group of J. Clarke in Berkeley [44] realised the Josephson array on top of a 2D electron gas, controlling the strength of damping through a gate voltage that allowed to vary the conductance of the 2DEG. Y. Takahide and coworkers [45], in the group of S. Kobayashi in Tokyo, instead engineered shunt resistances in parallel to each junction in a one-dimensional array (they had to fabricate several different samples with different shunts in order to observe the increasing effects of dissipation!). In Fig. 1 we show the results of the Berkeley experiment. The experimental setup

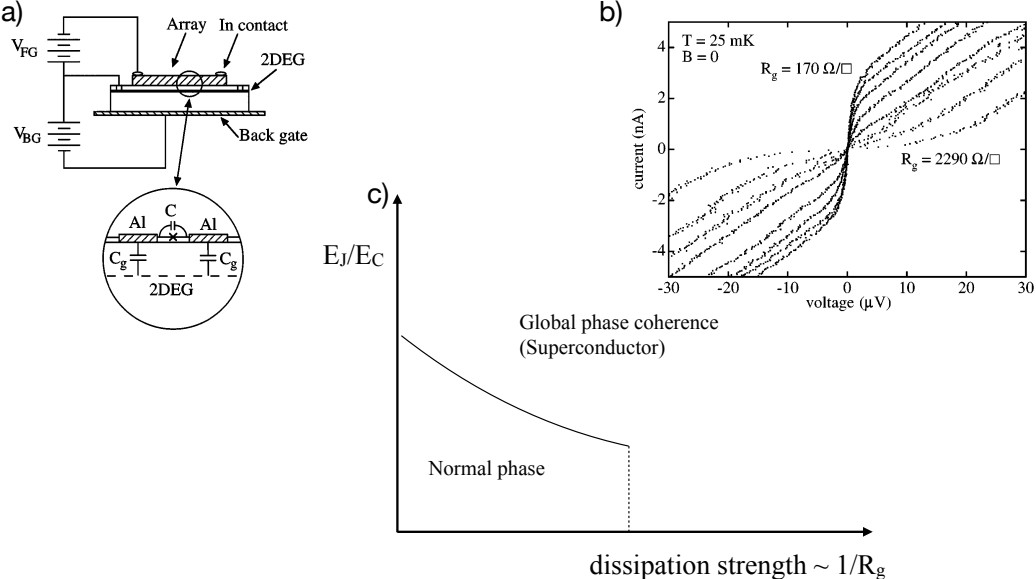

Figure 1: (a-b) Experimental measurement of a dissipative phase transition in a Josephson junction arrays, from [44] [Copyright (1997) by the American Physical Society]. (a) The Josephson array (on the top of the device) is in electrostatic contact with a two-dimensional electron gas (2DEG) that is responsible for a voltage noise on the superconducting islands. The level of the noise is modulated by changing the resistance $R_g$ of the 2DEG. (b) The current-voltage characteristics for different levels of the noise, i.e. different values of the resistance $R_g$. On increasing $R_g$, i.e. decreasing the dissipation, the current-voltage characteristics change from superconducting-like (finite current at zero voltage) to insulator-like (zero current below a threshold voltage). (c) Phase diagram—sketch of the zero-temperature phase diagram, above a certain value of the dissipation strength the array acquires global coherence for any value of the Josephson coupling [The phase diagram is sketched from [43]].

is shown in panel (a): dissipation is controlled by coupling the arrays to a two-dimensional electron gas and modulating its conductivity (hence the voltage noise acting on the array) by means of the back gate voltage. Panel (b) shows the current-voltage characteristics for different level of dissipation (i.e. for different values of the resistance of the 2DEG $R_g$). At low $R_g$ the current-voltage characteristics are that of a superconductor (with global phase coherence). On increasing $R_g$ there is a qualitative change to the current-voltage characteristics of an insulator (Coulomb blockade regime) with zero current below a threshold voltage. This is, to our knowledge, the first experimental observation of a dissipative phase transition in a quantum many-body synthetic system.

In the language of quantum information processing, Josephson junction arrays can be considered as the ancestors of synthetic systems, although with limited control on their quantum evolution due to the presence of imperfections and noise. The impressive technical advances over the last decades have enabled the design of new platforms to study open many-body systems. As we will see in the forthcoming Sections, these new systems offer an unprecedented control both in the realisation of synthetic quantum matter and in the study of its dynamical evolution (for instance, the coherent dynamics has always proven impossible to study in Josephson arrays).

An important feature characterising the early works and differentiating them from the most recent experiments, is that Josephson arrays operated in a regime where the detailed balance relation is always satisfied; as we discuss in Sec. 1.3, this is a hallmark of the global thermal equilibrium of the system and of its environment. As already mentioned, an excellent description of these systems could be based on Caldeira-Leggett models, provided they are properly generalised to the many-body context. Furthermore, in many cases the out-of-equilibrium properties of the superconducting order parameter of Josephson junctions would be described by dynamical equations typical of classical dynamical critical phenomena [46]. Instead, in the modern platforms that are currently being developed (funnily enough, some of them still use superconductors and Josephson junctions), the dynamics is governed by the external driving and by dissipation, so that the state cannot be found by considering an equilibrium partition function—not even including the environment within the partition sum. We discuss these points further in Sec. 1.3 below. In the Sec. 2 we will review many experimental platforms of more recent interest which are currently playing a major role in the study of synthetic quantum matter and which enable the study of open quantum many-body physics in controlled settings.

## 1.3 Relation with thermal equilibrium

In this section we discuss some key concepts about the distinction between quantum systems in equilibrium and driven-dissipative systems. Note we use the term driven-dissipative here to identify systems where there is either some driving term in the Hamiltonian along with coupling to a environment or coupling to multiple environments. The term "open quantum systems" is broader, and can include the case of a system coupled to a single environment. One point we discuss in particular here is to expand on the discussion in the previous section, distinguishing the behaviour of a system coupled to a single environment from the behaviour of a driven-dissipative system.

To set this discussion in context we first provide a brief summary of the key ideas that arise in equilibrium statistical physics, in order to then comment on how these are changed in driven-dissipative systems. For a full discussion, the reader may see textbooks on thermodynamics and statistical physics, e.g. [47, 48]. Equilibrium statistical physics starts from the postulate of equal equilibrium probabilities (PEEP), which states that all microstates occur with equal probabilities. This then leads to the idea that the probability of a macrostate—where some macroscopic variables are specified—is determined by the

number of ways in which such a macrostate can arise, i.e. the number of microstates corresponding to a given macrostate. Defining the entropy as the logarithm of this number of microstates then leads to the notion that the most probable macrostates are those that maximise entropy.

To get from the PEEP to the familiar Boltzmann distribution, one may consider a system that is weakly coupled to an environment in thermal equilibrium. In particular, we consider the case where the system plus environment is in a microcanonical state, i.e. the total energy of the system plus environment is fixed. Weak coupling allows one to consider the energy of the system and environment additively, i.e. to ignore any energy associated with their coupling. Thus, the energy in the environment is determined by the (fixed) total energy and the energy of the system. If the system exchanges only energy with the environment (as opposed to e.g. particles, or any other conserved quantity), then one can consider the entropy of the environment as being a function of the energy of the system. Since the environment can be assumed to be large, and thus weakly affected by this transfer of energy, one can consider the entropy of the environment to depend linearly on the energy of the system. The coefficient in this expression is the inverse temperature. This then leads to the standard Boltzmann distribution: $P \propto \exp(-\beta(E - TS_S))$ with $\beta = 1/(k_B T)$ and $S_S$ being the entropy of the system.

To understand what changes in driven-dissipative systems we highlight a key assumption to obtain the Boltzmann distribution, namely the existence of a well defined entropy cost to increasing the energy of the system. This concept applies for coupling to a single environment, but cannot apply when a system is coupled to multiple environments that are not in thermal equilibrium with each other. For multiple environments with multiple temperatures, the entropy cost of increasing the system energy depends on which environment is involved. As such, for systems coupled to multiple environments, the relative coupling rates and dynamics of the system will matter, and the probability that the system is in a given state cannot only depend on its energy.

While we have introduced the Boltzmann distribution from the assumption of equilibrium with an external environment, the same expression can also be derived for a large system: here, when considering one part of the system, one can regard the rest as acting as an effective environment. In a finite system there will be a distinction between the result derived above, which are correct for the canonical ensemble (ensemble at fixed temperature) vs the microcanonical ensemble (ensemble at fixed system energy). However, in the thermodynamic limit, these ensembles become equivalent for extensive quantities.

In the discussion so far we assumed weak coupling between the system and its environment. As long as one is restricted to a single environment (or multiple environments in thermal equilibrium), one can extend the above equilibrium analysis further. One may thus discuss how behaviour changes when a system is strongly coupled to the environment, as is of relevance to the discussion in the previous section. We discuss first an approach that works for static equilibrium properties, and then discuss dynamical responses.

For static properties, one may consider the Boltzmann distribution for the combined system and environment. This means the system density matrix is given by the combined Gibbs state:

$$\hat{\rho}_S = \frac{1}{\mathcal{Z}} \text{Tr}_E \left[ \exp \left( -\beta(\hat{H}_S + \hat{H}_E + \hat{H}_{S-E}) \right) \right]. \tag{1}$$

Here we have written the equilibrium system density matrix, $\rho_S$, in terms of the Hamiltonian of the system, $\hat{H}_S$, the environment, $\hat{H}_E$, and the system-environment interaction, $\hat{H}_{S-E}$. This expression generalises the Boltzmann distribution discussed above, by assuming internal equilibrium of the combined system and environment. The partition function $\mathcal{Z}$ appears here as normalisation, required to ensure $\text{Tr}[\rho_S] = 1$.

If the coupling to the environment is weak, then $\hat{H}_{S-E}$ can be neglected. In this case, after tracing over the environment, one recovers $\hat{\rho}_S \propto \exp(-\beta\hat{H}_S)$. Assuming $\hat{H}_{S-E}$ vanishes is reasonable when considering the equilibrium state of the system. However, one should note that if the system environment coupling vanishes, the timescale for the system and environment to come into equilibrium will diverge. The equilibrium state can generally be assumed to be equivalent to the long time average—this is the notion of ergodicity. Formally this equilibrium ensemble with the system Hamiltonian corresponds to first taking the limit of long times, and only then considering the limit of vanishing system-environment coupling.

If the coupling to the environment is not weak, then the equilibrium state defined by Eq. (1) will not equal $\exp(-\beta\hat{H}_S)/\mathcal{Z}_S$ (known as the system Gibbs state), but will be modified by the coupling to the environment. Since an equilibrium density matrix, $\hat{\rho}_S$, does however exist (as defined by Eq. (1)), one can however define an effective "Hamiltonian of Mean-Force" (HMF) [49, 50] by writing:

$$\hat{\rho}_S = \frac{e^{-\beta\hat{H}_{\mathrm{HMF}}}}{\mathcal{Z}_{\mathrm{HMF}}}. \tag{2}$$

The state $\rho_S$ as written here is sometimes referred to as the Mean-Force Gibbs state; as discussed here, this is equivalent to the combined Gibbs state. Mathematically the definition of $\hat{H}_{HMF}$ just corresponds to extracting the logarithm of the equilibrium density matrix. It is easy to verify, by using the identity

$$e^{-\beta(\hat{H}_S+\hat{H}_E+\hat{H}_{S-E})} = e^{-\beta(\hat{H}_S+\hat{H}_E)}T_\tau e^{-\int_0^\beta d\tau \hat{H}_{S-E}(\tau)},$$

with the coupling Hamiltonian expressed in the time-ordered interaction representation, that the HMF

$$\hat{H}_{\mathrm{HMF}} = \hat{H}_S - \frac{1}{\beta}\ln\langle T_\tau e^{-\int_0^\beta d\tau \hat{H}_{S-E}(\tau)}\rangle_E.$$

The form of $\hat{H}_{\mathrm{HMF}}$ may be complicated with environment-mediated interactions, and thus potentially nonlocal, and its form may depend on temperature. Nonetheless, this formulation identifies a natural generalised of the concept of potential of mean force [49, 50] sometimes studied in classical statistical physics. There can also exist simple forms that arise in the limit of strong system-environment coupling [51]. In many situations, it is also convenient to use a path integral formulation and consider the effective action deriving after tracing out the environmental degrees of freedom. [38]. Most of the early works studying dissipative phase transitions in Josephson arrays, see Section 2.5, approached the problem in this way.

While the above suggests an effective mean-force Hamiltonian can be found to determine static properties, the situation is more complicated when considering dynamic properties. No effective system Hamiltonian can correctly capture the dynamical response of a system strongly coupled to an environment. This is because the timescales of the dynamics of the environment become important. One can capture the effects of the environment through an effective Keldysh action [52] (a concept that we discuss further below). The Keldysh action however describes a frequency-dependent response of the environment, and this cannot be capture by any effect Hamiltonian. It is this physics that led to the interest in dissipative phase transitions discussed in the historical perspective above.

One question that arises when considering systems strongly coupled to their environment is how one can tell whether the system is in thermal equilibrium or not. As noted above, the Hamiltonian of mean force can always be defined from the logarithm of the density matrix, and since $\hat{H}_{HMF} \neq \hat{H}_S$, one cannot simply use the equilibrium density

matrix to practically determine whether a given system is in equilibrium or not. What can however be used is the fluctuation dissipation theorem [53,54] (FDT). The FDT is defined by considering the two-time correlation functions of some observable $\hat{A}(t)$. Specifically, one considers the power spectral density $S_A(\omega)$ and the response function $\chi_A(\omega)$, that are the Fourier transforms of the following two-time correlation functions:

$$S_A(\tau) = \frac{1}{2}\left\langle \{\hat{A}(t+\tau), \hat{A}(t)\}\right\rangle, \qquad \chi_A(\tau) = i\Theta(\tau)\left\langle [\hat{A}(t+\tau), \hat{A}(t)]\right\rangle, \qquad (3)$$

where $\{,\}$ indicates the anticommutator, $[,]$ indicates the commutator, and $\Theta(\tau)$ the step function. Note that in a stationary state, these expressions depend only on the time difference, $\tau$ and not the initial time $t$, hence the possibility of Fourier transforming only with respect to $\tau$. What the FDT proves [53,54] is that in thermal equilibrium, the Fourier transforms of these two functions obey:

$$\frac{S_A(\omega)}{\text{Im}[\chi_A(\omega)]} = \coth\left(\beta\omega\right). \qquad (4)$$

We have assumed here the operator $\hat{A}$ corresponds to an observable, so $\hat{A}$ is Hermitian and cannot be a fermionic operator. However, beyond this restriction, the expression is general: it holds for any observable $\hat{A}$ and for any form of system-environment coupling. One can understand that this expression works because $\chi_A(\omega)$ measures the density of states of the system, while $S_A(\omega)$ measures the noise due to thermally occupied modes. As a result, the combination given here allows the density of states (and thus effects of strong system-environment coupling) to "cancel out" from the expression.

Summarising, in systems coupled to a single environment, no matter how strongly, one can expect the FDT to hold in full at late enough times, i.e. once the system has reached equilibrium. In general open quantum systems, such as driven-dissipative setups, the FDT may approximately hold over some range of frequencies, but it will not do so in general, since such systems are not in thermal equilibrium. Other theoretical frameworks then need to be developed, and the Lindblad master equation is the simplest and most widespread that describes a system reaching a stationary state that is, in general, not at thermal equilibrium. The study of the varied phenomenology that can appear in these setups is the object of these Lecture Notes.

## 1.4   Notation

It is convenient to summarise in a table the acronyms and the notation principally used throughout these Lecture Notes; some exceptions exist and we highlight these when they occur. We set $\hbar = k_B = 1$ throughout.

| | |
|---|---|
| Asymptotic Decay Rate | ADR |
| Dynamical Mean-Field Theory | DMFT |
| Dissipative Phase Transition | DPT |
| Mean-Field | MF |
| Steady State | ss |
| | |
| Commutator | $[\hat{A}, \hat{B}] = \hat{A}\hat{B} - \hat{B}\hat{A}$ |
| Anticommutator | $\{\hat{A}, \hat{B}\} = \hat{A}\hat{B} + \hat{B}\hat{A}$ |
| Lattice sites | $i, j$ |
| Coordination number | z |
| Space coordinate in continuous systems | $\mathbf{r}$ |
| Time | $t$ |
| Number of particles | $N$ |
| Number of lattice sites | $N_s$ |
| Length of system | $L$ |
| System dimensionality | $d$ |
| Volume | $L^d$ |
| System dimensionality | $d_H$ |
| Cartesian components | $\alpha, \beta = x, y, z$ |
| Wave-vector in continuous systems | $\mathbf{k}, \mathbf{q}$ |
| Labels (not referring to space/lattice) | Greek letters $\delta, \ldots \mu, \nu, \ldots$ |
| Operators and Superoperators | $\hat{O}$ , $\hat{\mathcal{O}}$ |
| Density matrix of the system | $\hat{\rho}(t)$ |
| Hamiltonian of the system | $\hat{H}$ |
| Lindblad jump operators | $\hat{L}$ |
| Lindbladian superoperator | $\hat{\mathcal{L}}$ |
| $\mu$-th eigenvalue of the Lindbladian superoperator | $\lambda_\mu$ |
| non-Hermitian Hamiltonian | $\hat{H}_{\mathrm{nH}}$ |
| Bose operators | $\hat{a}, \hat{a}^\dagger$ |
| Fermi operators | $\hat{c}, \hat{c}^\dagger$ |
| Spin operators on a lattice | $\hat{\sigma}_i^\alpha$ |
| Field operators | $\hat{\psi}$ |
| Magnetic interaction | $J_{i,j}^\alpha$ |
| On-site interaction | $U$ |
| Hopping coupling constant | $t_H$ |
| Light-matter interaction | $g$ |
| Detuning | $\Delta$ |
| Dissipative rates | $\kappa, \gamma, \Gamma, \ldots$ |
| Asymptotic Decay Rate | $\lambda_{\mathrm{ADR}}$ |
| Control parameter for phase transition | $\eta$ |

## 2 Experimental Platforms

In this Section we briefly discuss some of the experimental setups which are relevant for the field of many-body open quantum systems. In line with the general goal of these Lecture Notes, we focus on synthetic quantum matter where many-body effects and open-system dynamics are particularly relevant: superconducting circuits and circuit QED arrays [28, 55], ultracold atoms with controlled dissipation [56], ultracold atoms in high-finesse cavities [57, 58], and trapped ions [59, 60]. These experimental platforms are often presented as paradigmatic examples of isolated quantum systems. However, in most experimental realisations there will be effects due to the presence of an environment, typically represented by the experimental apparatus itself. Although the effect of such an environment can often be made rather small, it is never completely negligible. On the other hand, what is very important here is that it can be made highly controllable.

We emphasise that there are several reviews already existing on the related topic of quantum simulation, such as Ref. [12, 13, 61], therefore we will limit ourselves to discuss the key aspects that are particular relevant in the present context.

### 2.1 Trapped Ions

Single trapped ions represent elementary quantum systems with degrees of freedom that are well isolated from the environment [59]. Experimentalists working with trapped ions have for long time manipulated the quantum state of a single ion using coherent fields, as well as using the opportunities offered by spontaneous emission (see e.g. Refs. [62–64]), which is one of the simplest open-quantum-system effects that can be considered. Using this, it is possible to bring the ions nearly to rest by laser sideband cooling, and to prepare the ion in a desired electronic quantum state by the use of optical pumping. [65]. This makes trapped ions well-suited for the study of open-quantum-system dynamics under well-controlled conditions.

As a simple example, we briefly summarise the key results reported in Ref. [62], which achieved the first experimental observation of the quantum Zeno effect [66], i.e. strong coupling to an environment inhibiting coherent transitions between the energy levels of a quantum system. Working with $Be^+$ ions, this work considered two hyperfine levels coupled by a radio-frequency field while a strong laser pulse coupled one of those levels to an excited unstable state which would undergo spontaneous emission (see the sketch in Fig. 2, left panel). By observing the spontaneously emitted photons, it was possible to quantitatively measure the reduction of coherent transitions between the hyperfine states.

The early results on the manipulation of single ions can be seen as foundational for contemporary research. Indeed, in quantum optics and atomic physics, the techniques of optical pumping and laser cooling are now routinely employed for the dissipative manipulation of quantum states, although on a single-particle level. The current studies on engineering dissipative dynamics that we discuss in Sec. 5 can be seen as a generalisation of these techniques to a many-particle context: it is not by accident that the quantum Zeno effect is currently experiencing a revival in the many-body setting, as we discuss more extensively in Sec. 7.2.

In more recent years, trapped ions have started to play a significant role in the context where the role of the environment is played by a measurement apparatus and the quantum system is *monitored*. The possibility of repeatedly and frequently measuring the ions—which is at the heart of the quantum Zeno effect—is also what places these setups on the forefront of this research field. As an example, in Ref. [67] the authors consider one-dimensional chains of ions of length spanning from 4 to 8 ions and engineer a unitary

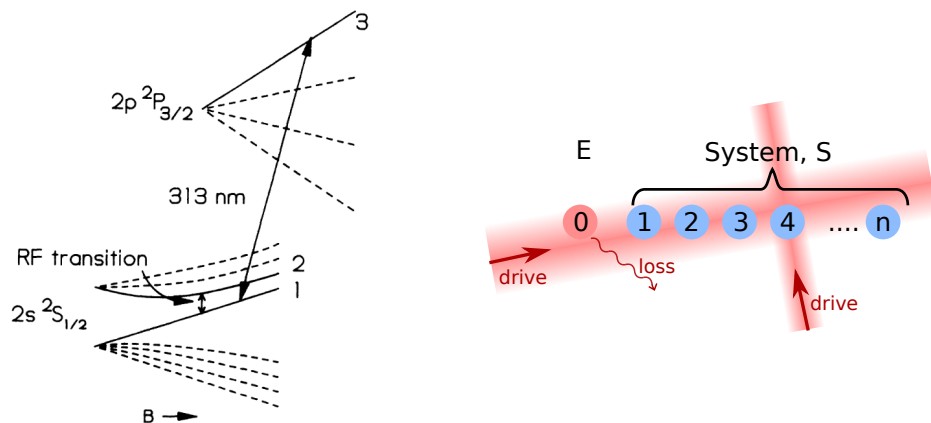

Figure 2: (Left) Energy levels of $^9$Be$^+$ ions in a magnetic field $B$ employed in the experiment on the quantum Zeno effect [62] [Copyright (1989) by the American Physical Society]. The two hyperfine levels, 1 and 2, are coupled by a radio-frequency field; a strong laser pulse couples the level 1 with the excited level 3, which undergoes spontaneous emission. In this way, the quantum mechanical properties of a single ion are manipulated using coherent and dissipative processes. From Ref. [62]. (Right) Experimental tools for the simulation of open quantum systems with ions. $n + 1$ ions are used to simulate the open-system dynamics of $n$ ions (the system, S) in a controlled fashion. Spontaneous emission is induced in a controlled fashion on the first ion (the environment, E), inducing the desired open-system-dynamics on the other $n$ ions. Figure based on experiment in Ref. [64].

dynamics (technically, a random circuit composed of unitary gates) with interspersed projective measurements. When the measurement rate is low, the evolution reaches a mixed phase: the multiple measurement outcomes create a mixed density matrix. On the contrary, for a measurement rate exceeding a given threshold, the evolution reaches a pure phase: measurements project the system on to a deterministic quantum state. We will discuss the theoretical and experimental considerations associated to monitored quantum systems in Sec. 8.

Regarding the development of experimental platforms, it is interesting to mention one early approach to creating a many-body synthetic quantum system based on trapped ions employing the coupling to an environment [64]. Building on a well-tested quantum computing architecture, which is designed to undergo purely unitary (i.e. closed-system) dynamics, this work combined multi-qubit unitary gates with optical pumping to implement dissipative processes. In particular, the experimental platform is composed of an array of $n + 1$ ions, with the first ion serving as an ancilla, and the remaining $n$ ions being the system (see the sketch in Fig. 2, right panel). The ancilla is coupled to the system through application of specific gates, but also undergoes dissipative dynamics ultimately due to spontaneous emission events. This in turn induces a controlled and tunable dissipative dynamics on the rest of the system. As an example, this platform was employed to prepare entangled Bell states composed of $n = 2$ ions or GHZ states composed of $n = 4$ ions. In the first case, the environment acts on two ions at the same time, and its effect depends on the reduced density matrix of the two particles. In the second case, the coupling to the environment is mediated by four ions. Given the relatively small number of qubits involved, it was possible to perform full quantum state tomography of the system and demonstrate that, thanks to the dissipative processes, the desired state had been realised.

In more recent years, several other experiments have been reported which have actively

exploited the judicious control of dissipation for the study of quantum matter or dynamics; we mention here a few of them. One example is the phonon laser realised in [68] where the phase diagram and the dissipative transition have been detected. More recently the dissipative generation of an entangled state of two trapped-ion qubits, based on a scheme developed in Ref. [69], has been experimentally realised in Ref. [70]. The possibility of measuring and resetting the ions with high fidelity and addressability is crucial in the results reported in Ref. [71], where entanglement entropy of a spin chain is directly measured. Finally, the possibility of using dissipation to simulate the quantum dynamics of electronic transfer, a process at the heart of several biochemical processes, has been reported in Ref. [72].

## 2.2   Ultracold atoms

In recent years, experiments with ultracold atoms have evolved towards the development of platforms where the coupling to an environment can be precisely characterised and tuned, so that it can be reduced or enhanced over several orders of magnitude; consequently, open-quantum-system dynamics can be studied with these platforms in the most genuine spirit of quantum simulation. One key example of this is cold atoms coupled to optical cavities, where the light in the cavity represents an open quantum system. In the absence of coupling to an optical cavity, there are two main open-quantum-system effects that can take place: losses of particles from the trapping potential, and heating due to the interaction with the electromagnetic fields that trap or control the gas. In the following we first present these experiments and then discuss in a separate Section the case of cavity QED with ultracold atoms. Finally, we will introduce Rydberg states of ultracold atoms.

**Atom loss.**   Losses have been a major concern from the early days of Bose-Einstein condensation. Several articles dating from the 1990's and early 2000's have characterised the dynamics of the number of bosons in the context of evaporative cooling and for Bose-Einstein condensates, proposing simple theories that are mean-field in spirit as they neglect correlations among the atoms [73–75]. Generally, there are three kinds of processes that can give rise to atom loss, which are classified according to the number of condensate particles that are involved. One-body losses are due to the interaction of condensate particles with thermal background particles that float in the vacuum chamber [76]. Two-body losses typically characterise experiments with atoms trapped in a metastable excited energy level or, in general, with molecules [77–80]. Other situations where the two-body loss process is relevant have also been reported in the literature, for instance with magnetic atoms like Chromium, or with alkali atoms trapped in the state of largest hyperfine spin—here this process is typically irrelevant for Rubidium but important for Caesium. In such cases, collisions between two particles can induce transitions towards internal levels with lower energy: the internal energy that is gained is converted into kinetic energy of the particles, that can then escape the trap. This contrasts to atoms in their lowest internal state; in such cases, even if a lower-energy bound state exists, the combined effects of energy and momentum conservation typically prevent relaxation to a bound state. Three-body losses can however always occur [74, 75, 81], and are the result of recombination processes where two atoms bind into a molecule and transfer part of their binding energy to a third atom in the form of kinetic energy, so that both the molecule and the atom can escape the trap. In recent years, there has been work focused on the engineering of loss processes, in order to have a higher degree of control, focusing in particular on one-body [82, 83] and two-body [84, 85] loss processes. A series of experimental works has also focused on the possibility of inducing losses in a localised region of space, typically a single site of an optical lattice [86].

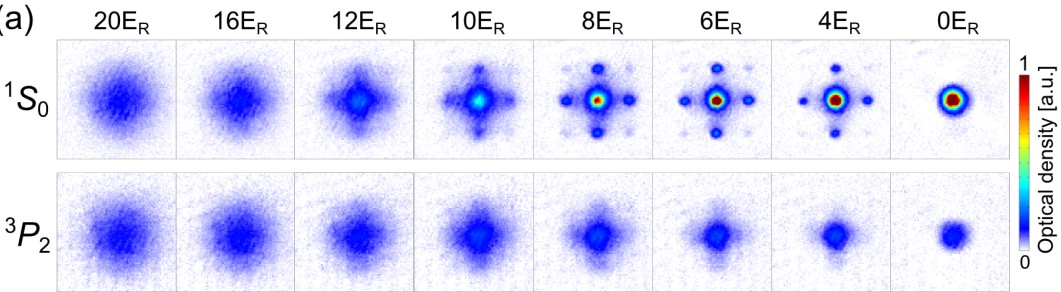

Figure 3: Time-of-flight images taken with different lattice depths, from the very deep case of 20 recoil energy $E_R$ to the case of no lattice 0 $E_R$; the first line corresponds to an atomic level that is not subject to dissipation, the second instead corresponds to a metastable energy level that is subject to strong two-body losses; the transition to a superfluid becomes less clear. From Ref. [89] [Copyright (2019) by the American Physical Society].

As a paradigmatic example, let us discuss the experiment reported in Ref. [87], where a bosonic Yb isotope is cooled in a three-dimensional optical lattice. Two-body inelastic atom losses at a controllable rate are induced via a single-photon association process: two atoms in the same lattice site are photoassociated into a molecular state and immediately dissociated. This process releases a high kinetic energy to the dissociated atoms, which eventually escape the optical lattice. By changing the intensity of the photoassociating laser, the rate of two-body losses is controlled with high fidelity. This then allows one to study the effect of two-body losses on the zero-temperature quantum phase transition between a Mott insulator and a superfluid, first reported in Ref. [88].

Figure 3 is taken from Ref. [89], where the authors present a series of time-of-flight absorption images obtained by changing the final lattice depth from the Mott insulator to the superfluid regime for two atomic energy levels characterised by a significantly different intrinsic two-body loss rate. For the dissipative energy level, the interference pattern that characterises the superfluid phase in the shallow lattice regime becomes blurred.

References [87,89] opened a new avenue for quantitative studies of the depletion of coherence in an atomic gas, as characterised by the formation of off-diagonal long-range order due to two-body loss events. We mention here Ref. [90] where a theoretical semiclassical description of the dissipative dynamics is proposed.

In Secs. 5.3.3 and 7.2 we will present a more thorough discussion of the theoretical and experimental developments associated to the open-quantum-system dynamics induced by losses.

**Atom heating.** Heating processes are the second open-quantum-system process that needs to be characterised, primarily because they drive the system towards featureless high-temperature states and wash out all interesting quantum effects. There are many microscopic processes that can give rise to heating effects. These include fluctuations of the laser field responsible for the trapping of the particles in an optical lattice, which can induce a noise dynamics of the trapping potential; collisions among the atoms, particularly those that release energy from excited internal states (the difference with loss processes is that here, after the collision, the reaction products are still trapped); incoherent scattering of the lattice light. The study of the heating dynamics due to incoherent light scattering for single atom systems is well understood [91, 92]. However, a key question of current interest is the study of how this heating dynamics occurs for many-body quantum states, where the interplay of few body heating processes and many-body physics can give rise to

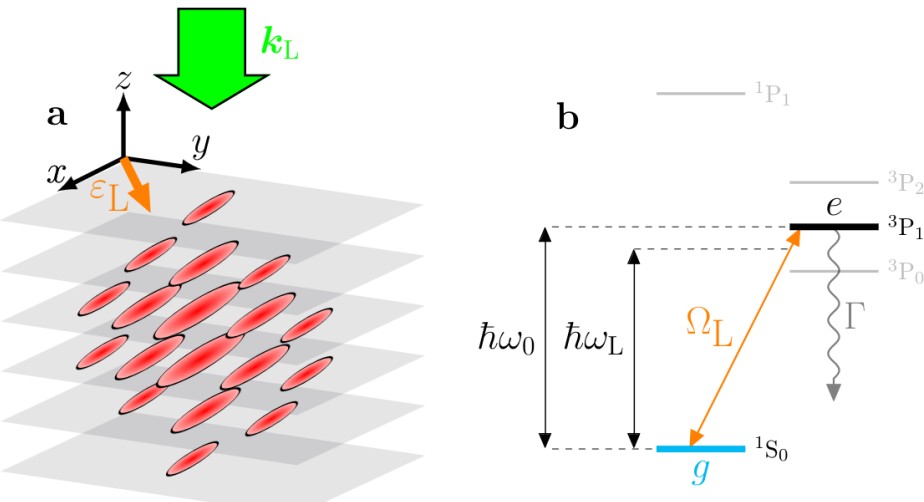

Figure 4: (a) An ultracold gas of bosonic $^{174}$Yb atoms is loaded in a two-dimensional array of one-dimensional systems. The green arrow represents a laser propagating through the systems and inducing spontaneous emission in a controlled fashion. (b) Details on the energy levels and laser couplings. The orange arrow represents a laser close to an atomic resonance that induces spontaneous emission at a rate $\Gamma \sim 520s^{-1}$. The atom is randomly recoiled and this destroys the quantum coherence of the gas. From Ref. [98].

interesting effects [93–95]. Related to that question is the study of the effect of making frequent measurements of the position of the atoms in optical lattices [96, 97]. Similar to the discussion of losses given above, there have recently been experiments where the heating dynamics could be studied in a controlled way. This can be done by illuminating the atoms with laser light that is resonant with an atomic transition, so that spontaneous emission can take place at a controlled rate.

As an example, we consider the experiments reported in Refs. [98, 99]. A bosonic Yb isotope is cooled in a stack of two-dimensional optical lattices, and exposed to a laser tuned close to an atomic resonance which induces spontaneous emission at a controllable rate. The spontaneous emission of photons induces a random recoil of the atom in the trap, that is responsible for the loss of coherence (see the sketch in the left panel of Fig. 4). The experiment shows the standard time-of-flight picture with coherent Bragg peaks that are washed out in time by the induced spontaneous emission processes. In this controlled experiment, the appearance of an anomalous decay time is also reported, and will be discussed in Sec. 7.1.

## 2.3 Cold atoms in optical cavities

As noted above, one particular case where many-body open quantum behaviour arises is with ultracold gases trapped in optical cavities. While there can be experiments where one uses the bare coupling between cold atoms and a cavity light mode [100], these coupling strengths are not tunable and it is hard to realise strong coupling. One can however use an alternate scheme, as discussed next, to realise a tunable form of matter-light interaction, which has led to a growing research field (see e.g. the reviews [57, 58]).

The key idea, proposed in Ref. [102], is to realise coupling between atoms and cavity modes via a two photon process, involving one cavity photon and one pump photon (see

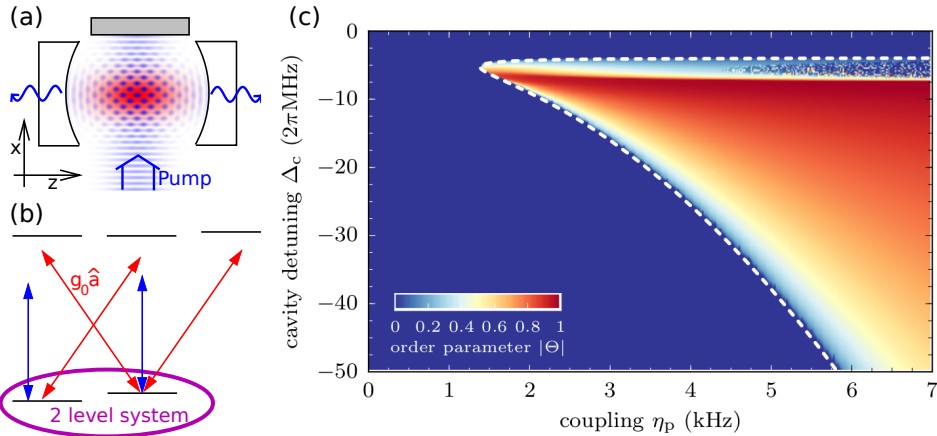

Figure 5: (a) Cartoon showing self organisation of atoms in an optical cavity, with transverse pumping. (b) Energy levels for Raman pumping scheme. Transitions between two low-lying atomic states are driven by a two photon process, involving scattering light between the pump and cavity. (c) Theoretical phase diagram of self organisation of atoms in an optical cavity, predicted by solving the Gross-Pitaevskii equation for atoms, coupled to a self-consistent cavity field. The coupling (horizontal axis) is controlled by the pump strength. The colour scale indicates the intensity of light in the cavity. From [101].

Fig. 5(a)). This idea has several advantages. Firstly, it allows control over the effective light-matter coupling strength, by tuning the laser pump strength. An additional benefit is that this scheme overcomes any "No-go theorem" associated with bounds on the strength of matter-light coupling compared to energy splitting [103]; this is because the coupling to light is controlled by the dipole matrix element of the virtual transition, rather than that of the transition between the two low-lying states. As a result it is possible to realise the Hepp–Lieb–Dicke superradiance transition [104, 105], where the cavity acquires a macroscopic photon field above a critical coupling strength. Such a transition was indeed observed in a realisation where the two low-lying atomic states correspond to different motional states of the atoms [106] (see Fig. 5(b)).

The Raman pumping scheme in fact permits even greater control over the effective Hamiltonian, beyond controlling the strength of matter-light coupling. In the rotating frame of the pump, the effective cavity frequency becomes the energy cost of scattering a photon from the pump to the cavity, so is tunable. Further, if one uses two separate pump beams for the two legs of the Raman transition, the effective energy splitting of the low-lying levels becomes the two-photon detuning [102], allowing control over this via the beat frequency between the two pump beams.

When combined with cavity geometries that permit control over multiple separate cavity modes [107], this Raman pumping scheme allows for considerable control. For example, experiments with two competing cavity modes have been able to realise a model with a continuous symmetry (associated with relative amplitude of the two cavity modes), and thus to explore spontaneous symmetry breaking of a continuous symmetry [108, 109]. When going from two cavity modes to many, further new features arise. By using a confocal cavity—a cavity where sets of transverse modes come in degenerate families— one can build transversely localised wavepackets from the different degenerate modes. By slightly adjusting the mirror spacing away from the confocal degeneracy point, one can then realise tunable-range interactions between atoms [110]. Recent work has further

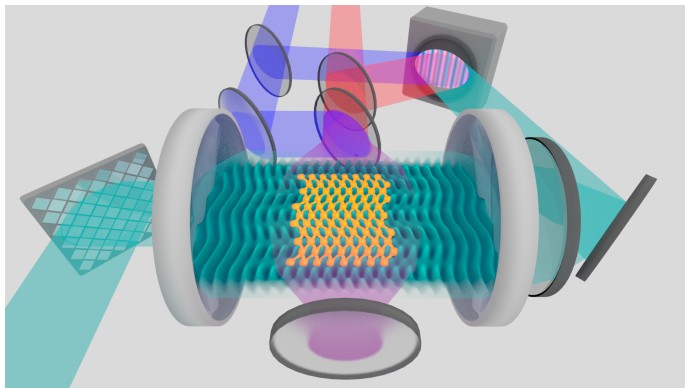

Figure 6: Cartoon of atomic gas in a multimode cavity. Self organisation into a density-wave pattern occurs, but because of the freedom allowed by the multimode cavity, the phase of the density wave can vary across the cavity. To allow full phase freedom for the density wave, a pump scheme with two colours of light is required (see Ref. [117] for details. [Image credit B. Marsh, Y. Guo. Based on experiment in Ref. [111]].

shown how the two ideas of short-range interactions and continuous symmetry breaking can be combined to realise a compliant optical lattice, that can distort, and thus has sound modes [111] (see Fig. 6). Several other interesting directions have been explored with similar experiments. By using positioning of atomic ensembles along the cavity axis and using a (nearly) concentric cavity, one can engineer programmable interactions between clumps [112], or one can use this to explore chaotic dynamics [113]. There are also proposals for how an associative memory and spin-glass physics can be realised with a confocal cavity [114–116].

In the context of a single mode cavity, when the atomic states involved are different motional states, the transition to a superradiant state is associated with the formation of an optical lattice. As such, there can also be a Mott insulator transition that occurs as the depth of this emergent lattice grows [118]. Very recently, there have also been work realising such spatial ordering using a degenerate Fermi gas [119].

As noted above, such experiments are intrinsically open quantum systems, due to the combination of the cavity loss, and the driving via the transverse pump. While many of the phenomena mentioned above (self organisation transitions in various geometries) can be understood equally in a closed system, there have also been clear experimental signatures that can only be understood by including dissipation. These include the observation of limit cycles [120, 121]. Much of the theoretical work on such systems—particularly those considering Bose-Einstein condensates in single-mode cavities—has been able to make use of mean-field treatments. However, there are particular questions—particularly when considering correlated states of atoms, such as Mott insulating states, where beyond-mean field approaches become necessary. In this context various approaches have been used, including matrix product states [122] and multi-configuration time-dependent Hartree (MCTDH) [123]. Beyond mean-field methods also become necessary when considering spatially extended systems in nearly confocal cavities, where fluctuations can significantly modify phase boundaries [124].

The design of new platforms may lead to the realisation of different forms of couplings to the bath that, while retaining the Lindblad form, involve more complex (long-range and/or correlated) jumps and feedback. A dissipative term in the Lindbladian with power-law-decaying correlations could in principle be realised in cold atoms inside a cavity. The

scheme, proposed theoretically in [125, 126], requires three-level atoms in the presence of a magnetic field gradient, and a Raman beam. The flexibility in the design of suitable jump operators means, in the context of quantum state preparation, the possibility to have access to a wider range of out-of-equilibrium many-body states or phases.

## 2.4   Rydberg Gases

Another special class of experiments with cold atoms is that involving atoms in highly excited "Rydberg states". When atoms are in highly excited states, this can significantly modify their optical properties as compared to ground state atoms. Most notably it leads to enhanced dipole-dipole interactions, due to the increased radius of the electronic orbit, and to enhanced lifetimes, as reviewed in Ref. [127].

Direct transitions from the atomic ground state to a Rydberg state are forbidden by selection rules, so that transitions are typically driven by a two-photon process, via driving with two laser frequencies. As discussed further below, one can also consider replacing one of these with an optical cavity mode. If the intermediate state is far off resonance, the fact that the process is a two-photon transition becomes unimportant, and one can consider an effective coherent driving of the ground state to Rydberg state. Because of the large dipole-dipole interaction between atoms in Rydberg states, the presence of one excited atom shifts the energy required to excite other atoms to the Rydberg state (see Fig. 7). When the drive frequencies are resonant with the optical transition, or detuned slightly below resonance, then the energy shift due to an excited atom moves the transition for a second atom further away from resonance. This leads to the "Rydberg blockade" [128], where the presence of one atom prevents others being excited over some range of distances. For a small atom cloud, this blockade means that only a single Rydberg excitation may be possible in the cloud: this allows one to gain the collective enhancement in matter-light coupling that comes from considering large numbers of atoms while still restricting to a two-level system of zero or one excitations. For larger clouds of atoms, multiple excitations will be allowed but separated by a blockade radius.

The ability to drive a single Rydberg transition in an ensemble of atoms provides a potential route to engineer many-body gates between atoms, and there have been proposals to use this for digital quantum simulation [130] of both closed-system and dissipative models. In addition to proposals for experiments that involve directly exciting a Rydberg state, there have been a number of proposals involving atoms in a controllable superposition of ground and Rydberg states. This induces "dressing" of the atom properties by the Rydberg state, providing a controllable route to enhance atom-atom interactions [131–133].

Including dissipative processes in Rydberg experiments is straightforward, by considering the effect of spontaneous radiative decay from the excited atomic states, including the Rydberg state. This gives a natural source of dissipation that is counterbalanced by the external laser drive. Depending on the parameter range considered (e.g. for Rydberg dressed atoms as noted above), one may sometimes adiabatically eliminate excited states to yield an effective dissipative model within a 'slow' subspace. This has allowed for a wide range of theoretical proposals using dissipative dynamics with Rydberg atoms [134, 135]. In several of these examples the role of the Rydberg blockade is inverted. This can be done by detuning the driving laser to frequencies above the transition frequency. In such cases, the presence of one excited atom moves the transition frequency for a subsequent atom closer to resonance. Both regimes (Rydberg blockade and facilitated dynamics) are a natural context to study "Kinetically constrained models" [134, 136–138], where the rate at which a process happens depends on the state of adjacent sites.

Experiments on arrays of Rydberg atoms [129, 139–141] have begun to explore these concepts . In the experiment in Ref. [141], a key role is played by dissipative processes

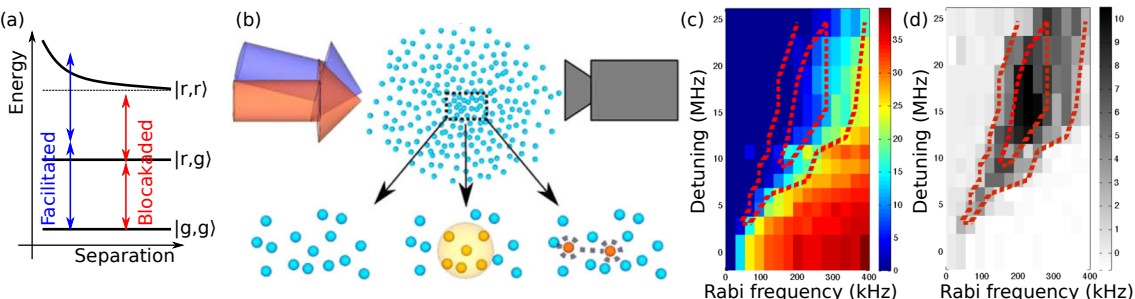

Figure 7: (a)Illustration of Rydberg blockade and facilitation for two atoms. The transition frequency between the state with a single Rydberg excitation, $|r, g\rangle$ and the state with two excitations, $|r, r\rangle$ depends on the spatial separation between the atoms. As a result, for red-detuned driving there is a blockade, as the transition is shifted out of resonance. In contrast for blue-detuned driving there can be a specific separation at which the transition is resonant, leading to facilitated dynamics. (b,c,d) Bimodal counting statistics for an ensemble of Rydberg atoms, from Ref. [129]. Panel (b) shows the experimental setup, with insets illustrating excitation processes for increasing pump frequency. Panels (c,d) show a measured phase diagram, found by measuring (c) the mean number of Rydberg excitations $\bar{n} = \langle n \rangle$, and (d) the Mandel Q parameter of the distribution of excitations $Q = \langle (n - \bar{n})^2 \rangle / \bar{n} - 1$. [Panels (b,c,d) adapted from Ref. [129] - Copyright (2014) by the American Physical Society]

which take atoms to inactive states. This leads to a dynamics that drives the system toward the critical point between a low excitation density "absorbing" state and an "active" state. These phenomena are discussed further in Sec. 6.2.4.

There has also been work that combines the use of Rydberg states with coupling to an optical cavity as described in the previous section. Here one of the two laser beams is replaced by coupling to an optical cavity, to create Rydberg polaritons [142]. By performing these experiments in a twisted optical cavity [143], this produced polaritons which behave as if in the presence of an effective magnetic field. Combined with the strong Rydberg interaction this led to the creation of a Laughlin state of polaritons. Since such experiments involve both (dissipative) cavities and excited states of atoms, there is again a clear potential to explore many-body physics of dissipative systems.

## 2.5   Superconducting circuits

Superconducting nano-circuits have played a fundamental role in the field of quantum technologies since the late 1990s, when the first designs based on charge/flux qubits where theoretically studied and experimentally realised. For an extensive review of the early developments in this platform, see Ref. [146]. Superconducting qubits consist of circuits which contain one or a few Josephson junctions in the quantum regime, with each Josephson junction (JJ) formed by two superconducting electrodes separated by a thin oxide layer. In the quantum regime a JJ is described by the Hamiltonian

$$\hat{H}_{JJ} = \frac{\hat{Q}^2}{2C} - E_J \cos \hat{\varphi}, \tag{5}$$

where $E_J$ is the Josephson energy associated to the superconducting flow through the junction, $C$ is the capacitance of the junction. The first term in the Hamiltonian is the electrostatic energy while the second is the phase-dependent contribution due to super-

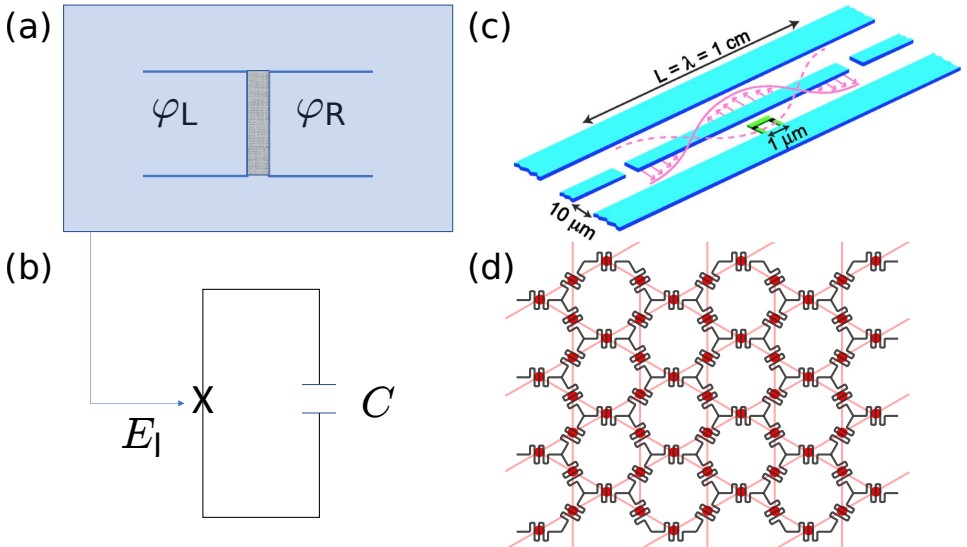

Figure 8: (a) A sketch of a Josephson junction, two superconductors with a difference in the phases of their order parameters $\varphi = \varphi_L - \varphi_R$ separated by a tiny insulating barrier. The junction (labelled by a cross, X) is embedded (b) in a circuit with additional elements. In this simple version only the contribution due to the finite capacitance of the junction is indicated. (c) the Josephson junction is embedded into a stripline resonator, leading to what is known as circuit-QED element. Adapted from [144]. (d) the c-QED element can be arranged in an array with the neighbouring cavities coupled by photon hopping. The gray lines indicate the arrangement of the transmission line realised in Ref. [28, 145]. The red circles indicate where transmon qubits would be placed, and the faint pink lines show the effective Kagome lattice.

currents. The non-trivial dynamics of this element arises from the commutation relation between the (canonically conjugate) charge $\hat{Q}$ and phase difference $\hat{\varphi}$

$$\left[\frac{\hat{Q}}{2e}, e^{\pm i\varphi}\right] = \pm e^{\pm i\varphi} . \tag{6}$$

A sketch of a JJ and the corresponding lumped circuit scheme as a non-linear LC-resonator is shown in Fig. 8 (panels (a) and (b)). A key feature of the JJ, important for quantum information processing, is that it is a non-linear, dissipationless element. The anharmonic spectrum of $H_{JJ}$ allows one to single out two levels that can be used to span the Hilbert space of the superconducting qubit. Different choices of the coupling parameters (i.e. the ratio $(2e^2/C)/E_J$) as well different ways to insert it in more complex structures have led to several different qubit designs (charge, flux, phase, ..., qubits). Restricting the dynamics to this computational subspace, the state of the qubit (whatever its origin is) will be labelled by the eigenstates of the Pauli matrix $\hat{\sigma}_z$.

The Josephson element illustrated in Fig. 8 (panels (a) and (b)), placed inside a stripline resonator, is at the heart of circuit-QED. The concept of circuit-QED is analogous to cavity-QED, but replacing optical cavities with microwave resonators, and replacing atoms with superconducting qubits. The microwave resonator, made of superconducting material, is shown in Fig. 8 (panel (c)) and the effective "giant atom" is realised by placing a superconducting nano-circuit inside the stripline resonator. Since the initial proposal in [144], circuit-QED has been a key enabling technology for the growth of solid-state quantum

information processing. A comprehensive description of the progresses and status of the art on circuit-QED can be found in the review by A. Blais *et al.* in [147].

Large arrays of Circuit-QED cavities have been theoretically proposed and experimentally realised as simulators of quantum many-body open systems. An sketch of a design for an array of cavities, realised in Ref. [28, 145] is shown in Fig. 8 (panel (d)). Here single cavities are arranged in a regular lattice in such a way to allow photons to hop between neighbouring sites. Note that the wiggly lines in this figure are the cavities, and the "nodes" are the coupling between different cavities.

A circuit-QED array can be modelled with a Hamiltonian that describes qubit-cavity coupling in each cavity, along with photon hopping between cavities. For a single cavity mode with frequency $\omega_c$, the corresponding Hamiltonian is $\hat{H}_c = \omega_c \hat{a}_i^\dagger \hat{a}_i$, where the operator $\hat{a}_i$ ($\hat{a}_i^\dagger$) annihilates (creates) a photon in the $i$-th cavity. The presence of Josephson elements inside each cavity leads to a strong effective non-linearity between photons. To account for it, it is enough to model the qubit as a few-level system, coupled to a cavity mode. The simplest such model is the Jaynes-Cummings model in which one mode of the cavity interacts with a two-level system:

$$\hat{H}_i^{(0)} = \frac{\varepsilon}{2}\hat{\sigma}_i^z + \omega_c \hat{a}_i^\dagger \hat{a}_i + g(\hat{\sigma}_i^+ \hat{a}_i + \hat{\sigma}_i^- \hat{a}_i^\dagger) \ , \tag{7}$$

$\hat{\sigma}_i^\pm$ are the raising/lowering operators for the two-level system and $\varepsilon$ denotes the energy difference between the two levels. In the rotating frame with respect to the uncoupled Hamiltonian the relevant quantity is the detuning $\Delta = \omega_c - \varepsilon$. The spectrum of Eq. (7) is anharmonic so that, effectively, the two-level system induces a repulsion between the photons in the cavity.

If the cavities are coupled, a kinetic hopping term $-t_H \sum_{\langle i,j \rangle}(\hat{a}_i^\dagger \hat{a}_j + \text{H.c.})$ should be added to the Hamiltonian, where $t_H$ is the hopping amplitude, and the sum over $\langle i,j \rangle$ denotes neighbouring sites. In the presence of hopping the Hamiltonian of the photons is still quadratic, but now describes photon modes delocalised over the whole lattice. In general, the Hamiltonian for an array of cavities is given by

$$\hat{H} = \sum_i \hat{H}_i^{(0)} - t_H \sum_{\langle i,j \rangle}(\hat{a}_i^\dagger \hat{a}_j + \text{H.c.}) \ . \tag{8}$$

The interplay between the different terms in the Hamiltonian is responsible for a rich dynamics. A strong effective nonlinearity between the photons turns the cavity into a turnstile device, where no more than one photon can be present on a given site at the same time. On the other hand, photon hopping between neighbouring cavities favours delocalisation and competes with photon blockade.

So far we discussed the Hamiltonian governing the unitary dynamics of the circuit-QED array. The open system dynamics arises because of losses, either photon leakage out of each cavity and/or decay of the qubit. Both these two terms are very well accounted for by a Lindblad term. The explicit form will be described in Sec. 3.1.5, Eq. (27). In the absence of external driving, the steady state will be the one in which all the cavities have no photons and the qubits are in the ground state. In order to reach a non-trivial steady-state, cavities should refilled by means of an external coherent or incoherent drive.

Additional details of the experimental realisation of coupled cavity arrays will be discussed in Sec. 5.3.4 and can be found in the review [55].

## 2.6 Polaritons and photons

In most of the platforms discussed so far, light has been considered as a source of external pumping or, via photon emission, as a loss process. However—as already mentioned in

Sec. 2.3 on cold atoms in optical cavities—light modes in an optical cavity can also play a role as part of the quantum system to be controlled and studied. Such an idea is key to the physics of polariton and photon condensates. This topic is particularly large, with comprehensive reviews covering a variety of aspects of polariton physics [16, 17, 148], and books e.g. [18, 19] covering key topics. As such, we will only provide a brief and partial summary of essential concepts here, to serve as context for places where we later discuss methods and ideas that were first developed for polariton and photon condensates. For a thorough review of the status of polariton physics, we refer the reader to one of these many reviews.

Microcavity polaritons are the hybrid particles that result from strong coupling between photon modes trapped in an optical microcavity and material excitations, generally excitons (bound electron-hole pairs). When light-matter coupling is strong, one can no longer consider excitons and photons separately as eigenstates, and instead one has new eigenstates, lower and upper polaritons, which are symmetric and anti-symmetric superpositions of photons and excitons. As a result, these particles have hybrid properties of both matter and light. This means that in contrast to bare photons they can show nonlinear interactions. Such strongly coupled polaritons have now been realised in a wide variety of material systems—such realisations have been reviewed for classes of materials including semiconductor quantum wells [16, 149], organic materials [150], two-dimensional materials [151], and recently hybrid Perovskites [152]. In addition to exciton-polaritons—on which we focus in this section—there can be polaritons involving a variety of other material excitations, for an overview, see [153].

As shown in Fig. 9(a), a typical microcavity experiment involves photons confined between mirrors formed of distributed Bragg reflectors (DBRs)—layers of alternating dielectric constant [18]. Such DBRs have high reflectivity, thus leading to trapped photon modes with long lifetimes. Because the relevant photon modes are confined in a microcavity, the photons have an effective quadratic dispersion for small in-plane momentum $k_\perp$. That is, one can write the photon dispersion as:

$$\omega_{k_\perp} = c\sqrt{k_\perp^2 + (n2\pi/L)^2} \simeq \omega_{k_\perp=0} + \frac{k_\perp^2}{2m_{\text{eff}}} + \mathcal{O}(k^4)$$

where $L$ denotes the effective length of the cavity (see Fig. 9(a)), and $n \in \mathbb{N}$ determines the mode index. When the effective cavity length is chosen to make $\omega_{k_\perp=0}$ close to resonant with typical exciton energies (i.e. close to the optical bandgap for semiconductors), then the resulting effective mass photon $m_{\text{eff}}$ is small, typically $10^{-4}$ times the electron mass. This small mass is inherited by the polariton (see Fig. 9(b)).

While DBR mirrors can have high quality factors (i.e. low losses), photons will ultimately leak out of the mirrors. As discussed below, this makes the polariton system an open system. This loss is also important in allowing access to information about the polariton system. One may either image the far field of the emission (i.e. angle distribution, as suggested in Fig. 9(a)), or use lenses to image the real space distribution of polaritons in the cavity.

**Polariton condensation.** Because polaritons are a superposition of a photon and a bosonic excitation, they have (approximately) bosonic properties. In particular, one can have a state in which a single polariton mode is macroscopically occupied, analogous to the equilibrium state of Bose-Einstein condensation. However, since polaritons are part photon, they are inevitably subject to losses, and so one requires external pumping to compensate this loss. In most experiments seeking condensation, pumping occurs via a non-resonant laser pump, to a highly excited state, followed by subsequent relaxation to

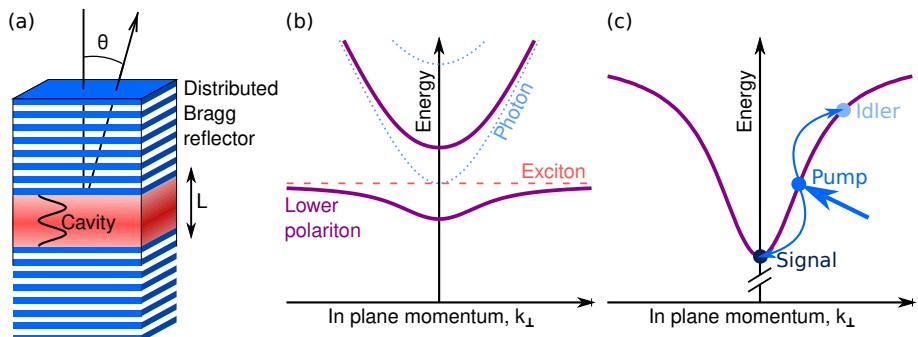

Figure 9: Microcavity polaritons. (a) Cartoon showing a microcavity structure, where an active layer is confined between two distributed Bragg reflectors. (b) Polariton dispersion, arising from strong coupling between photons and excitons. (c) Illustration of parametric scattering. An external pump resonantly excites polaritons at a particular wavevector and energy, and then scatters to signal and idler modes.

populate excitons. This means that rather than the thermal equilibrium Bose-Einstein condensate, polaritons generally adopt a non-equilibrium steady state set by a balance of drive and dissipation [154–156]. Such experiments have indeed allowed the observation of Bose-Einstein condensation of polaritons [157]. The extent to which the system is out of equilibrium can vary, with some experiments with very long polariton lifetimes showing behaviour far closer to equilibrium [158]. In thermal equilibrium, the critical temperature required for Bose-Einstein condensation depends inversely on the mass, hence the small effective polariton mass leads to relatively high critical temperatures, which can range from tens of Kelvin to room temperature depending on the material.

The desire to understand how the properties of a non-equilibrium condensate differ from those of an equilibrium condensate—particularly questions related to superfluidity and phase coherence [159]—helped motivate the development several of the theoretical tools we discuss below. Although the underlying theory is quantum mechanical, much of the phenomenology of polariton condensates—particularly spatial pattern formation and dynamics—is classical. That is, these systems can often be understood in terms of a classical theory of nonlinear waves. This theory takes the form of a driven-dissipative version of the Gross-Pitaevskii equation. We will discuss in Secs. 4.3.3 and 4.6.1 how such a theory can arise as the the classical saddle point of a path integral, or equivalently, the mean-field approximation of a Lindblad master equation. Given our focus on many body quantum systems, we will not however discuss further the varieties of behaviour that can result from this equation. For an extensive discussion see Ref. [17].

In addition to experiments with non-resonant excitation, there has also been significant work on optical parametric amplification and oscillation, with resonant drives [160, 161]. In such experiments, a resonant coherent pump directly excites polaritons, which can then scatter coherently to signal and idler modes (see Fig. 9(c)). While the pump here is resonant, this scattering process introduces a free phase, so the transition to an optical parametric oscillation state can be considered as a symmetry-breaking phase transition.

**Quantum polariton and polaritonic lattices.** As noted above, much of the physics of polaritons is essentially a theory of nonlinear classical waves. There are however some limits in which quantum effects can be seen for polaritons. Experiments on polaritons in highly confined geometries [162, 163] observed weak anti-bunching of polariton emission, resulting from polariton-polariton interactions. Such experiments involved confining polaritons by

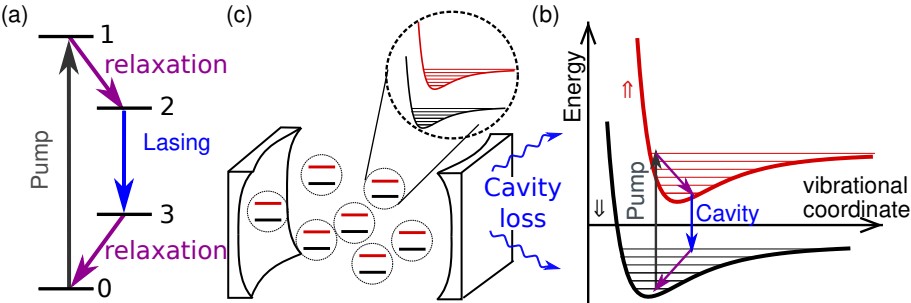

Figure 10: (a)Four-level lasing model. (b) Cartoon of dye laser/photon condensate experiments, with molecules in an optical cavity. Both the electronic state (ground/excited) and vibrational state of the molecules are relevant in coupling to light. (c) Effective lasing scheme using vibrational levels for a dye laser or photon condensate.

using one planar mirror, and a second mirror on the tip of an optical fibre.

Polaritons can also be confined by creating micropillars [164] by etching the edges of the structure sketched in Fig. 9(a). These pillars can then also be engineered into lattices, providing another platform for the coupled cavity arrays discussed in the previous section. Experiments with such arrays have demonstrated topological non-interacting band structures, and lasing in topologically protected modes [165–167]. For reviews of this topic, see Refs. [168, 169].

In addition to improving confinement of polaritons, there are various routes that can be used to increase the intrinsic polariton-polariton interaction strength. Several ideas have been explored in this direction. We mention three of these here. *Dipolar polaritons* [170, 171]—These involve three-way hybridisation between photons, direct excitons (electrons and holes in the same quantum well), and indirect excitons (electrons and holes confined in two different quantum wells). Spatial separation of electrons and holes leads to a permanent dipole moment, increasing interactions, but reduces the coupling to light. The three-way hybridisation allows one to control the trade-off between these two effects. *Rydberg polaritons* [172, 173]— These involve hybridisation of photons with Rydberg excitons, i.e. highly excited excitonic states, analogous to Rydberg states of atoms as discussed above. *Trion polaritons* [174]—These involve polaritons in electrically doped materials, so that the optical transitions involved are transitions from an electron to a trion (two electrons and a hole).

An alternative route toward stronger quantum effects is to consider cases which involve few emitters strongly coupled to light. Such a scenario can be realised for various platforms, including coupling a quantum dot to a cavity in a photonic crystal [175, 176]. For a review of similar semiconductor platforms, see Ref. [177]. More recently, single emitter strong coupling has been realised with organic molecules coupled to plasmon modes in metal nanoparticles [178].

**Photon condensation and lasing.** A class of experiments that are closely related to those studying polariton condensates is work on photon condensation. The distinction between the two is whether the coupling between light and matter is strong or weak. Photon condensation occurs in the regime of weak coupling, where one can regard the photon modes as the eigenstates. Nonetheless, photon condensation depends on coupling to matter, albeit through incoherent emission and absorption.

In discussing the case of photon condensation and weak light matter coupling, one comes very close to describing a far wider class of driven dissipative systems: photon

lasers. In most typical applications of lasers, a quantum description is not really relevant or necessary. As with polariton condensate, a classical theory of nonlinear waves (with nonlinearity due to the gain medium) can often be sufficient [179]. However, one always derive an underlying quantum theory of lasing. One such theory is a model of four-level emitters coupled to a single cavity mode (see Fig. 10(a)), where coherent driving occurs on the 0–1 transition, accompanied by incoherent relaxation on the 1–2 and 3–0 transitions, while lasing occurs on the 2–3 transition, due to coupling to a cavity mode. Such a theory—described by a Lindblad master equation—can then be treated in various levels of approximation, ranging from fully classical, to semiclassical (including spontaneous as well as stimulated emission), and fully quantum—see Refs. [35–37, 180] for a detailed discussion. Quantum treatments can be relevant for understanding photon statistics. One can further make a connection between such a model and some of the generalised Dicke models discussed in the context of cold atoms in cavities. These models can show distinct lasing and superradiant phases [181–183].

For experiments on photon condensation [184], one may understand such experiments as an atypical regime of a dye laser [185]. In a dye laser, the gain medium consists of organic molecules, where electronic transitions are dressed by vibrational modes (see Fig. 10(b,c)). The photon condensate results from operating in a parameter regime where most photons emitted by the molecules can be reabsorbed, as the process of repeated absorption and re-emission is what leads to thermalisation. In contrast, a standard dye laser is designed so that the absorption and emission are spectrally separated, so that laser light emitted by the photons will not be reabsorbed. When absorption and emission are strong enough compared to photon loss rates, the distribution of photons in a cavity can approach a quasi-equilibrium state. In contrast to true thermal equilibrium—in which the chemical potential for photons is stuck at zero—the photon condensate has a chemical potential set by the balance between external driving and loss. Combining this non-zero chemical potential with the effective quadratic dispersion of photons in a cavity leads to the possibility to realise a form of non-equilibrium Bose-Einstein condensation [184].

## 3 Theoretical frameworks

In this Section we set out the theoretical frameworks that will underpin these Lecture Notes. We first introduce the master equation—an equation of motion for the density matrix, and some of its various reformulations. We then discuss the Schwinger–Keldysh path integral, an alternative framework, and we show how the two are related. We also discuss the quantum trajectory approach, based on stochastic equations of motion for a pure system state, such that ensemble-averaged properties recover those predicted by the master equation. We conclude with a short summary of a number of other frameworks which are used, but which are not so relevant to the rest of these Lecture Notes.

### 3.1 Lindblad master equation

In an open quantum system, a key requirement is to find a way to model the dynamics of the system without having to explicitly represent the dynamics of the environment to which the system is coupled. Indeed, if one includes in the description the state of the environment as well as the system, one generally has an entangled wavefunction, which implies considerable difficulties when proceeding to its resolution. Using a Schmidt decomposition [186], this

state can always be written in the form

$$|\Psi\rangle = \sum_\mu \sqrt{p_\mu} \, |\psi_{S,\mu}\rangle \, |\phi_{E,\mu}\rangle \, , \tag{9}$$

where $p_\mu > 0$ is the real-valued and positive probability of a given state, $|\psi_{S,\mu}\rangle$ are a set of orthonormal states of the system $S$, and $|\phi_{E,\mu}\rangle$ are orthonormal states of the environment $E$ (i.e. $\langle\psi_{S,\mu}|\psi_{S,\nu}\rangle = \delta_{\mu,\nu}$ and $\langle\phi_{E,\mu}|\phi_{E,\nu}\rangle = \delta_{\mu,\nu}$). As noted above, except in very special cases, this sum contains multiple terms, and thus the state is entangled.

The open-quantum-system paradigm is to assume we are only interested in expectations of operators on the system. As such, one may see that using the state above, one has:

$$\langle\Psi|\hat{O}_S|\Psi\rangle = \sum_\mu p_\mu \langle\psi_{S,\mu}|\hat{O}_S|\psi_{S,\mu}\rangle = \mathrm{Tr}\left[\hat{O}_S\hat{\rho}_S\right], \qquad \hat{\rho}_S \equiv \sum_\mu p_\mu \, |\psi_{S,\mu}\rangle\langle\psi_{S,\mu}| \, . \tag{10}$$

This demonstrates that the expectation of system operators is determined fully by the system density matrix $\hat{\rho}_S$ as defined above, that is generically mixed: if one can find it, one can proceed to calculate all such observables. Since we are not interested in other density matrices, we will suppress the suffix $S$ and write $\hat{\rho}$ for the system density matrix in the rest of these Lecture Notes.

The key problem is then to find how $\hat{\rho}$ evolves in time without needing to first find the wavefunction of the system and environment. In general this is a challenging task, however for a Markovian open quantum system, we can describe the ensemble-average state of the system by the density matrix $\hat{\rho}$, which evolves under a time-local Liouville–von Neumann equation. For all the situations we consider in these Lecture Notes, the equation of motion for the density matrix takes Lindblad form [187]:

$$\frac{d\hat{\rho}}{dt} = -i[\hat{H}, \hat{\rho}] + \sum_\mu \left( \hat{L}_\mu \hat{\rho} \hat{L}_\mu^\dagger - \frac{1}{2}\left\{\hat{L}_\mu^\dagger \hat{L}_\mu, \hat{\rho}\right\} \right) \, , \tag{11}$$

where the coherent dynamics generated by the system Hamiltonian $\hat{H}$ competes with the dissipative processes described by a set of jump operators $\hat{L}_\mu$. We give a derivation of this equation in Appendix A. Here we review some of the key properties of the Lindblad evolution and provide few examples of systems of interest, such as lattice models of quantum spins, bosonic or fermionic particles with different types of jump operators. It is often convenient to introduce the shorthand:

$$\hat{\mathcal{D}}[\hat{L}_\mu, \hat{\rho}] = \hat{L}_\mu \hat{\rho} \hat{L}_\mu^\dagger - \frac{1}{2}\left\{\hat{L}_\mu^\dagger \hat{L}_\mu, \hat{\rho}\right\} \, , \tag{12}$$

to describe the dissipator corresponding to given jump operator $\hat{L}_\mu$. In most of the examples we will discuss, each of the dissipation terms acts on a single site, and so in general we will label jump operators by site label $i$ as well as a label $\mu$ will be used if multiple operators act per site.

The master equation as written has various notable properties: it preserves the normalisation of the density matrix $\mathrm{Tr}[\hat{\rho}] = 1$, because $\mathrm{Tr}[\frac{d}{dt}\hat{\rho}] = 0$. It preserves Hermiticity, i.e. the relation $\mathrm{Tr}[\frac{d}{dt}\hat{\rho} - (\frac{d}{dt}\hat{\rho})^\dagger] = 0$ is correct as long as $\hat{\rho} = \hat{\rho}^\dagger$. The Lindblad master equation also preserves positivity of the eigenvalues of $\hat{\rho}$, as well as a stronger constraint—complete positivity—which means that not only is $\hat{\rho}$ positive, but also any combined density matrix of $\hat{\rho}$ and another system remains positive. The Lindblad form of the master equation is in fact the most general form of Markovian master equation that can guarantee all three of these properties.

The master equation as written above is linear in the system density matrix, so can be formally considered to take the form $\frac{d}{dt}\hat{\rho} = \hat{\mathcal{L}}[\hat{\rho}]$, where $\hat{\mathcal{L}}[\cdot]$ is the Lindbladian superoperator (such Lindbladian superoperators are a subset of "Liouvillian" superoperators, the Liouvillian refers to any superoperator describing the density matrix evolution). This superoperator describes a linear operation mapping operators into operators and thus can in principle be represented as a rank four tensor, namely an object with four indices. The master equation is thus amenable to a formal solution introducing the exponential of the Lindbladian superoperator:

$$\hat{\rho}(t) = \exp\left(\hat{\mathcal{L}}t\right)[\hat{\rho}(0)]. \tag{13}$$

### 3.1.1 Symmetries

Symmetries play an important role in understanding the collective behaviour of many-body systems, particularly in regard to spontaneous symmetry breaking as discussed in Sec. 6. In closed systems, one can simply consider the symmetries of the Hamiltonian. That is, a symmetry corresponds to a unitary transform $\hat{S}$ under which the Hamiltonian is invariant, $\hat{S}^\dagger \hat{H} \hat{S} = \hat{H}$, or, equivalently, which commutes with the Hamiltonian $[\hat{H}, \hat{S}] = 0$, and is thus conserved under the closed-system dynamics thanks to the Ehrenfest theorem. In the case of continuous symmetries, one may also identify the generator of the symmetry as a conserved quantity, in accordance with Noether's theorem. When conserved quantities exist, one can divide the Hilbert space into different sectors, corresponding to eigenvalues of the conserved quantity. For an initial state within a given sector, the dynamics will always remain within that sector.

For open systems described by a Lindblad master equation, the properties of symmetries are more subtle [188,189]. One may distinguish two classes of symmetries:

**Weak symmetries.** In this case, there exists a unitary transform $\hat{S}$ such that $\hat{\mathcal{L}}[\hat{S}\hat{\rho}\hat{S}^\dagger] = \hat{S}\hat{\mathcal{L}}[\hat{\rho}]\hat{S}^\dagger$. That is, the total Lindbladian is invariant under such a symmetry.

**Strong symmetries.** In this case one has that the Hamiltonian and each of the jump operators are individually invariant under the symmetry. Using the notation of Eq. (11) this means that $[\hat{H}, \hat{S}] = 0$ and additionally $[\hat{L}_\mu, \hat{S}] = 0$.

Obviously all strong symmetries are also weak symmetries, but not vice versa. The most significant distinction between a weak and strong symmetry comes when considering the question of conserved quantities, which only truly exist for a strong symmetry. Indeed one can use the Lindblad equation (11) to write an equation of motion for the expectation value of any operator $\hat{O}$, $\langle\hat{O}\rangle = \text{Tr}\left[\hat{O}\hat{\rho}\right]$ to obtain

$$\frac{d\langle\hat{O}\rangle}{dt} = i\langle[\hat{H}, \hat{O}]\rangle + \frac{1}{2}\sum_\mu \langle\hat{L}_\mu^\dagger [\hat{O}, \hat{L}_\mu]\rangle + \frac{1}{2}\sum_\mu \langle[\hat{L}_\mu^\dagger, \hat{O}] \hat{L}_\mu\rangle, \tag{14}$$

from which it follows that in order to be associated to a conserved quantity of Lindblad dynamics an operator needs to commute with both Hamiltonian and each jump operator.

As a consequence of the above, for problems with a weak (but not strong) symmetry, one may find a unique steady state, while problems with a strong symmetry will have multiple steady states, corresponding to different values of the conserved quantity (see also Refs. [190–195] for a more in-depth discussion on the uniqueness of the steady state and Refs. [196–198] for master equations with multiple entangled steady states). Despite this distinction, a weak symmetry does still lead to a block-diagonal structure of the Lindbladian.

To illustrate the block-diagonal structure for weak symmetries, we may consider a simple example of a single bosonic mode, with incoherent pumping and dissipation:

$$\hat{H} = \omega_c \hat{a}^\dagger \hat{a}, \qquad \hat{L}_- = \sqrt{\kappa_-}\hat{a}, \quad \hat{L}_+ = \sqrt{\kappa_+}\hat{a}^\dagger. \tag{15}$$

Such a model has a weak symmetry under $\hat{S} = e^{i\theta\hat{a}^\dagger\hat{a}}$. One may see that additionally $\hat{S}$ commutes with the Hamiltonian for this example, but $\hat{S}$ does not commute with the jump operators $\hat{L}_{-,+}$. In the absence of the dissipative terms, the symmetry would imply conservation of the number operator, $\hat{n} = \hat{a}^\dagger\hat{a}$. However, the pump and decay processes change the number, so the number is not conserved by the master equation, as anticipated for a weak symmetry. One can however note that if one considers writing the density matrix in the number basis, $\hat{n}\ket{n} = n\ket{n}$, and one writes $\rho_n^{(k)} = \bra{n}\hat{\rho}\ket{n+k}$, one finds that the equation of motion for $\rho_n^{(k)}$ only involves other elements of the density matrix with the same $k$. Thus, there is a block-diagonal structure, which follows from the weak symmetry. The steady state corresponds to the $k=0$ sector, and there is a unique solution within this sector, since particle number is not conserved.

### 3.1.2 Vectorisation, eigenstates and steady states

As noted above, the Lindblad master equation is a linear differential equation for the system density matrix. As such its behaviour can be understood in terms of the eigenvalues and eigenvectors of this linear operator. Such analysis will be of particular importance in Sec. 6 where we study the nature of dissipative phase transitions [199,200], and in Sec. 7 where we study the approach to the steady state. Here we introduce these eigenvectors and values, by means of the procedure of "vectorisation" of the density matrix.

Vectorisation amounts to writing operators $\hat{\rho}$ as vectors $\ket{\rho}\rangle$, and superoperators $\hat{\hat{\mathcal{L}}}[\cdot]$ as operators $\hat{\mathcal{L}}$. These objects live in a vector space with a scalar product equivalent to a trace of the product of operators, $\langle\!\langle a|b\rangle\!\rangle = \text{Tr}[\hat{a}^\dagger\hat{b}]$. After vectorisation the master equation takes the form

$$\frac{d}{dt}\ket{\rho}\rangle = \hat{\mathcal{L}}\ket{\rho}\rangle.$$

It is important to note that the matrix $\hat{\mathcal{L}}$ appearing here is non-Hermitian and can be constructed directly from the form of the master equation, or can be done at the level of operators extending the standard second quantisation formalism. For examples, see [201–208].

In this vectorised form, we can now directly consider the eigenspectrum of $\hat{\mathcal{L}}$. As it is a non-Hermitian matrix it has distinct right-eigenvectors $\ket{r_\mu}\rangle$ and left-eigenvectors $\langle\!\langle l_\mu|$, with corresponding complex eigenvalues $\lambda_\mu$ (for a broad overview on non-Hermitian matrices, see Ref. [27]). These are defined as:

$$\hat{\mathcal{L}}\ket{r_\mu}\rangle = \lambda_\mu\ket{r_\mu}\rangle, \tag{16}$$

$$\langle\!\langle l_\mu|\hat{\mathcal{L}} = \lambda_\mu\langle\!\langle l_\mu|, \tag{17}$$

where $\langle\!\langle l_\mu|, \ket{r_\mu}\rangle$ are generally biorthogonal. The second equation can also be written in adjoint form as $\hat{\mathcal{L}}^\dagger\ket{l_\mu}\rangle = \lambda_\mu^*\ket{l_\mu}\rangle$. We choose a normalisation such that they are also biorthonormal:

$$\langle\!\langle l_\mu|r_\nu\rangle\!\rangle = \delta_{\mu\nu}. \tag{18}$$

To motivate the interest of this vectorised notation, we show here that within this framework it is very easy to prove that the Lindblad master equation must have at least one steady state. As noted earlier, the master equation is trace preserving, which means in superoperator form $\text{Tr}\left[\hat{\hat{\mathcal{L}}}[\hat{\rho}]\right] = 0$. In vectorised form this reads $\langle\!\langle\mathbb{1}|\hat{\mathcal{L}}\rho\rangle\!\rangle = 0$. As

this is valid $\forall \hat{\rho}$, it follows that $\langle\langle \mathbb{1} |$ is always a left-eigenvector of $\hat{\mathcal{L}}$ with zero eigenvalue, $\langle\langle \mathbb{1} | \hat{\mathcal{L}} = 0$, and we identify it with the $\mu = 0$ left-eigenvector: $\langle\langle \mathbb{1} | \equiv \langle\langle l_0 |$ and $\lambda_0 \equiv 0$. As the eigenvalues $\lambda_\mu$ are common for left and right-eigenvectors, there must be also at least one right-eigenvector with zero eigenvalue. We assume here that this right-eigenvector is unique and identify it with the vectorised form of the steady state density matrix, $\hat{\rho}_{ss} | r_0 \rangle\rangle \equiv | \rho_{ss} \rangle\rangle$. There are interesting cases in which the Lindbladian has multiple steady states (see for example Ref. [209]), to all of them should correspond a different left eigenvector of $\hat{\mathcal{L}}$. It follows from the orthonormality condition that, with the normalisation fixed by the definition $\langle\langle \mathbb{1} | \equiv \langle\langle l_0 |$, $\hat{\rho}_{ss}$ is normalised $\mathrm{Tr}[\hat{\rho}_{ss}] = \langle\langle l_0 | \rho_{ss} \rangle\rangle = 1$.

From the orthonormality condition (18) and from the assumption of unique steady state, it follows that all the right eigenstates different from the steady state $| r_{\mu\neq0} \rangle\rangle$ are traceless: $\langle\langle \mathbb{1} | r_{\mu\neq0} \rangle\rangle = \delta_{0,\mu\neq0} = 0$. We can now interpret the eigenstates of the Lindbladian different from the steady state as decay modes of deviations from the stationary state. Suppose now that at $t = 0$ the system starts in some state $| \rho(0) \rangle\rangle$ that is not the stationary state. After expanding it onto the basis of right eigenvectors of $\hat{\mathcal{L}}$, using the vectorised form of Eq. (13) one can deduce that at later times, the system density matrix will be given by

$$| \rho(t) \rangle\rangle = | \rho_{ss} \rangle\rangle + \sum_{\mu\neq0} c_\mu e^{\lambda_\mu t} | r_\mu \rangle\rangle , \qquad (19)$$

with $c_{\mu\neq0} = \langle\langle l_\mu | \rho(0) \rangle\rangle$ and $c_0 = \langle\langle \mathbb{1} | \rho_{ss} \rangle\rangle = 1$ by orthonormality (18). We can thus interpret each Lindbladian eigenmode $\mu \neq 0$ as a possible dynamical decay mode of some initial deviation from the steady state, with a decay rate given by $-\mathrm{Re}\,\lambda_\mu$. In general, a given decay mode will involve both diagonal elements of the density matrix in the energy-eigenstates basis (i.e. populations) as well as off-diagonal elements (i.e. coherences).

A special role is played by the eigenmode with the least-negative real part (excluding zero), which corresponds to the mode that decays most slowly. Thus, at long times (when all other modes have decayed) this mode controls the asymptotic decay to the steady state. As such, the real part of this eigenvalue is often referred to as the Asymptotic Decay Rate (ADR):

$$\lambda_{\mathrm{ADR}} = \min_{\mu>0} \left( -\mathrm{Re}\,\lambda_\mu \right). \qquad (20)$$

It is also known as the Lindbladian gap or Liouvillian gap (as illustrated in Fig. 11). The behaviour of this eigenvalue will play a key role in the discussion of how dark states are approached at long times (see Sec. 5.2.2), and to understand the behaviour of symmetry breaking for dissipative phase transitions (see Sec. 6).

### 3.1.3 Multi-time correlations

As discussed above, the Lindblad master equation can be used to find the steady state of the system, or to find how the system evolves toward its steady state. In addition, we will also want below to consider multi-time correlations. For example, the fluorescence spectrum (i.e. inelastic scattering of light) from a driven system is determined by knowing the two-time correlations of a system. We focus here on two-time correlations, and so the task we want to realise is to write:

$$\left\langle \hat{O}_\mu(t+\tau)\hat{O}_\nu(t) \right\rangle \equiv \mathrm{Tr}_{S+E} \left[ e^{i\hat{H}\tau} \hat{O}_\mu e^{-i\hat{H}\tau} \hat{O}_\nu \hat{\rho}_{S+E}(t) \right], \qquad (21)$$

where $\hat{H} = \hat{H}_S + \hat{H}_E + \hat{H}_{S-E}$ as discussed above. The challenge, discussed below, is how to write $\langle \hat{O}_\mu(t+\tau)\hat{O}_\nu(t) \rangle$ without needing to explicitly reintroducing the environment.

Calculating multi-time correlations from a master equation relies on the "quantum regression theorem" [180, 187, 211]. This starts by considering that the Lindblad master

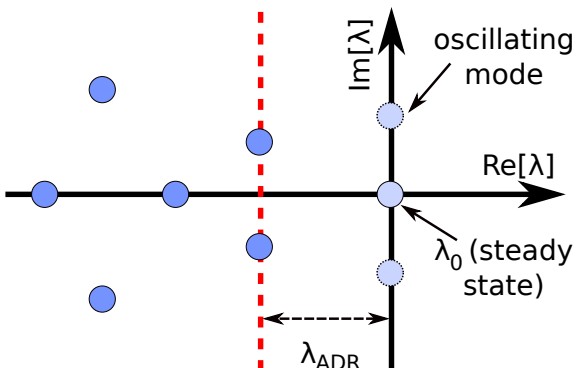

Figure 11: Sketch of the Lindbladian spectrum. We see the eigenvalues have negative real-part (describing decay modes, guaranteed by the complete positivity of the Lindblad map) and come in general in pairs of complex conjugate eigenvalues. The stationary state is characterised by a zero eigenvalue (zero real part and zero imaginary part) whose existence is also guaranteed by the mathematical structure of this superoperator [Based on figure from Ref. [210]].

equations means that one knows how to write the time-evolution of a single operator in the form $\langle \hat{O}_\mu(t+\tau) \rangle = \sum_\lambda f_{\mu\lambda}(\tau) \langle \hat{O}_\lambda(t) \rangle$. This condition is always true for the Lindblad master equation, as measurement of a sufficient number of operators $\hat{O}_\lambda$ at time $t$ (nb, $\hat{O}_\mu$ and $\hat{O}_\lambda$ are generically different operators) allows full tomography of the density matrix, and hence allows determination of $\langle \hat{O}_\mu(t+\tau) \rangle$. That is, by measuring enough operators one can determine the density matrix, and then evolve that density matrix with the Lindbladian, and thus find the expectation at time $t+\tau$. We will discuss this further in Sec. 3.4.2 in terms of the adjoint equation for operators. The quantum regression theorem then states one can write two-time correlation functions by using the same coefficients $f_{\mu\lambda}(\tau)$ to give:

$$\left\langle \hat{O}_\mu(t+\tau)\hat{O}_\nu(t) \right\rangle = \sum_\lambda f_{\mu\lambda}(\tau) \left\langle \hat{O}_\lambda(t)\hat{O}_\nu(t) \right\rangle. \tag{22}$$

An equivalent—but often more practical—statement of the same idea is to write:

$$\left\langle \hat{O}_\mu(t+\tau)\hat{O}_\nu(t) \right\rangle = \mathrm{Tr}\left[ \hat{O}_\mu \exp\left( \hat{\mathcal{L}}\tau \right) \left[ \hat{O}_\nu \hat{\rho}(t) \right] \right]. \tag{23}$$

That is, one can calculate two-time correlations by acting with the first operator, continuing to propagate the resulting object $\hat{O}_\nu \hat{\rho}$ under the Lindbladian, and then measuring the second operator. Multi-time correlations follow via the obvious generalisation. Given the discussion in the previous section, one may anticipate that all such correlation functions can be expressed in terms of the eigenvalues and vectors of the Lindbladian; we will discuss this point in detail in Sec. 6.1.3.

### 3.1.4   Thermalisation

As discussed in Sec. 1.3, when a system is coupled to a single environment, one should expect that at long times it will reach equilibrium with that environment. In this section, we discuss the extent to which the Lindblad master equation—considered throughout this review—is consistent with this expectation. We note that in this section we focus on systems that are not driven, and are not coupled to multiple baths; as discussed in Sec. 1.3, driven systems are not expected to reach thermal equilibrium, whereas systems in contact with a single bath are expected to.

In line with our previous discussion, we consider two aspects of this question: (1) Does the Lindblad master equation recover the correct steady state (i.e. the mean-force Gibbs state as given in Eqs. (1) and (2))? (2) Does the Lindblad master equation produce two-time correlation functions consistent with the fluctuation dissipation theorem? We discuss each of these in turn.

**Steady state properties.** For simple few-body systems—for example a single cavity mode—it is straightforward to produce a master equation that recovers the expected equilibrium Gibbs state as its steady state. If one considers the system Hamiltonian $\hat{H} = \omega_c \hat{a}^\dagger \hat{a}$, then the corresponding equilibrium Gibbs state should be a diagonal matrix in the photon number basis $|n\rangle$ with elements $\langle n|\hat{\rho}|n\rangle = e^{-\beta n \omega_c}/\mathcal{Z}$. If one considers a master equation with pumping and dissipation, i.e. with jump operators $\hat{L}_- = \sqrt{\kappa_-}\hat{a}$, and $\hat{L}_+ = \sqrt{\kappa_+}\hat{a}^\dagger$, the resulting steady state is diagonal with $\langle n|\hat{\rho}|n\rangle = (\kappa_-/\kappa_+)^n/(1 - \kappa_-/\kappa_+)$. Thus, if one takes $\kappa_-/\kappa_+ = e^{-\beta \omega_c}$ one recovers the the equilibrium Gibbs state. This ratio is consistent with what one would find from a standard weak-coupling derivation of the master equation (see appendix A), which gives $\kappa_- = \kappa(n_B(\omega_c) + 1), \kappa_+ = \kappa n_B(\omega_c)$ where $n_B(\omega_c)$ is the Bose–Einstein distribution, $n_B(\omega_c) = [e^{\beta \omega_c} - 1]^{-1}$. One may note that this recovered the Gibbs state for the system Hamiltonian, and not a mean-force Gibbs state: this is consistent with taking the limit of weak system-environment coupling.

While the simple example above does reproduce the equilibrium Gibbs state, this does not generally hold for more complex many-body problems. That is, the forms of master equations we discuss in this review (with local dissipation terms) does not reproduce the Gibbs state of many body Hamiltonians[1]. One key point in this is that the equilibrium state of a many body Hamiltonian will typically involve correlations between elements at different positions or different lattice sites. The number of such correlations that must be correctly determined scales with the dimension of the many-body Hilbert space, i.e. is exponential in the number of lattice sites. The steady state of a model with purely local dissipation terms will be restricted in the form of correlations that can be realised, and generally not compatible with giving the correct value for an exponential number of correlations.

In appendix A we discuss the standard Bloch-Redfield approach for deriving the master equation. For a problem on a lattice, this approach will (in general) produce a number of terms exponential in the number of lattice sites. This occurs because the system Hamiltonian generically has a non-degenerate spectrum with this number of eigenstates (i.e. a number of eigenstates equivalent to the dimension of the many-body Hilbert space). The decomposition of the system-bath coupling into eigenoperators of the system Hamiltonian introduces this large number of terms. Following such a derivation is however clearly challenging, exactly due to this exponential number of terms. This challenge can be avoided by using an approximate system Hamiltonian in the derivation, neglecting coupling between sites. This will lead to an approximate local dissipation term, with only polynomial numbers of terms. However, this cannot generally recover the true thermal state of the many body Hamiltonian.

There has been extensive discussion of the question of "global" vs "local" master equations [212–217], i.e. comparing the effects of deriving the master equation with the true global system Hamiltonian and nonlocal dissipation terms, vs. using the local system Hamiltonian. There has also been significant work on the related question of how one can construct master equations to reproduce mean-force Gibbs states, even beyond the

---

[1]There are some cases where this still works. For example, some of the problems we consider with only loss terms have a trivial empty state as their steady state. The empty state can be the thermal equilibrium state at zero temperature, and the assumption that only loss terms exist is consistent with assuming a bath at zero temperature.

weak coupling limit, see for example Refs. [217–219].

**Multi-time correlations.**   Even when the steady state of the system is correctly described by a master equation, multi-time correlations generally are not. A key point is that the quantum regression theorem in Eq. (23) is generally not consistent with the fluctuation dissipation theorem in Eq. (4). That is, if one uses the quantum regression theorem to calculate the two time correlation functions in (3), their Fourier transforms will not satisfy Eq. (4). This point, as discussed in Refs. [220, 221] holds even for the simple case of a single bosonic mode coupled to a thermal bath discussed above, i.e.

$$\hat{H} = \omega_c \hat{a}^\dagger \hat{a}, \qquad \hat{L}_- = \sqrt{\kappa_-}\,\hat{a}, \qquad \hat{L}_+ = \sqrt{\kappa_+}\,\hat{a}^\dagger. \tag{24}$$

If one uses this model and calculates the two-time correlation functions $S_x(\tau)$ and $\chi_x(\tau)$ of the operator $\hat{x} = \hat{a} + \hat{a}^\dagger$ one finds these have the ratio:

$$\frac{S_x(\omega)}{\mathrm{Im}[\chi_x(\omega)]} = \frac{\kappa_- + \kappa_+}{\kappa_- - \kappa_+} = \coth(\beta\omega_c) \neq \coth(\beta\omega), \tag{25}$$

where we used the forms of $\kappa_\pm$ quoted above. This expression makes clear exactly why the fluctuation dissipation theorem cannot hold: the correlation is determined by the rates $\kappa_\pm$ which in turn depend on the thermal occupation of the bath at frequency $\omega_c$. This is not sufficient to determine the occupation of the bath at all frequencies $\omega \neq \omega_c$, and so the Markovian Lindblad equation cannot correctly calculate the response of the system at an arbitrary frequency. For many-body systems, the situation can be more subtle, as the many-body Hamiltonian may lead to the bath being sampled at many frequencies, and interactions within the system might also lead to effective thermalisation, see e.g. Refs. [222, 223] for further discussion.

As noted in the introduction, in this review we will focus on Markovian (i.e. weak coupling) models. Moreover, since we focus on driven-dissipative systems and other nonequilibrium states, we will not generically expect to reach thermal equilibrium. As such, the rest of our discussion will focus on models with local dissipation.

### 3.1.5   Examples of models

Here we provide some examples of Lindblad master equations for the kinds of models discussed in the rest of these Lecture Notes.

**Dicke model.**   A heavily studied example of a driven-dissipative many-body system is the Dicke model [224]. Such a model can be used to describe many experiments on cold atoms in single-mode optical cavities, as introduced in Sec. 2.3. The Dicke Hamiltonian [104, 105] takes the form:

$$\hat{H} = \omega_c \hat{a}^\dagger \hat{a} + \sum_i \left[ \epsilon \hat{\sigma}_i^+ \hat{\sigma}_i^- + g \left( \hat{a} + \hat{a}^\dagger \right) \hat{\sigma}_i^x \right]. \tag{26}$$

Here $\omega_c$ is the cavity frequency (or sometimes cavity-pump detuning) for a cavity mode that couples to all emitters, $\epsilon$ is the energy of each emitter, and $g$ is the emitter-cavity coupling strength. Note that this differs from the Jaynes–Cummings model in Eq. (7) in that this model describes many two-level systems (labelled by $i$), and the matter-light coupling is not in the rotating-wave approximation. Other variants of this model have been studied in some cases. For example, the matter-light coupling can be split into two terms, $g \left( \hat{a}\hat{\sigma}_i^+ + \mathrm{H.c.} \right) + g' \left( \hat{a}\hat{\sigma}_i^- + \mathrm{H.c.} \right)$ where $g, g'$ denote coupling strengths for "co-rotating" and

"counter-rotating" terms. The Dicke model corresponds to $g' = g$. The Tavis–Cummmings model (the many-emitter version of the Jaynes–Cummings model) occurs in the rotating-wave approximation and corresponds to setting $g' = 0$, which then leads to a Hamiltonian with a symmetry $\hat{S} = \exp\left[i\pi(\hat{a}^\dagger\hat{a} + \sum_i \hat{\sigma}_i^z/2)\right]$.

Typical dissipation processes considered in such a model include:

$$\hat{L}_{c-} = \sqrt{\kappa_-}\hat{a}, \qquad \hat{L}_{i-} = \sqrt{\gamma_-}\hat{\sigma}_i^-, \qquad \hat{L}_{i+} = \sqrt{\gamma_+}\hat{\sigma}_i^+, \qquad \hat{L}_{i\phi} = \sqrt{\gamma_\phi}\hat{\sigma}_i^z. \qquad (27)$$

The model with only the cavity loss process, $\hat{L}_{c-}$, is most-commonly considered to describe cold atoms in a single-mode cavity [106], however other variants have been studied elsewhere [225, 226].

**XYZ and Ising spin models.**    Another class of models of interest will be lattice models described by short-range interactions. A class of models that has been extensively studied in this context is spin models. A fairly generic example of such models is the XYZ model:

$$\hat{H} = \sum_{\langle i,j\rangle} \sum_\alpha J_\alpha \hat{\sigma}_i^\alpha \hat{\sigma}_j^\alpha. \qquad (28)$$

Here $\hat{\sigma}_i^\alpha$ are Pauli operators, with $\alpha \in \{x, y, z\}$ denoting which operator, and $i$ denoting which site. The couplings $J_\alpha$ are allowed to be different for each vector component of the spins. As in the previous section, the label $\langle i, j\rangle$ on the sum indicates summation over all neighbouring sites. An extensively studied [227] open-system version of this model is formed by combining this Hamiltonian with jump operators:

$$\hat{L}_{i-} = \sqrt{\gamma}\hat{\sigma}_i^- \qquad (29)$$

describing loss—taking the spin toward the spin down state—acting independently on each site. In this model one may reasonably consider the case where there is only a single class of jump operator per site, so the sum over jump operators is the sum over sites. If $J_x = J_y$, the model with this Hamiltonian and dissipation will evolve to a trivial state with all spins pointing down. A non-trivial state will exist as long as $J_x \neq J_y$, as then spin-spin interactions include process that do not conserve total $\hat{S}^z = \sum_i \hat{\sigma}_i^z/2$, thus balancing the losses.

Another particularly widely studied spin model is the Ising model, and the Ising model in a transverse field. Various different versions of this model have been studied in different contexts. The Ising model Hamiltonian takes the form:

$$\hat{H} = J\sum_{\langle i,j\rangle} \hat{\sigma}_i^z \hat{\sigma}_j^z + h\sum_i \hat{\sigma}_i^z. \qquad (30)$$

A driven-dissipative version of this model can be realised by combining this Hamiltonian with jump operators [228]:

$$\hat{L}_{i-} = \sqrt{\gamma_-}\hat{\sigma}_i^- \qquad \hat{L}_{i+} = \sqrt{\gamma_+}\hat{\sigma}_i^+ \qquad \hat{L}_{i\phi} = \sqrt{\gamma_\phi}\hat{\sigma}_i^z. \qquad (31)$$

Here, because the Hamiltonian conserves the total $\hat{S}^z$, one must balance decay with incoherent pumping to achieve a steady state. This model also includes pure dephasing terms, $\hat{L}_{i\phi}$.

The transverse-field Ising model Hamiltonian includes a field transverse to the spin-spin coupling. Using the model as written above, this leads to:

$$\hat{H} = J\sum_{\langle i,j\rangle} \hat{\sigma}_i^z \hat{\sigma}_j^z + h\sum_i \hat{\sigma}_i^x. \qquad (32)$$

As with the XYZ model, since this Hamiltonian does not conserve $\hat{S}^z$, a non-trivial steady state can exist even for a master equation with only loss terms $\hat{L}_{i-}$ [229–231]. In such a model, while the Hamiltonian has a symmetry under $\pi$-rotation around the $x$ axis, $\hat{S} = \prod_i e^{i\pi\hat{\sigma}_i^x/2}$, the dissipation breaks this symmetry, so this becomes a weak symmetry.

An alternate form of the dissipative transverse-field Ising model can however be considered, where

$$\hat{H} = J \sum_{\langle i,j \rangle} \hat{\sigma}_i^x \hat{\sigma}_j^x + h \sum_i \hat{\sigma}_i^z, \tag{33}$$

along with $\hat{L}_{i-}$ as defined above [232]. This model has a weak symmetry under $\pi$ rotations about the $z$ axis, $\hat{S} = \prod_i e^{i\pi\hat{\sigma}_i^z/2}$.

**Dissipative Hubbard models.** Another archetypal example of short-range lattice models is the Fermi–Hubbard model,

$$\hat{H} = -t_H \sum_{\langle i,j \rangle} \sum_{\sigma=\uparrow,\downarrow} \left[ \hat{c}_{i,\sigma}^\dagger \hat{c}_{j,\sigma} + \text{H.c.} \right] + U \sum_j \hat{n}_{j,\uparrow} \hat{n}_{j,\downarrow}. \tag{34}$$

written in terms of fermionic annihilation operators $\hat{c}_{i,\sigma}$ describing a fermion on site $i$ with spin $\sigma \in \uparrow, \downarrow$, and of $\hat{n}_{i,\sigma} = \hat{c}_{i,\sigma}^\dagger \hat{c}_{i,\sigma}$, the corresponding number operator. Here $t_H$ defines the amplitude for hopping between neighbouring sites while preserving the spin, and $U$ defines an interaction between fermions on the same site. Because of the Pauli exclusion principle, the only way a site can be multiply occupied is with two fermions of opposite spin. When considered as a model of cold atoms, a relevant source of dissipation is two-body loss defined by the process:

$$\hat{L}_{j-} = \sqrt{\kappa_-^{(2)}} \hat{c}_{j\uparrow} \hat{c}_{j,\downarrow}. \tag{35}$$

This describes a pair of fermions on the same site colliding and thus escaping the trap— as discussed in Sec. 2.2 such a process can also be engineered by using a laser to drive photoassociation. As with the interaction term, Pauli exclusion means this contact process can only exist between fermions with opposite spins.

The bosonic version of this model has also been extensively investigated, since it can describe both cold-atom experiments and photonic lattice, both in equilibrium and in its driven-dissipative variant. The Hamiltonian of the Bose–Hubbard model reads

$$\hat{H} = -t_H \sum_{\langle i,j \rangle} \left( \hat{a}_i^\dagger \hat{a}_j + \text{H.c.} \right) + \frac{U}{2} \sum_i \hat{n}_i(\hat{n}_i - 1), \tag{36}$$

where each lattice site hosts a single bosonic mode with annihilation operator $\hat{a}_i$. Different kind of dissipative processes can be considered, from incoherent pumping $\hat{L}_{i+} = \sqrt{\kappa_+}\hat{a}_i^\dagger$, heating due to spontaneous emission $\hat{L}_{i\phi} = \sqrt{\gamma_\phi}\hat{n}_i$ or m-body losses $\hat{L}_{i-(m)} = \sqrt{\kappa_-^{(m)}}(\hat{a}_i)^m$ (including $m = 1$ as standard loss).

**Weakly interacting dilute Bose gas.** A model closely related to the Bose–Hubbard model arises when considering the open system of microcavity polaritons [17]. This is the weakly interacting dilute Bose gas (WIDBG). This is defined by a Hamiltonian written in terms of bosonic annihilation operators $\hat{a}_{\mathbf{k}}$ for particles with momentum $\mathbf{k}$:

$$\hat{H} = \sum_{\mathbf{k}} \frac{k^2}{2m} \hat{a}_{\mathbf{k}}^\dagger \hat{a}_{\mathbf{k}} + \frac{U}{2V} \sum_{\mathbf{k},\mathbf{k}',\mathbf{q}} \hat{a}_{\mathbf{k}-\mathbf{q}}^\dagger \hat{a}_{\mathbf{k}'+\mathbf{q}}^\dagger \hat{a}_{\mathbf{k}'} \hat{a}_{\mathbf{k}}, \tag{37}$$

with $m$ being the boson mass, $U$ the (contact) interaction strength, and $V$ the quantisation volume. This has been used to describe both cold atoms [233] as well as microcavity polaritons [17]. This can be thought of as the long-wavelength (i.e. continuum) limit of the Bose–Hubbard model above.

To consider the open system one then adds processes describing gain or loss [159, 234]. Such a model is more transparent to write in terms of operators in real space, $\hat{a}(\mathbf{r}) = \sum_{\mathbf{k}} \hat{a}_{\mathbf{k}} e^{i\mathbf{k}\cdot\mathbf{r}}/L^{d/2}$, which leads to a Lindblad master equation involving an integral rather than a sum of dissipation terms:

$$\frac{d}{dt}\hat{\rho} = -i[\hat{H}, \hat{\rho}] + \int d^d\mathbf{r}\,\hat{\hat{\mathcal{D}}}[\sqrt{\kappa_-}\hat{a}(\mathbf{r}), \hat{\rho}] + \hat{\hat{\mathcal{D}}}[\sqrt{\kappa_+}\hat{a}^\dagger(\mathbf{r}), \hat{\rho}] + \hat{\hat{\mathcal{D}}}\left[\sqrt{\frac{\kappa_-^{(2)}}{2}}\hat{a}(\mathbf{r})^2, \hat{\rho}\right]. \quad (38)$$

Here we have included particle loss at rate $\kappa_-$, gain at rate $\kappa_+$, and two-particle loss at rate $\kappa_-^{(2)}$. The first two of these processes could equivalently be written in momentum space, leading to jump operators $\sqrt{\kappa_-}\hat{a}_{\mathbf{k}}, \sqrt{\kappa_+}\hat{a}_{\mathbf{k}}^\dagger$. The third process is more complex in momentum space as it does not lead to a diagonal Lindblad term. This model will be used to illustrate various theoretical approaches in the following sections.

**BCS model of fermion pairing.** In the same way that the WIDBG model can be thought of as the continuum analogue of the Bose–Hubbard lattice model, we next discuss the continuum analogue of the (Fermi-)Hubbard model. The archetypal example of this is the Bardeen-Cooper-Schrieffer (BCS) model that describes superconductivity [235]. Here one considers fermions described by the Hamiltonian

$$\hat{H} = \sum_{\mathbf{k},\sigma} \frac{k^2}{2m} \hat{c}_{\mathbf{k},\sigma}^\dagger \hat{c}_{\mathbf{k},\sigma} - \frac{u}{L}^d \sum_{\mathbf{k},\mathbf{k}',\mathbf{q}} \hat{c}_{\mathbf{k}-\mathbf{q},\uparrow}^\dagger \hat{c}_{\mathbf{k}'+\mathbf{q},\downarrow}^\dagger \hat{c}_{\mathbf{k}',\downarrow} \hat{c}_{\mathbf{k},\uparrow}. \quad (39)$$

In contrast to the repulsive interaction in the Hubbard model, the standard BCS model corresponds to considering an attractive contact interaction, denoted here with strength $u$. This attractive interaction can lead to fermion pairing, and thus superconductivity for charged fermions.

The open-system dynamics of such models have been studied considering two-body loss in various works [236, 237]. This is relevant to experiments on ultracold fermionic atoms: as with the examples described above, two-body losses can occur, where pairs of particles escape from the trap. In the continuum limit, this process can be described by the master equation:

$$\frac{d}{dt}\hat{\rho} = -i[\hat{H}, \hat{\rho}] + \int d^d\mathbf{r}\,\hat{\hat{\mathcal{D}}}\left[\sqrt{\kappa_-^{(2)}}\hat{c}_\uparrow(\mathbf{r})\hat{c}_\downarrow(\mathbf{r}), \hat{\rho}\right]. \quad (40)$$

In practice, for numerical convenience, most investigation of dissipative fermion pairing models has used lattices, and so are closer to the dissipative Hubbard model above [236].

**Bose-Fermi models.** A closely related model that has also been studied in a driven dissipative regime is the Bose-Fermi model:

$$\hat{H} = \sum_{\mathbf{k}} \left[\omega_k \hat{a}_{\mathbf{k}}^\dagger \hat{a}_{\mathbf{k}} + \sum_{i=c,v} \epsilon_k^i \hat{c}_{i,\mathbf{k}}^\dagger \hat{c}_{i,\mathbf{k}}\right] + \frac{g}{L^{d/2}} \sum_{\mathbf{k},\mathbf{q}} \left(\hat{c}_{c,\mathbf{k}+\mathbf{q}}^\dagger \hat{c}_{v,\mathbf{k}} \hat{a}_{\mathbf{q}} + \text{H.c.}\right)$$
$$+ \sum_{\mathbf{k},\mathbf{k}',\mathbf{q}} \sum_{i,i'} \frac{U_{\mathbf{q}}}{L^d} \hat{c}_{i,\mathbf{k}-\mathbf{q},\uparrow}^\dagger \hat{c}_{i',\mathbf{k}'+\mathbf{q},\downarrow}^\dagger \hat{c}_{i',\mathbf{k}',\downarrow} \hat{c}_{i,\mathbf{k},\uparrow}. \quad (41)$$

As written here, this describes cavity photons $\hat{a}_{\mathbf{q}}$ interconverting with electron-hole pairs. Here this is written in terms of conduction and valence bands, so creation of an electron-hole pair involves creation of a conduction electrons $\hat{c}_{c,\mathbf{k+q}}^\dagger$ and a valence band hole $\hat{c}_{v,\mathbf{k}}$. This model also includes Coulomb repulsion between electrons. This model corresponds to describing microcavity polaritons [238, 239] in terms of their constituent electrons. A similar model has also been studied in the context of pairing of ultracold atoms, when focusing on the two-channel Feshbach regime [240–242].

Driven-dissipative version of this model or closely related models [154, 243, 244] have been studied in the context of nonequilibrium polariton condensation. For such models one typically considers jump operators:

$$\hat{L}_{a-,\mathbf{k}} = \sqrt{\kappa}\hat{a}_{\mathbf{k}}, \qquad \hat{L}_{c+,i,\mathbf{k}} = \sqrt{\gamma_{+,i,k}}\hat{c}_{i,\mathbf{k}}^\dagger, \qquad \hat{L}_{c-,i,\mathbf{k}} = \sqrt{\gamma_{-,i,k}}\hat{c}_{i,\mathbf{k}}. \tag{42}$$

This describes loss of cavity photons, and exchange of electrons with some reservoir. (The rates of the fermionic process depend on the occupation of the reservoir, hence the assumption of $k$ dependence).

## 3.2   Schwinger–Keldysh path-integral formalism

An alternate—and complementary—way to study an open quantum system is through the Schwinger–Keldysh path integral formulation. A thorough introduction to the Schwinger–Keldysh path integral can be found in the book by Kamenev [52], and a comprehensive discussion of the application of this to the study of driven-dissipative systems in the reviews by Sieberer *et al.* [31, 32] and Thompson and Kamenev [245].

The Schwinger–Keldysh approach is based on a path integral representation of a generating function for correlations. One may start by considering the time evolution of the density matrix in the superoperator form $\frac{d}{dt}\hat{\rho} = \hat{\hat{\mathcal{L}}}[\hat{\rho}]$, and so writing the formal time evolution:

$$\hat{\rho}(t_1) = \mathcal{T}\exp\left(\int_{t_0}^{t_1} dt' \hat{\hat{\mathcal{L}}}\right)\hat{\rho}(t_0), \tag{43}$$

where $\mathcal{T}$ is a time ordering operator. One may then perform the standard path-integral derivation, by dividing the time evolution into a sequence of $N$ steps of length $\delta t$, inserting resolutions of identity at each time step, and then taking the limit $N \to \infty, \delta t \to 0$ with $N\delta t = t_1 - t_0$ to produce a continuous time path integral.

Because the density matrix is an operator and $\hat{\hat{\mathcal{L}}}$ is a superoperator, the construction of the path integral requires inserting two resolutions of identity at each time step (or equivalently, working in a doubled Hilbert space). For purely Hamiltonian evolution, these two copies are associated with the two copies of time evolution seen in writing:

$$\hat{\rho}(t_1) = \exp\left(-i\hat{H}(t_1 - t_0)\right)\hat{\rho}(t_0)\exp\left(i\hat{H}(t_1 - t_0)\right), \tag{44}$$

thus associating the two copies with forward time evolution (for operators to the left of $\hat{\rho}(t_0)$) and backward time evolution (for operators to the right). This leads to a picture, shown in Fig. 12, where time evolution proceeds on two branches of a single closed-time-contour path. This contour is known as the Schwinger–Keldysh closed-time contour.

In the continuum limit, this yields a path integral expression:

$$\mathcal{Z} \equiv \text{Tr}\left[\hat{\rho}(t_1)\right] = \int \mathcal{D}[\Psi_f, \Psi_b]\exp\left(iS[\Psi_f, \Psi_b]\right), \tag{45}$$

where $\int \mathcal{D}[\Psi_f, \Psi_b]$ denotes an integral over all time dependent paths of fields

$$\Psi_f(t) = \begin{pmatrix} \psi_f(t) \\ \psi_f^*(t) \end{pmatrix}, \qquad \Psi_b(t) = \begin{pmatrix} \psi_b(t) \\ \psi_b^*(t) \end{pmatrix},$$

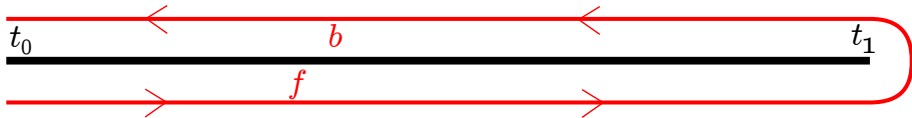

Figure 12: Schwinger–Keldysh closed-time-contour, illustrating labelling of two copies of time evolution labelled as forward (f) and backward (b) contours.

on the forward (f) and backward (b) branches respectively. The labels used to indicate these branches vary between different texts; we use the labels $f, b$ here to make clear the role of the branches as indicating forward and backward propagation.

The action, $S[\Psi_f, \Psi_b]$ can be determined from the Lindbladian by writing:

$$S[\Psi_f, \Psi_b] = \int_{t_0}^{t_1} dt \left( \psi_f^* i \frac{d}{dt} \psi_f - \psi_b^* i \frac{d}{dt} \psi_b - i\mathcal{L}(\Psi_f, \Psi_b) \right). \quad (46)$$

Here $\mathcal{L}(\Psi_f, \Psi_b)$ is defined by first writing the Lindbladian in normal-ordered form [52, 246] (i.e. creation operators to the left of annihilation operators), and then replacing operators acting to the left of the density matrix by fields on the forward contour, and operators to the right by fields on the backward contour.

Because density matrices are normalised to one, one finds that $\mathcal{Z} = 1$, so the value of $\mathcal{Z}$ does not give useful information about the system. However, one can calculate the expectation of observables from the path integral, by using generating functionals in terms of classical source fields $J$. Specifically one may write:

$$\left\langle \exp \left( i \int dt (J_f^T \hat{\Psi}_f - J_b^T \hat{\Psi}_b) \right) \right\rangle$$
$$= \frac{1}{\mathcal{Z}} \int \mathcal{D}[\Psi_f, \Psi_b] \exp \left( iS[\Psi_f, \Psi_b] + i \int dt (J_f^T \Psi_f - J_b^T \Psi_b) \right). \quad (47)$$

As above, we use $\Psi_{f,b}$ to represent the vectors $(\psi_{f,b}, \psi_{f,b}^*)$, and we use a similar notation for the source terms $J_{f,b}$. Because $\mathcal{Z} = 1$, the prefactor in the generating function can be ignored. Any correlation function of $\psi_{f,b}$ can then be found by taking derivatives with respect to the source fields $J_{f,b}$ and then ultimately setting $J_{f,b}$ to zero. While we introduced the quantities above on a finite time interval $[t_0, t_1]$, one can easily extend by taking $t_{0,1} \to \mp\infty$, and thus use this approach to find correlations at any time. In an open system, such results would then be expected to be independent of the initial state, $\hat{\rho}(t_0 \to -\infty)$.

To give a concrete example of the procedure defined above, we may consider the simple example of a bosonic mode with pumping and decay,

$$\frac{d}{dt}\hat{\rho} = -i[\omega \hat{a}^\dagger \hat{a}, \hat{\rho}] + \hat{\bar{\mathcal{D}}}[\sqrt{\kappa_-}\hat{a}, \hat{\rho}] + \hat{\bar{\mathcal{D}}}[\sqrt{\kappa_+}\hat{a}^\dagger, \hat{\rho}]. \quad (48)$$

Applying the rules above leads to a Schwinger–Keldyshpath integral with the action:

$$S = \int dt \left[ \psi_f^* \left( i\frac{d}{dt} - \omega + \frac{i}{2}(\kappa_+ + \kappa_-) \right) \psi_f - \psi_b^* \left( i\frac{d}{dt} - \omega - \frac{i}{2}(\kappa_+ + \kappa_-) \right) \psi_b \right.$$
$$\left. - i\kappa_+ \psi_f^* \psi_b - i\kappa_- \psi_b^* \psi_f \right]. \quad (49)$$

This example can be straightforwardly generalised to dissipative many-body models such as the ones discussed in Sec. 3.1.5. The Keldysh action of a dissipative Bose–Hubbard model for example reads

$$S = \int dt \sum_{\zeta=f,b} s_\zeta \left[ \sum_i \psi_{i\zeta}^* \, i\frac{d}{dt}\psi_{i\zeta} - H_\zeta \right] - i \sum_i \left[ L_{ib}L_{if}^* - \frac{1}{2}\left( L_{if}^*L_{if} + L_{ib}^*L_{ib} \right) \right], \quad (50)$$

where $s_{f,b} = \pm 1$, $H$ is the Bose–Hubbard Hamiltonian in Eq. (36) written for the boson fields $\psi_{i\zeta}, \psi_{i\zeta}^*$ and $L_{i\zeta}, L_{i\zeta}^*$ are the jump operators written in terms of the same bosonic fields. In the specific case of dephasing for example we have $L_{i\zeta} = \sqrt{\gamma_\phi}\psi_{i\zeta}^*\psi_{i\zeta}$, so that we get

$$S = \int dt \sum_{\zeta=f,b} s_\zeta \left[ \sum_i \psi_{i\zeta}^* i\frac{d}{dt}\psi_{i\zeta} + t_H \sum_{\langle i,j\rangle} \left( \psi_{i\zeta}^*\psi_{j\zeta} + hc \right) - \frac{U}{2}\sum_i \psi_{i\zeta}^*\psi_{i\zeta} \left( \psi_{i\zeta}^*\psi_{i\zeta} - 1 \right) \right] +$$

$$- i\gamma_\phi \sum_i \left[ \psi_{ib}^*\psi_{ib}\psi_{if}^*\psi_{if} - \frac{1}{2}\left( \psi_{if}^*\psi_{if}\psi_{if}^*\psi_{if} + \psi_{ib}^*\psi_{ib}\psi_{ib}^*\psi_{ib} \right) \right]. \quad (51)$$

Both examples above are bosonic problems and so the fields appearing in the path integral are complex variables. For fermionic problems, one instead finds a path integral over Grassmann variables (see [52, 246] for a discussion of this case). Problems that involves other operators, such as spins, generally require rewriting in terms of bosonic or fermionic operators before constructing a Schwinger–Keldysh action.

It is often convenient to rewrite the Schwinger–Keldysh path integral in terms of symmetric and anti-symmetric combinations of the forward and backward fields; these are known as the classical and quantum fields. As we will see, this rotation simplifies the structure of the quadratic part of the action. Going back to the example of the single bosonic mode with pumping and decay these fields read

$$\psi_{cl,q} = \frac{1}{\sqrt{2}}\left( \psi_f \pm \psi_b \right). \quad (52)$$

In terms of these rotated fields the non-interacting action in Eq. (49) becomes:

$$S = \int dt \begin{pmatrix} \psi_{cl}^* & \psi_q^* \end{pmatrix} \begin{pmatrix} 0 & [D_A]^{-1} \\ [D_R]^{-1} & [D^{-1}]_K \end{pmatrix} \begin{pmatrix} \psi_{cl} \\ \psi_q \end{pmatrix}, \quad (53)$$

where the retarded (R), advance (A), and Keldysh (K) components of the inverse Green's function are given by:

$$[D_{R,A}]^{-1} = i\frac{d}{dt} - \omega \pm \frac{i}{2}(\kappa_- - \kappa_+), \qquad [D^{-1}]_K = i(\kappa_- + \kappa_+). \quad (54)$$

The combination $\kappa_- + \kappa_+$ appearing in the inverse Keldysh Green's function can be thought as a "noise" strength. The fact that it is the sum of these two rates appearing here describes the fact that both gain and loss lead to noise. We will see further below that this same combination appears as a noise in many different approaches.

As a second example, we may consider the driven-dissipative WIDBG model, often used to describe polariton condensates [159, 234] written in Eq. (38). Using the same rules as noted above, this yields a Keldysh action of the form:

$$S = \int_{t_0}^{t_1} dt \int d^d\mathbf{r} \left\{ \begin{pmatrix} \psi_{cl}^* & \psi_q^* \end{pmatrix} \begin{pmatrix} 0 & [D_A^{(0)}]^{-1} \\ [D_R^{(0)}]^{-1} & [D^{(0)-1}]_K \end{pmatrix} \begin{pmatrix} \psi_{cl} \\ \psi_q \end{pmatrix} \right.$$
$$\left. - \left[ \left( \frac{U}{2} + i\frac{\kappa_-^{(2)}}{4} \right) \left( \psi_{cl}^{*2} + \psi_q^{*2} \right) \psi_{cl}\psi_q + \text{c.c.} \right] + i\kappa_-^{(2)} |\psi_{cl}|^2 |\psi_q|^2 \right\}, \quad (55)$$

where the bare Green's functions (i.e. neglecting corrections due to interactions), written here in real space, take the form:

$$[D_{R,A}^{(0)}]^{-1} = i\frac{d}{dt} + \frac{\nabla^2}{2m} \pm i(\kappa_- - \kappa_+), \qquad [D^{(0)-1}]_K = i(\kappa_- + \kappa_+). \quad (56)$$

This has a similar structure to the first model discussed above, but replacing the single-mode energy $\omega$ with the quadratic dispersion of the WIDBG model.

## 3.3 Quantum trajectories

The third framework for describing open quantum systems that we will discuss is that of quantum trajectories. This framework, also known as *unravelling*, describes the evolution of a pure quantum state due to a combination of Hamiltonian evolution and measurements which monitor the system [247–251]. Physically, the measurements can either be performed by an explicit experimental apparatus or by the environment (in this latter case we speak of an *unread* measurement). In quantum mechanics, a measurement is a stochastic process: the state of the quantum system changes depending on the outcome of the measure and the probability of each outcome depends on the state just before the measure is performed. When measurements are repeated multiple times, we speak of distinct *quantum trajectories*, each labelled by its own measurement record. As discussed below, the density matrix that we discussed so far is recovered by averaging over these trajectories. It is interesting to observe that different measurements can produce the same Lindblad master equation and thus the same density matrix $\rho(t)$, even if microscopically, at the level of each single quantum trajectory, the physical properties are extremely different. Indeed, there has been significant recent interest in the observation that different unravellings lead to different amounts of entanglement in the unravelled wavefunction [252–255]—this observation is closely related to the discussion of entanglement transitions which we will discuss in Sec. 8.

The unravelling method can also be seen as a theoretical technique for more efficiently studying a given Lindblad master equations: especially in many-body systems, the study of the density matrix can be numerically costly, as the size of the problem scales as the square of the total Hilbert space dimension. The cost of simulating pure states can present some advantages as it scales only as the Hilbert space dimension, even if the price to pay is that the evolution becomes stochastic and needs thus some form of repetition and averaging (which can often be parallelised) to decrease the statistical uncertainty. The previous statement implies that in these theoretical studies different forms of unravelling can be chosen, all leading to the study of the same density matrix.

We discuss here two frequently encountered ways of unravelling a Lindblad master equation. In both cases, we will produce a form which has both a stochastic noise term, and a non-unitary deterministic evolution (for example, the non-Hermitian Hamiltonian appearing in the quantum jump formalism). However, we emphasise that there can also be cases—where jump operators are Hermitian—where the unravelling leads to a unitary (but noisy) evolution [256–259], i.e. where no non-Hermitian Hamiltonian is required.

**Quantum jump trajectories.** "Quantum Jump" trajectories describe the case where the noise acts occasionally but abruptly on the quantum state to measure an operator $\hat{L}_\mu$

and is described by the following stochastic Schrödinger equation [248–251, 260, 261]

$$d\ket{\psi(\mathcal{N}_t, t)} = \left[-i\hat{H} - \frac{1}{2}\sum_\mu\left(\hat{L}_\mu^\dagger\hat{L}_\mu - \left\langle\hat{L}_\mu^\dagger\hat{L}_\mu\right\rangle_t\right)\right]dt\ket{\psi(\mathcal{N}_t, t)}$$
$$+ \sum_\mu\left(\frac{\hat{L}_\mu}{\sqrt{\left\langle\hat{L}_\mu^\dagger\hat{L}_\mu\right\rangle_t}} - 1\right)d\mathcal{N}_t^\mu\ket{\psi(\mathcal{N}_t, t)}, \tag{57}$$

where $\langle\circ\rangle_t \equiv \bra{\psi(\mathcal{N}_t, t)}\circ\ket{\psi(\mathcal{N}_t, t)}$, and $\mathcal{N}_t = \{\mathcal{N}_t^\mu\}$ are a set of statistically independent Poisson processes $d\mathcal{N}_t^\mu \in \{0, 1\}$ with average value $\overline{d\mathcal{N}_t^\mu} = dt\left\langle L_\mu^\dagger L_\mu\right\rangle_t$. This is written assuming $dt$ is sufficiently small that $\overline{d\mathcal{N}_t^\mu} \ll 1$. As such, for most timesteps $d\mathcal{N}_t^\mu = 0$ and so the evolution follows the form in the first line. However, occasionally $d\mathcal{N}_t^\mu = 1$ for some $\mu$, in which case the second line is non-zero, and will dominate. In those cases the state will change discontinuously, such that the state at the next time $t + dt$ becomes $\hat{L}_\mu\ket{\psi(\mathcal{N}_t, t)} / \sqrt{\left\langle\hat{L}_\mu^\dagger\hat{L}_\mu\right\rangle_t}$. The density matrix can be recovered by writing:

$$\hat{\rho}(t) = \sum_{\{d\mathcal{N}_{t'\leq t}\}} P(\mathcal{N}_t)\ket{\psi(\mathcal{N}_t, t)}\bra{\psi(\mathcal{N}_t, t)}, \tag{58}$$

summing over all measurement outcomes at previous times, with $P(\mathcal{N}_t)$ describing the probability of a given measurement record.

This relatively complicated equation can be made more intuitively understandable by discussing explicitly how it is solved in practice, for example using a first-order Monte Carlo wave-function scheme [261]. Let us consider the initial pure state of the dynamics $\ket{\psi(t)}$. First, we need to compute the wavefunction $\ket{\psi(t + dt)} = e^{-i\left(\hat{H} - \frac{i}{2}\sum_\mu\hat{L}_\mu^\dagger\hat{L}_\mu\right)dt}\ket{\psi(t)}$ obtained by applying the deterministic part of the evolution. The next step consists in the stochastic evolution. With probability $p_\mu = dt\bra{\psi(t + dt)}\hat{L}_\mu^\dagger\hat{L}_\mu\ket{\psi(t + dt)}$ the quantum state is abruptly modified to $\hat{L}_\mu\ket{\psi(t + dt)}$ and then normalised. Otherwise, with probability $1 - \sum_\mu p_\mu$, the state is unchanged and normalised. At this point, the process is then restarted applying the same procedure as before. Alternative sampling techniques are also possible, for example computing the average time till the next quantum jump and thus sampling the waiting-time distribution [261, 262]. The stochastic nature of the process creates a number of quantum trajectories, whose incoherent average gives the density matrix.

This form of stochastic unravelling often occurs for jump operators $\hat{L}_\mu = \sqrt{\kappa}\hat{a}$ corresponding to photon counting measurements: the measurement record consists of a set of times at which a photon was detected (i.e. times at which $\mathcal{N}_t^\mu = 1$).

**Quantum state diffusion.** A different unravelling occurs when measurements act continuously but weakly. This can occur for example when light from the sample is interfered with a strong reference beam in a balanced homodyne detection scheme. In this case photon counting will occur every time step, but most of those photons come from the reference, not the system, and thus do not represent a measure the system state. One can consider this case directly by using the same equation as above but replacing $\hat{L}_\mu \rightarrow \hat{L}_\mu + \beta$ and $\hat{H} \rightarrow \hat{H} + (i/2)\sum_\mu(\beta\hat{L}_\mu^\dagger - \beta^*\hat{L}_\mu)$, with $\beta$ denoting the strength of the reference beam. This substitution leaves the master equation unchanged, but clearly changes the probability of measurement at each time step.

In the limit of an infinitely strong reference beam, the system is described by the quantum state diffusion (QSD) equation [247]:

$$d\left|\psi(\xi_t, t)\right\rangle = \left[-i\hat{H} - \frac{1}{2}\sum_{\mu}\left(\hat{L}_{\mu}^{\dagger}\hat{L}_{\mu} + \left\langle\hat{L}_{\mu}^{\dagger}\right\rangle_{\xi_t, t}\left\langle\hat{L}_{\mu}\right\rangle_{\xi_t, t} - 2\left\langle\hat{L}_{\mu}^{\dagger}\right\rangle_{\xi_t, t}\hat{L}_{\mu}\right)\right]dt\left|\psi(\xi_t, t)\right\rangle +$$

$$+ \sum_{\mu}\left(\hat{L}_{\mu} - \left\langle\hat{L}_{\mu}\right\rangle_{\xi_t, t}\right)d\xi_t^{\mu}\left|\psi(\xi_t, t)\right\rangle, \tag{59}$$

where $\langle\circ\rangle_{\xi_t, t} = \langle\psi(\xi_t, t)|\circ|\psi(\xi_t, t)\rangle$ is the average over the quantum state. The first term in Eq. (59) represents the unitary evolution, while the remaining encodes the noise effects. The $d\xi_t^{\mu}$ are Îto increments of a Wiener process $\xi_t = (\xi_t^1, \xi_t^2, \dots)$, responsible for the stochastic nature of the trajectories, with zero mean $\overline{d\xi_t^{\mu}} = 0$ and fulfilling the exact property $d\xi_t^{\mu}d\xi_t^{\nu} = dt\delta^{\mu\nu}$. The second term in Eq. (59) describes a deterministic back-action from the measuring environment. We note the presence of a feedback mechanism, as the noise couples to the fluctuations of the measured operator $\delta\hat{L}_{\mu} = \hat{L}_{\mu} - \langle\hat{L}_{\mu}\rangle_{\xi_t, t}$. Mathematically, the role of this feedback is to preserve the norm of the state (and any of its cumulants) for any realisation of the noise.

It is important to stress the difference between the conditional and the mean state [263]. The conditional state is the quantum trajectory itself $\hat{\rho}(\xi_t, t) = |\psi(\xi_t, t)\rangle\langle\psi(\xi_t, t)|$, fixed by a realisation of the Wiener process. Instead, the mean state is given by

$$\hat{\rho}(t) = \int \mathcal{D}\xi_t P(\xi_t)\left|\psi(\xi_t, t)\right\rangle\langle\psi(\xi_t, t)|, \tag{60}$$

with $P(\xi_t)$ the probability distribution of the noise at all times up to $t$. Despite the fact the conditional state is always pure, the mean state is generically mixed, and evolves according to the Lindblad master equation. This difference is crucial, and highlights how the two states $\hat{\rho}(t)$ and $\hat{\rho}(\xi_t, t)$ provide rather different information on the statistical properties of the system. Although quantum trajectories have been traditionally introduced as a computational method to solve the Lindblad master equation by averaging over different noise realisations, more recently interest has shifted towards the understanding of the many-body physics encoded in the conditional state, as we will discuss more in detail in Sec. 8.

**Relating stochastic evolution to the Lindblad equation.** Monitored dynamics, as for example that governed by the stochastic Schrödinger equation in Eq. (57) or the quantum state diffusion in Eq. (59), are inherently non-linear: the evolution depends on the average value of the jump operators $\langle L_{\mu}\rangle$, $\langle L_{\mu}^{\dagger}\rangle$. Indeed, by following the dynamics along a given trajectory, information is acquired that conditions the subsequent evolution. By averaging over the trajectories, however, one should get $\hat{\rho}(t)$ which evolves according to the (linear) Lindblad equation. Let us now see how the Lindblad equation follows from the stochastic evolution; for simplicity, we will consider this derivation in the case of the quantum state diffusion (an analogous procedure applies similarly for the case of quantum jumps).

The starting point is the evolution of the density matrix along a given trajectory, $\hat{\rho}(\xi_t, t) = |\psi(\xi_t, t)\rangle\langle\psi(\xi_t, t)|$. The state at time $t + dt$ is given by

$$\hat{\rho}(\xi_t, t + dt) = \left[|\psi(\xi_t, t)\rangle + d|\psi(\xi_t, t)\rangle\right]\left[\langle\psi(\xi_t, t)| + d\langle\psi(\xi_t, t)|\right]. \tag{61}$$

By substituting the expression for $d|\psi(\xi_t, t)\rangle$ given in Eq. (59) and averaging over the stochastic variable $\xi_t$ one obtains the density matrix

$$\hat{\rho}(t) = \overline{|\psi(\xi_t, t)\rangle\langle\psi(\xi_t, t)|} \tag{62}$$

where the average over noise is written explicitly in Eq. (60). Some care should be taken in handling the expression in Eq. (61) because the term $d\left|\psi(\xi_t,t)\right\rangle d\left\langle\psi(\xi_t,t)\right|$ cannot be simply dropped, as it contains contributions of order $dt$. This can be seen by a close inspection of the right hand side of Eq. (59). When considering $d\left|\psi(\xi_t,t)\right\rangle d\left\langle\psi(\xi_t,t)\right|$, the parts of $d\left|\psi(\xi_t,t)\right\rangle$ proportional to $dt$ can be dropped, but the last term, proportional to $d\xi_t$ should be retained since $(d\xi_t)^2 = dt$. Direct substitution of Eq. (59) in Eq. (61) and averaging over the noise leads to the Lindblad equation.

## 3.4 Additional theoretical frameworks

The three frameworks described above are those on which we will focus in these Lecture Notes. However, there are a number of other frameworks that are sometimes used in describing many-body open quantum systems; here we provide a brief summary of some of them.

### 3.4.1 Phase space approaches

Phase space methods refer to representations of the density matrix in terms of distributions over continuous variables [264]. These have primarily been used to represent bosonic systems, but extensions to spin problems also exist [265, 266]. These techniques, originally developed for the study of single-atom and/or single-photon properties in the context of quantum optics, are currently being generalized and applied to the many-body context.

The earliest-studied phase space approach is the Wigner function, $W(z)$, defined in the context of a single bosonic mode $\hat{a}^{(\dagger)}$ by:

$$W(z) = \frac{1}{\pi^2} \int d^2w\, e^{-wz^* + w^* z} \mathrm{Tr}\left[\hat{\rho}\, e^{w\hat{a}^\dagger - w^*\hat{a}}\right], \qquad z \in \mathbb{C}; \tag{63}$$

where $d^2w$ indicates integration over the complex plane of $w$. Using such a representation, the Lindblad master equation translates into a partial differential equation for the Wigner function. The Wigner function can be interpreted as a quasi-probability density, since for general quantum states, $W$ can be negative. A frequently used approximation is the truncated Wigner approximation [267], which consists of neglecting third and higher-order derivatives in the partial differential equation. This then yields an equation that can be mapped to the Fokker-Planck equation for probability densities, and simulated by trajectories that sample $z(t)$.

Other phase space distributions correspond to different mappings from the density matrix to continuous functions. Among these, it is worth mentioning the positive-P distribution [268] (for a discussion on regimes of applicability see Ref. [269]). This works by creating a distribution over two continuous variables $P(z_1, z_2)$, which can be guaranteed to remain positive. Such methods were originally applied to closed system dynamics, for which they tended to undergo numerical instabilities. For open system dynamics, dissipation damps these instabilities, and so the positive-P distribution has recently been applied to the study of driven-dissipative Bose–Hubbard lattices [270].

### 3.4.2 Heisenberg–Langevin and adjoint equations

The Lindblad master equation, as well as quantum trajectory formalisms, are based on the time evolution of the state of the system. This is the open-system analogue of the Schrödinger picture for closed quantum systems, where states evolve and operators corresponding to observables are fixed in time. The opposite picture is the Heisenberg picture, where states are constant and operators evolve. This can be extended to open quantum systems in various ways.

The most straightforward way is the use of the adjoint equation for time-evolving an operator [264, 271]. The concept is based on the notion of the adjoint operator $\bar{\mathcal{L}}$, that satisfies the following identity for the expectation value of a generic observable $O$:

$$\langle O(t) \rangle = \text{Tr}\left[\hat{O}\,\hat{\rho}(t)\right] = \text{Tr}\left[\hat{O}\,\exp\left(\hat{\mathcal{L}}t\right)[\rho(0)]\right] \equiv \text{Tr}\left[\exp(\bar{\mathcal{L}}t)[\hat{O}]\,\hat{\rho}(0)\right]. \tag{64}$$

Note that in the final expression, the superoperator $\exp(\bar{\mathcal{L}}t)$ acts on $\hat{O}$ and not on $\hat{\rho}(0)$. For a generic Lindblad master equation in the form of Eq. (11), the action of the adjoint on an operator $\hat{O}$ takes the form:

$$\bar{\mathcal{L}}[\hat{O}] = i[\hat{H}, \hat{O}] + \sum_\mu \left(\hat{L}_\mu^\dagger \hat{O} \hat{L}_\mu - \frac{1}{2}\{\hat{L}_\mu^\dagger \hat{L}_\mu, \rho\}\right). \tag{65}$$

This is closely related to the quantum regression theorem [180, 187, 211], as introduced in Sec. 3.1.3 above. In terms of the adjoint representation, Eq. (23) can be rewritten as:

$$\left\langle \hat{O}_\mu(t+\tau)\,\hat{O}_\nu(t)\right\rangle = \text{Tr}\left[\exp(\bar{\mathcal{L}}\tau)[\hat{O}_\mu]\,\hat{O}_\nu\hat{\rho}(t)\right]. \tag{66}$$

It is tempting, but wrong, to regard this as giving the time evolution of the operator $\hat{O}(t)$: solving $\frac{d}{dt}\hat{a} = \bar{\mathcal{L}}[\hat{a}]$ for the damped harmonic oscillator with Hamiltonian $\hat{H} = \hbar\omega(\hat{a}^\dagger\hat{a} + 1/2)$ and quantum jump $\hat{L} = \sqrt{\kappa}\hat{a}$ gives $\hat{a}(t) = e^{-i\omega t - \kappa t/2}\hat{a}(0)$. The exponential decay $e^{-\kappa t/2}$ turns $\hat{a}(t)$ into an operator that does not satisfy the canonical commutation relations $[\hat{a}(t), \hat{a}(t)^\dagger] \neq 1$. However, as we just saw, it is perfectly legitimate to use the adjoint equation to determine the time evolution of *the expectation value* of an operator, $\frac{d}{dt}\langle\hat{O}\rangle = \langle\bar{\mathcal{L}}[\hat{O}]\rangle$. Using this, one can calculate both $\langle\hat{a}^\dagger(t)\hat{a}(t)\rangle$ and $\langle\hat{a}(t)\hat{a}^\dagger(t)\rangle$ via the adjoint equation and thus get the correct commutator.

In general, in order to solve this issue, one can consider the Heisenberg–Langevin equation (see e.g. Refs. [180, 264]), where the presence of dissipation introduces in the equation noise terms in addition to those describing directly the dissipation effects. As an example, for a simple model with a pumped and decaying bosonic mode $\hat{a}$, with dissipation terms

$$\hat{L}_- = \sqrt{\kappa_-}\hat{a}, \quad \hat{L}_+ = \sqrt{\kappa_+}\hat{a}^\dagger, \tag{67}$$

the Heisenberg–Langevin equations for the bosonic operators take the form:

$$\frac{d\hat{a}}{dt} = i[\hat{H}, \hat{a}(t)] - \frac{(\kappa_- - \kappa_+)}{2}\hat{a}(t) + \hat{F}_a(t). \tag{68}$$

$\hat{F}_a(t)$ is a quantum noise, that is, a stochastic operator which averages to zero, $\overline{\hat{F}_a(t)} = 0$ and has Gaussian correlations defined by:

$$\overline{[\hat{F}_a(t), \hat{F}_a^\dagger(t')]} = (\kappa_- - \kappa_+)\delta(t-t') \qquad \overline{\{\hat{F}_a^\dagger, (t')\hat{F}_a(t)\}} = (\kappa_- + \kappa_+)\delta(t-t'). \tag{69}$$

The role of the non-commuting noise here is crucial in preserving equal-time commutation relations. Without the noise, the bosonic operator would acquire an exponential decay $\exp(-(\kappa_- - \kappa_+)t/2)$, which is inconsistent with $[\hat{a}(t), \hat{a}(t)^\dagger] = 1$ remaining true at all times. The commutation relation of the noise term however provides a source that compensates this loss, see [180, 187] for details.

The form of Eq. (68) can be derived by considering an explicit model for the environment, writing coupled Heisenberg equations for the system and the environment, and then formally solving the equations for the environment and substituting into the system Heisenberg equation. In such an approach, the noise term arises from the initial value of environment operators. One may note a close relation between the structure of the Heisenberg–Langevin equation and the Schwinger–Keldysh Green's functions for this problem as given in Eq. (54).

### 3.4.3 Third quantisation

Third quantisation [201, 272] refers to constructing superoperators that describe the action of applying raising and lowering operators to a density matrix. This provides a specific formalisation of the vectorised form of the Lindblad master equation. Specifically it provides a set of superoperators (which become operators in the vectorised form) from which the Lindblad superoperator can be constructed. As discussed further below, for certain problems it also provides a route to constructing the eigenstates of the Lindblad superoperator.

Third quantisation was first discussed for fermionic problems, by introducing Majorana fermionic operators [201]. It was later extended to bosonic systems [272], for which a simpler representation exists:

$$\hat{\mathbf{a}}_f \, |\rho\rangle\rangle = |\hat{a}\hat{\rho}\rangle\rangle \, , \qquad\qquad \hat{\mathbf{a}}_b \, |\rho\rangle\rangle = |\hat{\rho}\hat{a}\rangle\rangle \, ,$$

$$\hat{\mathbf{a}}_f^\dagger \, |\rho\rangle\rangle = \left|\hat{a}^\dagger\hat{\rho}\right\rangle\rangle \, , \qquad\qquad \hat{\mathbf{a}}_b^\dagger \, |\rho\rangle\rangle = \left|\hat{\rho}\hat{a}^\dagger\right\rangle\rangle \, , \qquad (70)$$

where $f, b$ refer to forward (i.e. left-multiplication) and backward (right-multiplication) superoperators. Using such operators (and paying careful attention to which operators are closest to the density matrix in a given expression), one can rewrite the Lindbladian in terms of these third-quantised operators. This is particularly useful in the context of quadratic problems, as discussed further in Sec. 4.1.

An alternate form of third quantisation was introduced in Ref. [273], using quantum and classical superoperators, by direct analogy to the Keldysh action discussed above. For a bosonic problem these take the form:

$$\hat{\mathbf{a}}_{cl} \, |\rho\rangle\rangle = \frac{1}{\sqrt{2}} \, |\{\hat{a}, \hat{\rho}\}\rangle\rangle \, , \qquad\qquad \hat{\mathbf{a}}_q \, |\rho\rangle\rangle = \frac{1}{\sqrt{2}} \, |[\hat{a}, \hat{\rho}]\rangle\rangle \, ,$$

$$\hat{\mathbf{a}}_{cl}^\dagger \, |\rho\rangle\rangle = \frac{1}{\sqrt{2}} \, \left|\{\hat{a}^\dagger, \hat{\rho}\}\right\rangle\rangle \, , \qquad\qquad \hat{\mathbf{a}}_q^\dagger \, |\rho\rangle\rangle = \frac{1}{\sqrt{2}} \, \left|[\hat{a}^\dagger, \hat{\rho}]\right\rangle\rangle \, . \qquad (71)$$

Using such super-operators, one can rewrite the Lindbladian into a form that is directly analogous to the Keldysh action (with fields replaced by third-quantised operators). One can also use these operators to make connections to phase space methods, by using the eigenvectors of these third-quantised operators to form an orthogonal basis for the space of density matrices [273].

### 3.4.4 Effective non-Hermitian Hamiltonian

One early approach to the study of open quantum systems involved the study of non-Hermitian Hamiltonians, attempting to incorporate decay by adding imaginary terms to the Hamiltonian (see the extensive review in Ref. [27] for examples). This approach differs from that introduced above, based on the Lindblad master equation. Here we comment on how the two approaches can be related in some limiting cases. It is important to stress that studies based solely on non-Hermitian Hamiltonians cannot reproduce the full physics described by Lindblad master equations, and can generically be employed only to gain some qualitative information on the dynamics of the setup. However, the simulation of a non-Hermitian Hamiltonian may be computationally cheaper than that of the master equation.

To connect our Lindblad master equation, given in Eq. (11), to a non-Hermitian Hamiltonian, we start by performing the following splitting of the Lindbladian superoperator:

$$\hat{\hat{\mathcal{L}}}[\hat{\rho}] = \hat{\hat{\mathcal{L}}}_{\text{nH}}[\hat{\rho}] + \hat{\hat{\mathcal{L}}}_{\text{qj}}[\hat{\rho}], \qquad (72)$$

with

$$\hat{\bar{\mathcal{L}}}_{\mathrm{nH}}[\hat{\rho}] = -i\left(\hat{H} - \frac{i}{2}\sum_\mu \hat{L}_\mu^\dagger \hat{L}_\mu\right)\hat{\rho} + i\hat{\rho}\left(\hat{H} + \frac{i}{2}\sum_\mu \hat{L}_\mu^\dagger \hat{L}_\mu\right) = -i\left(\hat{H}_{\mathrm{nH}}\hat{\rho} - \hat{\rho}\hat{H}_{\mathrm{nH}}^\dagger\right),$$
(73a)

$$\hat{\bar{\mathcal{L}}}_{\mathrm{qj}}[\hat{\rho}] = \sum_\mu \hat{L}_\mu \hat{\rho} \hat{L}_\mu^\dagger.$$
(73b)

This splitting shows in a rather direct way that the Lindblad master equation can be interpreted as a dynamics with the effective non-Hermitian Hamiltonian $\hat{H}_{\mathrm{nH}}$, described by $\hat{\bar{\mathcal{L}}}_{\mathrm{nH}}[\hat{\rho}]$, and by a quantum jump term $\hat{\bar{\mathcal{L}}}_{\mathrm{qj}}[\hat{\rho}]$. If one considers an initial pure state $\hat{\rho} = |\psi\rangle\langle\psi|$ evolving under $\hat{\bar{\mathcal{L}}}_{\mathrm{nH}}$, one sees that this evolution is equivalent to evolving $|\psi\rangle$ under the Hamiltonian $\hat{H}_{\mathrm{nH}}$: this dynamics preserves its purity and thus we can deduce that $\hat{\bar{\mathcal{L}}}_{qj}[\hat{\rho}]$ is generically responsible for turning pure states into mixed states. At the same time, we may also note that the non-Hermitian dynamics will generically reduce the norm of a state; that is, time evolution of $|\psi\rangle$ under $\hat{H}_{\mathrm{nH}}$ does not preserve $\langle\psi|\psi\rangle$, and thus does not preserve the trace of $|\psi\rangle\langle\psi|$. The jump term $\hat{\bar{\mathcal{L}}}_{\mathrm{qj}}[\hat{\rho}]$ compensates for this, and enforces the trace preservation that is a generic property of the Lindblad master equation.

This discussion highlights that neglecting $\hat{\bar{\mathcal{L}}}_{\mathrm{qj}}[\hat{\rho}]$ compromises the possibility of describing the full time-evolution of the system. However, specific fine-tuned situations exist where some properties of the Lindblad master equation can be found using the non-Hermitian dynamics of $\hat{H}_{\mathrm{nH}}$ alone. Let us consider, for instance, an atom in an excited state $|e\rangle$ that can undergo a spontaneous emission process to the ground state $|g\rangle$, described by the jump operator $\hat{L} = \sqrt{\gamma}\,|g\rangle\langle e|$. If we are interested in the probability of detecting such atom in the excited state, described by the observable $|e\rangle\langle e|$, we can invoke the adjoint equation for the dynamics introduced in Eq. (65) and perform the analogous splitting $\bar{\mathcal{L}}[|e\rangle\langle e|] = \bar{\mathcal{L}}_{\mathrm{nH}}[|e\rangle\langle e|] + \bar{\mathcal{L}}_{\mathrm{qj}}[|e\rangle\langle e|]$. With a few steps of algebra it is possible to show that:

$$\bar{\mathcal{L}}_{\mathrm{qj}}[|e\rangle\langle e|] = 0;$$
(74)

by expanding formally the exponential $e^{\bar{\mathcal{L}}t}$ in its series, we can conclude that the Heisenberg dynamics of $|e\rangle\langle e|$ is entirely ruled by the effective non-Hermitian Hamiltonian $\hat{H}_{\mathrm{nH}}$. Although this is a simple one-atom problem, similar situations exist also in the many-body context when discussing the effects of atom loss, which we discuss further below. As we will also discuss in Sec. 5, $\hat{H}_{\mathrm{nH}}$ can additionally be used as a route to identifying the existence of "dark states" of open system dynamics.

The non-Hermitian dynamics we highlighted above is often referred to as "no-click dynamics". If one considers the quantum trajectory interpretation of the master equation, the non-Hermitian dynamics described by $\hat{\bar{\mathcal{L}}}_{\mathrm{nH}}[\hat{\rho}]$ is what one obtains if one performs a postselection on all the possible dynamics to select only those histories where no quantum jump has ever occurred—i.e. where the detector shows no "clicks". While these no-click configurations are in general exponentially rare, there can be situations in which they become relevant for the typical trajectory, as in the case of continuously monitored systems discussed in Sec. 8, in opposition to the averaged Lindblad dynamics.

## 4 Overview of Theoretical Methods

In this Section we give an overview of (analytical and numerical) methods used to study many-body open quantum systems. These include matrix-product states and tensor net-

works [274], the Corner-Space Renormalisation Method [275], Gutzwiller mean-field theories and their cluster extensions [276], Dynamical Mean-Field Theories [277], variational and neural network approaches [278–280] and Schwinger–Keldysh field theories [31, 32]. In addition, for small system sizes, there are software packages as QuTip [281, 282] that can be used to simulate the properties of open system. We emphasise that there are several reviews already existing on this topic, therefore we will limit ourselves to discuss some key aspects that are particularly relevant in the context of these Lecture Notes.

## 4.1 Exact solutions

We start by discussing classes of problems which can be solved exactly. In addition to being useful reference points against which to compare approximate methods, they can also form the starting point for perturbative expansions [283, 284]. In this regard it is worth noting that different notions of "exact solution" can exist. For some problems it is possible to find closed-form expressions for not only the steady state, but for all eigenvalues and right-eigenvectors of the Lindbladian superoperator in Eq. (16). For other problems it may be only possible to present exact solutions for the steady state.

**Quadratic (non-interacting) problems.** Quadratic problems—i.e. those in which the Hamiltonian is quadratic in bosonic or fermionic operators and the jump operators are linear—can be solved exactly. In general such problems take the form:

$$\hat{H} = \sum_{i,j} \left[ h_{ij}\hat{a}_i^\dagger \hat{a}_j + \frac{1}{2}\left(k_{ij}\hat{a}_i^\dagger \hat{a}_j^\dagger + \text{H.c.}\right)\right], \qquad \hat{L}_\mu = p_{\mu i}\hat{a}_i + q_{\mu i}\hat{a}_i^\dagger, \tag{75}$$

with $h, k, p, q$ as general matrices.

The fact that such problems can be exactly solved can be seen in various ways: in the context of the Schwinger–Keldysh path integral approach, such problems yield a quadratic action and thus involve Gaussian integrals. This means it is possible to exactly calculate any correlation functions from the Schwinger–Keldysh path integral by means of Wick's theorem. For bosonic problems, one can also use the Heisenberg–Langevin approach discussed above. This gives a linear equation that can be solved, and thus used to calculate any desired operator expectation value at later times.

One can also note that if the Lindbladian is quadratic, one can expect steady states to be of Gaussian form, and so one can use the well-known tools of Gaussian quantum information [285–288]. For a Gaussian state, one can express all correlation functions in terms of the covariance matrix. For our general problem, including anomalous terms, this reduces the problem to finding $\left\langle \hat{a}_i^\dagger \hat{a}_j \right\rangle, \left\langle \hat{a}_i^\dagger \hat{a}_j^\dagger \right\rangle$. All other correlation functions can be written in terms of these. For such Gaussian states, one may also note that phase space methods take a particularly simple form: the phase space distribution is also a Gaussian, with parameters determined by the covariance matrix. Furthermore one can use this exact solvability to classify the phases of quadratic open quantum systems [289].

Such quadratic problems can also be solved by third quantisation [201, 272, 273]. Using such an approach one can write explicit expressions for the left and right eigenvectors. These take the form of a ladder of states defined by the action of a set of third-quantised operators acting on the steady state $|\rho_{\text{ss}}\rangle\rangle$ for right eigenvectors, and a similar ladder of states starting from trace $\langle\langle\mathbb{1}|$. For the simplest case of a single bosonic mode with pumping and decay (i.e. $\hat{H} = \omega_c \hat{a}^\dagger \hat{a}, \hat{L}_- = \sqrt{\kappa_-}\hat{a}, \hat{L}_+ = \sqrt{\kappa_+}\hat{a}^\dagger$), and using the classical-quantum third-quantised operators given in Eq. (71), the eigenvectors and eigenvalues of

the Lindbladian take the form [273]:

$$|r_{\mu,\nu}\rangle\rangle = \frac{1}{\sqrt{\mu!\nu!}}\left(\hat{\hat{\mathbf{a}}}_q^\dagger\right)^\mu\left(\hat{\hat{\mathbf{a}}}_q\right)^\nu|\rho_{\text{ss}}\rangle\rangle\,,$$

$$\langle\langle l_{\mu,\nu}| = \frac{1}{\sqrt{\mu!\nu!}}\langle\langle\mathbb{1}|\left(\hat{\hat{\mathbf{a}}}_{cl} + \eta\hat{\hat{\mathbf{a}}}_q\right)^\mu\left(\hat{\hat{\mathbf{a}}}_{cl}^\dagger - \eta\hat{\hat{\mathbf{a}}}_q^\dagger\right)^\nu\,,$$

$$\lambda_{\mu,\nu} = \omega_c(\mu-\nu) - i\frac{\kappa_- - \kappa_+}{2}(\mu+\nu), \tag{76}$$

where $\eta = (\kappa_- + \kappa_+)/(\kappa_- - \kappa_+)$ gives a dimensionless noise strength[2]. For a general problem, the ladder operators that appear in defining the eigenstates are linear combinations of the original operators defining the problem. The linear combination involved can be found from the eigenvectors of an effective non-Hermitian Hamiltonian [273].

All the methods discussed above can be directly related by use of the classical and quantum third-quantised approach, as described in Ref. [273]. The connection between Keldysh action and the third quantised form of the Lindbladian was already mentioned above. The connection to phase space methods can be made by considering the eigenstates of the third-quantised operators (as a generalised form of coherent state), which shows:

$$\hat{\hat{\mathbf{a}}}_{cl}|z_{cl}\rangle\rangle = z|w_{cl}\rangle\rangle\,,\qquad \hat{\hat{\mathbf{a}}}_{cl}^\dagger|z_{cl}\rangle\rangle = z^*|z_{cl}\rangle\rangle\,,\qquad W(z) = \langle\langle z_{cl}|\rho\rangle\rangle\,. \tag{77}$$

Such an approach also makes clear the origin of a feature of the non-interacting system dynamics: the eigenvalues of the Lindbladian depend only on $\kappa_- - \kappa_+$, and not on the noise strength $\eta = (\kappa_- + \kappa_+)/(\kappa_- - \kappa_+)$. This independence is clear in the Schwinger–Keldysh formalism (see Eq. (54)). In the third quantised form, it can be seen that a non-unitary similarity transform on the vectorised problem:

$$\hat{\hat{\mathcal{L}}} \to \hat{\hat{\mathcal{V}}}^{-1}\hat{\hat{\mathcal{L}}}\hat{\hat{\mathcal{V}}}\,,\qquad \hat{\hat{\mathcal{V}}} = \exp\left(-\eta\hat{\hat{\mathbf{a}}}_q^\dagger\hat{\hat{\mathbf{a}}}_q\right)\,, \tag{78}$$

which maps the problem to an equivalent one with $\eta = 0$. This structure makes clear why the Lindbladian eigenvalues do not depend on $\eta$, but why $\eta$ does appear in the form of the eigenvectors.

**Quadratic Problems with Dephasing.** Another class of problems for which exact results can be obtained are quadratic systems of fermions and bosons with quadratic Hermitian jump operators, e.g.

$$\hat{H} = \sum_{i,j} h_{ij}\hat{a}_i^\dagger\hat{a}_j\,,\qquad \hat{L}_i = \sqrt{\gamma_\phi}\hat{a}_i^\dagger\hat{a}_i\,, \tag{79}$$

where $h$ is a general symmetric matrix $h_{ij} = h_{ji}$. The jump operators $\hat{L}_i$ here describe local dephasing [290–292]. Because the jump operators are quadratic, these models are described by a Lindbladian superoperator which is not of quadratic form. Still, because of the Hermitian nature of the jump operators, one can write down closed equations of motion for the correlation matrix $C_{i,j}(t) = \text{Tr}\left[\hat{\rho}(t)\hat{a}_i\hat{a}_j^\dagger\right]$. For example, in the specific case of one-dimensional nearest-neighbour hopping, $h_{ij} = J\left(\delta_{i,j+1} + \delta_{i+1,j}\right)$, we get [286, 288, 293, 294]

$$\frac{d}{dt}C_{i,j} = -iJ\left(C_{i-1,j}(t) + C_{i+1,j}(t)\right) + iJ\left(C_{i,j-1}(t) + C_{i,j+1}(t)\right) - \gamma_\phi(1-\delta_{ij})C_{i,j}(t)\,. \tag{80}$$

---

[2]If one writes $\kappa_- = \kappa_0(n_B+1), \kappa_+ = \kappa_0 n_B$, corresponding to pumping and decay rates for a state with average population given by the Bose-Einstein occupation $n_B(\omega_c)$, then $\eta = 2n_B(\omega_c) + 1 = \coth(\beta\omega_c/2)$.

In addition to $C_{i,j}(t)$, single particle dynamical correlation functions, i.e. Green's functions (see Sec. 3.2), can be obtained in closed form [294]. Furthermore, one can show that in general $k$th-order correlation functions also satisfy closed equations of motion. That is, in contrast to the generic case, the equations of motion for $k$th-order correlation functions do not couple to higher-order correlation functions. The exact solvability of these types of models can be seen also by looking at the properties of the Lindbladian (see Ref. [295] and discussion below) or by unravelling the Hermitian jump operator with a classical stochastic noise and resumming the resulting diagrammatic expansion exactly [296]. Because the jump operators are Hermitian, these models describe heating and thermalisation to infinite temperature as we will discuss in Sec. 7. They can also be used to describe diffusive transport when complemented with boundary driving terms [293, 297].

**Kerr-nonlinear oscillator.** A class of interacting problems for which exact solutions also exist is the Kerr nonlinear oscillator. Various different forms of this problem have been discussed [268, 271, 273, 298–305]. A general form of this model that covers most examples studied is given by the Hamiltonian:

$$\hat{H} = \omega \hat{a}^\dagger \hat{a} + \frac{U}{2} \hat{a}^\dagger \hat{a}^\dagger \hat{a}\hat{a} + \left( E^{(1)} \hat{a}^\dagger + E^{(2)} \hat{a}^\dagger \hat{a}^\dagger + \text{H.c.} \right), \tag{81}$$

along with dissipation terms

$$\hat{L}_{1-} = \sqrt{\kappa_-^{(1)}} \hat{a}, \qquad \hat{L}_{2-} = \sqrt{\kappa_-^{(2)}} \hat{a}\hat{a}, \qquad \hat{L}_{1+} = \sqrt{\kappa_+^{(1)}} \hat{a}^\dagger. \tag{82}$$

The first example considered is the linear driving of an anharmonic oscillator [299], so consists of the above model with $E^{(2)} = 0, \kappa_-^{(2)} = 0$. The key observation is that the complex-P distribution [268] for this model can be exactly found, because the corresponding equation of motion has a specific relation between the drift and diffusion terms [37]. Having found the density matrix, this then allows any properties of the steady state to be extracted.

The model with $\kappa_+^{(n)} = 0$ (so that dissipative loss is balanced by coherent pumping) has been studied in a number of recent works [271, 303], which have found ways to re-interpret how this exact solution arises. One approach involves the quantum absorber method [303, 306]. This involves considering losses from the system as being fed into a waveguide. If one can then devises a second system that perfectly absorbs all emission from the first system, this allows an exact solution of the dynamics. That is, if one considers a cascaded quantum system [264], with the original system and the second system connected via a chiral waveguide, the net effect is that the two systems reach a pure state, since there is no emission into the waveguide beyond the second system. This allows for an exact solution.

More recently, these results have been further re-interpreted as a signature of a hidden time-reversal symmetry [271]. This symmetry involves constructing a doubled version of the system by taking the steady state $\hat{\rho}_{\text{ss}} = \sum_\mu p_\mu |\psi_\mu\rangle\langle\psi_\mu|$ and using this to construct a state $|\Psi_T\rangle = \sum_\mu \sqrt{p_\mu} |\psi_\mu\rangle_A |\hat{T}\psi_\mu\rangle_B$. Here $\hat{T}$ is some generalised time-reversal operation, and $A, B$ refer to the two copies of the system. One then constructs $\hat{\rho}_{AB}(0) = |\Psi_T\rangle\langle\Psi_T|$ and time evolves this under the doubled Lindbladian, $\hat{\tilde{\mathcal{L}}}_{AB} = \hat{\tilde{\mathcal{L}}} \otimes \hat{\tilde{\mathbb{1}}}$. The generalised time-reversal symmetry occurs if there exists some operator $\hat{T}$ such that the evolution of $\hat{\rho}_{AB}(t)$ under $\hat{\tilde{\mathcal{L}}}_{AB}$ satisfies

$$\text{Tr}\left[ \hat{X}_A \hat{Y}_B \hat{\rho}_{AB}(t) \right] = \text{Tr}\left[ \hat{Y}_A \hat{X}_B \hat{\rho}_{AB}(t) \right] \tag{83}$$

for any pair of operators $\hat{X}, \hat{Y}$. Reference [271] shows that this condition is equivalent to demanding a duality condition of the Lindbladian under a class of mappings. This

identification provides an explicit construction of the required quantum absorber system, connecting to the method above. This approach also shows that the hidden time reversal symmetry is broken by thermal noise, so only survives for $\kappa_+^{(n)} = 0$.

A different special case is that without coherent pumping, i.e. where $E^{(n)} = 0$ [273, 301, 307]. In this case, the model has a weak symmetry corresponding to the phase of the photon mode, $\hat{S} = e^{i\theta\hat{a}^\dagger\hat{a}}$; this symmetry can be used to solve this model. As noted above, weak symmetries are sufficient to ensure the Lindbladian has a block diagonal structure; the key observation of Ref. [307] is that for some specific models (including the incoherently pumped Kerr model when $\kappa_-^{(2)} = 0$), each sector can be mapped to an equivalent non-interacting model. This allows one to find the complete Lindbladian eigenspectrum. The model can also be solved by transforming the third-quantised Lindbladian [273]. This has the effect of mapping the problem to a quadratic (and thus solvable model), but with a non-trivial time dependent transform for all operators. This nonetheless allows direct calculation of the time evolution of any initial state.

**Dissipative all-to-all Ising model.** Another example of exactly solvable models are certain versions of the Ising model, as written in Eqs. (30) and (31) [228, 307]. One way to understand the solution of this model is by noting it has many weak symmetries [307] under the operations $\hat{S}_i = e^{\theta\sigma_i^z}$. As with the example of the Kerr oscillator above, this means the Lindbladian has a block diagonal structure, and further elements of the problem make it possible to solve the model in each of these sectors.

There also exists an exact solution for a version of the transverse-field Ising model [308], but here with all-to-all coupling of the spins:

$$\hat{H} = J\sum_{i\neq j}\hat{\sigma}_i^z\hat{\sigma}_j^z + \sum_i\sum_\alpha h_i^\alpha\hat{\sigma}_i^\alpha, \qquad \hat{L}_0 = \sqrt{\Gamma}\sum_i\sigma_i^-, \qquad \hat{L}_i = \sqrt{\gamma}\sigma_i^-, \qquad (84)$$

with both collective and individual decay, and with both transverse and longitudinal fields $h_i^\alpha$ on each spin. For this model it is possible in certain cases to construct an operator $\hat{\Psi}$ such that in terms of the non-Hermitian Hamiltonian $\hat{H}_{\text{nH}} = \hat{H} - \frac{i}{2}\sum_\mu L_\mu^\dagger L_\mu$, one has $\hat{H}_{\text{nH}}^\dagger\hat{\Psi} = \hat{\Psi}\hat{H}_{\text{nH}}$ and also $\hat{L}_\mu\hat{\Psi} = \hat{\Psi}\hat{L}_\mu^\dagger$. This is a sufficient condition to mean that $\hat{\rho}_{\text{ss}} = \hat{\Psi}\hat{\Psi}^\dagger$ is a steady state solution. The operator $\hat{\Psi}$ can be found when either $\Gamma = 0$ or when $h_i^\alpha$ is uniform (independent of site).

**Integrable Lindbladians.** Finally, we close by mentioning recent developments in solving Lindblad problems with integrability techniques, most notably the Bethe ansatz. The idea behind these approaches is that the Lindblad superoperator, when written in vectorised form (as discussed in Sec. 3.1.2), takes the form of a non-Hermitian Hamiltonian which can in certain cases be solved by the Bethe ansatz. One example is again the problem of free fermions with dephasing discussed above. This can be mapped to a non-Hermitian Hubbard model with a purely imaginary interaction and solved by Bethe ansatz [295, 309]. A similar approach can be explored for models of two-level systems with collective dissipation, whose vectorised Lindbladian maps to the non-Hermitian Gaudin-Richardson model, which is again Bethe-ansatz solvable [310, 311]. Other examples involve mapping to non-Hermitian XXZ spin chains with different types of boundary conditions [312, 313]. In all these works one can obtain the full spectrum of the Lindbladian and sometime even obtain matrix elements of local operators from which the dissipative dynamics can be reconstructed.

## 4.2 Cluster expansions

Cluster expansion methods—as we will define below—can be relevant in cases where certain properties of many-body open systems can be extracted through a detailed knowledge of short-range correlations. This means that in some cases, one can extract useful results from exact numerical simulations of finite-sized systems by direct simulation of the master equation, which can be considered as the open-system analogue of exact diagonalisation.

To access behaviour of the infinite system from finite-sized simulations, one can use various scaling methods. A convenient technique is the linked-cluster expansion that has been successfully applied over several decades in classical and quantum statistical mechanics [314]. Powerful series expansions and resummation techniques [315, 316], developed over the years, allow to further determine long-range behaviour close to critical points from the scaling of a given observable for different cluster sizes, as discussed next.

The key idea behind the linked-cluster expansion is related to the fact that extensive quantities of lattice systems can be computed, in the thermodynamic limit, by means of series expansions whose different terms associated to clusters of increasing numbers of sites. The way the various terms appear in the series is dictated by the topology of the lattice and by the connectivity of the generator of the thermodynamics (the Hamiltonian, for the study of equilibrium properties) or of the dynamics (the Lindbladian, for the study of stationary properties); here, we will be interested in the latter situation. We briefly outline the method of numerical linked-cluster expansions (NLCEs), where the contributions to the series are obtained by means of exact diagonalisation techniques on finite-size clusters sites [315, 316].

We start with a presentation of the formalism following Ref. [317], to make evident its natural applicability to the study of driven-dissipative quantum systems governed by a Lindblad dynamics. In our discussion, we focus on the steady-state properties, but the approach is immediately adapted to the real-time dynamics as well. The goal is to compute expectation values of generic extensive observables $\hat{O}$ over the stationary density matrix $\hat{\rho}_{\rm ss}$.

The key to the method relies on the fact that the Lindbladian can be written as a sum of local terms; for the sake of simplicity let us assume that only couplings between at most neighbouring sites matter:

$$\hat{\hat{\mathcal{L}}} = \sum_{i,j} \alpha_{ij} \hat{\hat{\mathcal{L}}}_{ij}, \quad \text{with } \alpha_{ij} = 0 \text{ if } i, j \text{ are not neighbours.} \tag{85}$$

The observable $\langle \hat{O} \rangle = \text{Tr}\left[\hat{O}\hat{\rho}_{\rm ss}\right]$ can be always formally arranged in an expansion in powers of the $\alpha_{ij}$:

$$\langle \hat{O} \rangle (\{\alpha_{ij}\}) = \sum_{\{n_{ij}\}} O_{\{n_{ij}\}} \prod_k \alpha_{ij}^{n_{i,j}} \tag{86}$$

where $n_{,j}$ runs over all non-negative integers for each $(i, j)$, such that any possible monomial in the $\alpha_{ij}$ is included; the $O_{\{n_{ij}\}}$ are the coefficients of the resulting polynomial. The expansion (86) can be then organised in clusters:

$$\langle \hat{O} \rangle (\{\alpha_{i,j}\}) = \sum_c W_{[O]}(c), \tag{87}$$

where each $c$ represents a non-empty set of links between sites, which define the given cluster. Specifically, the so-called cluster weight $W_{[O]}(c)$ contains all terms of the expansion (86), which have at least one power of $\alpha_k$ if $k \in c$, and no powers of $\alpha_k$ if $k \notin c$. Vice-versa, all terms in Eq. (86) can be included in one of these clusters. Note that this writing is fully general, and can be used on finite sistems as well as on infinite ones.

Using the inclusion-exclusion principle, we can define $\langle O(c)\rangle = \mathrm{Tr}\big[\hat{O}\hat{\rho}_{\mathrm{ss}}(c)\big]$ as the steady-state expectation value of the observable $\hat{O}$ calculated for the finite cluster $c$ and using the definition given by the sum in Eq. (87) we obtain the recurrence relation:

$$W_{[O]}(c) = \langle O(c)\rangle - \sum_{s \subset c} W_{[O]}(s), \tag{88}$$

where the sum runs over all the sub-clusters $s$ contained in $c$, and $\hat{\rho}_{\mathrm{SS}}(c)$ is the steady state of the restricted Lindbladian $\hat{\mathcal{L}}(c)$ defined only over the cluster $c$. An important property of Eq. (88) is that, if $c$ is formed out of two disconnected clusters $c_1$ and $c_2$, its weight $W_{[O]}(c)$ is zero. This follows from the fact that $\langle O\rangle$ is an extensive property $\langle O(c)\rangle = \langle O(c_1)\rangle + \langle O(c_2)\rangle$ and $c = c_1 + c_2$. This means that in the expansion (87) one can simply focus on connected clusters, hence the name of *linked* cluster expansion. Obviously, this implies a significant calculation advantage.

The symmetries of the Lindbladian $\hat{\mathcal{L}}$ may drastically simplify the summation (87), since it is typically not needed to compute all the contributions coming from each connected cluster. This can be immediately seen, e.g., for situations where the interaction term $\alpha$ between different pairs of sites is homogeneous throughout the lattice. In such cases, it is possible to identify the topologically-distinct linked clusters, so that a representative $c_n$ for each class can be chosen and counted according to its multiplicity $\ell(c_n)$ per lattice site (the lattice constant of the graph $c_n$). Here the subscript $n$ denotes the number of $k$-spatial indexes that are grouped in the cluster, that is, its size. The property $\langle O\rangle$ per lattice site can be thus written directly in the thermodynamic limit $N \to \infty$ as:

$$\frac{\langle \hat{O}\rangle}{N} = \sum_{n=1}^{\infty} \left[\sum_{\{c_n\}} \ell(c_n)\, W_{[O]}(c_n)\right]. \tag{89}$$

The outer sum runs over all possible cluster sizes, while the inner one accounts for all topologically distinct clusters $\{c_n\}$ of a given size $n$. Let us emphasise that, if the series expansion (89) is truncated up to order $n = R$, only clusters $c$ at most of size $R$ have to be considered. Indeed each of them should include at least one power of $\alpha_{i,j}$, $\forall (i,j) \in c$. Therefore a cluster of size $R+1$ or larger does not contribute to the expansion, up to order $\alpha^R$.

As a matter of fact, dealing with open many-body systems significantly reduces our ability to compute large orders in the expansion, with respect to the *closed*-system scenario. The size of the space onto which the linear Lindblad superoperator $\hat{\mathcal{L}}$ acts scales as $d_H^{2n}$, where $d_H$ is the dimension of the local Hilbert space and $n$ is the number of sites of a given cluster. In isolated systems, one would need to evaluate the ground state of the cluster Hamiltonian, of size $d_H^n$. Therefore, for the case of spin-1/2 systems ($d_H = 2$), for clusters up to $n = 8$, the study of $\hat{\mathcal{L}}$ requires to work with linear spaces of dimension $2^{2\times 8} = 65536$. In addition to the calculation of the observable on a given cluster, algorithms exist to compute all topologically distinct clusters, for a given size and lattice geometry. A remarkable advantage of this method is that it enables a direct access to the thermodynamic limit by only counting the cluster contributions of sizes equal or smaller than a certain size, thus using a limited amount of resources). There is no small parameter which controls this expansion, the actual control parameter for the expansion is given by the amount of correlations that are present in the system. Needless to say, optimised algorithms for cluster generation are crucial to boost the power of this method, see [318] and references therein for additional details on the method with applications to Hamiltonian systems.

Numerical linked-cluster expansions have been used in driven-dissipative systems, for example, in [317, 319]. As an example of the method, we show how it performs for two-

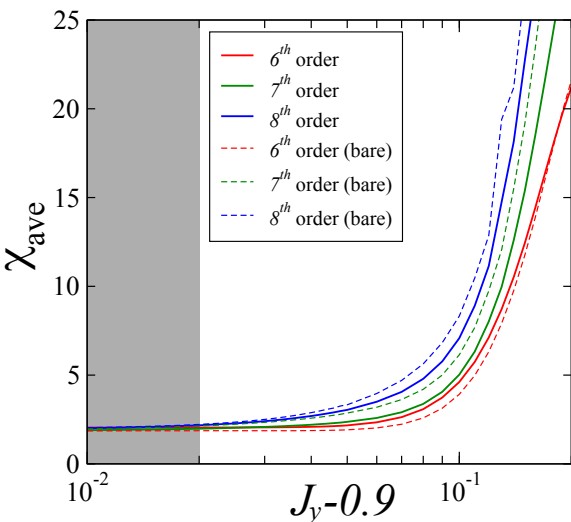

Figure 13: Numerical linked-cluster expansion (NLCE) calculation of the averaged magnetic susceptibility of the model in Eqs. (28),(29). Magnetic susceptibility is averaged over the $x-y$ plane, and plotted as a function of $J_y$ with other parameters set to $J_x = 0.9$, $J_z = 1, \gamma = 1$. Solid lines show the Euler resummed data of the NLCE up to the highest-achievable expansion order, $R = 8$. The dashed lines show the corresponding bare expansions. The shaded region, $0 < J_y - 0.9 < 0.02$ is where the NLCE converges. Adapted From Ref. [317] [Copyright (2018) by the American Physical Society].

dimensional dissipative quantum lattice models of interacting spin-1/2 particles, specifically an anisotropic Heisenberg spin-1/2 model with local (single-site) incoherent spin relaxation, as given by Eq. (28) and (29). We set $J_x = 0.9$, $J_z = 1$, and the dissipation $\gamma = 1$. For $J_y = 0.9$ the Hamiltonian conserves the magnetisation along the $z$ direction, and the steady state is the pure state with all the spins pointing down in the $z$ direction. Away from this singular point, for a certain $J_y > J_{y,c} > 0.9$ the system undergoes a second-order phase transition associated to the spontaneous breaking of the $\mathbb{Z}_2$. That is, a transition from a paramagnetic phase for $J_y < J_{y,c}$ to a ferromagnetic (FM) phase for $J_y > J_{y,c}$. In the FM phase, a finite magnetisation in the $x$-$y$ plane develops: $\langle \hat{\sigma}^x \rangle, \langle \hat{\sigma}^y \rangle \neq 0$, which also defines the order parameter of the transition. This phase transition has recently received a lot of attention, see for example [227, 276, 320, 321] and for this choice of parameters the critical point has been estimated to be $J_{y,c} \sim 1.0 - 1.1$. As an example of linked-cluster expansion at work, we show in Fig. 13 the angular-averaged susceptibility to an external field in the $x - y$ plane as a function of the coupling parameter.

## 4.3    Mean-field approximations

Mean-field theory can be a powerful approximation scheme to simplify many-body problems. In some cases it is a crude approximation, but can nonetheless help identify possible phases and the structure of the phase diagram (even if not the correct location of phase boundaries). In other cases, mean-field theory can prove to be a good, even exact, approximation. As explored further below, mean-field theory can also be the starting point for a variety of expansions that systematically increase the extent of correlations which are taken into account.

At its core, the key idea of mean-field theory is to replace an interacting many-body system with one where one part of a system evolves in a "mean field" arising due to the

effects of other parts of the system. The key approximation here is to neglect correlations between these parts of the system. Mean-field approximations can be formulated in a variety of contexts, including pure quantum dynamics (where it becomes an approximation for the wavefunction), classical problems, and as we discuss here, for open quantum systems. In this last case, the essential structure of a mean-field theory is to assume some form of factorisable density matrix:

$$\hat{\rho} = \bigotimes_{\mu} \hat{\rho}_{\mu} \tag{90}$$

where $\hat{\rho}_{\mu}$ is the density matrix of the $\mu$-th part of the system—for example, factorisation in real space (for which part $\mu$ means different sites), or momentum space (for which part $\mu$ means different momentum states). We discuss many different examples of this in the rest of this section. We note that this formal statement of the ansatz will hold for most, but not all, of the cases we discuss below; the conceptual structure of a factorised ansatz will however hold in all cases. Such a description is clearly abstract, in that there can be many ways to divide up the system. As such, there can often be more than one possible mean-field decoupling, as we discuss in the sections below.

In many of the cases we discuss below, there can be alternative statements of how one proceeds to derive a mean-field theory—for example, replacing operators by classical variables, factorising expectations of products of operators by products of expectations, or assuming only pairwise correlations among particles exist. While these rules may appear simpler than the statement of factorisation, they obscure the nature of the approximation that is actually being made, and can give the impression that different types of mean-field theory are distinct and unrelated. Our aim in this section is to make clear the links between different forms of mean-field theory (as well as their relation to beyond-mean-field approaches). These links come precisely from considering that mean-field theories are all related to assumptions of factorisation. We will nonetheless make clear in each of the following sections where an alternate approach can lead to the same end point.

When considering how to use mean-field approaches, two significant challenges arise. The first challenge is to decide which form of mean-field factorisation is most appropriate. We will discuss below some general considerations on when a given mean-field approach is likely to be accurate, which can be used to guide such a decision. Beyond this, one can consider known experimental behaviour of a system, or similarity to other known systems to propose a particular form of factorisation. The second challenge is how to then use the ansatz to find either the steady state or the dynamics. For an isolated system in its ground state, one can use a variational approach to minimise the ground state energy; i.e. for some variational ansatz $|\Psi_v\rangle$ one may vary parameters to minimise the energy $E_v = \langle \Psi_v | \hat{H} | \Psi_v \rangle$. For a system in thermal equilibrium at non-zero temperature, one may similarly consider a variational ansatz for the density matrix $\hat{\rho}_v$ and minimise the free energy, $F = \mathrm{Tr}\left[\hat{H}\hat{\rho}_v + k_B T \hat{\rho}_v \ln \hat{\rho}_v\right]$. For an open quantum system the equivalent procedure is less obvious since the steady state is not determined by optimising a thermodynamic potential. Ultimately one seeks a solution of the equation $\hat{\bar{\mathcal{L}}}[\rho] = 0$, however any restricted ansatz may be unable to exactly solve this equation. One can convert this condition into a minimisation problem in various ways; a popular choice involves taking the modulus squared of the vectorised form, i.e. $\left\|\hat{\bar{\mathcal{L}}}[\hat{\rho}]\right\|^2 = \langle \rho | \hat{\mathcal{L}}^{\dagger} \hat{\mathcal{L}} | \rho \rangle$.

In many cases, one is interested in the time evolution rather than just in the steady state. Here one can use an alternative variational approach, the time-dependent variational principle [322–324] (TDVP). This consists of assuming that one has a state $\hat{\rho}_t$ within the variational manifold at time $t$, and then finding the state at time $t + \delta t$ by minimising some distance measure $D\left[\hat{\rho}_{t+\delta t}, \hat{\rho}_t + \delta t \hat{\bar{\mathcal{L}}}[\hat{\rho}_t]\right]$, denoting the distance between the best variational

state at the new time and the time evolution of the previous state. In the analogous approach for pure quantum states it is clear that the distance measure to be used is the Euclidean norm $D\left[|\psi_1\rangle, |\psi_2\rangle\right] = \sqrt{\langle\psi_1 - \psi_2|\psi_1 - \psi_2\rangle}$; this will maximise overlap between the variational state at the next time step and its "correct" value. As with finding the steady state, the choice of correct distance function to minimise for density matrices is less clear, but approaches minimising the Euclidean norm [186] are often used.

### 4.3.1  Real space factorisation and Gutzwiller mean-field theories

In bosonic lattice models (such as the Bose–Hubbard or related models) or quantum spin chains, one natural mean-field factorisation is between different lattice sites. In this case $\hat{\rho} = \bigotimes_i \hat{\rho}_i$ holds with $i$ running over the site labels. Such an ansatz is closely related to the Gutzwiller ansatz first introduced in the context of the Fermi–Hubbard model [325]. However, as we will discuss later, for bosonic models this factorised ansatz becomes exact in the limit of infinite lattice connectivity in contrast to fermionic analogues. As such, we will refer to mean-field theories of this form as Gutzwiller mean-field theories [326].

An alternative context where similar factorisation holds comes for problems involving many emitters coupled to a single cavity [224]. Here, a common factorisation is between the state of the cavity modes and the state of the emitters, and so in this case Eq. (90) again holds with $i$ running over the cavity and the emitters.

To discuss this spatial factorisation more explicitly, suppose one considers a Lindblad master equation where the Hamiltonian takes the form $\hat{H} = \sum_i \hat{H}_i + \sum_{i \neq j} \hat{H}_{ij}$, describing terms that act on a single part of the system, $i$ and terms that couple two parts, $i$ and $j$. If we further assume that each jump operator involves operators on only one part of the system, $\hat{L}_\mu = \hat{L}_{\mu,i}$, then one may find a simplified equation of motion within the space of factorised density matrices:

$$\frac{d\hat{\rho}_i}{dt} = -i[\hat{H}_i + \hat{\Theta}_i, \hat{\rho}_i] + \sum_\mu \hat{\mathcal{D}}[\hat{L}_{\mu,i}, \hat{\rho}_i], \tag{91}$$

where $\hat{\Theta}_i = \sum_{j \neq i} \mathrm{Tr}_j(\hat{\rho}_j \hat{H}_{ij})$ describes the mean field of the other parts of the system, i.e. parts $j$, acting back on part $i$. In cases where the jump operators are not local, one can derive a more complex equation involving partial traces of these terms. The structure of this equation means that the evolution of different $\hat{\rho}_i$ are coupled, but only through a mean-field expectation. This coupling means the evolution of each $\hat{\rho}_i$ depends on expectations of terms on other sites.

To give concrete examples of such an approach, we may consider the XYZ spin model and dissipative Bose–Hubbard model that were introduced in Sec. 3.1.5. For the XYZ model, the Hamiltonian in Eq. (28) contained only pairwise interaction terms involving nearest-neighbour sites, while the dissipation, Eq. (29) acted on a single site. The mean-field decoupling then leads to

$$\hat{\Theta}_i = \sum_{j \in \partial_i} \mathrm{Tr}\left[\hat{\rho}_j \sum_\alpha J_\alpha \hat{\sigma}_i^\alpha \hat{\sigma}_j^\alpha\right],$$

where $j \in \partial_i$ indicates a sum over the sites $j$ that are nearest neighbours to site $i$. We can thus write the resulting mean-field equation of motion as:

$$\frac{d\hat{\rho}_i}{dt} = -i\left[\sum_\alpha h_i^\alpha \hat{\sigma}_i^\alpha, \hat{\rho}_i\right] + \hat{\mathcal{D}}[\sqrt{\gamma}\hat{\sigma}_i^-, \hat{\rho}_i], \qquad h_i^\alpha = J_\alpha \sum_{j \in \partial_i} \mathrm{Tr}\left[\hat{\rho}_j \hat{\sigma}_j^\alpha\right]. \tag{92}$$

In the limit of weak coupling $J^\alpha$, the steady state will be dominated by the dissipation, leading to a spin down state, with $h_i^x = h_i^y = 0$. When the couplings $J^x, J^y$ become large enough, one has a transition to a state where there is a self-consistent solution with non-zero $h_i^x, h_i^y$.

Applying the same logic to the Bose–Hubbard Hamiltonian given by Eq. (36) we find

$$\frac{d\hat{\rho}_i}{dt} = -i \left[ \phi_i \hat{a}_i^\dagger + \phi_i^* \hat{a}_i + \frac{U}{2}\hat{n}_i(\hat{n}_i + 1), \hat{\rho}_i \right] + \sum_\mu \hat{\mathcal{D}}[\hat{L}_{i,\mu}, \hat{\rho}_i], \quad \phi_i = -t_H \sum_{j\in\partial_i} \text{Tr}\left[\hat{s}\hat{\rho}_j \hat{a}_j\right]. \tag{93}$$

Here we have allowed for a general set of on-site dissipative processes, as described when introducing this model. For this mean-field factorisation, there is a transition between a superfluid state with non-zero $\phi_i$ and a normal state where $\phi_i = 0$. In the absence of dissipation the ground state with $\phi_i = 0$ is a Mott insulating state with integer occupation per site, however this is modified by any pumping or dissipation (or non-zero temperature).

We may note that this form of real-space mean-field factorisation is not appropriate for the Fermi–Hubbard model. This is because physical Hamiltonians always conserve fermion parity, whereas decoupling the hopping term for the Fermi–Hubbard model would give a model with single fermion operators, and would require a state with a non-zero expectation of a single fermion operator. We discuss an alternate approach that resolves this issue—dynamical mean-field theory—in section 4.4.

When considering Gutzwiller mean-field theories, it is worth distinguishing the mean-field approximation from two other additional approximations that are sometimes associated with mean-field factorisations in real space, but which are in fact additional approximations.

**Spatial Homogeneity.** The first is the assumption of spatial homogeneity. If one considers Eq. (91) for lattice model, one may note that in general the equation allows for $\hat{\rho}_i$ to evolve independently on each site. If one is focused on steady states, and one expects translational invariance, one can make an additional assumption that $\hat{\rho}_j = \hat{\rho}_i$ for all $i, j$, reducing to a single self-consistent equation of motion. One may though note that this is an additional assumption on top of mean-field theory. Note further that even when the model is translationally invariant, the steady state need not be. For example, antiferromagnetic coupling between sites might lead to a state with staggered order, and more complex forms of coupling can lead to other commensurate or incommensurate states, e.g. [227, 327–329].

**Semiclassical approximation.** The second approximation is the semiclassical approximation, when considering bosonic modes. This is the assumption that bosonic operators $\hat{a}$ can be replaced by classical fields $\psi$ throughout the equations of motion. There are some cases where mean-field theory in real space does imply a semiclassical approximation. One such example is the Dicke model, defined by Eqs. (26) and (27). If the mean-field factorisation introduced above is made between the cavity, $\hat{\rho}_c$, and the emitters, $\hat{\rho}_{e_i}$, i.e. $\hat{\rho} = \left(\bigotimes_i \hat{\rho}_{e_i}\right) \otimes \hat{\rho}_c$, then the mean-field equation for the cavity mode becomes:

$$\frac{d\hat{\rho}_c}{dt} = -i \left[ \omega_c \hat{a}^\dagger \hat{a} + \phi\left(\hat{a}^\dagger + \hat{a}\right), \hat{\rho}_c \right] + \kappa \hat{\mathcal{D}}[\hat{a}], \tag{94}$$

where the term $\phi$ comes from a trace over the emitters, i.e. $\phi = g \sum_i \text{Tr}\left[\hat{\sigma}_i^x \hat{\rho}_{e_i}\right]$ for the model in Eq. (26). One may easily see that the stationary solution of this equation is a coherent state, $\hat{\rho}_c = |z\rangle\langle z|$ with $|z\rangle = \exp\left(z\hat{a}^\dagger - z^*\hat{a}\right)|0\rangle$, $z = -\phi/(\omega_c - i\kappa/2)$. Thus, in

this specific case the mean-field factorisation implies that the state of the cavity bosonic mode is one for which the semiclassical approximation holds.

Despite examples such as that above, Gutzwiller mean-field factorisation is not in general equivalent to assuming a coherent state or that replacing operators with fields holds. For example, in the Bose–Hubbard model discussed above, the on-site Hamiltonian involves interacting terms, and as a result, the steady state is not a coherent state.

### 4.3.2    Momentum space factorisation

A second form of mean-field factorisation that often arises is that in momentum space. This corresponds to considering a density matrix of the form $\hat{\rho} = \bigotimes_{\mathbf{k}} \hat{\rho}_{\mathbf{k}}$, with each term involving only the pair of momenta $\mathbf{k}, -\mathbf{k}$. This structure is chosen to be compatible with states with overall zero momentum while allowing only correlations between excitations with equal and opposite momenta. Such approaches have been used to consider the dynamics of pairing in the BCS model (see e.g. [330, 331]), as well as for driven dissipative problems [236, 237] or for lossy gases [332].

Momentum factorisation is often accompanied by other special properties, that are sometimes thought of as intrinsic properties of mean-field approaches. As an example, we will next show that the standard BCS ground state ansatz can be recovered from two different starting points: one is from considering factorisation in momentum space, another is from considering translationally invariant Gaussian states. From the perspective introduced at the start of this section, the key feature of the BCS state is its factorised nature, rather than its Gaussian form, although (as we will show), both assumptions lead to the same end result. We consider the BCS ground state because this is the most famous example of such momentum state factorisation, and a particularly simple case to discuss. These considerations can be extended to open quantum systems with some additional work, but for simplicity and clarity we restrict this discussion to a closed-system example.

One may write down the momentum-factorised ansatz for the ground-state wavefunction of the BCS model by three considerations: assuming factorisation of the state in momentum space, $|\Psi\rangle = \prod_{\mathbf{k}} |\psi_{\mathbf{k}}\rangle$, requiring a state with definite fixed fermion number parity (i.e., not mixing odd and even fermion number states), and requiring a state with zero total momentum. These requirements yield the state:

$$|\Psi\rangle = \prod_{\mathbf{k} \geq 0} \left( \psi_{\mathbf{k}}^{(0)} + \sum_{\sigma, \sigma' \in \uparrow, \downarrow} \psi_{\mathbf{k}, \sigma\sigma'}^{(1)} \hat{c}_{-\mathbf{k}, \sigma}^{\dagger} \hat{c}_{\mathbf{k}, \sigma'}^{\dagger} + \psi_{\mathbf{k}}^{(2)} \hat{c}_{-\mathbf{k}, \downarrow}^{\dagger} \hat{c}_{-\mathbf{k}, \uparrow}^{\dagger} \hat{c}_{\mathbf{k}, \uparrow}^{\dagger} \hat{c}_{\mathbf{k}, \downarrow}^{\dagger} \right) |0\rangle . \tag{95}$$

Note that the notation $\mathbf{k} \geq 0$ in this product indicates a product over the half-space of momentum $\mathbf{k}$, since each term contains all contributions from both $\pm\mathbf{k}$. The corresponding density matrix $\hat{\rho} = |\Psi\rangle\langle\Psi|$ constructed frm this takes exactly the form of momentum-factorised ansatz discussed above.

If one further considers using this ansatz for the ground state of the BCS Hamiltonian (39) one finds that one can consider a restricted version:

$$|\Psi\rangle = \prod_{\mathbf{k}} \left( \cos\theta_{\mathbf{k}} + \sin\theta_{\mathbf{k}} e^{i\phi_k} \hat{c}_{-\mathbf{k}, \downarrow}^{\dagger} \hat{c}_{\mathbf{k}, \uparrow}^{\dagger} \right) |0\rangle . \tag{96}$$

Note that in this expression the product over $\mathbf{k}$ includes all terms. The simplification to this form can be understood by noting that after factorising over momentum, the non-vanishing terms in Eq. (39) are all quadratic in fermions, and the anomalous terms take the forms $\hat{c}_{-\mathbf{k}, \downarrow}^{\dagger} \hat{c}_{\mathbf{k}, \uparrow}^{\dagger}$ or $\hat{c}_{-\mathbf{k}, \downarrow} \hat{c}_{\mathbf{k}, \uparrow}$, leading to the form given. The use of $\theta_{\mathbf{k}}, \phi_{\mathbf{k}}$ to write the expression ensures normalisation. This quadratic form also means, as discussed next, that the BCS mean-field theory can alternatively be found from a Gaussian ansatz.

**Gaussian ansatz.** As in our discussion of real-space factorisation, there are features that arise in the momentum space factorisation of some problems (including the BCS ansatz) that are not generic properties of mean-field theory, but which frequently accompany with mean-field theory. In particular, one may note that the BCS ansatz written above can equivalently be recovered by assuming a Gaussian state. Such a state can be written either in real space or momentum space. In real space this means writing:

$$|\Psi\rangle = \exp\left(\iint d\mathbf{r}d\mathbf{r}'\phi(\mathbf{r}-\mathbf{r}')\hat{c}_\uparrow^\dagger(\mathbf{r})\hat{c}_\downarrow^\dagger(\mathbf{r}')\right)|0\rangle. \tag{97}$$

In writing this we have imposed translational invariance of this Gaussian ansatz. As a result, if we switch to momentum space operators by using

$$\hat{c}_\sigma^\dagger(\mathbf{r}) = \frac{1}{L^{d/2}}\sum_{\mathbf{k}}\hat{c}_{\mathbf{k},\sigma}^\dagger e^{-i\mathbf{k}\cdot\mathbf{r}}, \tag{98}$$

one finds the equivalent momentum-space representation:

$$|\Psi\rangle = \exp\left(\sum_{\mathbf{k},\mathbf{k}'}\tilde{\phi}_{\mathbf{k},\mathbf{k}'}\hat{c}_{\mathbf{k},\uparrow}^\dagger\hat{c}_{\mathbf{k}',\downarrow}^\dagger\right)|0\rangle, \tag{99}$$

with

$$\tilde{\phi}_{\mathbf{k},\mathbf{k}'} = \frac{1}{L^d}\iint d\mathbf{r}d\mathbf{r}'\phi(\mathbf{r}-\mathbf{r}')e^{i(\mathbf{k}\cdot\mathbf{r}+\mathbf{k}'\cdot\mathbf{r}')} = \chi_{\mathbf{k}}\,\delta_{\mathbf{k},-\mathbf{k}'}, \qquad \chi_{\mathbf{k}} \equiv \int d\mathbf{s}\phi(\mathbf{s})e^{i\mathbf{k}\cdot\mathbf{s}} \tag{100}$$

which then yields the state:

$$|\Psi\rangle = \exp\left(\sum_{\mathbf{k}}\chi_{-\mathbf{k}}\hat{c}_{-\mathbf{k},\uparrow}^\dagger\hat{c}_{\mathbf{k},\downarrow}^\dagger\right)|0\rangle = \prod_{\mathbf{k}}\left(1+\chi_{-\mathbf{k}}\hat{c}_{-\mathbf{k},\uparrow}^\dagger\hat{c}_{\mathbf{k},\downarrow}^\dagger\right)|0\rangle. \tag{101}$$

This recovers the BCS ansatz (up to normalisation) if we identify $\chi_{\mathbf{k}} = \tan(\theta_{\mathbf{k}})e^{i\phi_{\mathbf{k}}}$. Note that the combination of translational invariance and Gaussianity implies the state must factorise in momentum space. In contrast the state in Eq. (97) does not factorise in real space—i.e. it has correlations between the state at different positions in space.

As with our discussion of the relation between real-space factorisation and the semiclassical approximation, not all momentum space factorisations necessarily imply a Gaussian state. Indeed, the most general momentum space factorisation given in Eq. (95) is clearly not Gaussian. Further examples of non-Gaussian but momentum-factorised states are given by the Jastrow wave function for fermionic or bosonic Mott insulators [333, 334], which can be factorised in momentum space, when the system is translational invariant, yet does not satisfy Wick theorem because of the Jastrow projector.

### 4.3.3 Particle-based factorisation: Hartree–Fock theory and Gross-Pitaevskii equation

The above two approaches consider factorisation between different single-particle orbitals, i.e. different modes of the system. One can also consider factorisation between the states of different particles. If particles are distinguishable, this would have the straightforward meaning that, as a wavefunction, $\Psi(\mathbf{r}_1,\mathbf{r}_2,\ldots,\mathbf{r}_N) = \prod_{i=1}^N \psi_i(\mathbf{r}_i)$. For indistinguishable particles, this simple product must be replaced by a Slater determinant of a matrix of wavefunctions for fermions, or a matrix permanent for bosons. These constructs are required so that the wavefunction obeys the required anti-symmetry or symmetry. Such an

approach corresponds to Hartree–Fock theory [335]. Similar expressions can be written for density matrices as we discuss further below.

For bosons (but not for fermions) one can also consider the special case in which all particles are in the same state. In this case the matrix permanent takes a simple form: $\Psi(\mathbf{r}_1, \mathbf{r}_2, \ldots, \mathbf{r}_N) = \prod_{i=1}^{N} \psi(\mathbf{r}_i)$. Such an approximation is the basis for the widely used Gross–Pitaevskii equation that is the mean-field theory for the weakly interacting dilute Bose gas (WIDBG) model, introduced in Sec. 3.1.5

In second-quantised notation corresponding to Eq. (37), the product state ansatz takes the form:

$$|\Psi(t)\rangle = \frac{1}{\sqrt{N!}} \left( \sum_{\mathbf{k}} \tilde{\psi}_{\mathbf{k}}(t) \hat{a}_{\mathbf{k}}^{\dagger} \right)^N |0\rangle, \tag{102}$$

creating $N$ particles in the single-particle state defined by the Fourier transform, $\tilde{\psi}_{\mathbf{k}}$ of the wavefunction $\psi(r)$. The equation of motion corresponding to evolution of the function $\psi(r)$ under purely Hamiltonian evolution is then (in the large $N$ limit) the Gross-Pitaevskii equation:

$$i\frac{d}{dt}\psi(\mathbf{r}, t) = \left( \frac{-\nabla^2}{2m} + NU|\psi(\mathbf{r}, t)|^2 \right) \psi(\mathbf{r}, t). \tag{103}$$

It is worth noting that the same Gross–Pitaevskii equation can also be derived from an alternate picture, based on coherent states. Here one considers

$$|\Psi(t)\rangle = \exp\left( \sqrt{N} \sum_{\mathbf{k}} \tilde{\psi}_{\mathbf{k}}(t) \hat{a}_{\mathbf{k}}^{\dagger} \right) |0\rangle = \exp\left( \sqrt{N} \int d^d\mathbf{r}\, \psi(\mathbf{r}, t) \hat{a}(\mathbf{r}) \right) |0\rangle. \tag{104}$$

As such, this limit also corresponds to the semiclassical approximation for the WIDBG model.

One may extend this ansatz to consider a density matrix $\hat{\rho} = |\Psi\rangle\langle\Psi|$, which allows one to consider evolution under the Lindblad master equation written in Eq. (38). Using the ansatz $\hat{\rho} = |\Psi\rangle\langle\Psi|$ with $|\Psi\rangle$ as above one then finds the complex Gross–Pitaevskii equation:

$$i\frac{d}{dt}\psi(\mathbf{r}, t) = \left( \frac{-\nabla^2}{2m} + NU|\psi(\mathbf{r}, t)|^2 + \frac{i}{2}\left[ \kappa_+ - \kappa_- - N\kappa_-^{(2)}|\psi(\mathbf{r}, t)|^2 \right] \right) \psi(\mathbf{r}, t). \tag{105}$$

Such equations have been widely used to study the physics of nonequilibrium polariton condensates [17, 159]. They also can be derived as the saddle point of a Schwinger–Keldysh path integral, as we will discuss in Sec. 4.6

## 4.4 Large connectivity limit and Dynamical Mean Field Theory

As noted above, mean-field theory is an approximation. Since mean-field approximations involve neglecting correlations between different parts of the system, such approximations should best hold when such correlations are weak. A key property controlling this is the connectivity of sites, $z$, i.e. how many other sites affect a given site. When the connectivity is large, $z \gg 1$, the field seen by a given site comes from summing the contributions of many sites. That is, for large connectivity, a central limit theorem may apply [336], so that the sum over contributions can be replaced by its mean value. In the case of quantum systems one may show that the large connectivity limit for spins and bosons recovers mean-field theory in real space, as discussed above. For other systems—e.g. fermions or disordered systems—the large connectivity limit is more subtle, for example, for fermions the relevant limit is Dynamical Mean-Field Theory (DMFT) [337, 338]. Below we first discuss different ways of reaching the large connectivity limit where mean-field theory becomes valid, and then discuss DMFT for both bosons and fermions.

### 4.4.1 Large connectivity and validity of mean-field theories

As discussed above, when the number of neighbours $z$ of a given lattice site is large, statistical and quantum fluctuations induced by the neighbouring sites become small and can be treated in an approximate way, while the local, on-site physics must always be accounted for exactly. From this perspective, mean-field theories can be formulated by studying the large connectivity limit $z \to \infty$. As we discuss below, this large connectivity limit can be reached in different ways in different models.

**Bosonic and spin lattice models.** Let us consider a model on a regular lattice with connectivity $z$, such as the Bose–Hubbard model or its hard-core limit given by quantum spin chains. In such lattice models, high connectivity generally requires high dimensionality (e.g. for one-dimensional lattices, $z = 2$, while for cubic three-dimensional lattices $z = 8$). As we will see in the next section in more detail, the limit $z \to \infty$ reduces to Gutzwiller mean-field theory, i.e. to a factorised ansatz for the system density matrix [326, 339]. This is ultimately due to the fact that bosonic and spin operators which can take a non-zero expectation value (describing spin or bosonic coherent states that break a symmetry). Corrections to mean-field theory can be included by treating the leading $1/z$ corrections within Dynamical Mean-Field Theory [340, 341], as we will discuss in the next section for driven dissipative bosons.

**Fermionic lattice models.** For fermionic models on lattices with connectivity $z$, the limit $z \to \infty$ does not reduce to a static mean-field theory. This is due to the fact that fermionic parity cannot be broken, i.e. a single fermionic mode cannot condense [337], as we mentioned already in Sec. 4.3.1. Rather in the $z \to \infty$ limit a fermionic lattice model is mapped onto a self-consistent quantum impurity model describing a single site of the lattice embedded in a fermionic bath describing self-consistently the rest of the system. The problem therefore retains local quantum fluctuations even in the $z \to \infty$. In presence of dissipation has been studied for example in Ref. [342].

**Fully connected spin models and one-to-many modes models.** The mean-field approximation can be also formulated as an exact solution of a model on a fully connected lattice of $N$ sites, where each lattice site is coupled to $N - 1$ neighbours, in the thermodynamic limit $N \to \infty$. In these fully connected models the thermodynamic limit therefore coincides with the limit in which mean-field becomes exact, as opposed to lattice models discussed above in which fluctuations beyond mean-field survive in the thermodynamic limit.

Fully connected models can arise either in presence of all to all couplings or, effectively, in cases of one-to-many models such as the Dicke model, in which a single photon mode acts as a central site, coupling to many emitters as satellite sites. In such models, there are cases where a static mean-field theory can be shown to be exact in the limit that the number of emitters goes to infinity [343–346]. This holds as the model has a high connectivity (set by the number of emitters), and so the arguments noted above should apply. One may note however that counterexamples exist, particularly in cases where the the central site has a weak effect back on the other sites. In such cases, the mean-field effect of the satellite sites vanishes, and so the otherwise-subleading fluctuation terms are dominant [347, 348].

**Fully connected models with quenched disorder.** Here again the fully connected nature of the model makes mean-field theory exact in the thermodynamic limit. However the presence of quenched disorder makes mean-field theory non-trivial as for the fermionic case. Specifically, the large connectivity limit does not coincide with

a factorised ansatz, as discussed in Chapter 3, but is again described by a dynamical mean field [338]. In the context of open quantum systems examples have been recently studied of random Lindbladian with all to all couplings, for which DMFT provides the exact solution in the thermodynamic limit [349, 350].

### 4.4.2   DMFT for driven-dissipative bosonic lattices

To make concrete the ideas in this section we discuss DMFT for driven-dissipative bosons on a lattice with coordination number $z$ and follow the discussion presented in Ref. 277. In the following, since we are interested in the large connectivity limit we define the hopping as $t_H \equiv \tilde{t}_H/z$, and $\tilde{t}_H$ will be finite as $z \to \infty$, which is the correct bosonic scaling to have a well defined infinite connectivity limit [339–341]. For concreteness we consider the driven-dissipative Bose–Hubbard model introduced in Eq. (36). The Hamiltonian can be written by separating explicitly terms which are local (on-site) from those that couple neighbouring sites

$$\hat{H} = -\frac{\tilde{t}_H}{z} \sum_{\langle i,j \rangle} \left( \hat{a}_i^\dagger \hat{a}_j + \text{H.c.} \right) + \sum_i \hat{h}_{\text{loc}}[\hat{a}_i^\dagger, \hat{a}_i] \tag{106}$$

where the first term describes the hopping term while the second one accounts for generic local interactions. We will also consider a generic set of local jump operators, $\hat{L}_i$, defined on each site $i$.

The starting point to formulate DMFT for driven-dissipative bosons is the Keldysh action formulation of the Lindblad lattice problem, which following Sec. 3.2 we can write as

$$S = \int dt \sum_{\zeta=f,b} s_\zeta \left[ \sum_i \psi_{i\zeta}^* i \frac{d}{dt} \psi_{i\zeta} - \sum_i h_{\text{loc}}(\psi_{i\zeta}^*, \psi_{i\zeta}) \right] +$$
$$- i \int dt \sum_i \left[ L_{ib} L_{if}^* - \frac{1}{2} \left( L_{if}^* L_{if} + L_{ib}^* L_{ib} \right) \right] +$$
$$+ \frac{\tilde{t}_H}{z} \int dt \sum_{\zeta=f,b} s_\zeta \sum_{\langle i,j \rangle} \left( \psi_{i\zeta}^* \psi_{j\zeta} + \text{H.c.} \right), \tag{107}$$

where $s_{f,b} = \pm 1$ and $L_{i\zeta}, L_{i\zeta}^*$ are the jump operators written in terms of the bosonic fields $\psi_{i\zeta}^*, \psi_{i\zeta}$.

By taking the large connectivity limit one can formally map the effective action of the problem, obtained by integrating out all but one site of the lattice, onto a quantum impurity model. This describes an interacting Markovian single site, characterised by the same local Hamiltonian $H_i$ and local jump operators $L_{i\zeta}$ entering Eq. (91), coupled to a time-dependent field $\Theta_{i,\text{eff}}(t)$ acting as a coherent drive and a non-Markovian quantum bath (Fig. 14, top panel). These take into account the effect of the neighbouring sites and have to be determined self-consistently through the calculation of impurity properties. The DMFT effective Keldysh action reads (assuming for simplicity a normal phase where the bosons remain incoherent):

$$\mathcal{S}_{\text{eff}}[\psi_\zeta^*, \psi_\zeta] = S_{loc}[\psi_\zeta^*, \psi_\zeta] + \int dt \sum_{\zeta=f,b} s_\zeta \, \Phi_{\text{eff}\,\zeta}^*(t) \psi_\zeta(t) +$$
$$- \frac{1}{2} \iint dt dt' \sum_{\zeta,\zeta'=f,b} s_\zeta s_{\zeta'} \, \psi_\zeta^*(t) \Lambda^{\zeta\zeta'}(t,t') \psi_{\zeta'}(t'). \tag{108}$$

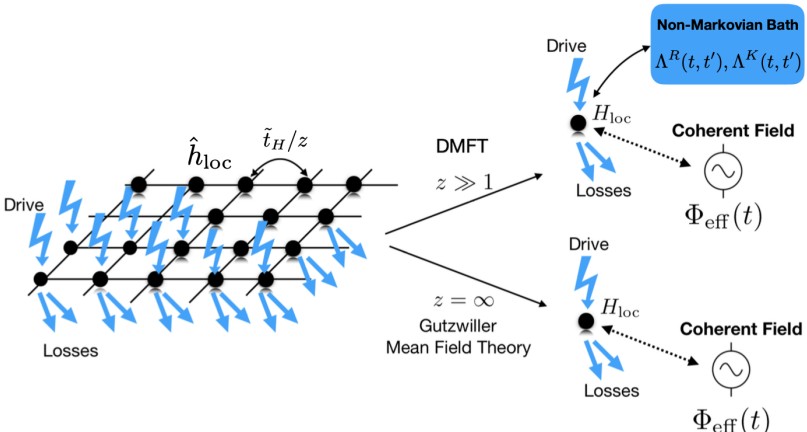

Figure 14: Dynamical Mean-Field Theory (DMFT) for bosonic open quantum systems. In the limit of infinite lattice connectivity, a driven-dissipative bosonic array is mapped via Gutzwiller mean-field theory onto a single-site in a self-consistent field. DMFT includes $1/z$ corrections from the neighbouring sites via the solution of a dissipative quantum impurity problem, where the single site is coupled to a non-Markovian (i.e. frequency dependent) bath, whose spectrum and distribution function, encoded in the retarded and Keldysh components of the bath hybridisation function $\Lambda^R(t,t'), \Lambda^K(t,t')$, are determined self-consistently by the local Green's function. Adapted from Ref. [277]

The first term in Eq. (108), $S_{loc}[\psi^*_\zeta, \psi_\zeta]$ is the local, on-site, contribution of the original lattice problem which can be read off from the first two terms in Eq. (107) and therefore includes interactions, as well as Markovian incoherent drive and dissipation leading to off-diagonal terms in Keldysh space. The second and third terms describe the feedback of the rest of lattice onto the site through its neighbours, in terms of an effective coherent drive $\Phi^*_{\text{eff}\,\zeta}(t)$ and an effective non-Markovian bath with hybridisation function $\Lambda^{\zeta\zeta'}(t,t')$. Both these quantities have to be determined self-consistently, in particular the effective coherent drive reads:

$$\Phi^*_{\text{eff}\,\zeta}(t) = \tilde{t}_H \Phi^*_\zeta(t) + \int dt' \sum_{\zeta'=f,b} s'_\zeta \Phi^*_{\zeta'}(t') \Lambda^{\zeta'\zeta}(t',t), \tag{109}$$

and has two contributions, the first coming from the average of the bosonic field as in Gutzwiller mean field theory $\Phi^*_\zeta = \langle \psi^*_\zeta \rangle_{S_{\text{eff}}}$ and the second one coming from the non-Markovian bath, a non-trivial finite $z$ correction accounting for the feedback of neighbouring sites on the local effective field [341, 351, 352]. The notation $\langle \ldots \rangle_{S_{\text{eff}}}$ appearing above indicates an integral weighted by the effective action, i.e.

$$\langle F(\psi_f, \psi_b) \rangle_{S_{\text{eff}}} = \int \mathcal{D}[\psi_f, \psi_b] F(\psi_f, \psi_b) \exp(iS_{\text{eff}}). \tag{110}$$

The DMFT self-consistency condition depends on the specific choice of the lattice and provides in general a functional relation between the hybridisation function of the non-Markovian bath to the impurity connected Green's function

$$G^{\zeta\zeta'}(t,t') = -i\langle \psi_\zeta(t)\psi^*_{\zeta'}(t') \rangle_{S_{\text{eff}}}. \tag{111}$$

For example, on a Bethe lattice with connectivity $z$ the DMFT self-consistency condition takes the form [352]

$$\Lambda^{\zeta\zeta'}(t,t') = \frac{\tilde{t}_H^2}{z}\, G^{\zeta\zeta'}(t,t';\Lambda). \tag{112}$$

Note here we have indicated that $G$ depends self-consistently on $\Lambda$ via the form of $S_{\text{eff}}$. The DMFT solution of the original Markovian lattice problem thus requires one to solve the Keldysh action (108), computing in particular the impurity Green's function (111) and the average of the bosonic field (109), for given values of the non-Markovian bath $\Lambda$ and effective field $\Phi$, and to iterate (109-112) until self-consistency.

It is instructive at this point to take explicitly the limit of infinite coordination number $z \to \infty$. In this limit, the DMFT effective action (108) becomes completely local in time

$$\mathcal{S}_{\text{eff}}[\psi_\zeta^*, \psi_\zeta] \xrightarrow{z=\infty} S_{loc}[\psi_\zeta^*, \psi_\zeta] + \int dt \sum_\zeta s_\zeta \Phi_{\text{eff}\zeta}^*(t)\psi_\zeta(t), \tag{113}$$

since the contribution from the non-Markovian bath scales as $1/z$ (see Eq. (112)), and as such can be unfolded back into a master equation for a single-site density matrix $\rho$, which satisfies Eq. (91) where $\mathcal{L}$ is the local part of the Lindbladian and the feedback from the neighbouring sites is carried by $\Phi(t) = \text{Tr}[\hat{a}\hat{\rho}(t)]$. This corresponds to a factorised Gutzwiller-like ansatz for the lattice many body density matrix. In other words, as for equilibrium or closed systems [341, 352] also for driven-dissipative lattice systems the infinite connectivity limit of bosons coincides with Gutzwiller mean-field theory. We note that when $\Phi = 0$, this mean-field describes completely uncoupled sites, while DMFT ($z < \infty$) captures the feedback from neighbouring sites through the self-consistent bath $\Lambda$.

To gain insights on the physics behind the DMFT approximation it is useful to compare it with the Gutzwiller mean-field theory discussed in Sec. 4.3.1. This approach corresponds to a factorized density matrix ansatz over different lattice sites and it is therefore equivalent to the solution of a single site problem in a self-consistent field (Fig. 14, bottom panel). By construction such a mean-field theory cannot describe short-range correlations and processes arising from coupling to neighboring sites, either of coherent or dissipative origin. This is particularly true for interacting normal phases such as Mott insulators of bosons or quantum Zeno phases which are featureless within Gutzwiller mean-field theory. The corrections due to DMFT in the large connectivity limit reintroduce part of these short-range fluctuations via the coupling to an effective self-consistent bath as in Eq. (108), and lead to a richer descriptions of isolated or dissipative Mott insulating phases [353, 354]. The price to pay is the solution of a dissipative quantum impurity model, and in particular the calculation of the its Green's functions, which although simplified with respect to the full master equation, still poses a major computational challenges, particularly in presence of non-linear dissipative processes. Different approaches have been developed to tackle these types of dissipative impurities, including exact diagonalisation of the impurity Linbdladian [342, 355, 356] or strong coupling self-consistent hybridisation expansions [277, 357, 358].

We may note that there have been other approaches to bosonic open quantum systems—such as the self-consistent projection operator approach [359]—which have close connections to DMFT.

### 4.4.3 DMFT for driven-dissipative fermions

We conclude our discussion of DMFT by briefly discussing its application to driven-dissipative fermionic problems, and the main differences from the bosonic version.

A first key point is that the large connectivity limit of fermions involves a different rescaling from the bosonic problem discussed above. For fermions one should instead rescale the hopping as $t_H \equiv \tilde{t}_H/\sqrt{z}$, with $\tilde{t}_H$ finite as $z \to \infty$. This different scaling can be traced back to the fact individual fermionic operators cannot have a non-zero expectation value; i.e. $\langle \hat{c} \rangle = 0$ due to conservation of fermionic parity in physical states. As such the trivial mean-field decoupling of the hopping—discussed above for bosons and which naturally led to the $1/z$ scaling—does not apply. As discussed in Ref. [337], this key difference changes the the structure of the large connectivity limit.

To give a concrete example we consider a Fermi–Hubbard-like mode with with next-nearest neighbour hopping and a local on-site interaction, with Hamiltonian:

$$\hat{H} = -\frac{\tilde{t}_H}{\sqrt{z}} \sum_{\langle i,j \rangle} \sum_{\sigma} \left( \hat{c}^\dagger_{i\sigma} \hat{c}_{j\sigma} + \text{H.c.} \right) + \sum_{i\sigma} \hat{h}_{\text{loc}}[\hat{c}^\dagger_{i\sigma}, \hat{c}_{i\sigma}]. \tag{114}$$

In addition we can consider a set of local jump operators $\hat{L}_i$, describing losses, dephasing or other local dissipative processes. Writing the Keldysh action associated to this Lindblad master equation one can follow the steps discussed above and obtain the effective quantum impurity model. The main difference with respect to the bosonic case is that now the coherent field—related to the expectation value of the impurity field—vanishes, and only the effective non-Markovian bath $\Lambda^{\zeta\zeta'}(t, t')$ survives. As previously, this bath must be found self-consistently by computing the Green's function of the fermionic impurity. A broad array of methods to solve these fermionic impurity models in and out of equilibrium have been developed, including for example exact diagonalization [342], diagrammatic Monte Carlo [360–364] and tensor-networks methods [365–367].

## 4.5 Systematic expansions beyond mean-field theory.

There are a number of approaches by which one may systematically improve on the results of mean-field approximations. By "systematic", we refer here to methods in which one can vary some convergence parameter, $P$, so as to interpolate between mean-field theories (as described above) in the limit $P = 1$, and numerically exact approaches in the limit $P \to \infty$ (or above some large but finite $P$ for a finite system). Each of these approaches described below corresponds to a systematic expansion in this sense and they all differ in the way they organise what kinds of correlations are kept in the calculation.

Within such expansions there are two further questions that determine the usefulness of an approach: first, is there uniform convergence on the exact solution? and second, is a sufficient level of accuracy achievable in practicable calculations? Regarding uniformity of convergence, this is defined by asking whether, for some error $\epsilon$, one can find a $P_\epsilon$ such that the error $E_P$ associated with calculations with parameter $P \geq P_\epsilon$ always satisfies $|E_P| < \epsilon$. It is worth noting that even when uniform convergence exists, there can be occasions where $E_P$ increases from one value of $P$ to the next—i.e. $E_P$ need not be monotonic. Uniform convergence though implies that for some sufficiently large $P_\epsilon$, all subsequent errors $E_P$ are sufficiently small. The fact that convergence need not be monotonic does imply the need for care when determining whether numerical results do or do not imply a converged result.

Regarding the practicability of calculations, we note that in some cases (such as matrix-product-state approaches) it has often been possible to reach convergence to reasonable error thresholds $\epsilon$. In such cases, one essentially has an exact result. In other cases, such as cluster mean field theory or cumulant expansions, reaching convergence has not generally been possible: the computational resources required grow too fast with $P$, or the convergence of error with $P$ is slow or non-uniform. Nonetheless, each of these approaches

shares the idea that in principle, tuning a control parameter of the method (as opposed to changing the problem) can lead to exact results. We next discuss in turn various examples of such expansions.

In most of the cases discussed below, one can formulate the approximation both in terms of a recipe of how to perform the calculation, and also in terms of a variational ansatz. The variational ansatz formulation can be useful in understanding exactly what approximation is being considered: what forms of correlations are kept and what are lost. In principle, once one has formulated such an approach, one can then use the Dirac–Frenkel time-dependent variational principle [322, 323] as introduced at the start of Sec. 4.3 to obtain a set of equations of motion for time evolution. In some cases there are however more direct approaches of how to perform time evolution, or to determine the steady state.

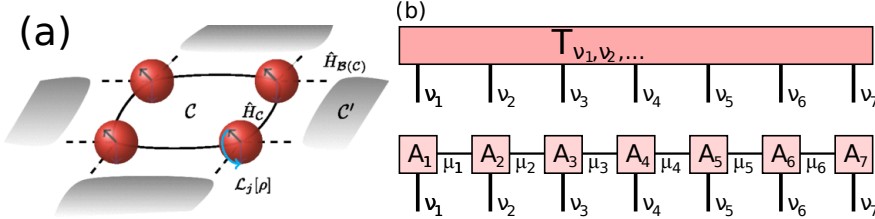

Figure 15: (a) Illustration of cluster mean-field theory. The four sites illustrated are considered as a single cluster, while the effect of other sites (other clusters) are represented by their mean fields. From [276]. (b) Matrix product state decomposition. The seven-index tensor (top) can be decomposed into multiple three-index tensors (bottom) as discussed in the text.

### 4.5.1 Cluster mean field theory.

In this approach, starting from the real-space mean-field theory described above, one systematically expands the size of what is considered as a "site" in the mean-field factorisation. The convergence parameter $P$ here is the size of the cluster. This means that one considers a cluster of several bare sites as one effective site, as shown in Fig. 15(a), and performs a factorisation of correlations between these different sites [276]. In the limit of an infinitely large cluster, this approach corresponds to including all correlations and so becomes exact. However, the cost of calculations increases exponentially with cluster size due to the growth of the size of Hilbert space. Such an approach would work well in cases where correlations are short ranged, so that including correlations between neighbouring sites captures these effects.

### 4.5.2 Matrix-product states.

Matrix-product states (MPS) provide a systematic expansion in terms of the degree of correlation between each site and other sites, using a structure which is efficient if the degree of correlation remains small, as discussed below [368–371].

The essential point of matrix-product-state methods is that a tensor with many indices (i.e. of high rank), $T_{\nu_1, \nu_2, \ldots \nu_{N_s}}$ can always be written as a product of lower rank matrices:

$$T_{\nu_1, \nu_2, \ldots \nu_{N_s}} = \sum_{\mu_1, \ldots, \mu_{N-1}} A_{1, \nu_1}^{\mu_1} A_{2, \nu_2}^{\mu_1, \mu_2} \ldots A_{N-1, \nu_{N-1}}^{\mu_{N-2}, \mu_{N-1}} A_{N, \nu_N}^{\mu_{N-1}}, \tag{115}$$

as illustrated in Fig. 15(b), where we assume the indices $\nu_i$ run over the values $\nu_i = 1 \ldots D$. The key benefit of such a rewriting can be seen by considering what happens if the indices

$\mu_1, \mu_2, \ldots, \mu_{N-1}$ can be restricted to run over a small number of values. If one writes $\mu_i = 1 \ldots M_i$ then $M_i$ is known as the bond dimension of the $i$th bond. One may then note that while specifying the original tensor requires $D^N$ values, specifying the matrix product state form requires $\sum_{i=1}^{N} D M_{i-1} M_i$ values (where we take $M_0, M_{N_s} \equiv 1$). If one can truncate $M_i$ to a small value, this replaces exponential scaling with $N$ with polynomial scaling. The maximum bond dimension $M_i$ is the convergence parameter for this approach. While such an approximation cannot hold in general, it turns out that in problems where correlations are bounded and relatively local, such an approximation can be good. That is, in many problems, the set of physically relevant states are all states that can be represented by low-rank matrix product states.

Such a representation can apply to many objects: wavefunctions, classical probability distributions, or density matrices [274, 372]. In the case of density matrices, these are often referred to as matrix product operator (MPO) methods, since the density matrix is an operator in Hilbert space. The structure of the matrix product state is most natural when considering one-dimensional lattice problems in real space, as then the adjacency of sites relates to the adjacency of matrices in the state. In the case of density matrices, this corresponds to writing the many-site density operator $\rho_{\nu_1, \nu_2, \ldots, \nu_{N_s}}$ in the form of Eq. (115), where the label $\nu_i = 1 \ldots d^2$ runs over the space of local density matrices corresponding to a $d$-dimensional Hilbert space on a given site. In the limit $M = 1$, where all indices $\mu_i$ are restricted to take a single value, this state becomes a product state, corresponding to the mean-field factorisation ansatz.

As an alternative to studying density matrices, one may also proceed by unravelling the Lindblad dynamics and then combine MPS with quantum jumps and study the evolution of the wavefunction along the different quantum trajectories [261]. As noted recently [254, 255, 373], different unravellings can lead to quite different bond dimensions in the resulting matrix product state. In some cases, the bond dimension for the unravelled approach can significantly exceed that for the density matrix approach [374].

If all sites are distinct, then, as noted above, the computational effort grows (albeit polynomially) with the number of sites. However, for translationally invariant problems (with periodic boundary conditions) one can assume all site tensors are equivalent, and directly access the infinite system limit. Higher dimensional extensions of such methods exist, such as projected entangled pair states (PEPS) in two dimensions [368, 370, 371]. These involve decomposing the many-body density matrix into a form that has a five-index tensor on each site, with four internal legs connecting each site to its neighbours. For applications to open systems, see e.g. [321, 375, 376]. The generic term "tensor network states" captures all such methods based on contracting networks of tensors.

In addition to lattice problems, there has also been work on one-to-many models such as the Dicke model [377, 378]. In these works, the star topology of a central cavity and satellite spins was mapped to a one dimensional chain, and swap operations were repeatedly used to move the cavity site to be adjacent to each satellite site in turn. Since the matrix-product-state geometry here does not reflect the physical geometry, there can be significant correlations between neighbouring sites required to capture the correlations between each satellite site and the central cavity, leading to large bond dimensions $M$.

### 4.5.3 Neural network ansatz

The tensor network approach described above makes clear the nature of the density matrix as a map from a pattern of indices to a complex number. The structure of the tensor network reflects how correlated the dependence on the indices should be. Recognising this structure suggests the possibility of alternative approaches, where a neural network is used to represent the map from input indices to the output value.

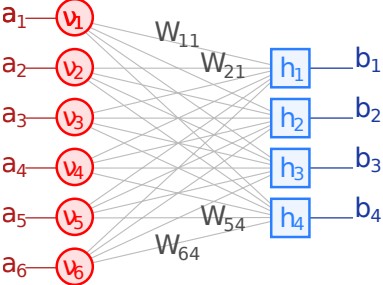

Figure 16: Restricted Boltzmann Machine ansatz for a wavefunction, i.e. Eq. (116). Plotted for $N_s = 6$ input states (red circles) and $M = 4$ hidden nodes (blue squares).

Neural-network approaches were introduced for ground-state wavefunctions in Ref. [379], specifically considering a Restricted Boltzmann Machine (RBM) ansatz. An RBM is a two-layer network, with links only between (not within) the layers. One layer, the "visible" layer consists of the input indices, $\nu_1, \ldots \nu_{N_s}$. For a wavefunction, these input indices correspond to the states of a set of a lattice of sites, i.e. $|\Psi\rangle = \sum_{\nu_1, \nu_2, \ldots, \nu_{N_s}} \Psi_{\nu_1, \nu_2, \ldots \nu_{N_s}} |\nu_1\rangle |\nu_2\rangle \ldots |\nu_{N_s}\rangle$. We assume the input indices should take the form $\nu_i = \pm 1$, corresponding to the states of a spin 1/2 on each site. The other "hidden" layer, $h_1, \ldots h_M$, with $M$ nodes, is traced over. For a wavefunction one thus writes:

$$\Psi_{\nu_1, \nu_2, \ldots \nu_{N_s}} = \sum_{h_1, \ldots, h_M = \pm 1} \exp\left( \sum_{i=1}^{N_s} a_i \nu_i + \sum_{j=1}^{M} b_j h_j + \sum_{i,j=1}^{N_s, M} W_{ij} \nu_i h_j \right)$$

$$= \exp\left( \sum_{i=1}^{N_s} a_i \nu_i \right) \prod_{j=1}^{M} 2 \cosh\left( b_j + \sum_{i=1}^{N_s} W_{ij} \nu_i \right). \tag{116}$$

See Fig. 16 for a sketch of this with $N_s = 6, M = 4$ showing the network before tracing out the hidden layer. The parameters $a_i, b_j, W_{ij}$ are all variational parameters, and various standard methods (e.g. stochastic gradient descent [380] or stochastic reconfiguration [381]) can be used to minimise the expectation of the Hamiltonian for such a state.

This RBM approach was later extended to density matrices. In Refs. [278–280, 382] an extension of the above ansatz was made to represent a purification of the density matrix (thus guaranteeing its positivity). Such an ansatz is written in terms of left indices $\nu_1^{(l)}, \ldots, \nu_{N_s}^{(l)}$ and right indices $\nu_1^{(r)}, \ldots, \nu_{N_s}^{(r)}$ for the density matrix. There are three hidden layers: one connected to just the left indices, one to the right indices, and one connecting the two indices together. To ensure Hermiticity the weight matrices connecting the left and right layers are related by complex conjugation, see Fig. 17. We thus have variational parameters $b_j, W_{ij}$ and their complex conjugates for the left and right layers respectively, with $i = 1 \ldots N_S$ and $j = 1 \ldots M_1$, and parameters $c_k, X_{ik}$ for the connecting layers, where $k = 1 \ldots M_2$. After tracing out the hidden layers this ansatz takes the form:

$$\rho_{(\nu_1^{(l)}, \ldots, \nu_{N_s}^{(l)}),(\nu_1^{(r)}, \ldots, \nu_{N_s}^{(r)})} = \exp\left( \sum_{i=1}^{N_s} a_i \nu_i^{(l)} + a_i^* \nu_i^{(r)} \right) \times$$

$$\prod_{j=1}^{M_1} 4 \cosh\left( b_j + \sum_{i=1}^{N_s} W_{ij} \nu_i^{(l)} \right) \cosh\left( b_j^* + \sum_{i=1}^{N_s} W_{ij}^* \nu_i^{(r)} \right) \times$$

$$\prod_{k=1}^{M_2} 2 \cosh\left( c_k + \sum_{i=1}^{N_s} (X_{ik} \nu_i^{(l)} + X_{ik}^* \nu_i^{(r)}) \right). \tag{117}$$

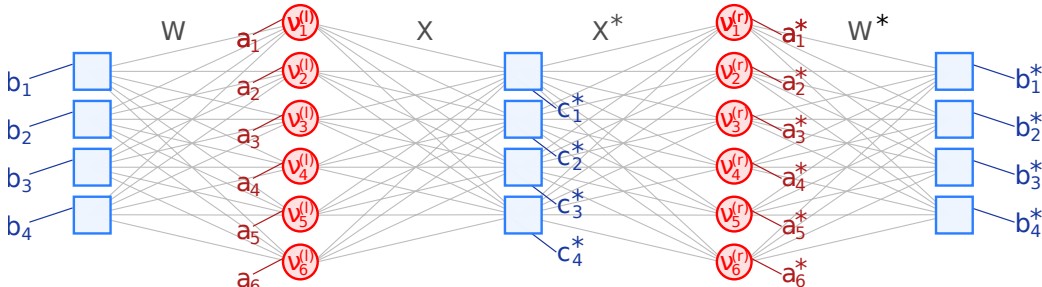

Figure 17: Restricted Boltzmann Machine ansatz for a density matrix, Eq. (117). Plotted for $N_s = 6$ input states and $M_1 = M_2 = 4$, i.e. four nodes for both types of hidden layer. The hidden layers (blue squares) are not explicitly labelled as Eq. (117) presents the result after summing over these layers.

Such a variational ansatz describes a restricted subset of the Hilbert space. For the open system, rather than minimising the expectation of the Hamiltonian, one must seek a solution of the steady-state condition $\hat{\mathcal{L}}[\hat{\rho}] = 0$. This can be turned into an optimisation problem by considering the minimisation of either $|\langle \rho | \hat{\mathcal{L}} | \rho \rangle|$ or of the Hermitian version $\langle \rho | \hat{\mathcal{L}}^\dagger \hat{\mathcal{L}} | \rho \rangle$. One can then apply standard methods to solve this equation.

An alternative way to write an RBM ansatz is to consider vectorisation of the indices on each site individually, and construct an RBM as a function of these indices [383, 384]. Because these vectorised indices now have more than two states, one cannot directly use the ansatz as written previously. However, as discussed in [384], if one labels the four states $\{\nu_i^{(l)}, \nu_i^{(r)}\} = (++, +-, -+, --)$ via a combined index $\nu_i = (+2, +1, -1, -2)$ one can write:

$$\rho_{\nu_1, \ldots, \nu_{N_s}} = \exp\left(\sum_{i=1}^{N_s} \sum_{\xi=1}^{3} a_i^{(\xi)} (\nu_i)^\xi\right) \prod_{j=1}^{M} 2 \cosh\left(b_j + \sum_{i=1}^{N_s} \sum_{\xi=1}^{3} W_{ij}^{(\xi)} (\nu_i)^\xi\right), \tag{118}$$

involving three different powers of the indices. This form of ansatz is better able to efficiently describe products of mixed states than the form in Eq. (118).

As defined so far, this ansatz does not have an obvious control parameter to systematically improve the approximation. The role of the convergence parameter here corresponds to the number of hidden neurons (or more specifically, the ratio of hidden neurons to visible neurons $M/N_s$). Increasing $M/N_s$ increases the expressivity of the ansatz, and convergence to the know exact result on increasing $M/N_s$ has been seen [379]. Another possibility one may also consider is to add further hidden layers to the neural network, i.e. going beyond the restricted Boltzmann machine.

### 4.5.4 Corner-space renormalisation

One further approach to truncating the full Hilbert space was introduced in Ref. [275] as "Corner-space renormalisation". This approach makes use of an iterative construction, first solving a problem on a small lattice, and then using that to determine an effective basis for a larger lattice.

This method can be understood by considering one step of the process. Suppose one has found a good approximation to the steady state of the master equation $\hat{\rho}_A$ on subsystem $A$, and $\hat{\rho}_B$ on subsystem $B$[3]. The aim is then to find an approximation on the system

---

[3]For generality this allows cases where $\hat{\rho}_A \neq \hat{\rho}_B$, however in translationally invariant problems, one expects $\hat{\rho}_A = \hat{\rho}_B$, since these are solutions to the same problem.

$A \cup B$. The density matrices can be written as

$$\hat{\rho}_A = \sum_{\mu} p_{\mu}^{(A)} \left| \phi_{\mu}^{(A)} \right\rangle \left\langle \phi_{\mu}^{(A)} \right|, \qquad \hat{\rho}_B = \sum_{\nu} p_{\nu}^{(B)} \left| \phi_{\nu}^{(B)} \right\rangle \left\langle \phi_{\nu}^{(B)} \right|. \tag{119}$$

The corner-space approach is to construct a basis for the Hilbert space $\mathcal{H}_{A \cup B}$ by considering the $M$ states $\left| \phi_{\mu}^{(A)} \right\rangle \left| \phi_{\nu}^{(B)} \right\rangle$ which have the largest values of $p_{\mu}^{(A)} p_{\nu}^{(B)}$. That is, one determines a set of indices $\mu_m, \nu_m$ for $m = 1 \ldots M$ such that $p_{\mu_m}^{(A)} p_{\nu_m}^{(B)} > p_{\mu_{m+1}}^{(A)} p_{\nu_{m+1}}^{(B)}$, and then uses these to construct the basis:

$$\mathcal{C}(M) = \left\{ \left| \phi_{\mu_1}^{(A)} \right\rangle \left| \phi_{\nu_1}^{(B)} \right\rangle, \left| \phi_{\mu_2}^{(A)} \right\rangle \left| \phi_{\nu_2}^{(B)} \right\rangle, \ldots \left| \phi_{\mu_M}^{(A)} \right\rangle \left| \phi_{\nu_M}^{(B)} \right\rangle \right\}. \tag{120}$$

One then solves the master equation within this reduced "Corner-space" basis $\mathcal{C}(M)$. The parameter $M$ serves as a convergence parameter, increasing $M$ until the solution is converged. This then yields a good approximation for the system $A \cup B$; the whole procedure can then be repeated to consider a larger system. We note that for this approach, one cannot straightforwardly formulate the approximation via a variational ansatz for the density matrix. One may note that the basic idea—of truncating a basis formed by combining the bases of two subparts of the system—is closely related to the original formulation of the density matrix renormalisation group (DMRG) [385, 386]. We note that DMRG approaches can be reformulated in terms of matrix product states [369], no similar reformulation is yet known for corner-space renormalisation.

### 4.5.5   Cumulant expansion.

Cumulant expansions provide an alternate approach to systematically include correlations between different parts of the system, by systematically including correlations between $M$ sites, $\langle \hat{O}_1 \hat{O}_2 \ldots \hat{O}_M \rangle$, where $\hat{O}_i$ is some operator on site $i$. One typically finds that the equations of motion for such correlations will involve terms arising from $M + 1$ or more sites. These can be related to correlations of $M$ or fewer sites by assuming that all cumulants beyond $M$th order vanish, and using this to write the expectations in terms of lower order terms [387]. Thus, $M$ serves here as the convergence parameter. The case where $M = 1$ is equivalent to the mean-field approximation described above. For example, for $M = 2$, setting 3rd order cumulants to zero $\langle \hat{O}_1 \hat{O}_2 \hat{O}_3 \rangle_c = 0$ means writing

$$\langle \hat{O}_1 \hat{O}_2 \hat{O}_3 \rangle = \langle \hat{O}_1 \hat{O}_2 \rangle \langle \hat{O}_3 \rangle + \langle \hat{O}_1 \hat{O}_3 \rangle \langle \hat{O}_2 \rangle + \langle \hat{O}_2 \hat{O}_3 \rangle \langle \hat{O}_1 \rangle - 2 \langle \hat{O}_1 \rangle \langle \hat{O}_2 \rangle \langle \hat{O}_3 \rangle. \tag{121}$$

In the limit of $M \to \infty$ all correlations are kept, so there is no approximation.

For open systems, the ansatz underpinning such an expansion is to assume a generalisation of density matrix factorisation, allowing two point correlations, e.g. for equivalent sites one may consider:

$$\hat{\rho} = \bigotimes_{i} \hat{\rho}_i^{(1)} + \sum_{i \neq j} \rho_{ij}^{(2)} \bigotimes_{k} \hat{\rho}_k^{(1)} + \sum_{i \neq j \neq k \neq l} \rho_{ij}^{(2)} \otimes \rho_{kl}^{(2)} \bigotimes_{m \neq i,j,k,l} \hat{\rho}_m^{(1)} + \ldots \tag{122}$$

where $\rho_{ij}^{(2)}$ are the traceless part of two-site density matrices. The second and third terms written here include correlations within a single pair of sites and within two pairs of sites respectively. The standard second-order cumulant approach (as discussed above) is equivalent to including all possible pairwise correlations. This would mean that e.g. a fourth-order term $\langle \hat{O}_1 \hat{O}_2 \hat{O}_3 \hat{O}_4 \rangle$ would have an expansion that includes terms with two pairwise correlations, consistent with what would be found by setting all 3rd and 4th order cumulants to zero.

This approximation has been considered in the context of problems with one-to-many and many-to-many couplings. For such problems, as noted above, the matrix product state approach is challenging. In such cases, where all satellite sites are equivalent, one may note that the computational effort to model $N$ sites at a given order of cumulant expansion is independent of $N$: the value of $N$ appears only as a parameter in the equations, and does not affect the number of equations to be solved. The cumulant expansion has been used also in the context of driven-dissipative spin chains, see for example Ref. [388] where it was benchmarked with matrix product state calculations. An open source implementation of the cumulant expansion to open quantum systems exists in Ref. [389]; see references therein for other applications.

As discussed so far, cumulant methods have been applied to simplify the master equation for the ensemble-averaged density matrix. Cumulant methods have also been combined with stochastic unravellings, to capture correlations within a single quantum trajectory [390].

### 4.5.6   Multi-configuration time-dependent Hartree (MCTDH).

This approach builds on the particle-based—i.e. Hartree–Fock—version of mean-field theory. As noted above, such a mean-field theory corresponds to considering a single configuration of single-particle states. to be occupied by the particles.

The multi-configurational approach is based on extending to consider a number $M > 1$ of different configurations, and increasing the parameter $M$ until numerical convergence is reached. This was originally introduced for describing wavefunctions of distinguishable particles [391, 392], and later extended to indistinguishable fermions [393–395] and bosons [396, 397]. In the context of bosons the MCTDH-B ansatz takes the form:

$$|\Psi(t)\rangle = \sum_{n_1, n_2, \ldots n_M} C_{\mathbf{n}}(t) \prod_{i=1}^{M} \frac{\left[\hat{b}_i^\dagger(t)\right]^{n_i}}{\sqrt{n_i!}} |0\rangle, \qquad \hat{b}_i^\dagger(t) = \int d^d r \psi_i(\mathbf{r}, t)\hat{a}(\mathbf{r}). \tag{123}$$

Here the vector $\mathbf{n}$ defines the (time-dependent) occupation of the different configurations, while $\psi_i(\mathbf{r}, t)$ describes the spatial profile of these time-dependent configurations. These different configurations are constrained to remain orthogonal, and then time evolution of $\mathbf{n}(t), \psi_i(\mathbf{r}, t)$ is found by using the Dirac–Frenkel time-dependent variational principle [322, 323] In the limit $M = 1$ this recovers the ansatz that leads to the Gross–Pitaevskii equation.

There are various approaches to extending MCTDH to density matrices [398, 399]. An open-source implementation exists [400] and has been used to study the dynamics of cold atoms in optical cavities [123].

## 4.6   Schwinger–Keldysh path integral methods.

In this section we discuss various approaches to solving problems based on the Schwinger–Keldysh formalism, as introduced in Sec. 3.2. For "non-interacting" problems, such as Eq. (53), where the action is quadratic in the fields, one can directly use this equation to evaluate any correlation functions of interest. For example, for the non-interacting single bosonic mode problem discussed above, one can calculate two point functions by inverting the the matrix in Eq. (53) to give

$$\begin{pmatrix} D_K & D_R \\ D_A & 0 \end{pmatrix}, \qquad D_K = -D_R[D^{-1}]_K D_A, \tag{124}$$

where $D_{R,A}$ are the simple inverses of the matrices above. In frequency space this means $D_{R,A}(\nu) = [\nu - \omega \pm \frac{i}{2}(\kappa_- - \kappa_+)]^{-1}$. By Fourier transforming back to the time domain

these Green's functions give (for bosons):

$$D_R(t) = -i\theta(t)\left\langle\left[\hat{a}(t), \hat{a}^\dagger(0)\right]\right\rangle, \qquad D_K(t) = -i\left\langle\left\{\hat{a}(t), \hat{a}^\dagger(0)\right\}\right\rangle, \qquad (125)$$

where $\theta(t)$ is the Heaviside step function. The two forms given here correspond to the commutator and anti-commutator respectively, thus allowing any particular operator ordering to be determined. For Gaussian problems (i.e. where the action has only quadratic terms), all higher-order correlation functions can be factorised into products of two-point correlation functions.

Physically the retarded Green's function corresponds to a response function—hence its causal structure, with non-zero value only for $t > 0$ in the time domain, and all poles in the lower half-plane in the frequency domain (assuming the stable configuration, $\kappa_- > \kappa_+$). In distinction, the Schwinger–Keldysh Green's function describes the noise spectrum. In a thermal equilibrium system, these noise and response functions are related by

$$D^K(\nu) = D^R(\nu)F(\nu) - F(\nu)D^A(\nu), \qquad (126)$$

where $F(\nu)$ is a distribution function, where $F(\nu) = 1 \pm 2n_{B,F}(\nu)$ for bosons or fermions respectively, with $n_{B,F}$ being the Bose or Fermi occupation function. This identity is a form of the fluctuation dissipation theorem [53,54](FDT), relating $D^K(\nu)$; it is also the frequency domain version of the Kubo–Martin–Schwinger (KMS) [401,402] condition on equilibrium Green's functions. In a general nonequilibrium context $F(\nu)$ will not adopt an equilibrium form, however one can often extract a low-frequency effective temperature [224,403] by fitting the behaviour as $\nu \to 0$; e.g. for a thermal bosonic system, $F(\nu) = \coth(\beta\nu/2) \simeq 2k_BT/\nu$, so one may define $k_BT_{\text{eff}} \equiv \lim_{\nu\to 0}\nu F(\nu)/2$.

For interacting problems, reformulating the problem in terms of such a path integral is not itself a method of solution, but provides a framework to then consider a number of theoretical approaches. In the remainder of this section we provide a summary of some of these approaches.

### 4.6.1   Saddle point and mean field theory.

One can often determine the value of a path integral approximately by considering the saddle point configuration—the path $\psi_{cl,q}(t)$ which makes the action stationary—and then expanding around this. One may show [52] that a "classical saddle point" will generally exist—i.e. a solution with $\psi_q(t) = 0$—at least in the absence of quantum sources, i.e. if $J_q = (J_f - J_b)/\sqrt{2} = 0$. Quantum saddle points may also exist when considering boundary conditions with different initial and final states: these can describe tunnelling as well as thermal activation connecting different initial and final states [52].

As with any path integral, this saddle-point approach coincides with a form of mean-field theory. For example, for the action for the weakly interacting dilute Bose gas given in Eq. (55), taking the saddle point with respect to the standard bosonic field recovers the complex Gross–Pitaevskii equation in Eq. (105). Specifically to recover this one should evaluate the variation $\delta S_{WIDBG}/\delta\psi_q^* = 0$ and then take $\psi_q \to 0$, corresponding to the classical saddle point. This yields:

$$i\frac{d}{dt}\psi_{cl} = \left(-\frac{\nabla^2}{2m} + \frac{U}{2}|\psi_{cl}|^2 + \frac{i}{2}\left[\kappa_+ - \kappa_- - \frac{\kappa_-^{(2)}}{2}|\psi_{cl}|^2\right]\right)\psi_{cl}. \qquad (127)$$

To recover Eq. (105) one should make the identification $\psi_{cl} \to \sqrt{2N}\psi$, corresponding to difference in normalisation between the single particle wavefunction $\psi$ used to derive Eq. (105) and the Keldysh-rotated many-particle field $\psi_{cl}$ appearing in the path integral.

A similar approach can be formulated to recover the Gutzwiller mean-field theory of Sec. 4.3.1 as a saddle point. Starting from the Keldysh action of the lattice and performing the Hubbard-Stratonovich decoupling [246] of the hopping in terms of a field $\phi_i, \phi_i^*$ one can obtain the effective action

$$S_{\text{eff}} = \int_{\mathcal{C}} dt \sum_{ij} \phi_i^* t_{ij}^{-1} \phi_j + \sum_i \Gamma[\phi_i^*, \phi_i]. \tag{128}$$

The second term here represents the generating functional of the Green functions of decoupled sites:

$$\Gamma[\phi_i^*, \phi_i] = -i \log \left\langle \mathcal{T}_{\mathcal{C}} \exp\left( i \int_{\mathcal{C}} dt \left( \phi_i^* \hat{a}_i + \hat{a}_i^\dagger \phi_i \right) \right) \right\rangle_0 \tag{129}$$

where $\mathcal{T}_{\mathcal{C}}$ indicates time ordering along the Keldysh contour $\mathcal{C}$ shown in Fig. 12. We note that this average is taken with respect to a density matrix where different sites are decoupled, therefore formally reducing the many-body problem to solving a driven-dissipative single site [354, 404]. By taking the saddle point of the effective action $\delta S_{\text{eff}}/\delta\phi_i = 0$ one obtains $\phi_j = \sum_i t_{ij} \langle \hat{a}_i \rangle$ thus recovering the Gutzwiller mean-field theory discussed in Sec. 4.3.1.

For other problems, it may be necessary to perform various transformations before taking a saddle point, such as using the Hubbard–Stratonovich decoupling [52, 246]. This allows one to introduce a field that decouples a specific interaction term. For example, this can be used to produce a saddle point theory equivalent to the site-based factorisation of the Bose–Hubbard model. For problems involving fermions, one cannot take a saddle point with respect to Grassmann variables. In such cases one must first perform whatever Hubbard–Stratonovich decouplings are required to yield a non-interacting fermion problem, and then integrate out the fermions. This will yield an action in terms of only complex fields, for which one can then take a saddle point. As a general principle, to find observable properties using the saddle point approach requires several steps. One should first add source currents $J_{cl,q}$ as noted above, evaluate the saddle point in the presence of those currents, and then take derivatives of the action with respect to the currents to evaluate physical observables. In many cases however a simpler approach can be realised, of identifying the saddle point value of a field $\psi$ with the expectation of the corresponding operator. For cases where operator correlation functions are required however, the full approach of evaluating the saddle point in the presence of the classical fields.

### 4.6.2 Quadratic fluctuations and ladder series expansions.

After having evaluated a saddle point, one can then systematically expand around this. Considering fluctuations up to quadratic order yields an effective action for which one can still exactly evaluate the path integral. As such, this allows one to find the effect of leading order fluctuation corrections to the mean-field result.

When such an approach is used to calculate correlations or response functions, the result of the saddle point and fluctuation approach is equivalent to re-summation of a diagrammatic expansion, typically corresponding to a ladder series in terms of the bare Green's functions. For example, this approach was discussed in Ref. [159] as a route to calculate the superfluid stiffness of a driven-dissipative WIDBG model.

Similarly, by expanding the effective action $S_{\text{eff}}$ in Eq. (128) around the mean-field configuration one can obtain the fluctuation contribution to the saddle point, controlled in this case by the local Green's functions of the driven-dissipative single-site problem. These objects can be computed exactly for example via a spectral decomposition of the single site Lindbladian [405, 406].

### 4.6.3    Renormalisation group.

The Schwinger–Keldysh action provides a useful starting point for many renormalisation group approaches [31,32,234,407–409]. The renormalisation group is a very broad area with many books and reviews covering it; introductions to this topic include Refs. [246,410,411]. The key idea is that by integrating over "fast" fields (i.e. the components of fields at high frequencies or momentum) one can find the effective action for slow modes, i.e. those at long wavelength and low frequency. After eliminating such modes, one then rescales the remaining momenta and frequencies to restore the original structure of the action, but with modified coefficients for each term. In almost all cases, nonlinearity of the action will mean that the fast modes couple to the slow modes, so that integrating out fast modes modifies the coefficients in the action that determine the slow modes. The goal of such analysis is to determine what form the low energy theory takes. This approach can also be particularly useful to identify phase transitions and study their critical behaviour, by identifying different fixed points of the renormalisation group flow that define either phases or critical boundaries between different phases.

The simplest approach one may consider to explore the likely consequences of renormalisation is to use "power counting" to determine which terms in the Keldysh action are relevant at long wavelengths and low energies [31, 32, 412]. This means looking at each term in the action, and counting powers of momentum, $k$, associated with each term. For example, considering the WIDBG action in Eq. (55), one has the following considerations: Spatial derivatives naturally scale as $k^1$, while the dispersion of the WIDBG means time derivatives scale as $k^2$. As such, the integral over both time and space gives a scaling $k^{-(d+2)}$ in $d$ dimensions. The overall action must be invariant, i.e. $k^0$, as must the constant factors in the quadratic terms $\kappa_\pm$. From these considerations we see the quantum field must scale as $\psi_q \sim k^{(d+2)/2}$. Considering the other quadratic terms in the action we see that $\psi_{cl} \sim k^{(d-2)/2}$. These considerations then lead to a useful observation regarding the quartic terms: the dominant terms there are those involving only one quantum field (and three classical fields). All other quartic terms will be smaller by a factor of $k^2$, and thus smaller in the long wavelength $k \to 0$ limit. This yields the effective long wavelength action:

$$
S_{WIDBG,SC} = \int_{t_0}^{t_1} dt \int d^d\mathbf{r} \Bigg\{ \begin{pmatrix} \psi_{cl}^* & \psi_q^* \end{pmatrix} \begin{pmatrix} 0 & [D_A^{(0)}]^{-1} \\ [D_R^{(0)}]^{-1} & [D^{(0)-1}]_K \end{pmatrix} \begin{pmatrix} \psi_{cl} \\ \psi_q \end{pmatrix}
$$
$$
- \left[ \left( \frac{U}{2} + i\frac{\kappa_-^{(2)}}{4} \right) \psi_{cl}^{*2}\psi_{cl}\psi_q + \text{c.c.} \right] \Bigg\}. \quad (130)
$$

This simplified action can also be considered as a semiclassical limit.

The power counting approach will allow one to determine which terms can survive in a low energy theory, but does not determine how the renormalised coefficients relate to the original coefficients.

To perform more complete calculations, one must use an explicit scheme to integrate out degrees of freedom and see how coefficients change. One fruitful approach there is the functional renormalisation group [411], which has been applied to Keldysh actions [31,32, 407]. This works by introducing a "regulator" into the effective action which suppresses long wavelength modes with $k < K$; one then studies an effective action as a function of $K$, and derives an equation for how the action evolves as $K$ changes. This ultimately yields equations for how the form of effective action changes with the momentum scale at which one works.

### 4.6.4 Stochastic interpretations.

As discussed in section 8, there are various methods by which one can transform a master equation for the ensemble averaged density matrix into equations for a set of individual stochastic trajectories. The Schwinger–Keldysh path integral—and in particular the semiclassical limit noted above—can be used as one such route to derive a stochastic equation of motion that captures the ensemble behaviour.

Following the notation of Ref. [412], one can consider deriving the stochastic equation by starting from a form of action that frequently arises in the semiclassical limit,

$$S = \int dt \int d^d r \left[ \psi_q^* \left( i \frac{d}{dt} \psi_{cl} + \frac{\delta f[\psi_{cl}]}{\delta \psi_{cl}} \right) + \text{c.c.} + i \Gamma |\psi_q|^2 \right] \tag{131}$$

This "semiclassical" form allows a general dependence on on the field $\psi_{cl}$, but restricts how $\psi_q$ appears. The $\psi_q$ dependence that occurs in this expression is in line with the terms that are relevant at long lengths scales according to the power-counting argument discussed above.

Starting from this action, the key idea is to perform a Hubbard–Stratonovich decoupling of the term $i \Gamma |\psi_q|^2$, by using the identity:

$$\exp \left[ -\int dt \int d^d r \Gamma |\psi_q|^2 \right] = \int \mathcal{D}(\xi) \exp \left[ -\int dt \int d^d r \left( \frac{|\xi|^2}{\Gamma} + i \xi^* \psi_q + i \xi \psi_q^* \right) \right]. \tag{132}$$

This puts the action into a form that is linear in $\psi_q, \psi_q^*$. As such the path integral over $\psi_q, \psi_q^*$ introduces a Dirac delta function requiring that $\psi_{cl}$ obey the Langevin equation:

$$i \frac{d}{dt} \psi_{cl} = -\frac{\delta f[\psi_{cl}]}{\delta \psi_{cl}} + \xi, \tag{133}$$

The dependence of the action of $\xi$ implies it has a white noise spectrum:

$$\langle \xi(\mathbf{r}, t) \rangle = 0, \qquad \langle \xi(\mathbf{r}, t) \xi^*(\mathbf{r}', t') \rangle = \Gamma \delta(t - t') \delta(\mathbf{r} - \mathbf{r}'). \tag{134}$$

The procedure here, extracting a Langevin equation from a Keldysh action is the inverse of the Martin–Sigia–Rose approach [52], which derives a Keldysh action starting from a Langevin equation. The interpretation of the above equation is that for such semiclassical actions, the resulting correlation functions can be equivalently found by simulating a set of trajectories of $\psi_{cl}$ from the Langevin equation in Eq. 133, and then averaging over noise.

As discussed in Refs. [407, 412], a further simplification occurs if the functional $f$ appearing above is purely imaginary. In such a case, one can give an explicit expression for the probability distribution of $\psi_{cl}$ that results from averaging over such trajectories:

$$P[\psi_{cl}] = \exp \left( \frac{i f[\psi_{cl}]}{\Gamma/2} \right). \tag{135}$$

(Note that the exponent here is real, since $f$ is imaginary.) In this case, despite the derivation from a nonequilibrium formalism, the behaviour can be understood as corresponding to an equilibrium Gibbs ensemble, with an energy controlled by the functional $-if[\psi]$, and a temperature set by the noise strength $\Gamma/2$.

Emergent equilibrium behaviour is in fact possible more widely, if $\text{Re}[f[\psi]] = r \, \text{Im}[f[\psi]]$ for some constant $r$; i.e. the real and imaginary parts are proportional to each other [31, 407, 413, 414]. The purely imaginary functional discussed above corresponds to the special case $r = 0$. This generalised equilibrium condition can be related [407, 414] to the KMS condition for equilibrium as discussed above.

In general, the functional $f[\psi]$ will not obey the conditions noted above, and so behaviour will not be equivalent to an equilibrium ensemble. However, in many cases (including the WIDBG model discussed above), a renormalisation group analysis of the action shows [234, 407, 408] that at long wavelengths, one reaches an effective action for which $\text{Re}[f[\psi]] = r \, \text{Im}[f[\psi]]$; i.e. this symmetry is an emergent property of the long wavelength theory.

One may observe that even without a full renormalisation analysis, the semiclassical limit of the WIDBG action, Eq. (130), takes the expected form of Eq. (131) with the noise strength $\Gamma = \kappa_- + \kappa_+$. As such, this yields a stochastic complex Gross–Pitaevskii equation, with the form:

$$
i\frac{d}{dt}\psi_{cl} = \left( -\frac{\nabla^2}{2m} + \frac{U}{2}|\psi_{cl}|^2 + \frac{i}{2}\left[ \kappa_+ - \kappa_- - \frac{\kappa_-^{(2)}}{2}|\psi_{cl}|^2 \right] \right)\psi_{cl} + \xi, \qquad (136)
$$

with the appropriate white-noise spectrum for $\xi$, with strength $\kappa_- + \kappa_+$

## 5  Dissipative State Engineering

After having discussed the general theoretical and experimental frameworks for the study of open quantum many-body systems, we now focus more specifically on one of its applications: the idea of *environment* or *reservoir engineering*. As already seen in several examples in these lecture notes, the steady state of a many-body system is generically mixed. Nevertheless, as we will extensively discuss in this section, under certain conditions, it is possible to use dissipation in order to engineer non-trivial, pure, many-body states in the stationary regime. This however is possible only through a careful tailoring of the coupling of the system to the environment. Originally proposed and successfully applied in the field of quantum optics and quantum information (see [415] for a recent review), where the focus was on single- or few-particle properties, be they atoms, molecules or photons, this idea is by now very well explored. The conceptual novelty that we want to discuss in this section is the recent extension to the realm of many-body quantum systems, which poses a set of novel problems, such as that of *locality* or of quantifying the *typical timescale* necessary to relax to the stationary state in the thermodynamic limit. This extension has been pioneered by two research articles appeared in the late 2000's [8, 9], which outlined the potential for studies of dissipation beyond single-particle effects, proposing the development of universal quantum computers in a truly multi-qubit setting solely based on dissipation.

After a brief review on optical pumping in Sec. 5.1, we present in Section 5.2 the theoretical aspects of dark-state engineering [8, 9, 416] and focus then on the dissipative preparation of Bose-Einstein condensates [416] (Section 5.3.1) and of topological states hosting Majorana zero modes [417–420] (Section 5.3.2). We then discuss two examples that are of particular experimental relevance: the creation of entangled states using two-body losses [421–423], in Section 5.3.3, and the preparation of correlated many-body states of photons, such as Mott insulators [424], in Section 5.3.4. We continue with a discussion in Sec. 5.4 on the use of dark-state engineering for quantum-information purposes [9] and in Sec. 5.5 on the use of measurements as a manipulation technique, the so-called quantum many-body steering [425]. Some space will finally be devoted in Sec. 5.6 to the discussion of their stability against perturbations [426, 427].

In this section we focus on only some aspects of dissipative state engineering. There are however many other topics that could well fit into it and that will not be discussed for the sake of brevity. For instance, an enormous literature has focused on the problem of

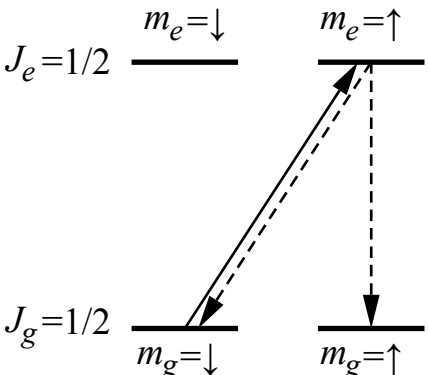

Figure 18: Sketch of the simple optical-pumping process described in the text. A ground state ($J_g = 1/2$) is connected to a short-lived excited state ($J_e = 1/2$) with a coherent field with right-handed circular polarisation (solid arrow). The excited state $m_e = \uparrow$ can decay to both ground states $m_g = \uparrow, \downarrow$ (dashed arrows). The state $m_g = \uparrow$ is a dark state of this dynamics.

entanglement generation assisted by dissipation, even with many-body quantum systems; the topic has been recently reviewed in Ref. [415] and will not be discussed here. In the context of many-body fermionic fluids, the possibility of creating $\eta$-pairing superconductivity has recently attracted a lot of attention [416, 428, 429], whereas other works have focused on the dissipative realisation of matrix-product-states [368, 416, 430, 431]. We refer the interested reader to the primary literature for those topics.

## 5.1  From optical pumping to dark states

In the context of quantum physics, the study of states that cannot absorb nor emit photons—so-called *dark states*—dates back to the early studies on three-level systems, where laser light was used to couple short- and long-lived atomic energy levels (Lambda systems) [180, 432, 433]. The existence of such states plays a crucial role in the appearance of dark absorption lines, and has eventually led to the study of phenomena such as *electromagnetic induced transparency*, *coherent population trapping*, and *slow light* [180, 434]. Before discussing dark states in the many-body context, it is thus useful to start with a discussion of a simple single-atom manipulation via techniques that leverage both closed- and open-system dynamics. Two paradigmatic examples can be mentioned: *sideband cooling* [435] and *optical pumping* [436]. Since this latter phenomenon is particularly simple, we briefly detail it here below.

The goal of optical pumping, in its simplest form, is to prepare a cloud of atoms in some chosen internal hyperfine state. It relies on the angular-momentum conservation of the light-matter interaction, but also crucially on the possibility of manipulating the atom using spontaneous emission, which is an open-quantum-system effect. Optical pumping is best explained considering the simple situation of a transition connecting a ground state with angular momentum $J_g = 1/2$ to an excited state with angular momentum $J_e = 1/2$ (see the sketch in Fig. 18). If such a transition is driven with coherent light that is right-hand circularly polarised, the only coupling that is possible is that between the states with $m_g = \downarrow$ and $m_e = \uparrow$ (here $m_g$ and $m_e$ are the magnetic quantum numbers of the ground and excited states, respectively; we will use $\downarrow, \uparrow$ to label the corresponding quantum numbers $\pm 1/2$). The excited state is short lived and decays via spontaneous emission to both ground-state levels, $m_g = \downarrow, \uparrow$. Once the laser light illuminates the atomic vapour, a part of the population decays to the $m_g = \uparrow$ and is trapped: only undesired effects (not shown

in the figure) can modify the state of those atoms. The atoms decaying to $m_g = \downarrow$, instead, are re-pumped to the excited state via the coherent field and the entire process is restarted. This eventually brings all the atomic population to the state $m_g = \uparrow$, which is a *dark state* of this dynamics. Optical pumping of this kind is routinely performed in many cold-atom experiments.

Mathematically, we can describe the dynamics with the following single-atom Lindblad master equation:

$$\frac{d\rho}{dt} = -i[\hat{H}, \rho] + \sum_{\sigma=\downarrow,\uparrow} \sum_{\sigma'=\downarrow,\uparrow} \left( \hat{L}_{\sigma,\sigma'} \rho \hat{L}_{\sigma,\sigma'}^\dagger - \frac{1}{2} \left\{ \hat{L}_{\sigma,\sigma'}^\dagger \hat{L}_{\sigma,\sigma'}, \rho \right\} \right) \tag{137}$$

with Hamiltonian

$$\hat{H} = \sum_{\nu=e,g} \sum_{\sigma=\downarrow,\uparrow} E_\nu \ket{m_\nu = \sigma}\bra{m_\nu = \sigma} + \left( \Omega e^{-i(E_e - E_g)t} \ket{m_e = \uparrow}\bra{m_g = \downarrow} + \text{H.c.} \right) \tag{138}$$

and jump operators

$$L_{\sigma,\sigma'} = \sqrt{\gamma_{\sigma,\sigma'}} \ket{m_g = \sigma}\bra{m_e = \sigma'}. \tag{139}$$

The Hamiltonian describes the four energy levels and the coherent field that has a right-handed circular polarisation and couples selectively between only two of them. Four decay channels are however possible, from both excited states $\sigma'$ to both ground states $\sigma$, and are described by four different jump operators. It is possible to show that the pure state $\ket{m_g = \uparrow}$ is a stationary state of the dynamics, as its time-derivative equals zero. This follows from the fact that the density matrix $\ket{m_g = \uparrow}\bra{m_g = \uparrow}$ commutes with the Hamiltonian and is annihilated by all four jump operators. In the next section we will see how this kind of reasoning can help us in the study of many-body problems.

## 5.2 Many-body dark states

The idea of generalising dark-state physics to the dissipative many-body scenario is more recent [8, 9, 437, 438] and while extensive theoretical work has been done, it has not yet been fully investigated experimentally. The starting point is a quantum many-body system coupled to an environment, whose dynamics is described by a Lindblad master equation, as introduced above. In the following we first discuss the nature of the steady (i.e. dark) states, and then discuss the asymptotic approach to such states.

### 5.2.1 Nature of the steady states

In the discussion presented in these Lecture Notes, the physics of dark states appears when we have one pure state $\ket{\Psi}$ such that (i) it is an eigenstate of the Hamiltonian, so that $\left[ \hat{H}, \ket{\Psi}\bra{\Psi} \right] = 0$, and (ii) it is in the kernel of all jump operators, $\hat{L}_\mu \ket{\Psi} = 0$. If conditions (i) and (ii) are satisfied, it is not difficult to show that $\ket{\Psi}\bra{\Psi}$ is a stationary state of the dynamics:

$$\frac{d}{dt} \ket{\Psi}\bra{\Psi} = -i \left[ \hat{H}, \ket{\Psi}\bra{\Psi} \right] + \sum_\mu \hat{L}_\mu \ket{\Psi}\bra{\Psi} \hat{L}_\mu^\dagger - \frac{1}{2} \left\{ \hat{L}_\mu^\dagger \hat{L}_\mu, \ket{\Psi}\bra{\Psi} \right\} = 0. \tag{140}$$

In such cases, $\ket{\Psi}$ is commonly dubbed "dark". This is exactly what happens in the master equation that we studied above in Sec. 5.1 on optical pumping.

This construction can be easily generalised to dark subspaces in the case of many linearly independent dark states but will not be further considered here.

The most general conditions for the appearance of dark states are broader than those defined above. It has been formulated in Ref. [416]. A pure dark state $|\Psi\rangle$ exists if and only if: (i) it is an eigenstate of the effective non-Hermitian Hamiltonian associated to the problem, $\hat{H}_{\mathrm{nH}} = \hat{H} - \frac{i}{2} \sum_\mu \hat{L}_\mu^\dagger \hat{L}_\mu$, with eigenvalue $\Lambda \in \mathbb{C}$, (ii) it is an eigenstate of all quantum-jump operators, $\hat{L}_\mu$ with eigenvalue $\lambda_\mu$, and (iii) the following relation holds: $\mathrm{Im}\,\Lambda = -\frac{1}{2} \sum_\mu |\lambda_\mu|^2$. We refer to the original article, Ref. [416], for the proof of this. The more-restricted statement presented above corresponds to the case $\lambda_\mu = 0$ and $\Lambda \in \mathbb{R}$.

The class of restricted dark states, i.e. $\lambda_\mu = 0, \Lambda \in \mathbb{R}$ is clearly less general, yet, it is of interest as it leads to various non-trivial many-body dark states. Let us first observe that any such dark state satisfies the identities $\langle\Psi|\hat{L}_\mu^\dagger \hat{L}_\mu|\Psi\rangle = 0$ for all $\mu$; thus, since every operator $\hat{L}_\mu^\dagger \hat{L}_\mu$ is positive semi-definite, $|\Psi\rangle$ must minimise $\langle\Psi|\hat{L}_\mu^\dagger \hat{L}_\mu|\Psi\rangle$. Consequently, $|\Psi\rangle$ is a zero-energy ground state of the *parent Hamiltonian*:

$$\hat{H}_p = \sum_\mu \hat{L}_\mu^\dagger \hat{L}_\mu. \tag{141}$$

This result is rather important because it gives a positive characterisation of the pure dark states corresponding to this situation. Indeed, Hamiltonians of the form (141) featuring a zero-energy ground state are well-known in the condensed-matter literature under the name of *frustration-free Hamiltonians*; they are the sum of non-negative operators that in general do not commute but that admit a common ground space. Notable examples include the toric-code Hamiltonian that plays a major role in the context of topological order [439] and the Rokhsar-Kivelson Hamiltonian for valence-bond states [440].

The concept of dark state has recently experienced a vigorous revival within the framework of *dissipative quantum state preparation*. If $|\Psi\rangle$ is the unique dark state of the dissipative dynamics and for some reasons one is interested in preparing it (for instance, it could be an entangled state with some relevance for quantum metrology), then the experimentalist may prepare the state via an accurate engineering of an appropriate environment, that will then create the state through dissipation [8, 9, 438]. This is what experimentalists working in quantum optics have long done at the level of single-particle systems with techniques like that of optical pumping, and that is currently being extended to the many-body realm. Clearly, interesting dark states do not appear in all quantum setups, but the set of systems where this takes place is wider than one could imagine at first sight. Before discussing explicit examples, however, it is important to stress a fundamental requirement that should be taken into account when dealing with many-body quantum systems: *locality*. A well-defined and physical many-body Lindbladian should have a local Hamiltonian and local jump operators, namely they should be the sum of operators with a support whose extension does not scale with the system size in the thermodynamic limit. Relaxing this condition would give rise to an uncountable number of forms of unphysical dissipative dynamics that could be used to engineer literally any quantum state. Even with this constraint of locality, the list of theoretical studies where dissipative quantum state preparation with a physical Lindbladian has been discussed is now rather long.

### 5.2.2   Asymptotic decay toward the dark states

Whereas statements on the stationary states can be obtained with relative ease, the characterisation of the dynamics, and in particular of the long-time dynamics, is more difficult. The scaling with the system size of the asymptotic decay rate, which is the gap of the Lindbladian operator, has attracted a lot of attention because it determines the time scale that is necessary for the system to relax to the stationary state that is being engineered. In this section we will use the length of the system, $L$ to denote system size—for lattice

based systems, one may note that $N_s \propto L^d$. When the asymptotic decay rate does not scale to zero with increasing system size, as in the example discussed in Sec. 5.3.2, the dissipative engineering is particularly efficient. In other situations, the asymptotic decay rate decreases polynomially with $L$, the size of the system; this is a less advantageous situations which can, nonetheless, be of interest.

There is no general recipe for the analytical calculation of the asymptotic decay rate in these setups. However, in several of the examples that will be presented in the remainder of the section we will detail specific techniques that have been designed for particular problems. Ideally, it would be desirable to link the asymptotic decay rate of the dissipative dynamics to the spectral properties of the parent Hamiltonian $\hat{H}_p$, which is a Hermitian operator, and for which, typically, more results are known. (Note we are only proposing here to use the parent Hamiltonian only as a simplified mathematical tool for establishing theorems, not as a fundamental description of the open system dynamics.) As an example of the possibilities of this approach, we remind here a result proven in Ref. [420] in the context of Lindbladians with no Hamiltonian. According to this work, when the parent Hamiltonian, Eq. (141), is gapless then the Lindbladian is also gapless. In particular, if $\hat{H}_p$ has a gap closing as $L^{-\beta}$, where $L$ is the size of the system, then the asymptotic decay rate of the Lindbladian closes at least as $L^{-\beta}$. The result is easily obtained by considering the first excited state of the parent Hamiltonian, dubbed $|\Psi_{\text{ex}}\rangle$, with energy $E_{\text{ex}}$. The linearly independent operators $|\Psi_{\text{ex}}\rangle\langle\Psi|$ and $|\Psi\rangle\langle\Psi_{\text{ex}}|$ are eigenoperators of the Lindbladian with eigenvalue $-E_{\text{ex}}/2$.

To conclude, we briefly comment on the accuracy of the many-body state by dissipative state engineering; as expected, waiting for longer times is beneficial to the quantum state preparation. If we define $p_0(t) = \text{Tr}\big[\hat{P}_0\,\hat{\rho}(t)\big]$, where $\hat{P}_0 = |\Psi\rangle\langle\Psi|$ is the projector onto the ground space of the parent Hamiltonian $\hat{H}_p$, then the following monotonicity property holds: $\frac{d}{dt}p_0(t) \geq 0$. Indeed, with an explicit calculation, $\frac{d}{dt}p_0(t) = \sum_\mu \text{Tr}[L_\mu^\dagger \hat{P}_0 L_\mu \hat{\rho}(t)]$, which is the sum of the expectation values of non-negative operators, and thus of non-negative numbers.

## 5.3  Examples

Having exposed some general considerations on dissipative state preparations, we now move to the presentation of several concrete examples. We consider dissipative quantum systems that leverage the notion of dark states, based on different experimental setups such as trapped ions [63, 64, 441], Rydberg atoms [130], ultracold atoms [8, 442]. We note that this is not an exhaustive summary of the important literature that has appeared in recent years.

### 5.3.1  BEC condensate

As a first example, we present here the dissipative preparation of Bose-Einstein condensates with long-range order [8, 416]; the fermionic case is discussed in Ref. [442]. Let us consider a lattice populated by bosons, described by the operators $\hat{b}_i^{(\dagger)}$ that satisfy canonical commutation relations, where $i$ labels the lattice site. For simplicity, we set the Hamiltonian to zero and consider only the dissipative dynamics induced by the following jump operators:

$$\hat{L}_{i,j} = \sqrt{\gamma}\left(\hat{b}_i^\dagger + \hat{b}_j^\dagger\right)\left(\hat{b}_i - \hat{b}_j\right), \tag{142}$$

acting on a pair of adjacent sites, where $\gamma$ is a decoherence rate. Intuitively, this dissipative operator eliminates from the lattice a boson that is antisymmetric component of the wavefunction on two neighbouring lattice sites and re-introduces it in a symmetric mode. Let

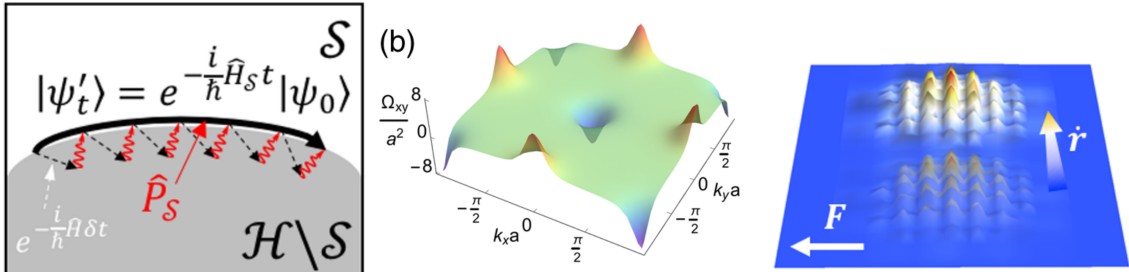

Figure 19: An example of topological physics induced by dissipation. (Left) When a Hilbert space $\mathcal{H}$ is dissipatively constrained to a subspace $\mathcal{S}$, the dynamics is governed by the projected Hamiltonian $\hat{H}_S = \hat{P}_{\mathcal{S}} \hat{H} \hat{P}_{\mathcal{S}}$, where $\hat{H}$ and $\hat{P}_{\mathcal{S}}$ are the bare Hamiltonian and the projection operator onto $\mathcal{S}$, respectively. (Middle) In the situation discussed here, the authors propose a dissipative mechanism that constraints the dynamics in the lower band of a lattice, that has a non-trivial Berry curvature $\Omega_{xy}$, plotted over the first Brillouin zone. (Right) As a final experimentally observable phenomenology associated to topological physics, it is shown that a wave packet undergoes transverse motion in response to a potential gradient $\mathbf{F}$. Adapted from Ref. [443] [Copyright (2017) by the American Physical Society].

us now consider the $N$-particle condensate state $|\text{BEC}_N\rangle = \frac{1}{\sqrt{N!}} \left( \hat{b}_{\mathbf{k}=0}^{\dagger} \right)^N |v\rangle$ where $\hat{b}_{\mathbf{k}=0} = \frac{1}{\sqrt{N_s}} \sum_j \hat{b}_j$ is the $\mathbf{k}=0$ bosonic mode and $|v\rangle$ the vacuum. The state $|\text{BEC}_N\rangle$ is the $N$-particle ground state of a problem of free bosons, and represents an exact idealised Bose-Einstein condensate where all particles have condensed in one bosonic mode. In order to show that $|\text{BEC}_N\rangle$ is a dark state of the dissipative dynamics induced by the jump operators in Eq. (142), it is sufficient to show that $\left( \hat{b}_i - \hat{b}_j \right) |\text{BEC}_N\rangle = 0$ for any pair of neighbouring sites. The thesis follows from the fact that the commutator $\left[ \hat{b}_i, \hat{b}_{\mathbf{k}=0}^{\dagger} \right] = \frac{1}{\sqrt{N_s}}$ does not depend on $i$ and can be proven by induction over $N$.

### 5.3.2  Topological states and Majorana dark states

Here we want to present the possibility of creating topological states of matter using dissipation, for which there are many examples. In Ref. [443], from which Fig. 19 is extracted, strong dissipation is exploited to confine the dynamics to a lattice band with non-trivial Berry curvature [443]. Related works have extensively discussed the possibility of using non-Hermitian dynamics, losses or other decoherence sources to engineer p-wave superfluids [444], integer or fractional quantum Hall states [445,446], as well as integer [447] or fractional Chern insulators [448–451]. The use of Rydberg atoms has been instead proposed for the dissipative engineering of Kitaev's toric code [130]; related to this subject is also the study of open quantum systems where strong dissipation is employed to create lattice gauge theories with cold atoms [452]. Recently, there has been very intriguing progress in realising Laughlin states composed of few photons, which is a setup that is intrinsically dissipative [142,453].

In the following we will discuss in some details the case of topological superconductors that host Majorana zero-energy modes [454] (although, as we will see, in this out-of-equilibrium scenario, it will be more appropriate to speak of *Majorana dark modes* or, equivalently, of *Majorana zero-damping modes*). We discuss in particular the results of Ref. [417] and use them to illustrate how the notion of dark state can be used to guide the

dissipative state preparation of Majorana zero modes.

The simplest model displaying zero-energy unpaired Majorana modes is the Kitaev model for a one-dimensional $p$-wave superconductor. We focus on one specific point of the phase diagram where its solution is particularly straightforward [454]. We introduce the fermionic operators $\hat{c}_j^{(\dagger)}$ that satisfy canonical anticommutation relations and describe the annihilation and creation of a spinless fermion at site $j$. The Hamiltonian reads

$$\hat{H}_{\mathrm{K}} = -t_H \sum_j \left[ \hat{c}_j^\dagger \hat{c}_{j+1} + \hat{c}_j \hat{c}_{j+1} + \text{H.c.} \right], \quad t_H > 0. \tag{143}$$

The model can be solved with the Bogoliubov–de Gennes transformation, and, when considered on a chain of length $L$ with open boundaries, it takes the form:

$$\hat{H}_{\mathrm{K}} = E_0 + \frac{t_H}{2} \sum_{j=1}^{L-1} \hat{\ell}_j^\dagger \hat{\ell}_j, \qquad \text{with} \qquad \hat{\ell}_j = \hat{C}_j^\dagger + \hat{A}_j, \quad \hat{C}_j^\dagger = \hat{c}_j^\dagger + \hat{c}_{j+1}^\dagger, \quad \hat{A}_j = \hat{c}_j - \hat{c}_{j+1}. \tag{144}$$

The modes $\hat{\ell}_j$ are fermionic operators that obey the canonical anticommutation relations, $\{\hat{\ell}_i, \hat{\ell}_j\} = 0$ and $\{\hat{\ell}_i, \hat{\ell}_j^\dagger\} = \delta_{ij}$, and that can be used to obtain the entire spectrum of the model. The ground state has energy $E_0$ and is two-fold degenerate: there are two linearly independent states $|\psi_e\rangle$ and $|\psi_o\rangle$ which satisfy $\hat{\ell}_j |\psi_\sigma\rangle = 0$ for all $j$. The quantum number distinguishing the two states is the parity of the number of fermions, $\hat{P} = (-1)^{\sum \hat{c}_j^\dagger \hat{c}_j}$, which is a symmetry of the model (indeed, the subscripts $e$ and $o$ stand for *even* and *odd*). Both states $|\psi_\sigma\rangle$ are $p$-wave superconductors, as can be explicitly proven by computing the expectation value of the corresponding order parameter: $\lim_{L\to\infty} \langle \psi_\sigma | \hat{c}_j \hat{c}_{j+1} | \psi_\sigma \rangle = 1/4$. The rest of the spectrum is obtained by subsequent application of the $\hat{\ell}_j^\dagger$ operator, knowing that in this precise and simple problem each excitation has an energy cost of $t_H/2$.

It is not difficult to design a master equation which features $|\psi_e\rangle$ and $|\psi_o\rangle$ as steady states and has local jump operators. This is possible because, as we have just seen, the two states lie in the kernel of the entire set of $\hat{\ell}_j$ operators. Introducing a Markovian dynamics with jump operators $\hat{L}_j = \sqrt{\gamma} \hat{\ell}_j$ ensures that the states $|\psi_\sigma\rangle$ are stationary and that in the long-time limit the system is brought into a subspace described in terms of $p$-wave superconducting states. Additionally, the parent Hamiltonian of this Markov process coincides with $\hat{H}_{\mathrm{K}}$ in Eq. (144) up to an additive constant. The obtained dynamics satisfies an important property, namely *locality*: the jump operators $\hat{\ell}_j$ only act on two neighbouring fermionic modes.

In such a model, there are two unpaired Majorana modes that are not affected by the dissipative dynamics:

$$\hat{\gamma}_L = \hat{c}_1 + \hat{c}_1^\dagger, \qquad \hat{\gamma}_R = i(\hat{c}_L - \hat{c}_L^\dagger). \tag{145}$$

In the Hamiltonian context, these boundary operators are typically called Majorana zero modes, because they commute with the Hamiltonian and force the entire spectrum of the Hamiltonian to be two-fold degenerate. This can be shown by creating a complex fermion $\hat{d} = \hat{\gamma}_L + i\hat{\gamma}_R$ that squares to zero ($\hat{d}^2 = 0$), satisfies canonical anticommutation relations ($\{\hat{d}, \hat{d}^\dagger\} = 1$), and that anticommutes with all the Bogoliubov modes $\hat{\ell}_j$ and $\hat{\ell}_j^\dagger$. Since it commutes with the Hamiltonian, it is said to be a zero-energy operator. This is the operator that differentiates the two ground states: $\hat{d} |\psi_e\rangle = 0$ and $\hat{d}^\dagger |\psi_e\rangle = |\psi_o\rangle$. More generally, the application of $\hat{d}^\dagger$ or $\hat{d}$ on one of the eigenstates listed above allows for the construction for another orthogonal state that has the same energy. It is thus responsible for the exact two-fold degeneracy of the entire spectrum of the Hamiltonian that we claimed, and of course also of the topologically protected degeneracy of the ground state.

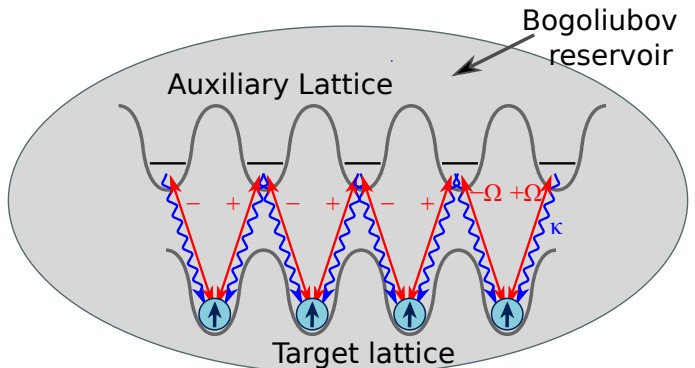

Figure 20: Proposed implementation scheme for the Lindblad master equation featuring Majorana boundary modes. The quantum wire is represented by the lower sites of an optical superlattice for spin polarised atomic fermions. They are coherently coupled to auxiliary upper sites by lasers with Rabi frequencies $\pm\Omega$, alternating from site to site. Dissipation results from spontaneous Bogoliubov phonon emission via coupling of the system to a BEC reservoir (light grey). As shown in the original article, this setting reduces to the jump operators $\hat{L}_j = \hat{C}_j^\dagger \hat{A}_j$ at late times, see Eq. (146). Inspired from Ref. [417].

In the dissipative context, the interpretation of the $\hat{\gamma}_{L,R}$ operators is slightly different, because now they represent operators that are not affected by the dissipation. As has been anticipated, it is more correct to speak of *dark modes* or of *zero damping modes*. Interestingly, as was also shown in the Hamiltonian context, Majorana zero damping modes also satisfy a non-trivial braiding statistics [417].

We may also mention that the jump operators $\hat{\ell}_j$ are linear in the fermionic operators, which yields a quadratic Lindbladian that can be exactly diagonalised, as we discussed in Sec. 4.1. A simple analysis shows that the asymptotic decay rate of the problem is $\gamma$ and that it does not scale with the size of the system.

The approach discussed so far suffers from the difficulty of an experimental implementation of a jump operator $\sqrt{\gamma}\hat{\ell}_j$ that does not conserve the number of particles. However, as was already noted in the original reference [417], another approach exists based on the experimental implementation of a number-conserving jump operator:

$$\hat{L}_j = \sqrt{\gamma}\hat{C}_j^\dagger \hat{A}_j. \tag{146}$$

Similarly to the jump operators in Eq. (142), these operators can be interpreted as the annihilation of an antisymmetric state and creation of a symmetric one, while the number of fermions is conserved by the ensuing dissipative dynamics.

It is possible to characterise the stationary states of the dynamics of this model [455]. We first observe that $\hat{\ell}_j |\psi_\sigma\rangle = 0$ implies $\hat{C}_j^\dagger |\psi_\sigma\rangle = -\hat{A}_j |\psi_\sigma\rangle$, so that:

$$\hat{L}_j |\psi_\sigma\rangle = \hat{C}_j^\dagger \hat{A}_j |\psi_\sigma\rangle = -\hat{C}_j^\dagger \hat{C}_j^\dagger |\psi_\sigma\rangle = 0. \tag{147}$$

Thus, the $|\psi_\sigma\rangle$ are also steady states of this new dynamics. In fact, we can say more: any projection of $|\psi_\sigma\rangle$ that has a well-defined number of particles is stationary:

$$|\psi_N\rangle = \frac{\hat{\Pi}_N |\psi_\sigma\rangle}{\sqrt{\langle\psi_\sigma| \hat{\Pi}_N |\psi_\sigma\rangle}}, \tag{148}$$

where $\hat{\Pi}_N$ is the projector onto the subspace of the global Hilbert (Fock) space with $N$ fermions. In Ref. [420] the authors show that there is only one dark state $|\psi_N\rangle$ once the value of $N$ is fixed. They further conclude that the dynamics induced by the jump operators in (146) conserves the number of particles and drives the system into a quantum state with the properties of a $p$-wave superconductor, that is, in the thermodynamic limit, $|\psi_e\rangle$ and $\hat{\Pi}_N |\psi_e\rangle$ have the same bulk properties. However, since the steady states of the system for open boundary conditions are unique, *they do not display any topological edge states*. Figure 20 illustrates some ideas about the experimental implementation of this scheme.

Finally, it is interesting to inspect the parent Hamiltonian associated to this alternate dissipative process:

$$\hat{H}_p = \sum_j \hat{L}_j^\dagger \hat{L}_j = \gamma \sum_j \hat{A}_j^\dagger \hat{C}_j \hat{C}_j^\dagger \hat{A}_j = -2\gamma \sum_j \left[ \hat{c}_{j+1}^\dagger \hat{c}_j + \hat{c}_j^\dagger \hat{c}_{j+1} + 2\hat{n}_j \hat{n}_{j+1} - \hat{n}_j - \hat{n}_{j+1} \right].$$
(149)

This parent Hamiltonian can be mapped—via a Jordan-Wigner transformation—to the ferromagnetic spin-1/2 Heisenberg Hamiltonian, which is gapless for any $N$ scaling linearly with $L$. We can thus employ results discussed above to conclude that the dissipative process is characterised by an asymptotic decay rate that scales at least as $L^{-2}$, the scaling of the gap of the parent Hamiltonian.

Following these foundational results reviewed here, the literature on this subject has been particularly rich. The two-dimensional case was also studied, and the dissipative creation of a vortex trapping a Majorana mode is discussed in Ref. [456]. A more general study of topological phases induced by dissipation in fermionic systems is presented in Ref. [418]. The number-conserving case is discussed in Ref. [420]. These results have opened the way to a more general investigation of the notion of topology in open quantum systems and for density matrices [457–474]. Finally, we mention that the topic of topological dark states is also intertwined with the long-standing quest of engineering fractional quantum Hall states in photonic setups [142, 419, 475–486].

### 5.3.3   Spin-Dicke states of fermionic gases

In this section we will focus on fermionic gases, where the interplay of statistics and spin produces intriguing steady states, as first pointed out in Ref. [421]. Our presentation follows in several points the more recent Ref. [487]. In contrast to the setup in the previous section that required careful engineering, it is interesting to discuss here the dark states associated to one of the most natural open-system processes in cold-atom experiments: atom losses.

We consider a $d$-dimensional hyper-cubic optical lattice in the single-band approximation populated by a fermionic spin-1/2 gas. We introduce the annihilation and creation operators $\hat{c}_{j,\sigma}^{(\dagger)}$ associated to the site $j$ and to the spin $\sigma$ that satisfy the canonical anticommutation relations. The Hamiltonian of the system is the standard Hubbard model:

$$\hat{H} = -t_H \sum_{\langle i,j \rangle} \sum_{\sigma=\uparrow,\downarrow} \left[ \hat{c}_{i,\sigma}^\dagger \hat{c}_{j,\sigma} + \text{H.c.} \right] + U \sum_j \hat{n}_{j,\uparrow} \hat{n}_{j,\downarrow}.$$
(34)

We will focus on the experimentally relevant case of local two-body losses with rate $\gamma$, described by the following set of jump operators:

$$\hat{L}_j = \sqrt{\gamma} \hat{c}_{j,\uparrow} \hat{c}_{j,\downarrow}.$$
(35)

Fermionic statistics here poses a constraint on the single-site loss process, that it can only remove two particles with opposite spins because it is not possible to have two particles

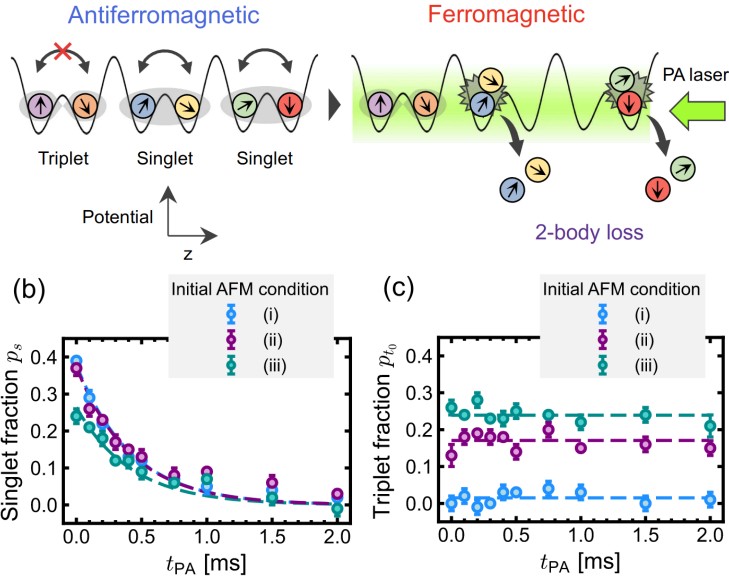

Figure 21: (Top) Sketch of the experiment on two-body losses. A one-dimensional array of double-well potentials is initially filled with one fermionic particle per site; six spin components are possible but in a double-well only two of them appear, so that the description of a single double-well can be carried out in terms of an effective spin-1/2 setup, that can be in a spin singlet or triplet state. Singlet states are characterised by a spatially symmetric wavefunctions, this allows the two particles to sit on the same site and to be lost. Triplet states, characterised by an antisymmetric wavefunction, cannot have a double site occupation, and are never lost. In the experiment, on-site two-body losses are induced with a photoassociation process. (Bottom) Dynamics of (b, left) the singlet fraction and of (c, right) the triplet fraction in all double-wells as a function of the time to which they are exposed to photoassociating light inducing losses. Whereas the singlet fraction decreases exponentially with time, the triplet fraction remains constant. The three set of data (i), (ii) and (iii) correspond to three different initial conditions Adapted from Ref. [423] [Copyright (2023) by the American Physical Society.].

with the same spin sitting at the same site. It follows immediately that the $z$ component of magnetisation of the gas is conserved during the dynamics. In fact, the model is SU(2) invariant and we could have expressed the jump operator in Eq. (35) in a basis with a different spin-quantisation axis, always with the result that the loss process removes two fermions with opposite spin. We can thus conclude that all components of magnetisation of the gas are a strong symmetry of the dynamics.

More remarkably, we can now observe that the two particles lost by the dissipative process in Eq. (35) are in a spin singlet. This follows easily from the fact that they share the same orbital wavefunction. A loss process that only annihilates spin singlets conserves the total spin of the system: the expectation value of $\hat{S}^2$ is conserved during the dynamics. Using the classification introduced in Section 3.1.1, it follows from what we just wrote that also $\hat{S}^2$ is a strong symmetry of the dissipative dynamics. We can visualise this by considering the dissipative dynamics of an initial state with only two fermions, that will generically be a linear superposition of a spin-singlet $S = 0$ state and of a spin-triplet $S = 1$. For the $S = 1$ state the orbital wavefunction must be antisymmetric with respect to the exchange of particles, and thus the two fermions can never occupy the same site: the $S = 1$

spin channel is thus not lossy. Conversely, the $S = 0$ spin channel can have particles on the same site and can dissipate them. Thus, as the average number of particles decreases, the spin $\langle \hat{S}^2 \rangle$ of the system remains constant. This simple observation has recently been the object of an experimental measurement [423]. As shown more clearly in Fig. 21, the authors have prepared a fermionic gas into several double wells, each filled with exactly two atoms, and have observed a decrease with time of the number of atoms in the singlet channel, while the number of atoms in the triplet channel was held constant.

Such spin-selective decay properties can be generalised to an arbitrary number $N$ of initial particles. In this case the relation between spin and dynamics is more complicated, but we can easily draw some general conclusions from the simple observation that during the loss dynamics the number of particles decreases while the spin of the gas remains constant. In fact, due to the spin conservation the number of particles of the gas cannot decrease arbitrarily, as a gas with $N$ particles can have a spin quantum number $S$ that is upper bounded by $N/2$ and thus:

$$\langle \hat{S}^2 \rangle \leq \left\langle \frac{\hat{N}}{2} \left( \frac{\hat{N}}{2} + 1 \right) \right\rangle. \tag{150}$$

Since $\langle \hat{S}^2 \rangle$ is conserved, this equation places a lower bound on the number of particles. Stationary states thus are expected to be incoherent mixtures of states with maximal spin, whose wavefunction has a spin part that is a Dicke state. It is known that Dicke states have a fully symmetric spin wavefunction: as a consequence, the orbital wavefunction is completely antisymmetric and cannot have double occupancies.

We can give a more rigorous description of these dark states of the loss dynamics. We begin by constructing the set of the $N$-particle eigenstates of the Hubbard Hamiltonian in Eq. (34) that are fully polarised. In an hypercubic plane with periodic boundary conditions, the eigenmodes of the kinetic energy part of the Hamiltonian (34) are plane waves with wavevector $\mathbf{k}$. We thus introduce the operator:

$$\hat{c}_{\mathbf{k},\sigma}^\dagger = \frac{1}{\sqrt{N_s}} \sum_j e^{i\mathbf{k}\cdot\mathbf{r}_j} \hat{c}_{j,\sigma}^\dagger \tag{151}$$

and the states:

$$\hat{c}_{\mathbf{k}_1,\uparrow}^\dagger \hat{c}_{\mathbf{k}_2,\uparrow}^\dagger \ldots \hat{c}_{\mathbf{k}_N,\uparrow}^\dagger |0\rangle\,, \tag{152}$$

where $|0\rangle$ is the empty state, $\hat{c}_{\mathbf{k},\sigma} |0\rangle = 0$. These states are the dark states of the loss dynamics, as can be proven by showing that they are eigenstates (with real eigenvalue) of the non-Hermitian Hamiltonian associated with the Hamiltonian in Eq. (34) and the jump operators in Eq. (35):

$$\hat{H}_{\text{nH}} = \hat{H} - \frac{i}{2} \sum_j \hat{L}_j^\dagger \hat{L}_j = -t_H \sum_{\langle i,j \rangle} \sum_{\sigma=\uparrow,\downarrow} \left[ \hat{c}_{i,\sigma}^\dagger \hat{c}_{j,\sigma} + \text{H.c.} \right] + \left( U - i\frac{\gamma}{2} \right) \sum_j \hat{n}_{j,\uparrow} \hat{n}_{j,\downarrow}. \tag{153}$$

This non-Hermitian Hamiltonian is a Hubbard model with complex interaction, where the interacting term has a conservative and non-conservative parts, and plays an important role in the physics of fermionic gases with two-body losses. The states in Eq. (152) have wavefunctions that factorise the orbital and spin parts. The orbital part is a Slater determinant constructed with the single-particle eigenmodes of the kinetic energy. The dark states that are not fully polarised in the $z$ direction can be obtained by applying the spin-lowering operator

$$\hat{S}^- = \sum_{\mathbf{k}} \hat{c}_{\mathbf{k},\downarrow}^\dagger \hat{c}_{\mathbf{k},\uparrow} \tag{154}$$

to the states in Eq. (152). The SU(2) invariance of the non-Hermitian Hamiltonian can then be invoked to argue that they are also dark states of the dynamics.

These results have been recently generalised to SU(3) gases subject to two-body losses, where the stationary states are discussed using the representation theory of the SU(3) symmetry group and are dubbed generalised Dicke states [422, 487].

The study of the asymptotic decay rates of the loss dynamics that we described so far are actually related to the spectral properties of the non-Hermitian Hubbard model in Eq. (153). This is a general property of Lindblad master equations with only losses that follows from the fact that the Lindbladian operator can be put in a block triangular form [488]. In a few words, this result can be understood in the following way. We first observe that loss operators are characterised by a weak symmetry, generalising that of the toy model introduced in Eq. (15), namely symmetry under the transformations $e^{i\theta \sum_{j,\sigma} \hat{n}_{j,\sigma}}$. As noted in Sec. 3.1.1, weak symmetries mean the Lindbladian has a block diagonal structure. If one labels blocks of the density matrix as $\hat{\rho}_{n_L,n_R}$, where $n_L, n_R$ correspond to eigenstates of the total-particle-number operator $\hat{N} = \sum_{j,\sigma} \hat{n}_{j,\sigma}$, the dynamics of the part of the density matrix with $n_L = n_R$ decouples from those parts where $n_R \neq n_L$. To reach the triangular form, a further requirement is needed: the fact our dissipation consists of only loss terms. This means that the master equation has the following block form:

$$\frac{d}{dt}\hat{\rho}_{n,n} = \hat{\hat{\mathcal{L}}}_n^{(0)}[\hat{\rho}_{n,n}] + \hat{\hat{\mathcal{L}}}_n^{(+2)}[\hat{\rho}_{n+2,n+2}]. \tag{155}$$

Physically this states that the time evolution of the block corresponding to $n$ particles depends only on itself, or on a source term from the block with $n+2$ particles. This yields a block triangular form as noted above. The number conserving part, $\hat{\hat{\mathcal{L}}}_n^{(0)}$ corresponds to the non-Hermitian Hamiltonian, while the jump operators lead to the coupling from the $n+2$ block. Because $\hat{\hat{\mathcal{L}}}_n^{(+2)}$ are just loss processes, they appear on the upper half of the triangle. This means that the spectrum of the Lindbladian is determined only by the diagonal blocks, i.e. the non-Hermitian Hamiltonian. Note that this behaviour would not arise if there were incoherent pumping to balance losses, even if the pumping respected the weak symmetry $\hat{S}$.

Using the above, one can show that for one-dimensional gases, the Lindbladian is gapless and the asymptotic decay rate scales as $1/L^3$ for states with total momentum close to $K \sim 0$ and with a finite density of excitations. We refer the reader to the specialised publications for further details [487, 489]. This result is derived analytically exploiting the fact that the Hamiltonian in (153) can be solved with the techniques of Bethe ansatz (see also Refs. [309, 313, 490, 491] for other examples).

### 5.3.4 Dissipative preparation of Mott insulators of photons

A recent example of successful experimental dissipative state preparation has been the realisation of a Mott insulator of microwave photons using superconducting circuits [55, 424]. Although this case does not deal with an exact dark state of the dissipative dynamics, the result has many analogies with the spirit of the discussion presented out so far. Crucially, it relies on dissipation to achieve the state preparation and is of significant experimental relevance.

A major challenge in realising strongly correlated fluids of light is the fact that photons are not conserved particles whose density can be tuned by an external chemical potential. Starting from early results on lasing and polariton condensation (See Sec. 2.6) and the weakly interacting Bose-Einstein condensates of photons, first reported in Ref. [184], several theoretical proposals have been put forward to realise an effective chemical potential for strongly interacting photons using tailored dissipative reservoirs [492–495]. In Ref. [424]

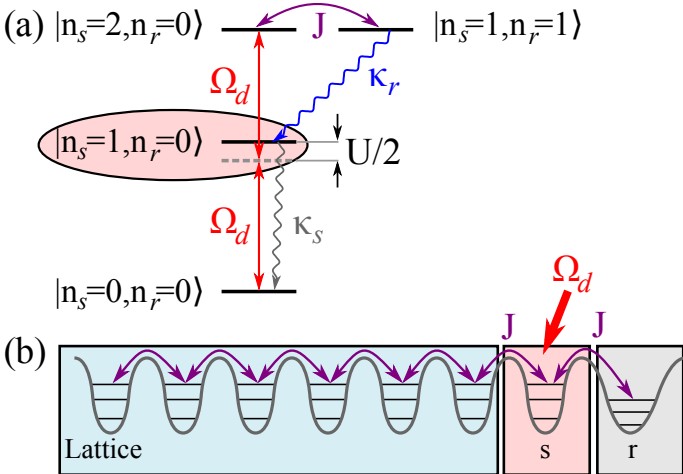

Figure 22: Scheme for the one-transmon dissipative stabiliser implemented in the experiment. (a) Level scheme showing a two-photon drive resonant with the $0 \leftrightarrow 2$ transition of the stabiliser site, $s$. A reservoir site $r$ is tuned so that the $n_s = 2 \rightarrow n_s = 1$ transition of the stabiliser site is resonant withe $n_r = 0 \rightarrow n_r = 1$ transition. The reservoir site then quickly decays at rate $\kappa_r$. The overall effect is to stabilise the $n_s = 1, n_r = 0$ state, as highlighted in pink. (b) Integration of the stabiliser into a Bose–Hubbard lattice. Inspired by Ref. [424].

the authors realised a 1D chain of eight coupled superconducting qubits (transmons [496] in their implementation), realising a Bose–Hubbard Hamiltonian. Each site has intrinsic losses which would lead to an empty state. In order to stabilise a Mott insulating phase with exactly one photon on each site, the authors implemented two versions of a *dissipative stabiliser*, respectively involving one and two auxiliary transmons. We will describe the working principle of the stabiliser, in the scheme involving one auxiliary transmon for simplicity, first discussing the case in which the system consists of a single site coupled to the auxiliary transmon.

We start by considering the situation where we want to prepare exactly one photon on a single site, assuming for simplicity that all the $n$-photon states have a long but not infinite lifetime. The solution that is routinely applied in isolated quantum systems cannot be applied: a direct transition via a $\pi$ pulse to the $n = 1$ state cannot work because the state will decay at a certain unknown time, so that it is not obvious a priori when one should reapply the $\pi$ pulse. The envisaged solution is sketched in Fig. 22 and is based on coherent transitions from the $n = 0$ state the $n = 2$ state with a classical field of frequency $\omega_{02}/2$: the $n = 2$ state is thus populated via a two-photon process. Thanks to the on-site Hubbard interaction $U$, the drive frequency is detuned from the transition from the $n = 0$ to the $n = 1$ state by $U/2$: $\omega_{02}/2 \neq \omega_{01}$ and hence the $n = 1$ state remains unpopulated. The single site is close to an auxiliary and strongly dissipative transmon that has energy $\omega_{12}$: the two-photon state is thus resonant with the state where one photon is in the site to be stabilised and one photon in the auxiliary site. Once the coherent transition to this state has taken place, the auxiliary transmon dissipates its photon in the environment and the target site is left with exactly one photon. The interesting point of this process is that it is initiated by a classical driving field that is left switched on during the entire procedure. When the $n = 1$ state is populated, the classical drive is off-resonance and cannot induce any transition in the system. However, as soon as the $n = 1$ state eventually decays, the classical drive becomes resonant again and rapidly refills the one-photon state.

When considering the whole chain instead of a single site, the scheme is generalised stabilising only the first site of the chain, so that it acts as a narrow-band photon source for the rest of the chain. Photons from this site tunnel into the other sites until filling them all. When all the sites are filled with one photon, then the stabiliser will be unable to inject additional photons, as this would require an energy higher then the source energy, because of the incompressibility of the Mott phase. If instead a photon is lost from the chain, then the stabilised site pumps one photon in the chain. The experimental results are presented in Ref. [424] show that within a few microseconds, the photon fluid has reached a steady state with near-unity occupancy.

A driving scheme in the same spirit of having an energy selective source of particles was proposed in [493], which is part of a general set of studies on reservoir engineering for many-body photonic states [492, 495, 497, 498] and on the theoretical characterisation of the dissipatively prepared Mott state [354, 499]. Other experiments involving strongly correlated photonic states have been performed [500, 501] and more experiments are to be expected; for instance, similar approaches are currently under discussion for the realisation of topological strongly correlated photonic fluids [142, 419, 475–486] (see also the previous section).

## 5.4   Quantum computation using dissipation

The notion of dark states is more versatile than what we have discussed so far and has applications that reach beyond the idea of dissipative state preparation. In a seminal article, F. Verstraete, M. M. Wolf and J. I. Cirac have discussed the computational power of the dissipative creation of many-body dark states, and have shown that the techniques explained above can be used to perform universal quantum computation [9]. The core of the reasoning proposed by the authors is rather simple and goes as follows. A set of $N$ qubits is considered over which a set of unitary gates $\{\hat{U}_t\}_{t=1}^{T}$ needs to be applied; each of the unitary gates $U_t$ is assumed to act only on two neighbouring spins. The system is initialised in $|\psi_0\rangle = |0\rangle_1 \otimes |0\rangle_2 \ldots \otimes |0\rangle_N$, the state where all $N$ qubits are in the state $|0\rangle$. The final outcome of the quantum computation is obtained by the ordered application of all gates, $|\psi_T\rangle = \hat{U}_T \ldots \hat{U}_2 \hat{U}_1 |\psi_0\rangle$. The main point of the authors is that $|\psi_T\rangle$ can be obtained as a dark state of a properly engineered Lindblad master equation.

The Lindbladian proposed to realise this desired state does not have any Hamiltonian part but only a set of jump operators that need to be carefully tailored to implement the gates $\hat{U}_t$. In order to define the jump operators, it is necessary to introduce an additional register, i.e. an additional quantum system, that in this case is defined on a $T$-dimensional Hilbert space with basis $\{|t\rangle_r\}_{t=0}^{T}$. The quantum jump operators are split into two sets and read:

$$\hat{L}_i = |0\rangle\langle 1|_i \otimes |0\rangle\langle 0|_r \qquad\qquad\qquad i = 1, 2 \ldots N; \qquad (156)$$

$$\hat{L}_t = \hat{U}_t \otimes |t\rangle\langle t-1|_r + \hat{U}_t^{\dagger} \otimes |t-1\rangle\langle t|_r\,, \qquad t = 1, 2 \ldots T. \qquad (157)$$

The $N$ operators $L_i$ act on the register and on the $i$-th qubit. The $T$ operators $L_t$ act on the register and those qubits on which the gate $\hat{U}_t$ acts.

The stationary state of such a Lindbladian reads:

$$\hat{\rho}_0 = \frac{1}{T+1} \sum_{t=0}^{T} |\psi_t\rangle\langle\psi_t| \otimes |t\rangle\langle t|_r \qquad \text{with} \qquad |\psi_t\rangle = \hat{U}_t \ldots \hat{U}_2 \hat{U}_1 |\psi_0\rangle\,. \qquad (158)$$

In order to prove this, let us first consider the jump operators $\hat{L}_i$. With a simple calculation $\hat{L}_i \hat{\rho}_0 \hat{L}_i^{\dagger} = 0$; moreover $\hat{L}_i^{\dagger} \hat{L}_i = |1\rangle\langle 1|_i \otimes |0\rangle\langle 0|_r$, and thus $\{\hat{L}_i^{\dagger} \hat{L}_i, \hat{\rho}_0\} = 0$. This shows that

the state $\hat{\rho}_0$ is stationary with respect to the action of the first set of jump operators. Let us now consider the second set of $\hat{L}_t$ operators. With a bit of algebra, one shows that:

$$
\begin{aligned}
\hat{L}_t \hat{\rho}_0 \hat{L}_t^\dagger =& \hat{U}_t \, |\psi_{t-1}\rangle\langle\psi_{t-1}| \, \hat{U}_t^\dagger \otimes |t\rangle\langle t|_r + \hat{U}_t^\dagger \, |\psi_t\rangle\langle\psi_t| \, \hat{U}_t \otimes |t-1\rangle\langle t-1|_r \\
=& \, |\psi_t\rangle\langle\psi_t| \otimes |t\rangle\langle t|_r + |\psi_{t-1}\rangle\langle\psi_{t-1}| \otimes |t-1\rangle\langle t-1|_r \, .
\end{aligned}
\tag{159}
$$

After having computed that $\hat{L}_t^\dagger \hat{L}_t = I \otimes (|t\rangle\langle t| + |t-1\rangle\langle t-1|)$ it is not difficult to show that the anticommutator in the Lindblad master equation from this jump operator takes the same form as Eq. (159):

$$
\frac{1}{2} \left\{ \hat{L}_t^\dagger \hat{L}_t, \hat{\rho}_0 \right\} = |\psi_t\rangle\langle\psi_t| \otimes |t\rangle\langle t|_r + |\psi_{t-1}\rangle\langle\psi_{t-1}| \otimes |t-1\rangle\langle t-1|_r \, .
\tag{160}
$$

It follows that the state $\hat{\rho}_0$ is stationary also with respect to the second set of jump operators.

A remarkable result is that the asymptotic decay rate for reaching the stationary state decays as $T^{-2}$ and thus the computation time diverges polynomially with the number of gates to be applied during the computation. The complexity of the computation is therefore hidden in the scaling of the number of gates $T$ with the number of qubits $N$. The uniqueness of the steady state can be proved by diagonalising the Lindbladian.

It is interesting to conclude this Section by making contact with earlier works on quantum information processing assisted by open-quantum-system techniques, as for instance the work by J.P. Paz and W.H. Zurek dating from 1998 where the idea of *continuous quantum error correction* is put forward [502]. They consider a quantum error-correction scheme where the system is repeatedly corrected and show that as the correction rate goes to infinity the dynamics of the system is actually described by a Lindblad master equation. Recent advances in the development of synthetic systems have revitalised these ideas, both from the viewpoint of error correction [503, 504] or of quantum memories [505]. Moving beyond the problem of safely storing quantum information with the help of dissipation, it was proposed in the early 2000's to perform quantum computation assisted by the presence of an environment [437, 506]. The idea differs from that of Ref. [9] in that the dark subspace is used to confine the entire quantum computation process, rather than engineering a dark state that contains the answer to the quantum computation.

## 5.5   Quantum many-body steering

Within this Section on dissipative state preparation we may also consider a different, although related, protocol based on quantum steering. The idea of steering is as old as quantum mechanics. Originally introduced by Schrödinger [507], its meaning evolved progressively to indicate questions related to the understanding of non-locality and the existence (or rather impossibility) of hidden-variable models. Quantum steering refers to the non-classical correlations that can be observed between the outcomes of measurements on entangled state and the properties of the state after the measurement, see Ref. [508] for a comprehensive review on this topic.

In a broader sense, but in line with the initial formulation, quantum steering protocols can be also related to quantum state preparation. The underlying idea is to control the dynamics of a quantum system by means of a sequence of measurements; as a result, the system is forced to evolve towards a desired quantum state. There is a rather large variety of cases depending on how the preparation is achieved; for instance, the measurement does not need to be projective but it can be a generic positive operator-valued measure (POVM) [186]. The protocol can either involve post-selection or be non-selective, as in the rest of this section. Here we briefly discuss the (non-selective) approach developed in

Ref. [425] because of its connection with the Lindblad dynamics and its applications to many-body state preparation. In the language we used throughout these Lecture Notes, the steering protocol designed in [425] can be considered as a (time-periodic) Floquet-Lindblad engineered dynamics.

Following [425], the protocol consists in a sequence of discrete steps in which the detector is initialised and then unitarily coupled to the system for a given time and then decoupled. The detector is then re-initialised. In spirit, this protocol is equivalent to the collisional models of system-bath dynamics (for further discussion of this topic see the review [509]). Indicating the system-detector interaction Hamiltonian with $\hat{H}_I$, and the initial state of the detector by $\hat{\rho}_d$, the state of the system (after tracing over the detector degrees of freedom) evolves as

$$\hat{\rho}(t + \delta t) = \text{Tr}_d \left[ e^{-i\hat{H}_I t} \, \hat{\rho}(t) \otimes \hat{\rho}_d \, e^{i\hat{H}_I t} \right] \tag{161}$$

This elementary step plus the re-initialisation reduces, in the continuous time limit, to a Lindblad dynamics (the proof is done with techniques that are reminiscent of the standard methods that are used to derive the Lindblad master equation, as we do in appendix A).

It is useful to consider first the case of a spin-1/2 system. By choosing a $\hat{H}_I = J\hat{\sigma}_s^+ \hat{\sigma}_d^- +$ H.c. (the indices $d, s$ refer to the system and detector qubits) it is possible to show that the state of the system will converge to an eigenstate along the $z$-direction in the Bloch sphere.

The same scheme, with a more complex choice of the number of detector qubits and their interaction with the system, can be followed to generate the ground state of the AKLT Hamiltonian. As detailed in Ref. [425], for each pair of neighbouring spins, one has to couple five detector qubits initialised in a fully polarised state. The number of detector qubits is dictated by the complex form of the interaction Hamiltonian that will lead to transition in the different spin sectors. Without entering into details, it is worth recalling that the choice of the interaction Hamiltonian $\hat{H}_I$ should satisfy the following condition

$$\langle \Psi | \hat{\rho}(t + \delta t) | \Psi \rangle > \langle \Psi | \hat{\rho}(t) | \Psi \rangle$$

meaning that at each step of the protocol, the state of the system get closer to the target state. This guarantees that the state approaches progressively the target state but does not determine the speed at which this occurs. To this end, active control strategies that choose the appropriate system-detector interaction, based on the measurement record, may speed-up the convergence [510].

To conclude, it may be useful to draw some comparison of this steering protocols with the dissipative state preparation discussed in the previous sections. In all the approaches, the system is driven towards the target states by a sequence of measurements, these are either performed by a measurement apparatus (as discussed here) or by the environment (previous sections). While dissipative engineering is entirely based on the identification of the form of the jump operators that have the desired (many-body) state as a target state, in steering protocols the measurement is generically adaptive (in some sense the Kraus operators may depend on time). Furthermore, another difference is that bath engineering is a non-selective protocol (the state is progressively purified), whereas steering may require post-selection. When this last condition applies, steering becomes a special case and a first example of monitored quantum system, whose dynamics will be addressed in more detail in Section 8.

## 5.6 Stability to perturbations

We conclude this section by discussing an important problem associated to the physics of many-body dissipative state engineering, namely that of the stability of the protocol

to perturbations. In Ref. [426] the authors discuss the effect of a perturbation on the preparation of a Bose-Einstein condensate discussed in Sec. 5.2, finding a phase diagram that displays a dissipative phase transitions. Dissipative phase transitions are the subject of Sec. 6, so this subsection, that summarises some of the results in Ref. [426], constitutes a bridge toward the material that will be presented there.

We consider the situation in which the purely dissipative dynamics generated by the jump operators in Eq. (142) is complemented by the unitary evolution associated with the standard Bose–Hubbard Hamiltonian for a $d$-dimensional lattice, first introduced in Eq. (36):

$$\hat{H} = -t_H \sum_{\langle i,j \rangle} \left( \hat{a}_i^\dagger \hat{a}_j + \text{H.c.} \right) + \frac{U}{2} \sum_i \hat{n}_i (\hat{n}_i - 1). \tag{162}$$

In this case, the interacting term proportional to $U$ breaks the conditions discussed above for the appearance of a pure dark state: $|\text{BEC}_N\rangle$ is not an eigenstate of $\hat{H}$. In this situation one cannot expect a pure dark state to appear, yet, as shown by the authors of Ref. [426] with a mean-field time-dependent Gutzwiller analysis, for weak enough values of $U$ and for weak loss rates the system remains in a condensed phase with a macroscopic occupation of the $\mathbf{k} = 0$ bosonic mode. When the interaction parameter $U$ is increased, however, the system is driven in a thermal phase that does not display off-diagonal long-range order. The result of this study also provides a qualitative understanding of the stability of the system: although the perturbation breaks the condition that is necessary for having a pure dark state, the mixed state that is obtained displays the same features. When the perturbation exceeds a critical value, the stationary state changes structurally and becomes a trivial thermal phase.

Note however that the thermodynamic limit and the $t \to \infty$ limit do not commute, as we will discuss in more detail in Section 6. Indeed, if the system is discussed at any finite time, the properties of the state $\rho(t)$ evolve smoothly in the phase diagram as a function of the parameters. Yet, if one looks at the properties of the stationary states, they feature sharp discontinuities in the expectation values of the $\mathbf{k} = 0$ mode.

Other articles have addressed this stability issue in other models; we mention here Ref. [427], which has addressed the competition between two dissipative mechanisms featuring incompatible many-body dark states. Calling $\eta$ the ratio of the decay rates of the two dissipative mechanisms, it is found that the steady state is pure only in the two limiting cases $\eta = 0$ and $\eta \to \infty$. In between, the stationary state is mixed and thus one cannot speak of dark state. Focusing on dissipative mechanisms that engineer a paramagnetic spin state and ferromagnetic spin states, a mean-field analysis reveals the existence of a nonequilibrium phase diagram composed of two phases separated by a second order phase transition that parallels the properties of the thermal phase diagram.

# 6 Dissipative Phase Transitions

An essential feature of the Lindbladian many-body dynamics is the competition between coherent evolution, generated by the Hamiltonian, and incoherent processes generated by the dissipation. This competition—as well as competition amongst different dissipative or coherent processes—can result in sharp, non-analytic changes in the properties of the stationary state of the system. This leads to the concept of Dissipative Phase Transitions (DPT), corresponding to non-analytic changes to the steady state as one varies some parameter, $\eta$, through a critical value $\eta_c$. We specifically assume the Lindbladian is a smooth function of $\eta$ (as would occur if $\eta$ controls the strength of some specific term in the Hamiltonian or dissipation). A non-analytic change of the steady state despite

smooth evolution of the Lindbladian is the open-quantum-system analogue of the behaviour well known in equilibrium phase transitions. For most dissipative phase transitions one can define an order parameter $\mathcal{O}$ in terms of an observable $\hat{O}$ computed on the steady state density matrix, $\hat{\rho}_{\text{ss}}$ for given $\eta$, i.e. $\mathcal{O}(\eta) = \text{Tr}\left[\hat{\rho}_{\text{ss}}(\eta)\hat{O}\right]$. A DPT occurs when $\left(\frac{d}{d\eta}\right)^k \mathcal{O}(\eta)$ shows singular behaviour (for some value of $k$) at $\eta \to \eta_c$. Since the observable is independent of the control parameter $\eta$, a DPT must imply a change in the structure of the stationary state.

As with other phase transitions, DPT can be first order (discontinuous, see e.g. [230,276, 388,412,511–515]) or second order (continuous, see e.g. [224,227,426,516,517]). Examples of these are illustrated in Fig. 23, through bifurcation diagrams describing the order parameter for different stationary states (fixed points) of the system. Figures such as these occur in the theory of dynamical systems [518]; they can also apply to describing the stable, metastable, or unstable points of a many-body system, often found from mean-field or semiclassical analyses. Here we use "unstable" to refer to points where any small fluctuation will grow exponentially. The distinction between "stable" and "metastable" is more subtle, and regards the relative rates of switching between such states—we will discuss this in detail in Sec. 6.1.

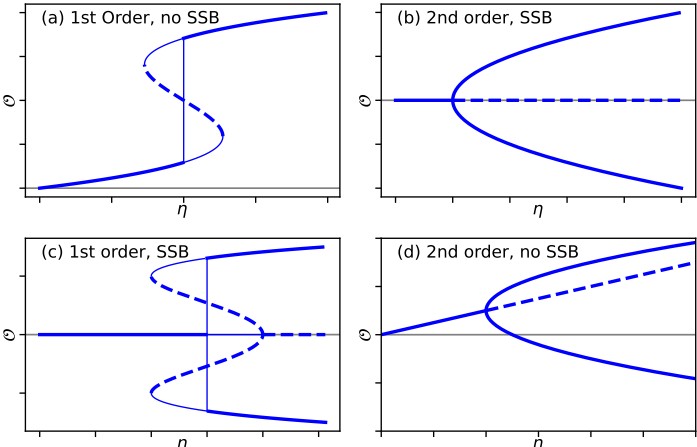

Figure 23: Examples of types of discontinuous behaviour, in terms of order parameter $\mathcal{O}$ as a function of parameter $\eta$. Thick solid lines indicate stable states, thin solid lines metastable states, while dashed lines indicate unstable fixed points. (See text below for further discussion). Case (a) presents the prototypical first-order transition without spontaneous symmetry break (SSB). Case (b) is the prototypical second-order transition with SSB. However, other scenarios can arise, such a first-order transition with SSB, shown in panel (c), or a second-order transition without SSB, shown in panel (d).

The form of curves shown in Fig. 23 are very typical for mean-field theories of dissipative phase transitions, showing multiple solutions at each value of $\eta$. Mean-field theory can distinguish stable and unstable fixed points (i.e. the dashed and solid lines in the figure). However, mean-field approaches do not distinguish between the stable (thick) and metastable (thin) lines, thus predicting bistability when multiple stable and metastable states exist. We will discuss below how the spectral theory of phase transitions refines this understanding.

As with other transitions, DPT can be associated with spontaneous symmetry breaking (SSB) in the thermodynamic limit. In the open quantum system context this means there

are symmetries under which the equation of motion is invariant but the steady state is not. We will discuss further below how this relates to the spectral properties of phase transitions, as well as how it emerges as one approaches the thermodynamic limit. The prototypical first-order phase transition, analogous to optical bistability, shown in Fig. 23(a) does not involve SSB, while the prototypical second-order transition, Fig. 23(b) does. However, as illustrated in Fig. 23(c,d), one can also have first-order transitions with SSB and second-order transitions without, for examples see [519]. Moreover, phase boundaries can change between first- and second-order through tricritical points, i.e. Fig. 23(b) can evolve to Fig. 23(c); see Refs. [520–522] for some examples of this.

A feature of DPT that is not generally available [523] for other types of transition is the possibility of cases where there is no stationary state, but instead persistent oscillations. This corresponds to a spontaneous symmetry breaking of continuous time-translation symmetry, as in the case of Boundary Time Crystal phases [524] or dissipative limit cycle phases [277, 525–528].

Dissipative phase transitions have been the subject of experimental investigations in various platforms. Notable examples include solid-state quantum optics [529–533], ultracold atoms in optical cavities in the context of Dicke superradiance [57, 106, 534], and Rydberg gases demonstrating an absorbing state transition and self-organised criticality [141]. Transitions to time crystal or limit cycle phases have been observed in experiments with ultracold atoms in cavities [535] or free space [536, 537] as well as nanolasers [538].

In this Section we review DPT from different theoretical perspectives, we provide several key examples and discuss experimental evidence in different platforms. A general classification of DPT has been proposed in terms of the properties of the Lindblad super-operator generating the dynamics, see Refs. [199, 200]. A complementary perspective on DPT can be obtained using Schwinger–Keldysh Field Theory [31, 32] which also allows to assess their relation with field-theoretic classification of other—i.e. classical (thermal) and quantum—phase transitions.

## 6.1    General Classification: Spectral Theory and Schwinger–Keldysh Field Theory

The formulation of the Lindblad master equation as an eigenvalue problem for the Lindbladian superoperator is suggestive of similarities with the spectral theory of quantum phase transitions: in equilibrium at zero temperature the existence of a quantum phase transition (QPT) is associated to the closing of an energy gap upon changing a control parameter. If this is the case the ground-state becomes degenerate with the first excited state and the nature of the system can change abruptly, either through a first order or higher order non-analyticity. As we discuss below, the same can be said for DPT in terms of eigenvalues of the Lindbladian; in particular, we will see this relates to the properties of eigenvalues with zero real part. This spectral theory of DPT has been the subject of significant interest recently, see for example Refs. [199, 200] and references therein.

As we will also discuss below, DPT (like other phase transitions) exist only in a thermodynamic limit of large system size. We will discuss below how finite-sized systems still show features of the thermodynamic-limit phase transition, and how the spectral properties depend on system size.

### 6.1.1    Transitions without symmetry breaking

We start by considering the case of a first-order DPT at $\eta = \eta_c$ without symmetry breaking, associated to the existence of a discontinuity in some physical observable, e.g. Fig. 23(a), and discuss the consequences of this on the eigenspectrum of the Lindbladian. We first

discuss behaviour in the thermodynamic limit and focus on the ensemble-averaged steady-state density matrix. We then consider effects of finite size and finite duration of experiments. As we discuss further below, there are subtle points about how these two limits—which do not commute—behave.

**Spectral theory in the thermodynamic limit.** It is tempting to try to naively apply the construction that works for a ground-state QPT, where the critical point is associated with the exchange of two eigenstates—i.e. the ground-state and the first excited state swap their role at the critical point—with the closing of the energy gap. Adapting this to the Lindbladian however requires care, since eigenstates of the Lindbladian do not generally correspond to states of the system. Specifically, while $\hat{\rho}_0 = \hat{\rho}_{\mathrm{ss}}$ is the physical steady state, all other Lindbladian eigenvectors $\hat{\rho}_{\mu \geq 1}$ are traceless and so, as discussed in Sec. 3.1.2, do not describe physical density matrices. The appropriate picture, as developed in Ref. [199], addresses this issue as we discuss below.

One may start by considering a transition in which there are two different stationary states, one for $\eta < \eta_c$ and one $\eta > \eta_c$. We define these states to be called $\hat{\rho}^<$ and $\hat{\rho}^>$ as follows:

$$\hat{\rho}^< = \mathrm{Lim}_{\eta \to \eta_c^-} \ \hat{\rho}_{\mathrm{ss}}(\eta), \qquad \hat{\rho}^> = \mathrm{Lim}_{\eta \to \eta_c^+} \ \hat{\rho}_{\mathrm{ss}}(\eta), \tag{163}$$

and crucially $\hat{\rho}^< \neq \hat{\rho}^>$. This suggests an expression such as

$$\hat{\rho}_{\mathrm{ss}}(\eta) = \theta(\eta_c - \eta)\hat{\rho}^< + \theta(\eta - \eta_c)\hat{\rho}^>. \tag{164}$$

We discuss the behaviour exactly at $\eta = \eta_c$ further below, as the behaviour at this point will only become clear when we consider the effects of finite size and finite time.

In order that the steady state can change discontinuously at $\eta = \eta_c$ it must be the case that at this point there are two steady states of the Lindbladian, i.e. $\hat{\mathcal{L}}_{\eta=\eta_c}[\hat{\rho}^<] = \hat{\mathcal{L}}_{\eta=\eta_c}[\hat{\rho}^>] = 0$. We will discuss below how the eigenvalues must behave as one moves away from this point. However, based on the statements already made, we may make statements about the behaviour of the eigenstates of the Lindbladian, $\hat{\rho}_0, \hat{\rho}_1$ in the vicinity of $\eta = \eta_c$.

As we have already seen, the steady state density matrix, $\hat{\rho}_0$ is discontinuous across the transition, jumping between $\hat{\rho}^<$ and $\hat{\rho}^>$ either side of the transition. As we will argue below, exactly at the transition we can expect $\hat{\rho}_0$ is a mixture of these states, such as the equal weight mixture $(\hat{\rho}^< + \hat{\rho}^>)/2$. One may note that the equal-weight mixture is not the only possible solution, and whether this steady state takes exactly this form may vary from problem to problem. In contrast to the discontinuous behaviour of $\rho_0$, the first excited state $\hat{\rho}_1$ is continuous, as $\hat{\rho}_1 = \hat{\rho}^> - \hat{\rho}^<$ is the only traceless form that can be made out of the two density matrices $\hat{\rho}^{<,>}$.

**Behaviour of Lindbladian eigenvalues.** By definition, the steady state $\hat{\rho}_0$ is an eigenvector of the Lindbladian with eigenvalue $\lambda_0 = 0$, while $\lambda_1$ determines the asymptotic decay rate as defined above, $\lambda_{\mathrm{ADR}} = -\operatorname{Re}\lambda_1$. For simplicity, we will focus here on the case where $\lambda_1$ is real; see Sec. 6.2.3 below for the case where this is not true. Near to the critical point $\eta = \eta_c$ one can in general expect that these two eigenvectors $\hat{\rho}_{0,1}$ fully control the long time dynamics [539]. That is, one can expect that no other eigenvectors have such small eigenvalues. Further away from $\eta_c$, one may have that the eigenvalue $\lambda_{\mathrm{ADR}}$ associated with $\hat{\rho}_1$ grows to be comparable with other modes, and the two lowest modes can no longer be considered in isolation.

A key question regarding the behaviour near $\eta = \eta_c$ is whether $\lambda_{\mathrm{ADR}}$ vanishes only at a point, or vanishes over an extended region. In Fig. 24 we have suggested the latter case. This would correspond to allowing bistability, where e.g. a system prepared in $\hat{\rho}^<$

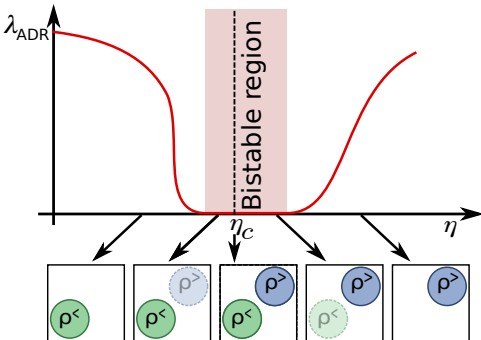

Figure 24: Dissipative phase transition without symmetry breaking, showing evolution of the Lindbladian gap, $\lambda_{\text{ADR}}$ (top) and the steady state density matrix (bottom) in the thermodynamic limit. At the critical point $\eta = \eta_c$ the steady state density matrix becomes a mixture of two states, corresponding to the steady state right before and after the transition. Over a range near this transition, the Liouvillian gap vanishes in the thermodynamic limit (see Fig. 28 for an actual example). This leads to a bistable regime, where a system prepared in the wrong state can persist for a long time (a time diverging as system size approaches the thermodynamic limit). Inspired by Ref. [199].

can survive even when $\eta > \eta_c$ or vice versa. Bistability is a specific case of metastability, where there exists one stable point and one metastable point. We will discuss below how such metastability can be consistent with a unique steady state density matrix $\hat{\rho}_{\text{ss}}$ by considering large but finite system sizes. We further illustrate this with a specific example taken from Ref. [199] shown in Fig. 28. We note that that this argument differs from that in Ref. [199], which suggests that the spectral gap vanishes only at the point $\eta = \eta_c$.

Exactly at the critical point, one may note that the arguments in the previous section imply that both $\hat{\rho}_0$ and $\hat{\rho}_1$ have eigenvalue zero. As such, this means one cannot uniquely determine what is the steady state. As we discuss below, this is an issue with considering the Lindbladian in the thermodynamic limit. We thus next turn to consider finite system size and finite time, which help resolve some points raised in the discussion so far.

**Effects of finite system size and finite time.** In the discussion so far, we have focused on the behaviour in the thermodynamic limit, since it is in that limit that phase transitions are strictly defined. We comment here on how this relates to behaviour in finite systems. This discussion will also clarify the significance of long-lived metastable states as introduced above.

In a finite system we may note that in general, even at $\eta = \eta_c$, one can expect a unique steady state to exist. Exceptions to this exist for models with strong symmetries where conserved quantities exist, in this case even finite-sized models have multiple steady states. To understand how this unique state arises from the two states $\hat{\rho}^{<,>}$, and the evolution of the eigenvalues of the Lindbladian in the finite-sized case, one can consider switching rates between the two states. The term "switching rates" imply a considering in terms of a trajectory-based approach (as discussed in Sec. 3.3). As noted there, different approaches to unravelling can lead to different results; here we assume an approach such as unravelling to describe homodyne detection. In such an approach one would see trajectories that stay around one of the two states, $\hat{\rho}^{<,>}$, but then occasionally switch between these two states—for an example, see Ref. [540]. By assuming this switching is a Poissonian process, one can then extract the switching rates from each state $<,>$ to the alternate state. As we

discuss below, in the large system size limit, switching rates become small, but also the *ratios* of switching rates between the states also become small or large. Metastable states arise when switching from a state has a low rate, so that the state can live for a long time, but when the ratio of switching to and from a state means that the metastable state is disfavoured. In contrast, stable states are those favoured by the switching process.

The trajectory picture is not however required to understand the rates. To make the discussion more concrete, one can consider a classical rate equation the time evolution of probabilities of two states $p_{<,>}$ determined by:

$$\frac{d}{dt}\begin{pmatrix} p_> \\ p_< \end{pmatrix} = \begin{pmatrix} -r_1 & r_2 \\ r_1 & -r_2 \end{pmatrix}\begin{pmatrix} p_> \\ p_< \end{pmatrix}, \tag{165}$$

where $r_1$ describes the rate of switching from $>$ to $<$, and $r_2$ vice-versa. This has the solution:

$$\begin{pmatrix} p_> \\ p_< \end{pmatrix} = \frac{1}{r_1 + r_2}\begin{pmatrix} r_2 \\ r_1 \end{pmatrix} + C\begin{pmatrix} 1 \\ -1 \end{pmatrix}e^{-(r_1+r_2)t}, \tag{166}$$

where $C$ depends on initial conditions. This can be compared to the ensemble-averaged density matrix, which will decay to the stable state at a rate $\lambda_{\mathrm{ADR}}$. This simple model thus has $\lambda_{\mathrm{ADR}} = r_1 + r_2$, and a steady state determined by the ratio $r_1/r_2$.

If one considers how these switching rates vary with system size, one expects the switching rates to decay with increasing system size, so that all rates vanish in the thermodynamic limit. Physically this occurs as states become macroscopically distinct in the large system size limit, thus suppressing transition rates between them. To give a specific example of how this might occur, consider the case where $r_{1,2} = \exp(-c_{1,2}N)$ where $N$ denotes system size. In this case, both $r_1, r_2$ (and thus $\lambda_{\mathrm{ADR}}$) vanish as $N \to \infty$, while $r_1/r_2 = \exp(-(c_1 - c_2)N)$ will either vanish or diverge as $N \to \infty$ depending on the sign of $c_1 - c_2$. In this scenario, for $c_1 < c_2$, $r_1/r_2 \to \infty$, so $\hat{\rho}_<$ is the stable state, while $\hat{\rho}_>$ is metastable, while for $c_1 > c_2$ the situation is reversed. This example explains how a vanishing Lindbladian gap $\lambda_{\mathrm{ADR}}$ over a finite region—which thus implies metastability—can be compatible with a unique steady state. That is, in the thermodynamic limit, metastability does not imply any continuous transfer of weight between the two states or any bimodality. Indeed, only if $r_{1,2}$ have the same functional dependence on $N$ would one expect $r_1/r_2$ to be finite as $N \to \infty$. The distinct behaviour exactly at $\eta = \eta_c$ is that at this point the switching rates become equivalent, making the steady state a mixture of the two competing states. For the example here, where $c_1 = c_2$ one has $r_1 = r_2$ and thus has exactly the equal weight mixture at this critical point.

Considering this example, we may also note that this shows that the limits of long times and large system size do not commute. We may consider the two cases:

**Large system limit first.** In the large system limit, $N \to \infty$ the switching rates between metastable and stable states vanish. Thus there can be multistability where multiple stable or metastable states exist. That is, for any finite $t$ at $N \to \infty$, the solution of Eq. (166) depends on the initial conditions via the constant $C$

**Long time limit first.** In this case, as long as $\lambda_{\mathrm{ADR}} = r_1 + r_2$ is non-zero one will reach a unique steady state, independent of the initial conditions. The discussion of the thermodynamic limit given above (based on Ref. [199]) assumes this limit.

There are several consequences of the arguments above that are worth noting:

- The fact that the switching rates vanish in the thermodynamic limit means that multistability can occur in experiments, i.e. bistability in this specific example. That

is, if one prepares a state $\hat{\rho}^<$ but for parameters where $\eta > \eta_c$ (or vice versa), the state $\hat{\rho}^<$ will persist for a long time, and this timescale will diverge as the system size increases. This means that the bistability predicted by mean-field approaches can have a physical meaning. Metastable states are distinguished from unstable states in that they can appear as long-lived states in a finite-sized system, with a lifetime growing with system size.

- This observation also highlights an important point to note about the density matrix. The density matrix represents the ensemble average over many realisations. This can be related to time averages for ergodic systems. The notion of ergodicity means that the dynamics of a system allow it to explore all possible states. As such, for ergodic systems, long time averages of the dynamics will be equivalent to ensemble averages. However, even when ergodic, the timescale required for the system to explore all states may become large—i.e. the time to switch between distinct solutions may grow (even exponentially) with the size of the system. As such single experiments do not necessarily correspond to the ensemble-averaged density matrix. We discuss other consequences of the distinction between ensemble averages and individual trajectories in Sec. 8.

- Even when considering ensemble-averaged measurements, there are still signatures of such metastability. These arise if one considers the properties of two-time correlations. These allow one to ask the question: "If I measure the system in one state at time $t_0$, how likely am I to measure it in the same state at time $t_1$?". This correlation function can give direct access to the Lindbladian gap $\lambda_{\mathrm{ADR}}$.

- For a finite system, the finite switching rates (in both directions) lead to a smooth crossover in the finite-size steady state. The width (i.e. range of parameter $\eta$) over which this crossover occurs will shrink as the system size increases. One can thus extrapolate to find the expected behaviour in the thermodynamic limit from finite-size experiments.

**Extracting $\hat{\rho}^{<,>}$ from finite-size simulations.** While the finite-size system has a unique steady state, it can be useful to have an estimate for the metastable states $\hat{\rho}^{<,>}$. Reference [199] provides a way to extract this from the first non-zero Lindbladian eigenvector, $\hat{\rho}_1$. This is based on writing this eigenvector as follows:

$$\hat{\rho}_1 = \sum_\mu p_\mu \, |\psi_\mu\rangle\langle\psi_\mu| \simeq \hat{\rho}^< - \hat{\rho}^>. \tag{167}$$

The first expression is just the eigendecomposition of any Hermitian operator, while the second is the expected form of $\hat{\rho}_1$ in terms of the metastable states. Since $\hat{\rho}_1$ is a traceless operator, some of its eigenvalues $p_\mu$ must be negative, and some positive. However, since $\hat{\rho}^{<,>}$ should be valid density matrices, they must both have a positive spectrum. This then suggests separating the sum over alpha into those terms for which $p_\mu$ is positive (or negative) and identify this with $\hat{\rho}^<$ (or $\hat{\rho}^>$). That is:

$$\hat{\rho}^< = \sum_{\mu|p_\mu>0} p_\mu \, |\psi_\mu\rangle\langle\psi_\mu|, \qquad \hat{\rho}^> = \sum_{\mu|p_\mu<0} (-p_\mu) \, |\psi_\mu\rangle\langle\psi_\mu|. \tag{168}$$

After normalisation, these will allow one to extract two valid density matrices $\hat{\rho}^{<,>}$ from a calculation at finite $N$.

A concrete example of finite-size scaling of the Lindbladian gap, the evolution from a crossover to a sharp transition, and the extraction of $\hat{\rho}^{<,>}$ is given in Sec. 6.2.2, in particular Fig. 28.

### 6.1.2 Symmetry-breaking phase transitions

We now turn to phase transitions that show spontaneous symmetry breaking, such as the examples shown in Fig. 23(b,c). When discussing symmetries in open quantum systems, one must distinguish between strong and weak symmetries, as discussed in Sec. 3.1.1. In this section we focus on cases with weak symmetries. Once again we first discuss the behaviour in the thermodynamic limit and then discuss how this emerges from the behaviour at large but finite system sizes.

**Spectral theory in the thermodynamic limit.**  We first discuss discrete symmetries, such as are shown in Fig. 23(b,c) which corresponds to breaking a $\mathbb{Z}_2$ symmetry. Here, for $\eta < \eta_c$ there is a unique steady state, but for $\eta > \eta_c$ the symmetry is spontaneously broken, leading to multiple steady states. While Fig. 23(b,c) suggests two steady states, which we denote $\hat{\rho}^{\pm}$, the full density matrix solution allows any linear combination of these states. Hence, a degenerate family of steady states is possible. This in turn implies that for $\eta > \eta_c$ the Lindbladian must have at least one additional zero eigenvalue (additional to the zero eigenvalue that always exists associated with there being a steady state). This extra zero eigenvalue corresponds to an eigenvector that describes moving within the space of possible steady states.

This argument is similar to that predicting the existence of a Goldstone mode when a continuous symmetry is broken [246, 410], however in the open quantum system a zero eigenvalue exists even if a discrete symmetry is broken. When considering systems with continuous symmetries, a similar situation arises, although now one must consider the possibility of infinitesimal changes to the order parameter symmetry, rather than just switching between discrete states. This leads to a larger degenerate subspace, and thus the possibility of multiple zero eigenvalues.

To summarise, whenever symmetry breaking occurs, the Lindbladian gap must vanish throughout this phase. This is shown in Fig. 25. The vanishing gap implies that all symmetry broken states are equally likely; this contrasts with the bistable system where (apart from exactly at $\eta = \eta_c$) one or other solution is favoured. One may also note that this is what distinguishes the similar structures in Fig. 23(b,d); in panel (b) the two solutions are equivalent so must have equal tunnelling rates, while in panel (d) they are not equivalent[4]. Figure 23(c) shows both symmetry breaking and a possible metastable regime. Here zero eigenvalues can be expected to arise from both these features.

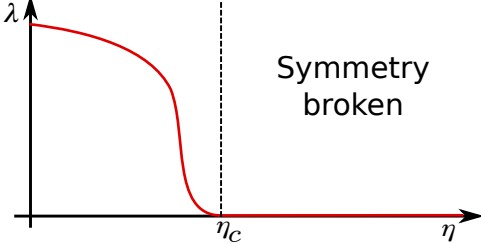

Figure 25: Dissipative Phase Transition with symmetry breaking. In this case, multiple steady states (corresponding to different ways to break the symmetry) exist beyond the critical point. This means the Lindbladian gap vanishes throughout the ordered phase [Inspired by Ref. [199]].

---

[4]We are not aware of particular examples that show the behaviour of panel (d), however there is no obvious reason they should not exist.

**Effects of finite system size.** Similarly to transitions without symmetry breaking discussed above, we may note that in a finite size system, for weak symmetries, there is a unique steady state solution. Moreover, this unique state always respects the symmetries of the equation of motion. This means that in a finite system, as viewed from the properties of the steady state density matrix, there is no spontaneous symmetry breaking. We can however again gain further insight into how the thermodynamic limit emerges by considering the behaviour of the Lindbladian spectrum.

For the case of discrete symmetry breaking we may consider the behaviour of the two lowest eigenvectors of the Lindbladian. As noted above, the steady state will respect the symmetry, and so must take the form $\hat{\rho}_0 = (\hat{\rho}^+ + \hat{\rho}^-)/2$ in terms of the two symmetry-broken solutions $\hat{\rho}^\pm$ introduced above. The other mode (that would have zero eigenvalue in the thermodynamic limit) is the traceless anti-symmetric combination $\hat{\rho}_1 = (\hat{\rho}^+ - \hat{\rho}^-)/2$. This mode corresponds to an eigenvalue that vanishes as system size diverges. The consequence of this mode is that if one prepares one symmetry-broken state at a point where symmetry breaking would occur in the thermodynamic limit, then the finite-size system will only recover the symmetric state very slowly. One can think of this in the same way as for the first-order transition, with switching rates between the two symmetry broken states. A key difference in the symmetry breaking case is that due to symmetry, the switching rates between different metastable states $\hat{\rho}^{<,>}$ are exactly equal. This means that $\hat{\rho}_0$ is always the symmetric sum of symmetry-broken states for all $\eta > \eta_c$, not only at the critical point.

As with the case without symmetry breaking, multi-time correlations can again be used to directly probe the Lindbladian gap. Here, in the ordered state, one can consider preparing one symmetry broken state at $t_0$ and asking whether one finds the same state at a later time $t_1$. A paradigmatic example of this (for continuous symmetry breaking) is the Schawlow–Townes linewidth of a laser [180, 541] (See also Sec. 2.6). Here, the thermodynamic limit of a laser corresponds to breaking a continuous symmetry corresponding to the phase of the emitted light; the linewidth depends on how phase correlations decay with time, thus are a signature of the Lindbladian gap. The fact the linewidth is non-zero indicates a finite correlation time, as expected away from the thermodynamic limit.

### 6.1.3 Critical behaviour and relation to Schwinger–Keldysh field theory

Schwinger–Keldysh field theory—as introduced in Sec. 3.2 and Sec. 4.6 provides a complementary approach to understanding dissipative phase transitions, in particular continuous (i.e. second-order) transitions, and a powerful formalism to study the resulting critical behaviour. Such an approach is reviewed extensively elsewhere [31, 32]. Here we briefly mention some key points, as well as discussing how such an approach links to the spectral theory discussed above.

The key approach to describing second-order phase transitions is the renormalisation group (RG) as introduced in Sec. 4.6.3. The idea underlying this is that at a critical point (i.e. at a second-order phase transition), correlation lengths and timescales diverge [246, 410]. Correlation and response functions (as discussed below) typically depend on a dimensionless length or time, set by the ratio of separation to a typical length and timescale. As such, when these scales diverge, correlation functions take universal forms. The RG approach describes these universal forms on long length and timescales, by integrating out the effects of short wavelength modes. Furthermore, one finds that under the RG, the asymptotic long wavelength form the Keldysh action is defined by the symmetry of the system and the dimensionality. This leads to a notion of Universality: the long wavelength effective actions can be divided into "universality classes" (determined by symmetry and dimensionality), and these in turn determine critical behaviour such as critical

exponents [32].

In discussing critical exponents and correlation lengths, these refer specifically to properties of correlation and response functions. That is, the field theory is considered as a route to calculate the (long wavelength) form of a set of response functions. Because of universality, there are only a finite set of response functions that need to be found to fully determine the behaviour. One such correlation function is susceptibility $\chi$, which determines how the order parameter $\mathcal{O}$ behaves as a function of a symmetry breaking term in the Hamiltonian, $\hat{H} \to \hat{H} + f\hat{O}$, i.e. $\chi = (\partial\mathcal{O}/\partial f)|_{f\to 0}$. For a second-order transition, this susceptibility diverges algebraically as $\eta \to \eta_c$, allowing one to define a critical exponent $\gamma$ via $\chi \propto (\eta_c - \eta)^{-\gamma}$. Further exponents can be defined by considering time-dependent fields and the frequency dependence, or finite-size systems, and the dependence on lengthscales. Other susceptibilities may also be important.

To connect such critical behaviour to spectral theory, we must consider how correlation functions or response functions such as susceptibility are related to the Lindbladian spectrum. For correlation functions, we may note that (as discussed above) multi-time correlation functions can in principle reveal information about the Lindbladian gap. Using the quantum regression theorem [180, 187, 211], a correlation function $\left\langle \hat{X}(t)\hat{X}(0) \right\rangle$ in steady state can be found by taking an initial state $\hat{X}\hat{\rho}_{\text{ss}}$, time evolving this state to time $t$ and then evaluating the expectation value. If one has a full eigensystem of the Lindbladian, i.e. eigenvalues and left- and right-eigenvectors $\lambda_\mu, \langle\langle l_\mu|, |r_\mu\rangle\rangle$, as described in Sec. 3.1.2, the time evolution will take the form:

$$\left\langle \hat{X}(t)\hat{X}(0) \right\rangle = \sum_\mu e^{\lambda_\mu t} \left\langle\left\langle \mathbb{1} \middle| \hat{X}r_\mu \right\rangle\right\rangle \left\langle\left\langle l_\mu \middle| \hat{X}\rho_{\text{ss}} \right\rangle\right\rangle. \tag{169}$$

One thus sees that only those eigenvalues $\lambda_\mu$ for which $\langle\langle l_\mu|\hat{X}\rho_{\text{ss}}\rangle\rangle \neq 0$ will contribute. For response functions, such as susceptibility, these can be calculated using (non-Hermitian) perturbation theory, again making use of the eigensystem of the Lindbladian. If one considers the vectorised Lindbladian in the form $\hat{\mathcal{L}} \to \hat{\mathcal{L}} + \hat{\mathcal{L}}_{pert}$, and calculates the expectation of an observable $\hat{O}$, this gives a steady state expectation:

$$\left\langle \hat{O} \right\rangle = \left\langle\left\langle \mathbb{1} \middle| \hat{O}\rho_{\text{ss}} \right\rangle\right\rangle - \sum_{\mu \geq 1} \frac{1}{\lambda_\mu} \left\langle\left\langle \mathbb{1} \middle| \hat{O}r_\mu \right\rangle\right\rangle \left\langle\left\langle l_\mu \middle| \hat{\mathcal{L}}_{pert}\rho_{\text{ss}} \right\rangle\right\rangle. \tag{170}$$

This shows directly how diverging susceptibility is related to a vanishing Lindbladian gap. Naively this might suggest the critical exponent is determined by how $\lambda_{\text{ADR}}$ depends on $\eta_c - \eta$. However, in general this expression can contain a contribution from many Lindbladian eigenvalues that vanish (or become small) at the critical point. Summing over their contributions can lead to non-trivial (non-integer) critical exponents.

It is worth noting that the spectral theory of transitions discussed in previous sections focuses on the behaviour of a few eigenvectors of the Lindbladian; in particular the properties of *the* eigenvector corresponding to the Lindbladian gap. This single eigenvalue will determine the critical properties transitions in zero-dimensional systems, or those where mean-field theory applies. In general, the critical properties will depend on the spectrum of eigenvalues corresponding to low energy effective modes—i.e. on a set of many eigenvalues, with the number diverging with system size.

One well studied example of this occurs when considering the long-wavelength modes of the driven-dissipative weakly-interacting dilute Bose gas (WIDBG), Eq. (38). These can be found in various ways, including by considering fluctuations around the steady state of the complex Gross-Pitaevskii equation, Eq. (105), which—as discussed earlier—can also be seen as the saddle point of the Schwinger–Keldysh path integral. To see this, one starts

by writing the Bogoliubov–de Gennes parameterisation

$$\psi(r, t) = \left( \psi_0 + \sum_{\mathbf{k}} u_{\mathbf{k}} e^{-i\nu_k t + i\mathbf{k}\cdot\mathbf{r}} + v_{\mathbf{k}}^* e^{i\nu_k^* t - i\mathbf{k}\cdot\mathbf{r}} \right) e^{-i\mu t},$$

where $\psi_0 e^{-i\mu t}$ is a stationary solution, so satisfies $\mu = NU|\psi_0|^2$ and $\kappa_+ - \kappa_- - \kappa_-^{(2)} N|\psi_0|^2 = 0$. One then substitutes this into Eq. (105) and expands up to linear order in the fluctuations $u_{\mathbf{k}}, v_{\mathbf{k}}$ giving coupled equations for the fluctuations:

$$\nu_k u_{\mathbf{k}} = \left( \frac{k^2}{2m} + \mu - i\kappa_{\text{eff}} \right) u_{\mathbf{k}} + (\mu - i\kappa_{\text{eff}}) v_{\mathbf{k}} \tag{171}$$

$$-\nu_k v_{\mathbf{k}} = \left( \frac{k^2}{2m} + \mu + i\kappa_{\text{eff}} \right) v_{\mathbf{k}} + (\mu + i\kappa_{\text{eff}}) u_{\mathbf{k}} \tag{172}$$

where $\kappa_{\text{eff}} = \frac{1}{2}\kappa_-^{(2)} N|\psi_0|^2 = \frac{1}{2}(\kappa_+ - \kappa_-)$. This then leads to the complex dispersion:

$$\nu_k = -i\kappa_{\text{eff}} \pm \sqrt{\frac{k^2}{2m}\left( \frac{k^2}{2m} + 2\mu \right) - \kappa_{\text{eff}}^2}. \tag{173}$$

In the limit $\kappa_{\text{eff}} = 0$ this recovers the standard Bogoliubov dispersion of the equilibrium WIDBG. For the dissipative case, one finds that the dispersion at small $k$ is purely imaginary and diffusive, i.e. $\nu_k \simeq -iD_{\text{eff}} k^2 + \mathcal{O}(k^4)$ for the + root, and $\nu_k \simeq -2i\kappa_{\text{eff}}$ for the − root. As $k$ increases, there is an exceptional point, beyond which the imaginary parts of the two roots coalesce, and a real part of the dispersion arises [154, 155]. Such behaviour is characteristic of the Goldstone mode in many driven-dissipative systems.

Recent experiments on exciton polaritons have observed such a diffusive Goldstone mode [542]. Similar results also apply for cold atoms in multimode optical cavities, where the the Goldstone mode associated with crystallisation—i.e. a phonon mode—has been observed [111]. The distinct properties of the diffusive Goldstone mode can change the critical behaviour associated with symmetry breaking transitions in such driven dissipative systems.

Renormalisation group approaches are a key tool to account for the effects of this continuum of low energy modes. This is discussed further in the reviews [31, 32, 159].

## 6.2   Examples

Having discussed the general properties of dissipative phase transitions, we here give a few illustrative examples of models in which dissipative phase transitions have been studied.

### 6.2.1   Dissipative Phase Transitions in Spin Chains

The first example we consider is spin-chain models. A notable feature these models illustrate is that DPT can display patterns of symmetry breaking which are completely different from the equivalent Hamiltonian in thermal equilibrium. A particularly clear application of this is presented in Ref. [227] for a dissipative XYZ model. This model (as already introduced in Sec. 3.1.5) is given by the Hamiltonian

$$\hat{H} = \sum_{\langle i,j \rangle} \sum_{\alpha} J_\alpha \hat{\sigma}_i^\alpha \hat{\sigma}_j^\alpha \tag{28}$$

along with jump operators

$$\hat{L}_{i-} = \sqrt{\gamma}\hat{\sigma}_i^- . \tag{174}$$

The ground state of this Hamiltonian is a standard example of quantum magnetism in condensed matter physics. Except in one dimension, the ground state of this model shows long-range magnetic order. The nature of that order—ferromagnetic vs antiferromagnetic—is determined by the sign of the dominant coupling $J_x, J_y, J_z$. While some patterns of coupling can lead to more complex phenomena such as ferrimagnetic order, small changes to ratio of $J_x/J_y$ do not generally change the nature of the order unless the signs change. As we discuss below, and as illustrated by the phase diagram in Fig. 26, this is quite different for the dissipative model.

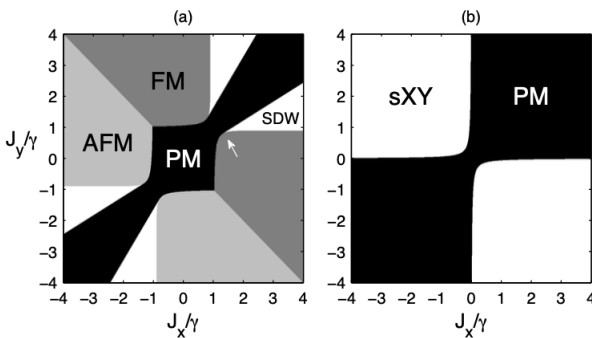

Figure 26: Nonequilibrium mean-field phase diagram of the XYZ model as a function of the anisotropy $J_x \neq J_y$, obtained respectively with $J_z = \gamma$ (left) and $J_z = 0$ (right). We see that along the line $J_x = J_y$ the model displays a regular paramagnetic phase which gives rise to a rich pattern of broken symmetry phases in presence of anisotropy. From Ref. [227] [Copyright (2013) by the American Physical Society].

In Ref. [227], this model was studied using a real-space mean-field factorisation. This leads to the following equations of motion for the local magnetisation $\langle \hat{\sigma}_i^\alpha \rangle$:

$$\frac{d\langle \hat{\sigma}_i^x \rangle}{dt} = -\frac{\gamma}{2}\langle \hat{\sigma}_i^x \rangle + \sum_j \left( J_y \langle \hat{\sigma}_i^z \rangle \langle \hat{\sigma}_j^y \rangle - J_z \langle \hat{\sigma}_i^y \rangle \langle \hat{\sigma}_j^z \rangle \right), \tag{175a}$$

$$\frac{d\langle \hat{\sigma}_i^y \rangle}{dt} = -\frac{\gamma}{2}\langle \hat{\sigma}_i^y \rangle + \sum_j \left( J_z \langle \hat{\sigma}_i^x \rangle \langle \hat{\sigma}_j^z \rangle - J_x \langle \hat{\sigma}_i^z \rangle \langle \hat{\sigma}_j^x \rangle \right), \tag{175b}$$

$$\frac{d\langle \hat{\sigma}_i^z \rangle}{dt} = -\gamma \sum_i \left( 1 + \langle \hat{\sigma}_i^z \rangle \right) + \sum_j \left( J_x \langle \hat{\sigma}_i^y \rangle \langle \hat{\sigma}_j^x \rangle - J_y \langle \hat{\sigma}_i^x \rangle \langle \hat{\sigma}_j^y \rangle \right). \tag{175c}$$

This dynamics always admits a steady state, corresponding to $\langle \hat{\sigma}_i^x \rangle = \langle \hat{\sigma}_i^y \rangle = 0$ and $\langle \hat{\sigma}_i^z \rangle = -1$. That is, a state with all the spins down along the z axis, with this axis chosen by the dissipative processes. However linear stability analysis around this steady state reveals a rich pattern of symmetry broken phases, distinct from those in the ground state of the Hamiltonian.

As discussed in Sec. 3.1.5, to reach a non-trivial state requires we consider $J_x \neq J_y$. This is because for $J_x = J_y = J$ the Hamiltonian can be rewritten as

$$\hat{H} = \sum_{ij} J \left( \hat{\sigma}_i^+ \hat{\sigma}_j^- + \text{H.c.} \right) + J_z \hat{\sigma}_i^z \hat{\sigma}_j^z \tag{176}$$

which conserves the total number of excitations $\sum_i \hat{\sigma}_i^z$. Since the dissipation leads to

loss of excitations, this means that for $J_x = J_y$ the only steady state is the *dark* state $\hat{\rho} = |\downarrow \ldots \downarrow\rangle\langle\downarrow \ldots \downarrow|$.

For $J_x \neq J_y$ excitation number is not conserved because of the reduced symmetry. This allows coherent excitation creation to balance the loss. This process can be seen from the mean-field dynamics which can be described as a competition between dissipative processes, taking the local spin toward pointing down, and precession around a time-dependent field. The effective field seen by each site comes from the neighbouring sites and takes the form

$$h_i^{\text{eff}} = \sum_{j \in \partial_i} (J_x \langle \hat{\sigma}_j^x \rangle, J_y \langle \hat{\sigma}_j^y \rangle, J_z \langle \hat{\sigma}_j^z \rangle),$$

using the notation of Sec. 4.3.1. Now, the key point is that for isotropic exchange and homogeneous order this field is always parallel to the spin direction itself, so there is no precession and dissipation always wins. However for $J_x \neq J_y$ the spin can have an arbitrary angle with respect to the field and therefore counterbalance the losses, so that new stationary states can emerge.

The discussion so far has been based on the mean-field theory, which, as discussed in Sec. 4.4 is expected to be valid in sufficiently high dimensions. Following Ref. [227] there have been many attempts to explore the phase diagram of this model beyond mean-field theory. One such example, Ref. [276], used the Cluster Mean-Field techniques described in Sec. 4.5.1. This revealed a striking effect arising from the nonequilibrium nature of the problem: short-ranged correlations, disregarded in the single-site mean-field approximation leading to Eq. (175a)–(175c), induce crucial changes to the topology of the phase diagram. In particular, the ferromagnetic phase that previously extends to arbitrarily large $J_y, J_x$ is instead found only for a finite range of couplings, and instead a re-entrant paramagnetic phase is seen at larger couplings. This is shown in Fig. 27.

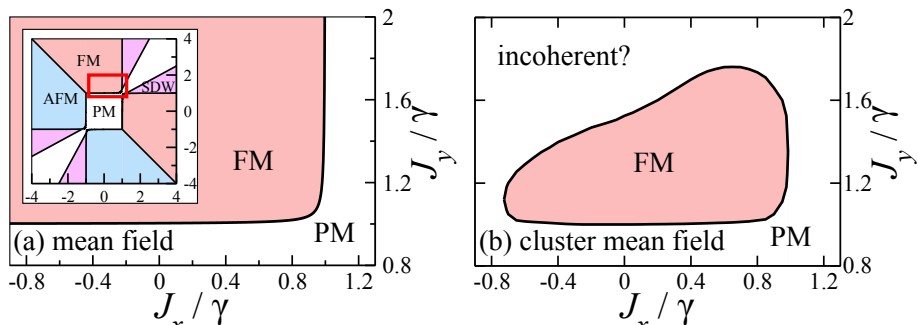

Figure 27: Role of short-range correlations on the phase diagram of the dissipative XYZ model, comparing (a) mean-field predictions vs (b) cluster mean-field theory in two dimensions. Adapted from Ref. [276].

Such significant changes due to short-range interactions would be highly unusual in ground-state or thermal-equilibrium phase diagrams. The origin of this effect can be seen to be rooted in the driven dissipative nature of the problem. In fact, in the limit of $J_y \to \infty$, the steady-state of the system becomes fully mixed and the purity drops to zero. This in turn leads to suppression of the magnetisation.

Further works on the dissipative XYZ model have used other methods to investigate the DPT in one or two dimensions, beyond mean-field. The Keldysh approach combined with renormalization group was used in [543] to study the critical state of an Ising model with long-range losses. References [321,375] used a tensor network technique: infinite projected entangled pair operators (iPEPO) while Ref. [320] used the corner-space renormalisation

group. In addition, Ref. [540] discussed the dissipative transition using quantum trajectories by extracting the Lindbladian gap and discussing signatures of critical slowing down. All these studies have highlighted how the DPT is washed out in one dimensional systems while it survives in two-dimensional lattices, although a conclusive results on the location of the phase boundaries is yet to be found.

Another interesting limit of the model we have discussed corresponds to $J_z = 0$, where mean-field theory predicts a staggered XY phase for any finite value of the anisotropy and a featureless paramagnetic phase when the couplings $J_{x,y}$ have the same sign. The effects of beyond-mean-field fluctuations on this result have been explored using Keldysh field-theory techniques [412] which predict a washing out of the quasi-long range order in two-dimensions. In addition to the features seen for the model discussed above, other models of dissipative spin chains have been discussed, which show a wider variety of phases. In particular, several such models [230, 327, 329, 357, 525, 544] observed transitions to a state which, in mean-field theory, corresponds to a periodic dynamics. As discussed further below, such a state corresponds to the occurrence of a time crystal, see Sec. 6.2.3.

### 6.2.2   Coherent Drive and Bistability

We now discuss the cases in which there are first-order, i.e. discontinuous, DPTs without symmetry breaking. As noted at the start of this section, semiclassical analysis suggests bistability in such cases, but the full treatment reveals a unique steady state at each point in the thermodynamic limit. Here we show some concrete examples of such behaviour.

**Kerr nonlinear oscillator.**   A paradigmatic example of a first order DPT occurs in the context of a driven-dissipative non-linear oscillator, as discussed in Ref. [199]. This is described by Hamiltonian and loss terms:

$$\hat{H} = -\Delta \hat{a}^\dagger \hat{a} + \frac{U}{2}\hat{a}^\dagger \hat{a}^\dagger \hat{a} \hat{a} + F\left(\hat{a}^\dagger + \hat{a}\right), \qquad \hat{L} = \sqrt{\kappa}\hat{a}. \tag{177}$$

While this is a single-mode problem without an immediately apparent thermodynamic limit, it has been shown that a well-defined thermodynamic limit can be still defined by taking $F \to \infty$ at fixed $UF^2$. In this limit one finds that the number of bosons has a discontinuity as one tunes the ratio of driving and loss.

To explore the approach to this thermodynamic limit, one can define $U = \tilde{U}/N, F = \tilde{F}\sqrt{N}$. The behaviour of the photon number and eigenvectors of the Lindbladian is shown in Fig. 28, showing how a sharp transition emerges as $N$ increases. One important feature of this model is that it shows an extended range of $F$ over which $\lambda_{\text{ADR}}$ decreases with increasing system size. While $\lambda_{\text{ADR}}$ is minimum at $\eta = \eta_c$, it shows strong dependence on system size over an extended region. This figure strongly suggests that $\lambda_{\text{ADR}}$ vanishes over an extended region.

**First-order transitions in spin lattices.**   Classical bistability, and the associated first-order transition in the full dynamics, has also been explored in models of spins on lattices, which more directly provide a many-body context and thermodynamic limit. Here we discuss two examples. The first is a coherently driven XY quantum spin model on a $d-$dimensional hypercubic lattice, with Hamiltonian

$$\hat{H} = -J \sum_{\langle i,j \rangle} \left(\hat{\sigma}_i^+ \hat{\sigma}_j^- + \text{H.c.}\right) + \sum_i \left(\frac{\Delta}{2}\hat{\sigma}_i^z + \Omega\hat{\sigma}_i^x\right) \tag{178}$$

where $\Omega, \Delta$ are respectively the strength of the local drive and the detuning, and local jump operators describing dissipation $\hat{L}_{i-} = \sqrt{\gamma}\hat{\sigma}_i^-$. This model was studied in Ref. [388]

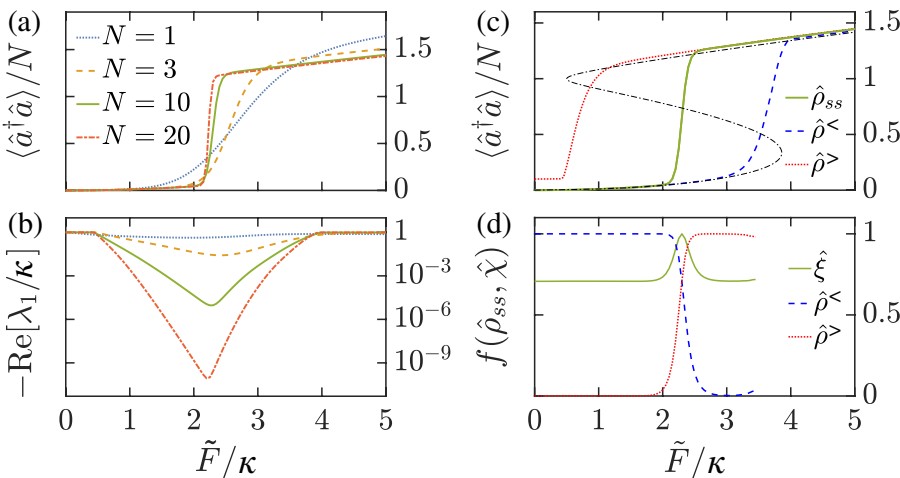

Figure 28: Dissipative phase transition in a Kerr non-linear resonator, as defined in Eq. (177). Left panels, (a,b) shows the system-size dependence (i.e. $N$ dependence) of the evolution vs rescaled pump strength $\tilde{F}$. Panel (a) shows the number of photons $\langle \hat{a}^\dagger \hat{a} \rangle$ in the cavity, and panel (b) shows the Lindbladian gap $\lambda_1 \equiv \lambda_{\mathrm{ADR}}$ plotted on a logarithmic scale. As discussed above, there is an extended range of $\tilde{F}/\kappa$ over which $\lambda_1$ is strongly dependent on system size, and for which one can anticipate $\lambda_1 \to 0$ in the limit $N \to \infty$. The right panels, (c,d) compare the true steady state solution to the two solutions $\hat{\rho}^{<,>}$ extracted from finite-size calculations according to Eq. (167). Panel (c) shows the evolution of the photon number in comparison to the semiclassical solution, which is shown as a dashed line. One may see that over some range of $\tilde{F}$, $\hat{\rho}^{<,>}$ follow the two branches of the semiclassical solution. Panel (d) shows the fidelity between the steady-state density matrix $\hat{\rho}_{\mathrm{ss}}$ and the states $\hat{\rho}^{<,>}$ and $\hat{\xi} = (\hat{\rho}^< + \hat{\rho}^>)/2$, using the fidelity $f(\hat{\rho}_A, \hat{\rho}_B) = \mathrm{Tr}\left[\sqrt{\sqrt{\hat{\rho}_A}\hat{\rho}_B\sqrt{\hat{\rho}_A}}\right]$. From Ref. [199] [Copyright (2018) by the American Physical Society].

using matrix product operator (MPO) simulations, in $d = 1, 2$, and an approximation scheme that accounts for fluctuations beyond mean-field (MFQF) in $d = 1, 2, 3$, equivalent to the cumulant expansion discussed in Sec. 4. Both methods were compared with mean-field theory, which predicts a bistable region for the steady-state magnetisation $\mu_\alpha = \frac{1}{N_s}\sum_i \mathrm{Tr}\left[\hat{\rho}_{ss}\hat{\sigma}_i^\alpha\right]$, as a function of the detuning $\Delta$. In one dimension both the MPO and the MFQF results show that bistability is destroyed by quantum fluctuations, a result which is consistent with that found in other coherently driven-dissipative models (see below). In two dimensions however bistability is found to survive in MFQF, although in a reduced region of parameter space, as shown in the left panel of Fig. 29. The MPO results are obtained on small cylinders, mapping the two-dimensional problem to a one-dimensional chain. These compare well with MFQF over a large region of detunings, however they show a smooth crossover instead of true bistability, see left panel of Fig. 29.

A second example is the transverse-field Ising chain with Hamiltonian and jump operators

$$\hat{H} = \frac{V}{4}\sum_{\langle i,j \rangle} \hat{\sigma}_i^z \hat{\sigma}_j^z + \frac{g}{2}\sum_i \hat{\sigma}_i^x, \quad \hat{L}_{i-}^z = \frac{\sqrt{\gamma}}{2}\left(\hat{\sigma}_i^x - i\hat{\sigma}_i^y\right), \quad \hat{L}_{i+}^x = \frac{\sqrt{\gamma}}{2}\left(\hat{\sigma}_i^y + i\hat{\sigma}_i^z\right). \quad (179)$$

This model was studied in Ref. [546], comparing mean-field results to cluster-mean-field theory in two dimensions, as shown in the right panel of Fig. 29. Once again mean-field

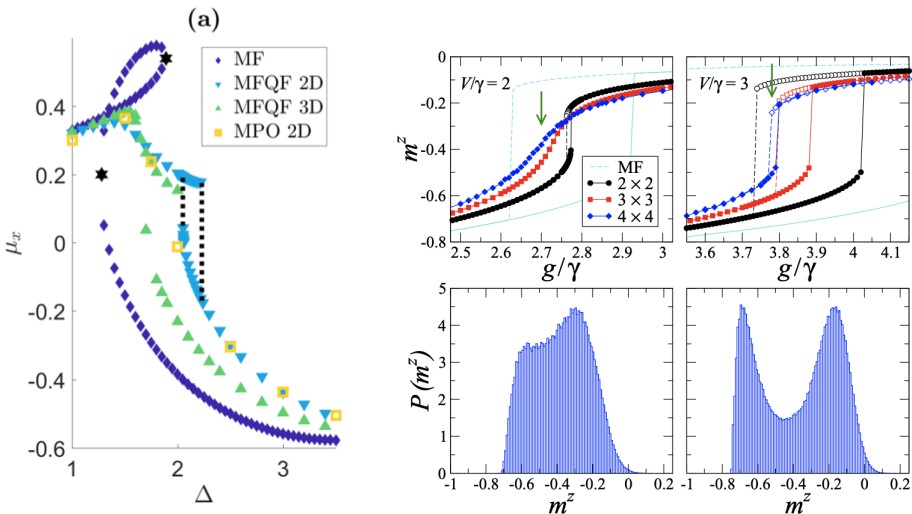

Figure 29: Bistability in coherently driven spin chains. Left Panel: XY model with longitudinal and transverse field. Average steady-state magnetisation $\mu_x$ for mean-field (MF), mean-field plus quantum fluctuations (MFQF) in $d = 2, 3$, and matrix product operator (MPO) on small two-dimensional lattices of size $12 \times 4$. From Ref. [545]. Right Panel: Ising Model in a Transverse Field. In both cases MF bistability is reduced by the addition of quantum fluctuations. From Ref. [546].

theory predicts bistability, while the cluster-mean-field theory shows either a crossover in place of a transition, or a sharp first-order transition.

As discussed in Sec. 6.1, the survival of bistability beyond mean-field theory can be understood as a consequence of considering the large system size limit before the long time limit. That is, when the large time limit is taken first, the smooth crossover at finite size turns into a sharp first order phase transition in the thermodynamic limit as discussed in [199]. However, in the opposite case, where the large system size limit is taken first, the lifetime of the two metastable states diverges, leading to bistability. Signatures of such bistable behaviour appear in the unique steady state at finite $N$, which display a bimodal distribution of local observables [388, 546].

There have been a large number of works, using a variety of methods, studying other examples of spin lattices (as well as other lattice based problems). Such problems provide a natural context to compare the predictions of real-space mean-field theory to beyond-mean-field treatments. In one-dimensional lattices with nearest-neighbour interactions, the mean-field bistability is found to be replaced by a crossover driven by large quantum fluctuations [515, 547–550]. That is, in one-dimensional spin-chains, there generally is often no DPT (however see Sec. 6.2.4 for a notable counter example). For two-dimensional lattices, a DPT does exist beyond mean field, but (as discussed above), mean-field bistability is replaced by a first-order transition. Examples of this include nonlinear bosons using a truncated Wigner approximation [515, 550], Ising spins using a variational ansatz accounting for short-range correlations [547], cluster mean-field approaches [546], and two-dimensional tensor networks (iPEPO) [321, 375, 376]. In addition to finding the steady state, one can also study the dynamics, and it has been seen that around the DPT the rate of convergence towards the steady state slows down [515, 550], as expected from the spectral theory outlined above [199, 539, 551].

### 6.2.3    Boundary Time-Crystal and Dissipative Limit-Cycles

We next discuss the DPT to a state that breaks time-translation symmetry, i.e. a *time crystal* or *limit cycle state*. As explained in Ref. [524], time-crystal states in a dissipative system can be considered as corresponding to a "boundary time crystal", a name referring to the fact that the open quantum system can be thought of as the "boundary" of the environment. Such a consideration is important because there is a well established no-go theorem [523] that prevents a time crystal arising in the ground state of a quantum system. This no-go theorem should apply to the combined system and environment, but the reason why a time crystal is nonetheless possible in the open system is that the no-go theorem prevents periodic dynamics for macroscopic observables in the system; the boundary however corresponds to a vanishing fraction of the combined system and environment. As such, periodic dynamics of the boundary does not violate the no-go theorem. For a complete discussion of this subtle point see Ref. [524].

In order to make a connection with the discussion presented in Sec. 5, time-crystals prepared via dissipation can be considered as an example of state preparation where in the steady state a dark-subspace (rather than a dark state) is selected. This fact, that we are now going to extensively discuss, opens very interesting research perspectives due to the possible relations of boundary time crystals with decoherence-free subspaces and noiseless subsystems [552]. Recent experimental detection of dissipative time-crystals has been reported in [553, 554]

**Collective spin model.**    As a first example we consider the following Hamiltonian and jump operator:

$$\hat{H} = \omega_0 \hat{S}^x + \frac{\omega_x}{S}(\hat{S}^x)^2 + \frac{\omega_z}{S}\left(\hat{S}^z\right)^2, \qquad \hat{L} = \sqrt{\frac{\kappa}{S}}\hat{S}^- \tag{180}$$

describing a collection of two-level systems, with $\hat{S}^\alpha = \sum_{j=1}^{N}(1/2)\hat{\sigma}_j^\alpha$ with $\alpha = x, y, z$. The total spin $S$ is conserved by the master equation and we fix it equal to the maximum value $S = N/2$. For simplicity we focus here on the case $\omega_x = \omega_z = 0$ where there is only one control parameter in the problem, namely the ratio $\eta \equiv \omega_0/\kappa$. The general case is discussed in the supplement of Ref. [524].

Figure 30 shows the Lindbladian eigenspectrum, and the real-time dynamics of the spin. The spectrum (calculated for finite N) shows signatures of what will become a phase transition at $\eta = \eta_c \equiv 1$. Below the transition, a unique steady state exists, corresponding to an isolated zero mode in the spectrum. Above the transition, there is instead persistent oscillations. At finite $N$, this appears at modes at finite frequency (finite imaginary part of $\lambda$) and small real part. As $N \to \infty$ the real part vanishes and the oscillations become long lived. The same behaviour can be seen by viewing the dynamics in the time domain as shown in the lower panel of Fig. 30.

One can understand what happens in the thermodynamic limit, corresponding to the limit of large spin $S = \to \infty$, by using a semiclassical approximation, replacing expectations of products of operators by products of expectations. Writing $S_\alpha = \left\langle \hat{S}^\alpha \right\rangle$ we obtain classical equations of motion:

$$\frac{d}{dt}S_x = \frac{\kappa}{S}S_x S_z \tag{181a}$$

$$\frac{d}{dt}S_y = -\omega_0 S_z + \frac{\kappa}{S}S_y S_z \tag{181b}$$

$$\frac{d}{dt}S_z = \omega_0 S_y - \frac{\kappa}{S}\left(S_x^2 + S_y^2\right) \tag{181c}$$

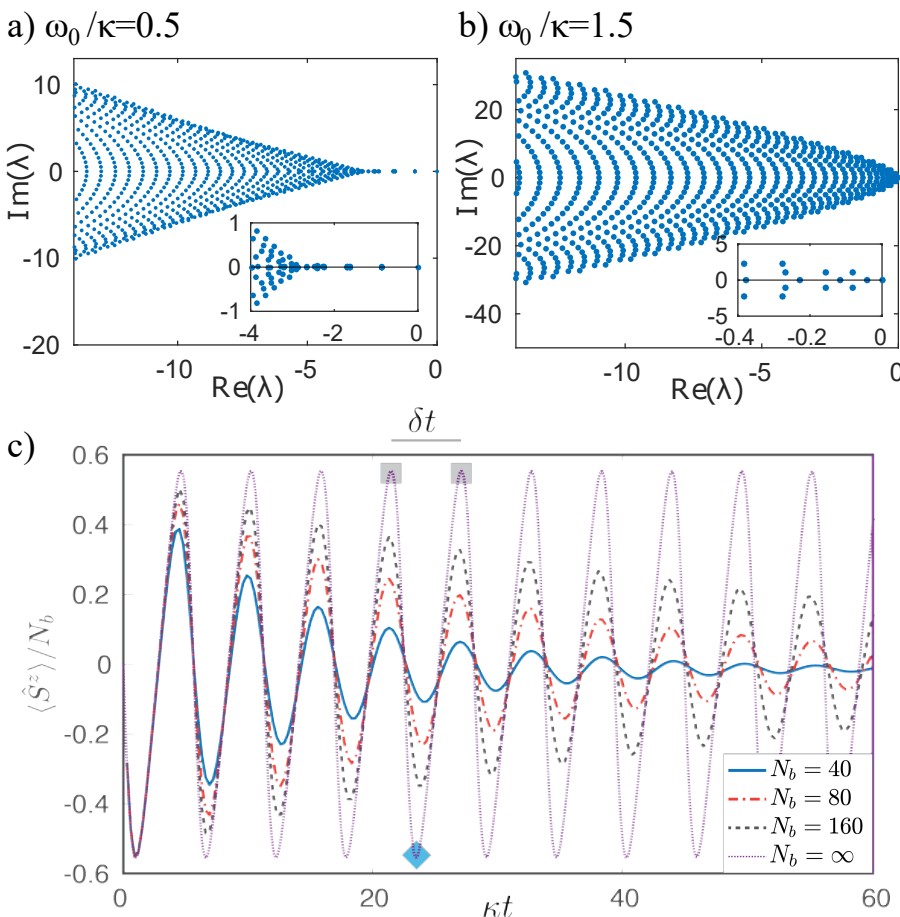

Figure 30: Boundary Time-Crystal. Top: Lindblad spectrum across a dissipative time crystal phase transition, where time-translation invariance is spontaneously broken. In panel (a), for $\omega_0/\kappa = 0.5$ we see a finite dissipative gap, corresponding to a well defined steady state and a dynamics which leads to damping. In panel (b), for $\omega_0/\kappa = 1.5$ the Lindbladian spectrum becomes dense near $\lambda = 0$, acquiring also a finite imaginary part. In the thermodynamic limit this gives rise to spontaneous oscillations. Bottom: Time evolution of $\left\langle \hat{S}^z \right\rangle$ showing long-lived oscillations at finite $N$, and the corresponding thermodynamic limit, plotted for $\omega_0/\kappa = 1.5$. From Ref. [524].

In addition to the conserved magnitude of spin (as discussed above), there is a second conserved quantity $M = S_x/S_y - \omega_0/\kappa$. This conservation law has important consequences on the dynamics of the system.

Considering the stationary sates, i.e. $\frac{d}{dt}S_\alpha = 0$ we find that for $\eta = \omega_0/\kappa \leq 1$ the steady states are

$$(S_x, S_y, S_z) = S(0, \eta, \pm\sqrt{1 - \eta^2}).$$

Linear stability analysis shows that the solution $S_z = -\sqrt{1 - \eta^2}$ is stable. Such stability analysis also gives the time to reach the stationary state, $\tau = 1/\kappa\sqrt{1 - \eta^2}$. These results suggest a transition at $\eta = 1$. For $\eta > 1$ one can find fixed point solutions

$$(S_x, S_y, S_z) = S(\pm\sqrt{1 - 1/\eta^2}, 1/\eta, 0)$$

however these are *not stable*. Linear stability analysis show that fluctuations about these

have imaginary eigenvalues! This corresponds to a Hopf bifurcation, with a finite-frequency instability. Typically such Hopf bifurcations lead to dynamics with a limit cycle as an attractor. While we indeed see a periodic attractor, it is not a limit cycle [555], in that there is not a unique periodic attractor, but instead multiple periodic attractors. Nonetheless, as seen in Fig. 30(c), there is periodic behaviour seen at late times (which survives to longer and longer times as $N \to \infty$), hence this state can be understood as a time crystal.

**Dissipative Bose–Hubbard model.** A different example of DPT with breaking of time-translation invariance occurs for models in which the semiclassical equations display a limit cycle phase, and exploring how this phase is affected by quantum fluctuations. Examples that fall into this class include semiclassical models of lasing (see Sec. 2.6) and optical parametric oscillators (See for example [556, 557]). More recently the focus has shifted on lattice models of driven-dissipative systems, such as the driven-dissipative Bose–Hubbard model which we discuss now. There have been many forms of driven-dissipative Bose–Hubbard models studied, see for example Refs. [426, 558, 559]. In this section we focus on a particular form of driving and dissipation introduced in Ref. [277] . As discussed previously the Bose–Hubbard model model has the Hamiltonian:

$$\hat{H} = -t_H \sum_{\langle i,j \rangle} \left( \hat{a}_i^\dagger \hat{a}_i + \text{H.c.} \right) + \frac{U}{2} \sum_i \hat{n}_i(\hat{n}_i - 1), \tag{182}$$

and the driven-dissipative version we consider here is formed by considering two kinds of jump operators for each lattice site $i$,

$$\hat{L}_{i+(2)} = \sqrt{\kappa_-^{(2)}} \, (\hat{a}_i)^2 \,, \qquad \hat{L}_{i-} = \sqrt{\kappa_+^{(1)}} \, \hat{a}_i^\dagger. \tag{183}$$

Such a model has both time translation invariance and a weak $U(1)$ symmetry $\hat{S} = \prod_i e^{i\theta \hat{n}_i}$.

The semiclassical equations of motion for this model—which should hold for large numbers of bosons per site—correspond to a lattice version of the Gross–Pitaevskii equation. The semiclassical equations predict a coherent phase for any non-zero pumping, $\kappa_+^{(1)} > 0$, independent of $t_H/U$. The coherent phase corresponds to phase locked oscillations of the coherent field on each site, i.e. $\langle \hat{a}_i \rangle = |\psi_0| e^{i\mu t}$. As such, this coherent phase breaks both time translation and the $U(1)$ symmetry, and so this model is a candidate to understand how quantum fluctuations modify this phase.

Figure 31 shows results from Ref. [277], calculating the phase diagram of this model using DMFT as introduced in Sec. 4.4. This shows a phase transition from a normal state at small $t_H$ to the superfluid state which breaks time translation symmetry at large $t_H$. The incoherent normal state at small $t_H$ is consistent with what is known [298, 560, 561] for a individual nonlinear bosonic modes with one-photon pumping and two-photon loss. As discussed in Sec. 4.4, we rescale the hopping as $t_H = \tilde{t}_H/z$, and show results as a function of the coordination number $z$. In addition to DMFT results, the Gutzwiller mean-field phase boundary (i.e. assuming real-space factorisation of the density matrix) is also shown. As expected, one sees that the DMFT results converge on this result in the $z = \infty$ limit. The DMFT results show a strong dependence on the connectivity $z$ indicating a strong effect of fluctuations; one may note that this dependence is considerably stronger than has been seen for the Bose–Hubbard model in equilibrium. [341, 351, 352].

### 6.2.4 Absorbing state phase transitions

An intriguing class of dissipative phase transitions is that involving absorbing states [138, 521, 562]. This idea, first discussed in classical statistical mechanics [563], describes models

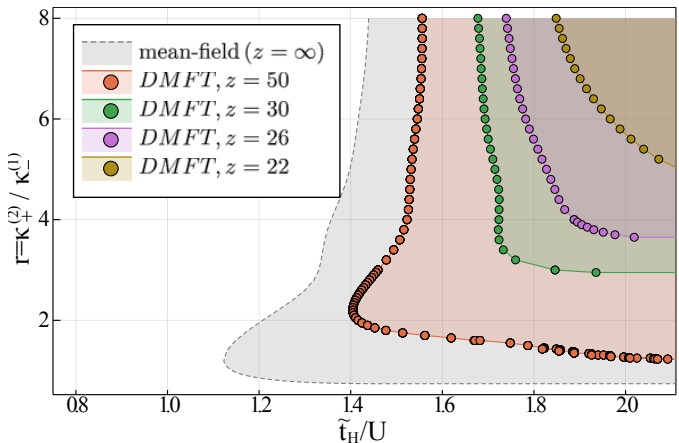

Figure 31: Phase Diagram of the driven-dissipative Bose–Hubbard model found using DMFT, as a function of pump/loss ratio $r = \kappa_+^{(1)}/\kappa_-^{(2)}$ and hopping/interaction ratio $\tilde{t}_H/U$ where $t_H = \tilde{t}_H/z$, for different values of the lattice connectivity $z$. These results are compared to the Gutzwiller mean-field theory that should hold at $z = \infty$. DMFT results are shown only for large values of $z$, as for smaller $z$, the computation becomes costly at large pumping, due to the large Hilbert space dimension. Adapted from Ref. [277].

where there is a state that, once reached, can never be left. Such states have close connections to dark states as discussed above, and in some cases are equivalent. We discuss below an example extending this concept to dissipative quantum systems.

Reference [521] introduced a quantum model that extends a standard classical model of absorbing state transitions. The classical model involves a lattice of sites in a ground or excited state, with processes where an excitation may decay, as well as "activated" processes where an excitation may split in two ("branching") or two neighbouring excitations may merge into one ("coagulation"). Activated processes refer to changes to the state of one site conditional on the site of other sites.

A quantum extension of this model can be written in terms of a lattice of two-level systems represented by Pauli operators. It is convenient to define $\hat{n}_i = \hat{\sigma}_i^+ \hat{\sigma}_i^-$, the projector onto the excited state of site $i$, so one may then write the Hamiltonian and jump operators as:

$$\hat{H} = \Omega \sum_{\langle i,j \rangle} \hat{n}_j \hat{\sigma}_i^x, \quad \hat{L}_{i,-} = \sqrt{\gamma}\hat{\sigma}_i^-, \quad \hat{L}_{\langle i,j \rangle, C+} = \sqrt{\kappa}\hat{n}_j \hat{\sigma}_i^-, \quad \hat{L}_{\langle i,j \rangle, C-} = \sqrt{\kappa}\hat{n}_j \hat{\sigma}_i^+. \quad (184)$$

The Hamiltonian describes a spin flip on site $i$, conditioned on the states of its neighbours $j$, i.e. a quantum version of an activated process. The three types of jump operators describe single-site decay $\hat{L}_{i,-}$ and two activated processes that act on pairs of nearest-neighbour sites. Process $\hat{L}_{\langle i,j \rangle, C+}$, is the classical branching process, creating an excitation on site $i$ conditioned on there being excitations on a neighbouring site $j$. The inverse coagulation process $\hat{L}_{\langle i,j \rangle, C-}$ destroys an excitation conditioned on an excitation on neighbouring site. These processes are illustrated in Fig. 32(a).

The model in Eq. (184) includes the classical model [563] in the limit $\Omega = 0$. It is clear to see that in general this model has a dark state $\hat{\rho} = \prod_i |\downarrow\rangle\langle\downarrow|_i$, which is equivalent to the absorbing state for the corresponding classical model. For some parameters, there can also be a second non-trivial state with a finite density of excitations, so $n = \langle \hat{n}_i \rangle$ serves as order parameter for this phase transition. It is notable that $\prod_i |\downarrow\rangle\langle\downarrow|_i$ is always a dark state of

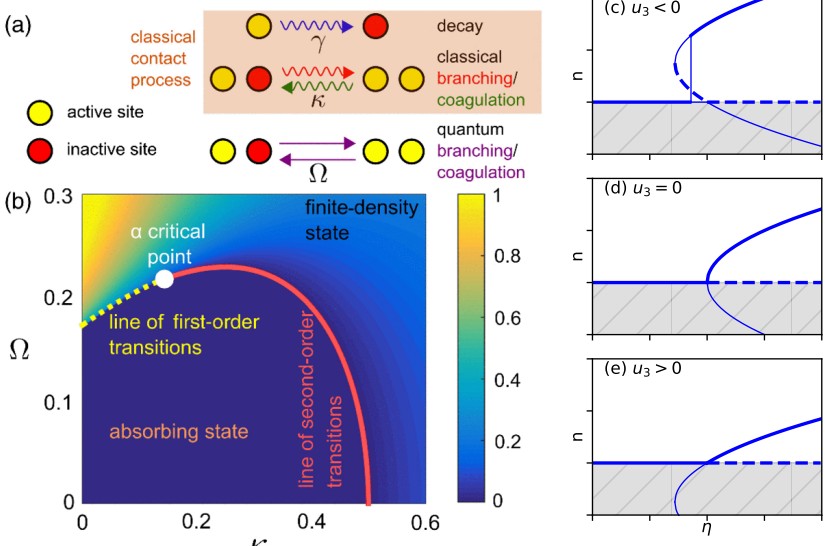

Figure 32: Illustration of absorbing state phase transition. Panel (a) illustrates the processes involved in the model, as defined in Eq. (184). Panel (b) shows the phase diagram of the resulting model. The axis $\Omega = 0$ corresponds to the purely classical model. As illustrated this shows a critical point in which the phase diagram changes from first order to second order. Panels (c,d,e) show the order parameter (excitation density) as a function of the tuning parameter $\eta$ (i.e. the analogue of Fig. 23) for three scenarios corresponding to (c) discontinuous transition, (d) critical point, and (e) continuous transition. As discussed in the text, only positive values of $n$ are physical [Panels (a,b) from Ref. [521] - Copyright (2016) by the American Physical Society.].

the model for any parameters. This therefore differs from the generic spectral theory of phase transitions discussed earlier.

As discussed in Ref. [521], the phase transition at general $\Omega$ can be understood from a saddle-point analysis of a Keldysh action. This reveals an effective action that can be written in the form $\Gamma(n) = -\eta n^2 + u_3 n^3 + u_4 n^4$ with $\eta, u_3, u_4$ being combinations of the model parameters $\Omega, \kappa, \gamma$ along with the lattice coordination number $z$. When $u_3 = 0$, this model has a continuous phase transition with a critical point at $\eta = 0$, of the form as shown in Fig. 23(b). There is however a difference in that only $n \geq 0$ corresponds to physical states, hence the model does not show any symmetry breaking, even in this apparently symmetric case. When $u_3 > 0$ the transition remains continuous at $\eta = 0$, but for $u_3 < 0$ multiple saddle points appear indicating a discontinuous transition. These three scenarios are illustrated in Fig. 32(c,d,e). Using the form of the coefficients $\eta, u_3$ in terms of the bare parameters (as given in Ref. [521]), one finds that in the classical limit $u_3 > 0$, but increasing $\Omega$ and decreasing $\kappa$ can make $u_3$ change sign, leading to the phase boundary shown in Fig. 32(b).

A remarkable feature of absorbing state phase transitions is that they can still show phase transitions in one-dimensional systems. This was studied in detail in Ref. [138], which considered the $\kappa = 0$ limit of the model defined above. The saddle-point analysis summarised above suggests a first-order transition would occur in this limit for any dimension. Numerical analysis in Ref. [138] using matrix product state methods (specifically infinite time evolving block decimation [564]) showed the existence of a transition even in one dimension. Remarkably, the numerical results indicate a continuous transition with critical exponents matching those at the critical point [562] corresponding to $u_3 = 0$. This was interpreted as a result of strong fluctuation effects in one dimension driving the system toward the critical point.

The ability to realise such transitions even in one dimensional systems makes these promising models for realisation in experiments. Indeed, as discussed above, results have already been presented for a similar model in a system of Rydberg atoms [141].

# 7   Dynamics of approach to the steady state

In this Section we will go beyond the simple interest in the steady state of the dissipative many-body process and discuss several of its dynamical aspects. For this reason, we will focus on situations in which the interplay between system and environment leads to a simple and featureless stationary state (i.e. without any particular quantum or classical correlation) and the interesting behaviour is mainly found in the way the system approaches its stationary properties.

## 7.1   Heating Dynamics Under Dephasing

We begin our discussion by focusing on the heating dynamics under dephasing, which is for instance what happens when ultracold-atomic gases evolve under controlled spontaneous emission (see Sec. 2.2). More generically, in this subsection we will study Lindblad master equations on a lattice with Hermitian site-local jump operators $\hat{L}_i^\dagger = \hat{L}_i$, so that we can write:

$$\frac{d}{dt}\hat{\rho} = -i[\hat{H}, \hat{\rho}] + \sum_i \left( \hat{L}_i \hat{\rho} \hat{L}_i - \frac{1}{2}\left\{ \hat{L}_i^2, \hat{\rho} \right\} \right) = -i[\hat{H}, \hat{\rho}] - \frac{1}{2}\sum_i [\hat{L}_i, [\hat{L}_i, \hat{\rho}]]. \tag{185}$$

The second expression makes very explicit that in the presence of Hermitian jump operators the identity matrix, normalised in order to have unit trace, is a steady state of the evolution.

This state can be identified with the infinite-temperature state, corresponding to the limit $\lim_{\beta \to 0^+} e^{-\beta \hat{H}} / \mathcal{Z}$: the dynamics therefore describes an heating process towards infinite temperature. The subject that we will explore is how such a stationary state is approached and the role of interactions and many-body effects on this transient dynamics. Among the many works that have studied this problem, in Sec. 7.1.1 we will address dephasing problems in spin chains, whereas bosonic systems will be the focus of Sec. 7.1.2.

It is however important to stress that the dynamics of heating towards infinite temperature has been the subjects of many other works that we cannot review in details here. The heating dynamics of dissipative Luttinger liquids and their lattice realisations have been studied in a number of works [565–567]. A generalised hydrodynamic picture of this heating process in integrable systems has been developed in Ref. [568]; while Ref. [569] has discussed the dissipative dynamics in the framework of a non-Hermitian linear response theory. Furthermore there have been studies focusing on the spreading of correlations in interacting and non-interacting models under dephasing [570, 571] as well as the growth of operator entanglement, which determines the efficiency of the Matrix-Product Operator representation for the density matrix [572].

Finally, we note that even if the stationary state is featureless—as occurs in the examples above where the stationary state is the infinite temperature state—the dynamical correlations developed in the approach to the steady state contain rich information about the thermalisation process. These features have been investigated in the case of a free fermionic chain under dephasing [294, 296]. For strongly correlated systems with charge and spin degrees of freedom, such as the Fermi–Hubbard model, selective dephasing of the spin sector has been shown to lead to a non-trivial charge order in the stationary state; this is related to the so called $\eta-$pairing state [428]. Slow dynamics and anomalous diffusion in open quantum many-body systems can also emerge in presence of long-range hopping [573] or kinetic constraints [134, 136–138] (see also discussion in Sec. 6.2.4).

### 7.1.1 Quantum spin chains under dephasing

We begin our discussion by considering a spin chain subject to a dephasing noise. One such example is the dissipative XX chain studied in Ref. [551], as described by the Hamiltonian and jump operators

$$\hat{H} = -J \sum_i \left( \hat{\sigma}_i^+ \hat{\sigma}_{i+1}^- + \text{H.c.} \right) + h \sum_i \hat{\sigma}_i^z, \qquad \hat{L}_i = \sqrt{\gamma} \hat{\sigma}_i^z. \tag{186}$$

We note that such dephasing processes can be physically realised by considering the interaction of spins with a magnetic field pointing in the $z$ direction with strength fluctuating in time. The Lindblad master equation follows from averaging such a model over noise realisations, assuming the field correlations are those of white noise. When $J = 0$ the system is composed of decoupled sites and the single-site matrix element of the density matrix, quantified by the expectation value of $\hat{\sigma}_i^+$, decays in time as $e^{-2\gamma t}$.

If we consider $J \neq 0$ a few simple considerations help understanding the main aspects of the physics of the many-body problem. We begin by focusing on the total magnetisation of the chain $\hat{S}^z = \sum_i \hat{\sigma}_i^z$; the adjoint Lindblad equation introduced in Sec. 3.4.2, that for Hermitian Lindblad operators reads: $\frac{d}{dt} \hat{S}^z = +i[\hat{H}, \hat{S}^z] - \frac{1}{2} \sum_i [\hat{L}_i, [\hat{L}_i, \hat{S}^z]]$, allows us to conclude that the total magnetisation of the chain is conserved during the dynamics because $[\hat{H}, \hat{S}^z] = 0$ and $[\hat{L}_i, \hat{S}^z] = 0$. On the other hand, the local magnetisation is not and spreads through the spin chain, its dynamics being constrained by the continuity equation $\frac{d}{dt} \hat{\sigma}_j^z = +i[\hat{H}, \hat{\sigma}_j^z]$. For closed systems, the local magnetisation evolves by spreading in a ballistic way. For open systems, dephasing enters through the equation of motion for the spin current: the end result is the emergence of a diffusive transport at long times [293, 294, 574].

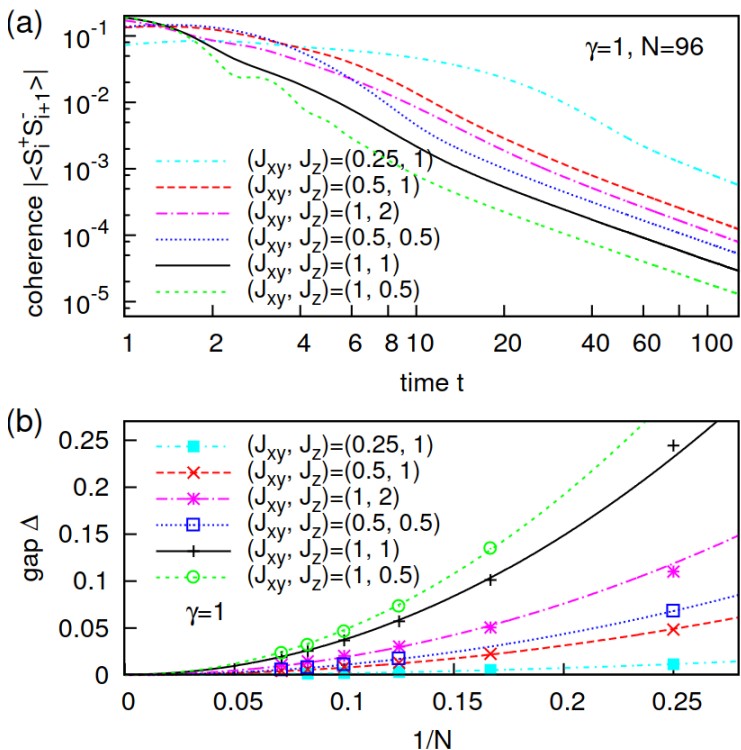

Figure 33: Numerical study of the dissipative XXZ model in Eq. (187). (a) Algebraic decay of the inter-site coherences for several values of the couplings parameters $J_{xy}$ and $J_z$ obtained with MPS-based techniques; the authors fit a late-time decay as $t^{-1.58}$ independent of the parameters of the system. (b) Lindbladian gap obtained with exact-diagonalization techniques; in all parameter regimes considered, the authors fit a decay as $1/N_s^2$. From Ref. [551] [Copyright (2013) by the American Physical Society].

As such one can expect a power law decay of the magnetisation at long times. This has been discussed extensively in the fermionic language. Indeed the Hamiltonian in Eq. (186) can be reduced via a Jordan-Wigner transformation to a quadratic fermionic Hamiltonian, which is non-interacting; however, because of the dephasing Lindblad operators, the Lindbladian is not quadratic. As discussed in Sec. 4.1, two-point fermionic observables can nonetheless be solved exactly; thanks to this property one can study for instance the dynamics of the local magnetisation without further approximations. We refer to Refs. [286, 288] for a detailed study in the more general case of the XY model and of several aspects of its dynamics, among which the ADR. The evolution of this model towards its steady state can be understood qualitatively since the dynamics of diagonal matrix elements is expected to rapidly become effectively classical and diffusive [290–292]. This result can be also understood from the spectral properties of the associated Lindbladian. For a tight-binding chain the Lindbladian spectrum can be exactly diagonalised using Bethe-ansatz techniques [295].

A different dynamical behaviour is expected for quantities which do not coincide with the conserved quantity, nor are a trivial function thereof, such as for example the coherences. If we consider the operator $\hat{A} = \sum_i \left( \hat{\sigma}_i^+ \hat{\sigma}_{i+1}^- + H.c. \right)$, it commutes with the Hamiltonian. Interestingly, one can prove by direct calculation that $\sum_i [\hat{L}_i, [\hat{L}_i, \hat{A}]] \propto \hat{A}$, which implies an exponential decay of its expectation value with a decay rate that does not depend on the chain length, $\langle \hat{A} \rangle \propto e^{-4\gamma t}$. If translational invariance of the initial state

is assumed, a similar result holds for $\hat{\sigma}_i^+ \hat{\sigma}_{i+1}^-$. Several works have focused on the role of inter-site couplings in changing the dynamical behaviour of the system and giving rise to power-law decays in the coherences $\langle \hat{\sigma}_i^+ \hat{\sigma}_{i+1}^- \rangle$ [551, 575]; for this to occur, a minimum requirement is that there must be a vanishing gap of the Lindbladian in the thermodynamic limit, represented by a set of many Lindbladian eigenvalues corresponding to slow decay. This is well illustrated by the case of a XXZ spin chain with dephasing:

$$\hat{H}_{XXZ} = \sum_i \left( \frac{J_{xy}}{2} \left( \hat{\sigma}_i^+ \hat{\sigma}_{i+1}^- + \hat{\sigma}_i^- \hat{\sigma}_{i+1}^+ \right) + J_z \hat{\sigma}_i^z \hat{\sigma}_{i+1}^z \right), \qquad \hat{L}_i = \sqrt{\gamma} \hat{\sigma}_i^z. \tag{187}$$

This model was studied via DMRG and exact-diagonalization techniques in Ref. [551], and was found to display a power-law decay of the inter-site coherences as a function of time, and a Lindbladian gap vanishing as $\Delta \sim 1/N_s^2$, as reported in Fig. 33.

This result can be understood by writing down an effective Lindbladian in the strongly dissipative regime, $\gamma \gg J_{xy}, J_z$ and by splitting the Lindbladian into two parts:

$$\hat{\mathcal{L}}_0[\hat{\rho}] = -i[J_z \sum_i \hat{\sigma}_i^z \hat{\sigma}_{i+1}^z, \hat{\rho}] + \sum_i \hat{\mathcal{D}}[\hat{L}_i, \hat{\rho}], \tag{188}$$

$$\hat{\mathcal{L}}_1[\hat{\rho}] = -i[(J_{xy}/2) \sum_i \left( \hat{\sigma}_i^+ \hat{\sigma}_{i+1}^- + \hat{\sigma}_i^- \hat{\sigma}_{i+1}^+ \right), \hat{\rho}]; \tag{189}$$

the idea is to treat $\hat{\mathcal{L}}_1$ as a perturbation. The steady state of $\hat{\mathcal{L}}_0$ is highly degenerate, with the degenerate steady states taking the form $\rho_{ss} = |\{\sigma\}\rangle\langle\{\sigma\}|$, with $|\{\sigma\}\rangle = \prod_i |\sigma_i\rangle$ and $|\sigma_i\rangle$ is an eigenstate of $\hat{\sigma}_i^z$. The perturbation $\hat{\mathcal{L}}_1$ has non-vanishing matrix elements between these terms. Treating $\hat{\mathcal{L}}_1$ at second order leads to an effective Lindbladian that takes the form [551]

$$\hat{\mathcal{L}}_{\text{eff}} [|\{\sigma\}\rangle\langle\{\sigma\}|] = - \sum_{\{\tau\}} \langle\{\tau\}|\hat{K}|\{\sigma\}\rangle \times |\{\tau\}\rangle\langle\{\tau\}| \tag{190}$$

and lifts the degeneracy. The operator $\hat{K}$ here is an effective Hamiltonian that describes a ferromagnetic Heisenberg model

$$\hat{K} = - \frac{J_{xy}^2}{\gamma} \sum_i \left( \frac{1}{2} \left( \hat{\sigma}_i^+ \hat{\sigma}_{i+1}^- + \hat{\sigma}_i^- \hat{\sigma}_{i+1}^+ \right) + \hat{\sigma}_i^z \hat{\sigma}_{i+1}^z - \frac{1}{4} \right), \tag{191}$$

whose low-energy spectrum is gapless, with a gap closing as $1/N_s^2$ due to the fact that it is composed of magnons with wavevectors $k$, energy proportional to $k^2$, and minimal wavevector $k \propto N_s^{-1}$. This provides a simple theoretical framework to explain the numerical results reported in the bottom panel of Fig. 33. To the best of our knowledge, the superoperator $\hat{L}_{\text{eff}}$ in Eq. (190) has never been used to theoretically compute the late-time decay in the top panel of Fig. 33 although a calculation of such decay exponent, equal to $3/2$, is presented in Ref. [295] for the case $J_z = 0$.

Finally, we mention the study in Ref. [551] of the dissipative quantum Ising model in a transverse field coupled to dephasing noise; here the total magnetisation is not conserved by the dynamics and several of the previous considerations need to be modified accordingly. Here the on-site spin-flip coherence of the state, $\langle \hat{\sigma}_i^+ \rangle$, decays exponentially in time, although with a decay rate that depends on the parameters of the model. We refer to the original article for further details.

### 7.1.2 Bose–Hubbard model under dephasing

Another well-studied example of interaction-induced anomalous dissipative dynamics is provided by the Bose–Hubbard model with dephasing that was introduced in Sec. 3.1.5. In one dimension, we consider the Hamiltonian introduced in Eq. (36) and the jump operator proportional to the local density operator:

$$\hat{H} = -t_H \sum_{\langle i,j \rangle} \left( \hat{a}_i^\dagger \hat{a}_j + \text{H.c.} \right) + \frac{U}{2} \sum_i \hat{n}_i(\hat{n}_i - 1), \qquad \hat{L}_{i\phi} = \sqrt{\gamma_\phi}\, \hat{n}_i. \tag{192}$$

Here we focus on the strongly-interacting regime and follow the discussion of Refs. [576,577]. In the absence of hopping, $J = 0$, any density matrix that is diagonal in Fock space is a steady state of the dynamics. A small hopping removes this degeneracy and when the initial state has a well-defined number of bosons it drives the system towards a unique steady state: the completely-mixed state of all Fock states with total number of particles given by the initial condition. The long-time dynamics can be captured by treating the small off-diagonal matrix elements of the density matrix at second order in perturbation theory and writing down a rate equation for the diagonal ones in this basis.

It is instructive to follow Ref. [576] and to discuss the case $N_s = 2$ of the dissipative Bose–Hubbard model in Eq. (192): two coupled sites with dephasing and interaction. Since the total number of bosons, $N$, is fixed, we can define a basis of two-site states $\{|n\rangle\}$ labelled by the number $n = 0, 1, .., N$ of bosons in the left site (the number of bosons in the right site being $N - n$). We can then parameterise the density matrix as $\hat{\rho} = \sum_{n,m} \rho_{n,m} |n\rangle\langle m|$. Writing the equation of motion for the diagonal $\rho_{n,n}$ and off-diagonal $\rho_{n,n+1}$ matrix elements of the density matrix, and then eliminating the latter, one thus obtains:

$$\frac{d}{dt}\rho_{n,n} = \left( \frac{J^2 \gamma_\phi}{2N^2 U^2} \right) 2N^2 \left( W_{n+1}\rho_{n+1,n+1} - [W_n + W_{n+1}]\rho_{n,n} + W_n \rho_{n-1,n-1} \right) =$$

$$= \left( \frac{J^2 \gamma_\phi}{2N^2 U^2} \right) (2W_{n+1}\Delta\rho_n - 2W_n\Delta\rho_{n-1}), \tag{193}$$

where $W_{n+1} = (n+1)(N-n)/(n - N/2 + 1/2)^2$ and $\Delta\rho_n = N^2 (\rho_{n+1,n+1} - \rho_{n,n})$. Interestingly, all the dependence on the physical parameters enters only in the rescaled time $\tau = t/t^*$, with $t^* = 2N^2 U^2/J^2 \gamma_\phi$.

In the limit of large bosonic occupation, $N \gg 1$, one can map Eq. (193) into a diffusion equation for the probability distribution $p(s,\tau) = N\rho_{n,n}(\tau)$ with $s = n/N - 1/2 \in [-1/2, 1/2]$ that takes the form:

$$\frac{\partial}{\partial \tau}p(s,\tau) = \frac{\partial}{\partial s}\left( D(s)\frac{\partial}{\partial s}p(s,\tau) \right). \tag{194}$$

The diffusion function reads $D(s) = \frac{1}{4s^2} - 1$ and thus features a strong spatial dependence which diverges at $s = 0$, corresponding to the symmetric occupation of the two sites, and vanishes at the boundaries $s = \pm 1/2$. The slow diffusion at the boundaries corresponds to the slow rate of populating energetically-costly configurations. While continuing our discussion, it is important to keep in mind that the variable $s$ is not a true spatial variable, as it represents a sort of density of bosons in the left site. Importantly, one can find a scaling solution for this non-Brownian motion which, at short-times ($\tau \ll 1$), reads:

$$p(s,\tau) = \frac{\sqrt{2}}{\Gamma(1/4)} \frac{1}{\tau^{1/4}} \exp\left( -\frac{s^4}{4\tau} \right). \tag{195}$$

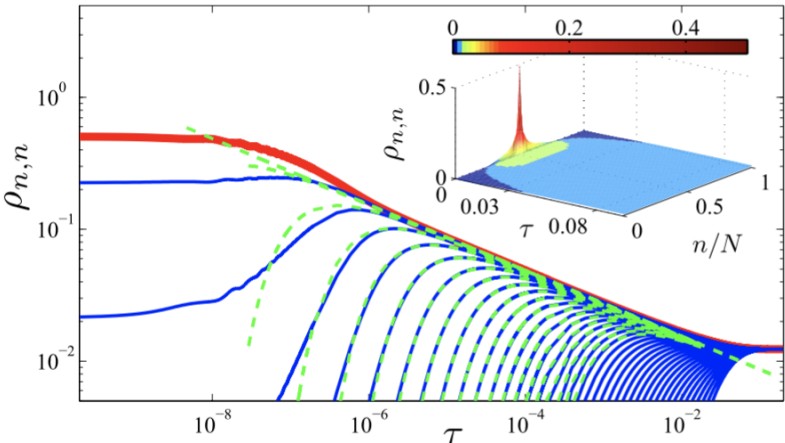

Figure 34: Left: Dephasing dynamics and anomalous diffusion in the two-site Bose–Hubbard model: we see the dynamics of the diagonal matrix-elements $\rho_{n,n}$ of the density matrix showing long tails as time grows. The simulation is performed for $N = 80$ and the red line corresponds to $n = 40$, the situation where both sites are equally populated. The thin blue lines correspond to larger values of $n$, from 41 to 80. The dashed green lines correspond to the anomalous diffusion in Fock space of the equation (195). The inset shows a three-dimensional plot of the evolution. From Ref. [576] [Copyright (2012) by the American Physical Society.].

This analytical solution shows that the system displays anomalous diffusion in Fock space and matches the numerics well, as shown in Fig. 34. Given this form of the local occupation probabilities, several physical consequences follow. In particular, one can show that the coherence between two neighbouring sites $C(t) = \langle \hat{b}_1^\dagger \hat{b}_2 + \text{H.c.} \rangle_t$ can be written as $C(t) \simeq \frac{J}{U} \mathcal{C}(t/t^*)$ with

$$\mathcal{C}(\tau) = \int ds \, \frac{\partial}{\partial s} p(s, \tau) \frac{s^2 - 1/4}{s}. \tag{196}$$

In the power-law regime, when $\tau \ll 1$ one obtains $C(\tau) \propto 1/\sqrt{\tau}$ for the coherence.

The anomalous decay of the coherence was shown to survive beyond the two-site case in the strong-interaction limit of the Bose–Hubbard model, using a time-dependent Gutzwiller ansatz [577], similar to that discussed in Section 4.3. Such anomalous decay has been also observed in experiments with ultracold atoms under controlled dissipative processes [98, 578]. We follow Ref. [98], where the authors report the preparation of a nearly-ideal Bose-Einstein condensate of Yb atoms with broad spatial coherence loaded in a two-dimensional array of one-dimensional tubes with a superimposed optical lattice.

The gas is exposed to dissipation by shining a near-resonant laser, leading to a spontaneous emission from the excited state. The effect of such a drive can be modelled as a dephasing process and the loss of spatial coherence is associated to the fact that after spontaneous emission the atom recoils in a random direction. By measuring the dynamics of momentum distribution, and in particular the decay in time of the central peak and its broadening, one could track the decay in time of the coherence (see Fig. 35, top panels, for three snapshots at different times). The result shows a behaviour that is qualitatively different from the exponential decay expected for non-interacting atoms, and which is consistent with a power-law decay. With a procedure that we do not want to detail here, the authors extract from these data the coherence function $C_s(t) = \langle \sum_j \hat{b}_j^\dagger \hat{b}_{j+s} + \text{H.c.} \rangle_t$; the bottom panels of Fig. 35 shows the experimental data of $C_1(t)$ and $C_2(t)$. If we simply

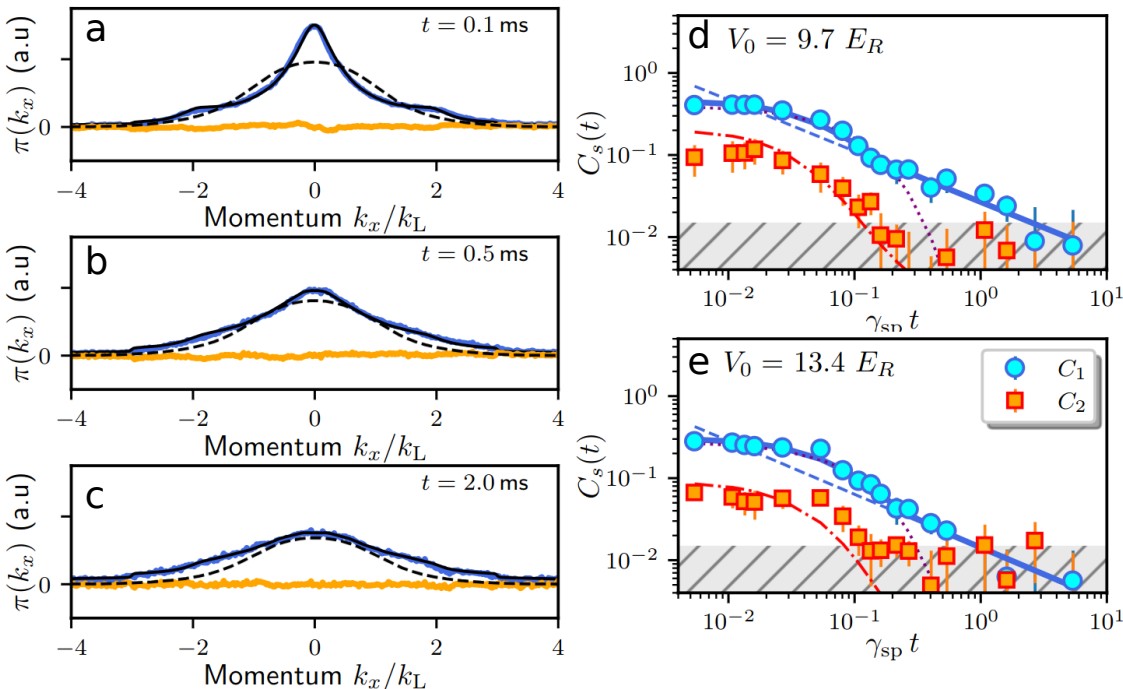

Figure 35: Spatial coherence of a one-dimensional many-body bosonic gas in an optical lattice and subject to dephasing due to induced spontaneous-emission processes. (Panels a,b,c) Snapshots of the momentum distribution function $\pi(k_x)$ as a function of time; the initial spatial coherence, reflected by a peaked distribution, is lost as times goes by. (Panels d,e) Dynamics of spatial coherence of the gas; $C_1(t)$ is well fitted by the solid blue line that crosses over from an initial exponential decay to a long-time algebraic decay. From Ref. [98].

focus on the function $C_1(t)$, plotted with blue circles, the experimental data are well fitted by a function that crosses over from an initial exponential decay to a long-time algebraic decay. By adapting the reasoning proposed at the very beginning of this section on the heating of the XX model, for $U = 0$ the value of $C_1(t)$ is expected to decay exponentially in time. A long-time decay that is not exponential is the hallmark of correlations among the bosons created during the dissipative dynamics, and hence is a genuine many-body effect.

## 7.2 Depletion Dynamics of Bosonic Gases under Many-Body Losses

We now address the specific kind of open-system dynamics induced by atom losses. In contrast with what was discussed in Section 5.3.3, we will not focus here on the existence of intriguing dark states: our goal is to characterise dynamically how losses decrease the population of the gas, and the kind of nonequilibrium states through which the gas is driven during the process. This task is required in order to produce a quantitative description of the experiments which are always plagued by these effects (see Ref. [74] for an early study). Usually, these kinds of theoretical studies require the ability to fully treat the nontrivial interplay of the classical fluctuations induced by the loss events and the quantum fluctuations induced by the Hamiltonian dynamics; as such, understanding this physics has a broader relevance. For simplicity, we will focus on situations where the only stationary state is the vacuum and thus we discuss only the lossy dynamics of bosonic gases.

In general, the population of a bosonic gas subject to $m$-body losses evolves according

to an equation that links the density to the local quantum correlations of the gas [579]. We consider for simplicity a gas on a lattice described by the Bose–Hubbard Hamiltonian $\hat{H}_{\mathrm{BH}}$ in Eq. (36) and by the jump operators describing $m$-body losses,

$$\hat{L}_{j,(m)} = \sqrt{\kappa^{(m)}}\, \hat{a}_j^m. \tag{197}$$

One may compute the time evolution of the *expectation value* of the number operator $\hat{N} = \sum_j \hat{a}_j^\dagger \hat{a}_j$ using the adjoint equation as described in Sec. 3.4.2. This takes the form:

$$\frac{d}{dt}\left\langle \hat{N} \right\rangle = i\left\langle [\hat{H}_{\mathrm{BH}}, \hat{N}] \right\rangle + \sum_j \left\langle \hat{L}_{j,(m)}^\dagger \hat{N} \hat{L}_{j,(m)} - \frac{1}{2}\{\hat{N}, \hat{L}_{j,(m)}^\dagger \hat{L}_{j,(m)}\} \right\rangle$$

$$= \frac{1}{2}\left\langle \sum_j \hat{L}_{j-(m)}^\dagger [\hat{N}, \hat{L}_{j-(m)}] + \mathrm{H.c.} \right\rangle \tag{198}$$

where we used the fact that the Hamiltonian conserves the number of particles. If we now consider a basis of the Hilbert space composed of states that have a definite number of bosons, it is not difficult to show that $[\hat{N}, \hat{L}_{j,(m)}] = -m\hat{L}_{j,(m)}$ and thus:

$$\frac{d}{dt}\left\langle \hat{N} \right\rangle = -\kappa^{(m)} m \sum_j \left\langle \hat{a}_j^{\dagger m} \hat{a}_j^m \right\rangle. \tag{199}$$

If we assume spatial homogeneity and introduce the density $n(t) = \langle \hat{N} \rangle / N_s$ and the normalised zero-distance $m$-body correlation function $g_m(0,t)$, the equation reads:

$$\frac{d}{dt}n(t) = -\kappa^{(m)}\, m\, g_m(0,t)\, n(t)^m, \quad \text{with} \quad g_m(0,t) = \frac{1}{N_s}\frac{\left\langle \sum_j \hat{a}_j^{\dagger m} \hat{a}_j^m \right\rangle}{n(t)^m} = \frac{\left\langle \hat{a}_0^{\dagger m} \hat{a}_0^m \right\rangle}{n(t)^m}. \tag{200}$$

This equation clearly shows that the dynamics of the population contains important information on the correlations among the particles of the gas. For this reason, already in the early days of Bose-Einstein condensates, three-body loss rates have been employed as a sensitive probe of the statistical correlations between atoms [73]. More recently similar techniques have been employed in the first experiments on Efimov physics, where the appearance of the Borromean three-body bound state was signalled by an abrupt modification of the three-body loss rate [580].

In the remainder of this section, we will concentrate on two-body losses ($m = 2$), which arguably induce the simplest non-trivial loss dynamics. Before we focus on the $m = 2$ case, we make a few comments about two other heavily studied case, $m = 1$ and $m = 3$.

The case of one-body losses, $m = 1$, can be treated exactly as discussed in Sec. 4.1, as they correspond to a simple non-interacting limit. Furthermore, since for $m = 1$ it is always true that $g_1(0,t) = 1$, Eq. (200) does not depend on particle correlations, and instead shows a simple exponential decay (although, as discussed in Refs. [579,581], it can have non-trivial effects on the rapidity distribution of the integrable and one-dimensional Bose gas). We note however that there can be non-trivial behaviour with one-body losses. For example, several recent studies have considered the effect of one-body losses that do not act on the entire sample, but only on a part of it, e.g. a single site of an optical lattice, finding interesting non-trivial results [86,582–601]; moreover, one-body losses have applications in the detection of topological invariants [602,603]. More recently, the study of non-local one-body loss processes in correlated bosonic gases has also proved to be the source of intriguing effects, such as critical behaviour [604]. There has also been significant

interest in the study of three-body losses in quantum simulation and in characterising many-body states [73–75, 445, 605–611].

Two-body losses have been significantly studied both theoretically and experimentally. One point that contributes to their appeal is the fact that several aspects of this dynamics can be understood using a non-Hermitian Hamiltonian with complex interaction constant, describing at the same time elastic and inelastic interactions. Here below, we want to briefly describe the theoretical ideas and the experimental results that are employed in the two limiting regimes of weak and strong two-body losses. We conclude with a very concise list of other notable results presented in the literature.

### 7.2.1 Ideal Bose gas and weak two-body losses

We start by characterising the dynamics of a one-dimensional non-interacting Bose gas under weak two-body losses in a lattice of $N_s$ sites with periodic boundary conditions; our discussion here is based on Ref. [579] (see also the review in Ref. [612]). The Hamiltonian for this system can be written in momentum space as: $\hat{H} = -2t_H \sum_k \cos(k)\hat{a}_k^\dagger \hat{a}_k$ using standard Fourier-transform techniques. If we look at the theoretical approach that we are going to introduce from the viewpoint of the methods detailed in Sec. 4, the solution of the dynamics is based on an ansatz that displays a mean-field factorisation in momentum space, as discussed in Sec. 4.3.2; moreover, we will also see that this state is a bosonic Gaussian state and thus that it is particularly simple to treat since it satisfies the Wick's theorem. Rather than following this abstract approach, however, we are going to introduce the ansatz for the density matrix from a more physical viewpoint.

It is well known that the properties of a non-interacting Bose gas can be defined by its momentum occupation function, $n_k = \langle \hat{a}_k^\dagger \hat{a}_k \rangle$. Under the Hamiltonian dynamics, the operators $\hat{a}_k^\dagger \hat{a}_k$ are constants of motion because they all commute with the Hamiltonian (and in fact, they all mutually commute); one may use them to construct a generalised Gibbs ensemble (GGE) state characterised by a set of Lagrange multipliers $\lambda_k$:

$$\hat{\rho}_{\text{GGE}} = \prod_k \frac{e^{-\lambda_k \hat{a}_k^\dagger \hat{a}_k}}{Z_k}, \qquad \text{with} \quad Z_k = \text{Tr}[e^{-\lambda_k \hat{a}_k^\dagger \hat{a}_k}]. \qquad (201)$$

The interest of this quantum state is that it can reproduce the *local* (in real-space) properties of a pure state that has evolved under a long unitary time-evolution with the non-interacting bosonic Hamiltonian; in order to do that it is simply necessary to tune the $\{\lambda_k\}$ so that $\text{Tr}\left[\hat{\rho}_{\text{GGE}}\, \hat{a}_k^\dagger \hat{a}_k\right]$ matches the value of the pure state under consideration [613, 614].

In the presence of losses, the operators $\hat{a}_k^\dagger \hat{a}_k$ are no longer conserved quantities of the dynamics. Nonetheless, when losses are weak (sometimes referred to in literature as *adiabatic losses*), one can introduce a *time-dependent* GGE that accounts for the time evolution of $\langle \hat{a}_k^\dagger \hat{a}_k \rangle$ by making the Lagrange multipliers depend on time [615, 616]:

$$\hat{\rho}_{\text{tGGE}}(t) = \prod_k \frac{e^{-\lambda_k(t)\hat{a}_k^\dagger \hat{a}_k}}{Z_k(t)}, \qquad \text{with} \quad Z_k(t) = \text{Tr}[e^{-\lambda_k(t)\hat{a}_k^\dagger \hat{a}_k}]. \qquad (202)$$

We will discuss the evolution of this ansatz under the Lindblad master equation with the jump operators as defined in Eq. (197) for $m = 2$.

The state $\hat{\rho}_{\text{tGGE}}(t)$ is Gaussian and once it is used as an ansatz in the Lindblad master equation, it leads to simple mean-field dynamical equations for the momentum occupation functions and for the total density $n(t) = \sum_k n_k(t)$:

$$\frac{d}{dt}n_k(t) = -2\kappa_2 n(t) n_k(t) \qquad \Rightarrow \qquad \frac{d}{dt}n(t) = -2\kappa_2 n(t)^2. \qquad (203)$$

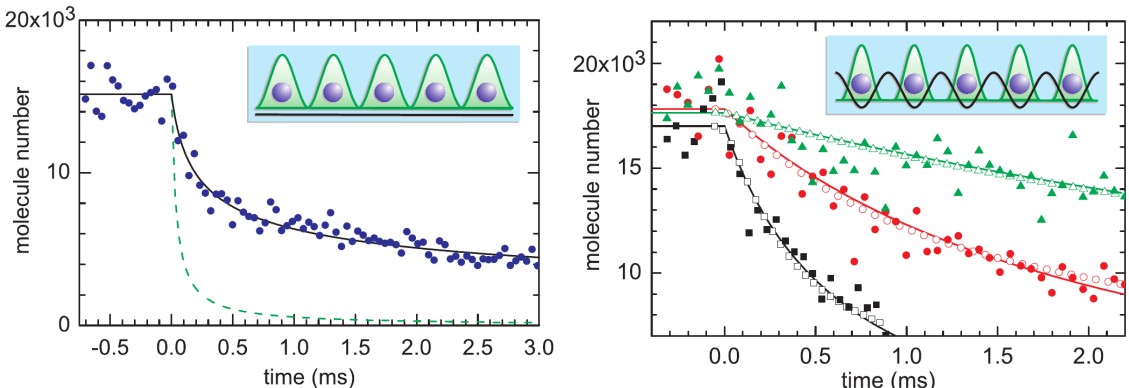

Figure 36: Strong two-body losses in a one-dimensional molecular bosonic gas. (Left) Dynamics of the molecular population; losses begin at $t = 0$. The dashed line shows the expectation of an uncorrelated system. The observed loss is much slower than the dashed line because of strong correlations. (Right) Loss dynamics in presence of an optical lattice. The strength of the optical lattice, and thus the strength of the bare loss rate increases from black to red to green. We observe a clear signature of the quantum Zeno effect: for strong rate effects, the life time of the gas increases instead of decreasing. From Ref. [84] [N. Syassen *et al.*, Science **320**, 1329 (2008)]. Reprinted with permission from AAAS.

By comparing the second equation of Eq. (203) to the expression in Eq. (200), one sees that this implies the gas maintains a zero-distance 2-body correlation function equal to one during the dynamics, $g_2(0, t) = 1$. The solution of the dynamics of the density of the gas is equally simple:

$$n(t) = \frac{n(0)}{1 + 2\kappa_2 n(0)t}, \qquad n_k(t) = \frac{n_k(0)}{1 + 2\kappa_2 n(0)t}. \qquad (204)$$

This shows a decay at late times as $1/t$. For a non-interacting gas, thus, weak losses do not appear to induce any peculiar quantum properties, although it is important to note that even when $n_k(0)$ corresponds to a Bose-Einstein distribution, $n_k(t)$ does not remain an equilibrium distribution.

### 7.2.2 Fermionised regime and quantum-Zeno effect

We now focus on the limit of strong two-body losses, following the theoretical discussion in Refs. [332, 579, 581, 617]. We start from a simple observation that is valid when one neglects hopping between lattice sites: lattice sites with double or higher occupancies undergo some lossy dynamics, whereas lattice sites with at most one particle are stable. The Fock space is thus the direct sum of two orthogonal subspaces, $\mathcal{H}_{stable}$ and $\mathcal{H}_{loss}$, whose different dynamical properties are best highlighted by the different action of the non-Hermitian Hamiltonian associated to the process:

$$\hat{H}_{\mathrm{nH}} = -t_H \sum_j \left[ \hat{a}_j^\dagger \hat{a}_{j+1} + \text{H.c.} \right] + \left( U - i\frac{\gamma}{2} \right) \sum_j \hat{n}_j(\hat{n}_j - 1). \qquad (205)$$

In the limit $t_H = 0$ of no hopping processes, $\mathcal{H}_{stable}$ is the kernel of the Hamiltonian, whereas $\mathcal{H}_{loss}$ is the complement.

We consider the effect on $\mathcal{H}_{stable}$ of a non-zero but small hopping rate, $0 < t_H \ll \gamma$. The hopping process can connect states with two singly-occupied neighbouring lattice sites

with an unstable state where two bosons sit on the same site, a state that belongs to the dissipative space $\mathcal{H}_{loss}$. We thus conclude that also the states belonging to $\mathcal{H}_{stable}$ become unstable and a second-order perturbative calculation allows us to determine the strength of the effective complex interaction:

$$\frac{t_H^2}{U + i\frac{\gamma}{2}} = \frac{t_H^2}{U^2 + \left(\frac{\gamma}{2}\right)^2} \left(U - i\frac{\gamma}{2}\right). \tag{206}$$

This simple analysis contains the whole essence of the typical timescales of strongly lossy gases: the effective loss rate $\Gamma = \frac{\gamma}{2} \frac{t_H^2}{U^2 + \left(\frac{\gamma}{2}\right)^2}$ scales as $1/\gamma$ and thus the lifetime of the gas increases as the loss-rate of the gas is increased. This counterintuitive result has been the object of several experimental analyses and Fig. 36 displays some of the main results obtained in the first reported experiment [84]. This perturbative analysis can also be performed also for the Lindbladian superoperator and not just for non-Hermitian Hamiltonians, we refer the reader to the original literature for more details [617].

Before continuing, it is interesting to briefly discuss the name *"quantum Zeno effect"* that is commonly employed to describe this phenomenology. The quantum Zeno effect was originally introduced to describe the fact that a frequently monitored unstable quantum system experiences a longer decay time than an unmonitored one [66] (see also Sec. 2.1). This is exactly what happens in the system described above, as long as the action of the environment is considered as an unread measurement. Strong two-body losses can be interpreted as an action of the environment that repeatedly checks whether in the system any lattice site is doubly occupied. By frequently performing this check, the environment effectively prevents two particles from occupying the same site and thus increases the lifetime of the state where all lattice sites have at most one particle [618].

We now show that the method of the GGE ansatz—as employed above for the discussion of weak losses—can also be employed to solve the full dynamics of the system in the regime of strong losses. The crucial point is the observation that the system is characterised by an effective separation of time scales. In this section we have already assumed that $\gamma \gg t_H$. Inspection of Eq. (206) allows us to also write that $t_H \gg \Gamma$. It is important in the following to keep in mind these two inequalities, which state that after the fast loss of all doubly occupied sites, with a rate $\gamma$, the system is a weakly lossy gas of hard-core bosons, with a rate $\Gamma$.

Based on the above, the study of strong two-body losses is performed via an effective perturbative master equation that only acts on $\mathcal{H}_{stable}$. It is customary to say that a gas described by a state in $\mathcal{H}_{stable}$ is *fermionised* because it is composed of hard-core bosons and can only have at most a particle per site. Note however that the operators creating such excitations on different sites commute, so these are not true fermions. To avoid confusion, we use the Pauli matrices to represent the hard-core bosons. The Hamiltonian and the quantum jump operators of the effective master equation read:

$$\hat{H} = -J \sum_j \left[ \hat{\sigma}_j^+ \hat{\sigma}_{j+1}^- + \text{H.c.} \right]; \qquad \hat{L}_j = \sqrt{\Gamma} \hat{\sigma}_j^- \left( \hat{\sigma}_{j-1}^- + \hat{\sigma}_{j+1}^- \right). \tag{207}$$

In Fig. 37 we compare Eq. (204) with a numerical solution of the hardcore boson model, Eq. (207) obtained with state-of-the-art techniques based on quantum trajectories [261] for sizes up to $L = 14$. These numerical simulations do not rely on physical approximations and serve here as a benchmark. We propose a first comparison with the solution of the simple equation that appears in the right side of Eq. (203), that is routinely employed to describe two-body losses: $\frac{d}{dt}n(t) = -2\Gamma n(t)^2$. This equation does not assume any spatial

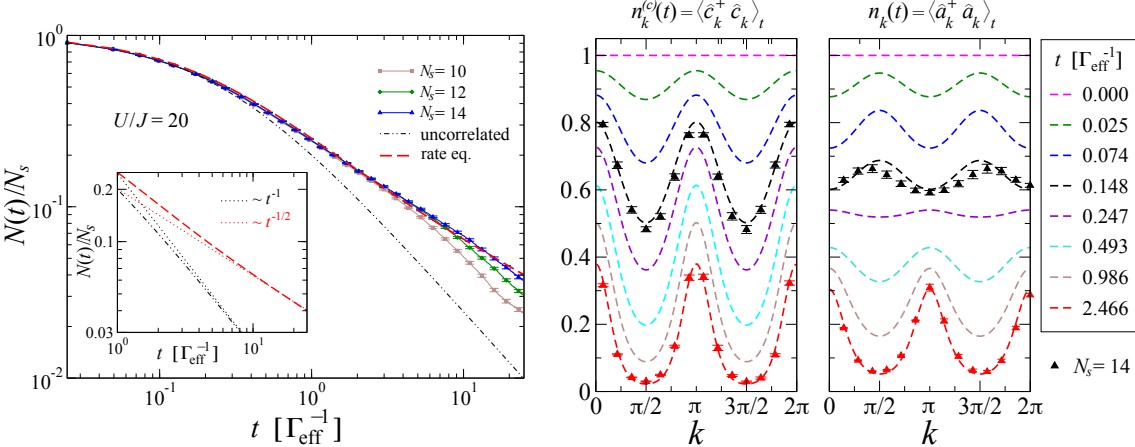

Figure 37: (Left) Time evolution of the number of atoms in the strongly lossy regime according to the rate equations (208) for the initial state with one boson per site (dashed red line). The result is benchmarked with simulations based on quantum trajectories for $N_s = 10$, 12 and 14 (each point is averaged over $10^4$ trajectories). The dot-dashed black line represents the solution $N(t)/N_s$ in Eq. (204) using $\Gamma$ instead of $\kappa_2$. The inset highlights the different long-time decay as $t^{-1}$ for the spatially-uncorrelated solution and as $t^{-1/2}$ for the rate equation. (Centre) Fermionic rapidity distribution and (right) bosonic quasi-momentum distributions. Dashed lines are the predictions using the rate equation. Data from quantum-trajectory simulations for $N_s = 14$ (symbols) are presented for two times. From Ref. [332].

correlation of the gas and unsurprisingly the solution, dubbed "uncorrelated" in the figure, agrees with the numerics only at short times because the initial state is uncorrelated. Increasingly strong deviations appear at long times, indicating the build-up of correlations that the mean-field model fails to capture.

An alternate analytic approximation for the hard-core bosons can be found by using a Jordan-Wigner transformation [246] to map the hard-core bosons into fermions. The corresponding Hamiltonian can be diagonalised in momentum space and reads $\hat{H} = \sum_k -2J \cos(k)\hat{c}_k^\dagger \hat{c}_k$, where $\hat{c}_k^{(\dagger)}$ are fermionic operators obeying canonical anticommutation relations. Since the operators $\hat{c}_k^\dagger \hat{c}_k$ mutually commute and are constants of the motion under the Hamiltonian dynamics, we may again characterise the local properties of the lossy gas using a time-dependent GGE that is a fermionic Gaussian state which factorises in momentum space, $\hat{\rho}_{\text{tGGE}}(t) = \prod_k \frac{1}{Z_k(t)} e^{-\lambda_k(t)\hat{c}_k^\dagger \hat{c}_k}$, with $Z_k(t) = \text{Tr}\left[e^{-\lambda_k(t)\hat{c}_k^\dagger \hat{c}_k}\right]$. Writing the Lindblad master equation in the fermionic formulation we obtain, after some algebra, the following rate equations for the expectation values $n_k^{(c)}(t) = \text{Tr}[\rho_{\text{tGGE}}(t)\hat{c}_k^\dagger \hat{c}_k]$. Note that this quantity is the *fermionic* occupation number, and is also referred to as the *rapidity distribution* of the gas:

$$\frac{\mathrm{d}}{\mathrm{d}t} n_k^{(c)}(t) = -\frac{4\Gamma}{L} \sum_q \left[\sin(k) - \sin(q)\right]^2 n_q^{(c)}(t)\, n_k^{(c)}(t). \tag{208}$$

These equations are easily solved numerically and the solution, dubbed "rate equation", is plotted in Fig. 37, where we observe an excellent agreement between the predictions of the rate equation and the numerical simulations for all times shown. Unlike the spatially-uncorrelated solution, which predicts the scaling $\sim t^{-1}$ at long times, the rate equations (208) predict that $n(t)$ decays to zero as $t^{-1/2}$. This result is highlighted in the

inset of Fig. 37 and can be analytically proven. This algebraic decay is the hallmark of the spatial correlations that build up after dissipation is enabled.

In addition to considering the total population, Fig. 37(centre) also shows the time evolution of the fermionic population (rapidity distribution) $n_k^{(c)}(t)$. In the long-time limit $t > \Gamma^{-1}$ this is well-approximated by

$$n_k^{(c)}(t) \approx \frac{1}{(8\pi\Gamma t)^{1/4}} e^{-\sin^2(k)\left(\frac{8\Gamma t}{\pi}\right)^{1/2}}. \tag{209}$$

Although at initial times the population is uniformly spread among the different momenta, a double-peaked distribution emerges for long times, with maxima at $k = 0, \pi$. The interplay between two-body losses and coherent free-fermion dynamics has thus created a nonequilibrium fermionic gas where the notion of Fermi sea is completely lost. Standard time-of-flight measurements give instead access to the *bosonic* momentum distribution function $n_k(t) = \langle \hat{a}_k^\dagger \hat{a}_k \rangle_t$, discussed above. The link between $n_k^{(c)}(t)$ and $n_k(t)$ is known explicitly but is complicated so not written here. The result of the calculation is shown in Fig. 37(right).

At this point instead it is interesting to observe the fact that these results are obtained using a density matrix that is of mean-field form in the sense that it displays momentum-space factorisation. However, differently from what done in the discussion of the non-interacting Bose gas, the density matrix is not a bosonic Gaussian state; it is a fermionic Gaussian state, but this property does not trivially transfer to the bosonic language. As an example, this means that the bosonic correlation function $g_2(0, t)$ is different from zero and thus a non-trivial dynamics can take place.

### 7.2.3   Related results

The study of the dynamics of lossy gases has been the object of several other publications that have addressed aspects of the problem that we did not discuss here. For example, the effect of an harmonic confinement [581, 619, 620], the nonequilibrium properties of the gas [621] or the effect of spin in the fermionic case [422, 622, 623]. Recently, the same problem has been addressed with Keldysh field theory, obtaining similar results [624, 625], which have raised interesting questions on the universal properties of the gas decay discussed also in Refs. [309]. For superfluid fermionic gases with two-body losses the dynamics of particle density depletion, together with the destruction of the superfluid order parameter, have been studied in Ref. [237].

The theoretical results presented above are based on the notion of time-dependent GGE, that is possible when an open- or closed-system dynamical effect perturbs a closed system with conserved quantities. A general discussion of this theory and of some applications beyond lossy gases can be found in Refs. [615, 616, 626]. As a notable example, we mention here the recent development of the quantum theory of reaction-diffusion processes, that employs exactly the same formalism [627, 628]. There have also been works studying random Lindbladians, and the observation of phase transitions in the relaxation dynamics between slow and fast relaxation [629].

## 8   Monitored Quantum Systems

In this last section we return to the description of open quantum systems introduced in Sec. 3.3, based on the *unravelling* the Lindblad master equation into stochastic quantum trajectories [247–250]. As discussed in that section, the physical interpretation of the

resulting stochastic dynamics is that of a system evolving under continuous monitoring of certain observables. The quantum trajectory produced this way represents the evolution of the state conditioned on a given measurement record. Although such ideas are well established in the field of few body quantum optics [248,250,251,260,264], these tools have recently attracted much attention in the context of open many-body systems [261] as well as in hybrid quantum circuits [630] (that is, circuits with both unitary gates and measurement processes). In the context of these Lectures Notes, we will focus only on monitored quantum many-body systems which are described by a continuous-time Lindblad master equation, or the corresponding unravelling. The particular setting we will consider in this section is a system, described by a given Hamiltonian $H$, which is coupled on each lattice site to a monitored environment, as illustrated in Fig. 38. As such, we will not consider the role of projective measurements and of random unitary circuits, for which other reviews exist [630–632].

## 8.1   Trajectory-resolved averaging

A key point in the field of monitored quantum systems is the idea that there may be new physics contained in the quantum trajectories, i.e. in the conditional state, that is not captured by the average state. To appreciate this point it is helpful to consider a specific choice of unravelling from among those introduced in Sec. 3.3, such as Quantum State Diffusion. (Similar considerations can be drawn for other unravellings). As we have mentioned, given the evolution of the conditional state:

$$d \left| \psi(\xi_t, t) \right\rangle = \left[ -i\hat{H} - \frac{1}{2} \sum_\mu \left( \hat{L}_\mu^\dagger \hat{L}_\mu + \left\langle \hat{L}_\mu^\dagger \right\rangle_{\xi_t, t} \left\langle \hat{L}_\mu \right\rangle_{\xi_t, t} - 2 \left\langle \hat{L}_\mu^\dagger \right\rangle_{\xi_t, t} \hat{L}_\mu \right) \right] dt \left| \psi(\xi_t, t) \right\rangle +$$

$$+ \sum_\mu \left( \hat{L}_\mu - \left\langle \hat{L}_\mu \right\rangle_{\xi_t, t} \right) d\xi_t^\mu \left| \psi(\xi_t, t) \right\rangle , \tag{210}$$

one can show that by averaging over the noise, the conditional density matrix $\hat{\rho}(\xi_t, t) = |\psi(\xi_t)\rangle\langle\psi(\xi_t)|$ gives back an average state evolving with a Lindblad master equation, with jump operators given by the monitored operator $\hat{L}_\mu$. As a consequence of the linearity of the average over the noise it is sufficient to know the mean density matrix $\overline{\hat{\rho}(\xi_t, t)}$ to compute all observables which are linear in the state $O[\hat{\rho}] = \hat{O}\hat{\rho}$. Specifically, for any function $f(\hat{O})$, we have

$$\overline{\mathrm{Tr}(\hat{\rho}(\xi_t, t) f(\hat{O}))} \equiv \mathrm{Tr}(f(\hat{O})\hat{\rho}(t)). \tag{211}$$

This is no longer the case if one is interested in the noise-average of quantities which depend *non-linearly* on the conditional density matrix [263]. A simple example is the purity $\mathcal{P}(\hat{\rho}) = \underline{\mathrm{Tr}\hat{\rho}^2}$: for the conditional state $\mathcal{P}(\hat{\rho}(\xi_t, t)) = 1$ (and hence its average over trajectories $\overline{\mathcal{P}(\hat{\rho}(\xi_t, t))} = 1$), while for the mean state we have $\mathcal{P}(\hat{\rho}(t)) < 1$. Another particularly relevant example is provided by the entanglement entropy. Significant work and attention has been devoted to understanding the dynamics of this quantity under continuous monitoring. Given a partition of the system into two parts, $A \cup B$, the (conditional) reduced density matrix $\hat{\rho}_A(\xi_t) = \mathrm{Tr}_B \hat{\rho}(\xi_t)$ encodes the bipartite entanglement, via the entanglement entropy:

$$S(\xi_t) = -\mathrm{Tr}_A \left[ \hat{\rho}_A(\xi_t) \ln \left( \hat{\rho}_A(\xi_t) \right) \right]. \tag{212}$$

If the overall state is pure this is a well defined measure of entanglement, which yields the amount of Bell pairs that can be distilled from the quantum state. On the other hand

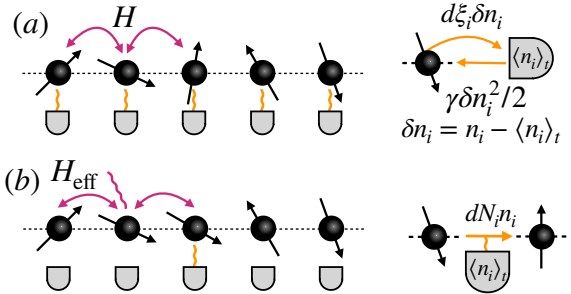

Figure 38: Cartoon of the monitored Quantum Ising Chain. (a) Quantum state diffusion protocol: on each lattice site a two-level system (Ising spin) interacts with its neighbours and is subject to a weak continuous measurement of the up-polarised state. (b) Quantum jump protocol: the spins in the chain interact through a non-Hermitian Hamiltonian. Occasionally a quantum jump take place, projecting the measured degree of freedom in the up-polarised state. The no-click limit corresponds to post-selecting only trajectories without jumps. Adapted from Ref. [633].

for a mixed state, such as the averaged state $\overline{\hat{\rho}(\xi_t, t)}$, entanglement with the environment also enters into the entropy of the subsystem density matrix. Since the unravelled system density matrix corresponds to a pure state, there is however no ambiguity about bipartite entanglement vs entanglement with the environment. After evolving the state under Eq. (210), one can compute the average entropy given by

$$\overline{S} = \int \mathcal{D}\xi_t P(\xi_t) S(\xi_t), \tag{213}$$

which is clearly a non-linear functional of the conditional state and therefore can be sensitive to physics not captured by the average state.

In general, similar issues of trajectory-resolved averaging arise when one is interested in more subtle statistical correlation functions of the stochastic process, for example in the so called overlaps [115] $\overline{\langle O_\mu(t)\rangle\langle O_\nu(t)\rangle} = \overline{\text{Tr}(\hat{O}_\mu\hat{\rho}(\xi_t, t))\text{Tr}(\hat{O}_\nu\hat{\rho}(\xi_t, t))}$. We note that these questions are closely related to the difference between annealed and quenched averages in disordered systems [634].

## 8.2   Measurement-Induced Phase Transitions

Based on the previous discussion we can expect that, as a function of the monitoring rate, the entanglement entropy of a monitored system might display non trivial behaviour which one would not see in the averaged dynamics (see Fig. 39). This statement can be understood intuitively: one can expect that while the unitary dynamics creates entanglement the effect of monitoring or measurements tends to project the system into an eigenstate of the (site-local) measurement operator, thus reducing the entanglement. Therefore one can quite generically expect a crossover or phase transition in the entanglement content. The prediction of a phase transition was put forward in two theoretical works that focused on random unitary circuits with projective measurements [635, 636] leading to the concept of Measurement-Induced Phase Transitions (MIPT), separating a volume-law entangling phase from an area-law disentangling phase. The latter can be interpreted as arising from a many-body Zeno effect, where quantum measurements prevent the spreading of quantum information. Such MIPT have been discussed in the context of purification transitions

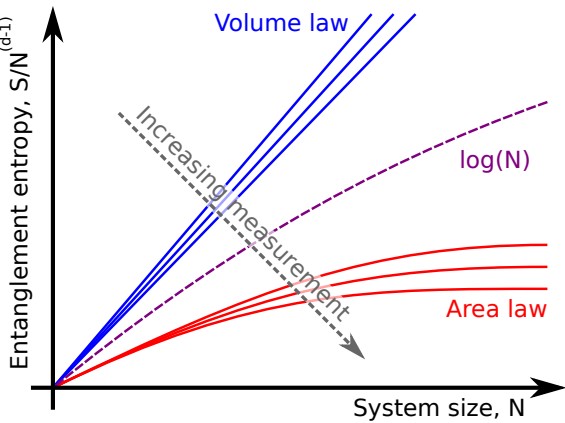

Figure 39: Sketch of different entanglement entropy scalings with system size in monitored quantum systems undergoing a Measurement-Induced Phase Transition.

for mixed states [637, 638]: for weak monitoring an initial mixed state purifies in a time that diverges with system size, typically exponentially for chaotic dynamics, while for large monitoring the environment succeeds in locally purifying the system at a constant rate, independent of system size. This robust mixed phase, which corresponds to the volume-law entanglement phase for the pure state protocol discussed above, can be also interpreted from a quantum information perspective as an error correcting phase, where unitary dynamics protects quantum information from external measurements and errors.

It was later realised that MIPT are not restricted to random circuits with projective measurements but can generically occur in some classes of open quantum many-body systems under suitable unravelling. One such example is non-interacting systems under continuous monitoring of the particle density, which have been extensively studied both numerically [633, 639–647] —taking advantage of the Gaussianity of the state—and analytically—by developing a replica field theory approach [648–650]. Despite the non-interacting nature of the problem the phenomenology is extremely rich and still under active investigation. Monitored one-dimensional free fermions with conserved particle number for example are believed to display a crossover from a logarithmic entanglement phase to an area law [639, 649, 651]. When the symmetry of the model is reduced to a discrete symmetry, such as for example in the monitored Ising chain [633] or for random Majorana fermions, the transition is believed to remain sharp [650]. Sharp entanglement transitions are also known to occur in the fully post-selected case of no-click evolution [652–655] and the role of quantum jumps and deviations from full post-selection on these transitions has been recently discussed [656, 657]. Finally, the dependence of the entanglement dynamics and the associated MIPT on the specific unravelling scheme has been also discussed [373, 633, 639, 644, 658], as well as the role of the monitored operator, in particular in the case of particle losses [659–662].

While the vast majority of theoretical and numerical works have focused on non-interacting, Gaussian, monitored systems there are few works that have discussed the interacting case [663–668] with projective or weak measurements. Here of course, due to the genuine many-body nature of the problem, numerical simulations are challenging and so only much smaller systems can be accessed. As such, understanding of such models is still at its infancy.

Finally, although MIPT are examples of entanglement phase transitions, they are not however limited to the entanglement. Indeed as discussed in the previous section there are

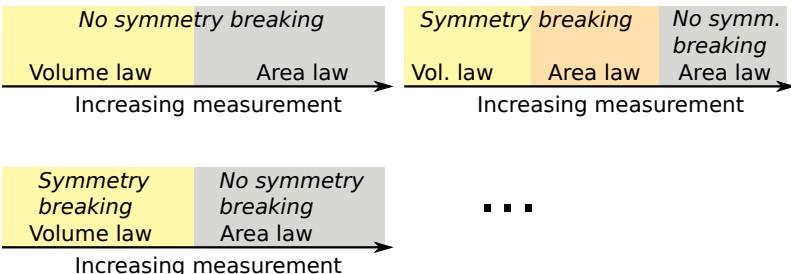

Figure 40: Interplay between measurement-induced phase transitions and symmetry-breaking dissipative phase transitions.

other quantities that can be sensitive to the conditional state. Among these we mention connected correlation functions [639], as well as histograms of local observables and full counting statistics [641, 669, 670].

## 8.3 Post-selection Problem

As we have discussed so far, monitored quantum systems allow one to explore the effect of quantum measurements on the system dynamics. There is however a fundamental challenge associated to achieving this goal in a genuine many-body context, which is rather fundamental in nature and referred to as the "post-selection problem". This problem arises as follows: in order to perform averages of observables along a given trajectory experimentally one should be able to reproduce the same sequence of measurements outcomes. However, the number of possible measurement records grow exponentially with both system size and time. As such the probability of acquiring the same measurement record multiple times is very low. For this reason, experimental results have so far been limited to a few sites, and considerable efforts were required to increase the lattice length.

From a theoretical point of view it is important to find robust mitigation techniques for the post-selection problem. Different strategies have been suggested and explored experimentally, including using space-time duality [671, 672] as has been experimentally achieved in [673] with the Google processor. A different strategy has proposed to use adaptive quantum circuits with measurements and feedback [674, 675], *i.e.*, conditioning the evolution on the measurement record to make the MIPT visible at the average (density matrix) level. However feedback has been shown to alter crucially the dynamics of the system and to lead to a different type of phase transition as compared to the original MIPT [676–679].

Finally, a recent theoretical proposal suggested an implementation of MIPT using atomic ensembles of laser-driven atoms and collective dissipation [680]. Here the slow disentangling dynamics of collective jump operators leads to a saturation time for the entanglement entropy growing only as $\tau \sim \log(N)$, with $N$ the number of atoms, leading to a polynomial overhead for post-selection (quantum state diffusion for this model has been previously studied in [681]).

The post-selection problem remains a formidable hurdle to overcome, and the search for cases where it can be mitigated is necessary for experimental progress in monitoring quantum many-body systems.

## 8.4 Experimental Evidence for MIPT

Despite the challenges associated to the post-selection problem, there has been recent progress on the experimental side to detect signatures of MIPT on small quantum devices.

The first quantum simulation experiment with trapped ions [67] used the idea of probing the transition via a reference qubit, in the spirit of a purification transition [638, 682], as discussed earlier. Specifically the experimental set-up comprised of a chain of trapped ion qubits, used as the system, reference qubit and measurement ancilla qubits. After initialisation, the reference qubit is entangled with a randomly selected system qubit and then undergoes unitary evolution interspersed with projective measurement. Measurements were performed at the end of the circuit, using some ancilla qubits. Postselection on the reference qubit is performed in order to obtain the value of the reference qubit Pauli operators, conditioned on the measurement outcomes. Finally, the reference qubit density matrix is reconstructed and from this the entropy can be obtained, revealing the purification transition. Subsequent works followed by the IBM and Google teams, exploring MIPT using superconducting qubits. In a first experiment explicit post-selection was performed on a small superconducting quantum processor characterised by random unitary evolution and measurements. Differently from the trapped ion experiment, here the measurements were performed mid-circuit, on a fast time-scale and with high-fidelity. The density matrix was reconstructed via tomography and the Renyi entropy estimated, showing a transition from volume to area law scaling [683]. Finally, the experimental realisation from the Google group took advantage of space-time dualities [673] to avoid mid-circuit measurements and thus probe the MIPT for a chain of 70 qubits. A key step of this implementation was the use of classical post-processing of the measurement record to characterise the structure of the phases. These pioneering experiments represent a first step to explore MIPT but it is clear that in the future more experimental evidence is needed, particularly in the regime of large number of qubits.

## 9 Summary and Outlook

The aim of these Lectures Notes was to give, to the best of our abilities, a self-contained introduction to the properties of open many-body synthetic systems as realised in the experimental platforms where the coupling to an environment can be tuned at will, and that could be also viewed as open-system quantum simulators. As briefly summarised in Section 2, such platforms range from cold atoms to trapped ions, from cavity arrays in superconducting networks to BECs in cavities. A common feature of all these systems is that all of them are well described by the assumption that the coupling to the environment is Markovian, i.e. memoryless: in Section 3 we described the main theoretical and conceptual frameworks in which such dynamics can be formulated. After introducing the main analytical and numerical theoretical tools to study the many-body Lindblad master equation in Section 4, we dealt with several consequences of the above-mentioned interplay between Hamiltonian and dissipative contributions on the evolution of the systems.

In general, both the steady state and the approach to it are the result of the interplay between the unitary evolution—dictated by the system Hamiltonian—and the sources of dissipation and decoherence, such as photon and atom losses, or local dephasing processes. Concentrating first on the steady state, in Section 5 we showed that appropriately designed baths and Hamiltonians can represent powerful ways to engineer many-body states: this is a well-known technique in quantum optics that was more recently extended to many-body systems. The realisation of many-body dark states may however require some fine tuning of coupling constants. By tuning the relative ratios of the parameters of the setup, the steady-

state changes and the system can enter different phases separated by transition points or lines: these are the dissipative phase transitions that we analysed in Section 6. In contrast to thermal or quantum phase transitions, dissipative phase transition are of nonequilibrium nature, so that the stabilisation of different type of ordering is not based on (free-)energy arguments: new phases, not allowed at equilibrium, are possible, and the critical behaviour may be governed by new critical exponents. A rich, interesting phenomenology appears also in the approach to the steady-state where the relaxation dynamics can be dictated by cooperative effects, as we discussed in Section 7; this is the case, for example, of the depletion dynamics of interacting Fermi/Bose particles subject to loss events. Finally, in Section 8 we focused on the striking phenomenology that can appear when it is possible to follow the stochastic dynamics along quantum trajectories: measurement-induced phase transitions can then occur that are not detectable via any averaged physical observable.

There are several very important aspects of the physics of open quantum many-body systems that we did not cover in these Lecture Notes. The general perimeter of this work has been presented in the introduction and in each specific section we have made reference to subjects and works that were not explicitly discussed, either for pedagogical reasons or simply for keeping the discussion limited in space. Instead of re-listing them here, we prefer to mention some topics that have not been discussed and that are likely to flourish in the next years; the following list reflects our personal perspective and is not meant to be a thorough review of these areas.

i) Non-Markovian dynamics. While there is a huge literature on this theory and its application to a few qubits or few-level systems [11, 684], less is known in the many-body case; the development of experimental platforms that we are witnessing will surely motivates further studies in this direction.

ii) Interplay between boundary and bulk dissipation. In these Lecture Notes we considered the case of dissipative terms in the bulk, whereas an extensive discussion of the role of boundary dissipative baths can be found in [685]. The simultaneous presence of boundary terms in driven-dissipative systems can be a very interesting avenue for exploring how quantum transport could take place in a non-equilibrium phase, or is affected by a dissipative phase transition.

iii) Adiabatic dynamics and the Kibble–Zurek mechanism in open systems. An early introduction to the topic in the context of closed quantum systems can be found in [686]. Whereas the discussion of adiabatic dynamics in open systems has significantly developed in the context of few-body physics, the extension to Markovian many-body systems looks like a natural one.

iv) Time-dependent Lindbladians. This topic embraces several different, very interesting, situations. These include the case of dissipative Floquet dynamics (see for example Ref. [687]), the whole topic of quantum control [688, 689] and related work on shortcuts to adiabaticity [690] in many-body open systems. A recent example of this topic, based on the variational approach of [691] can be found in Ref. [692].

v) Integrability and chaos in open quantum systems. Although the topics have already developed and the community has produced some first significant results, the number of studies devoted to this subject is small with respect to those addressing closed systems. The question of what it means to have an integrable Lindbladian, and what are the consequences for physical properties, is still beginning to be understood. Additional references can be found, for example in Ref. [312, 693] and the

references cited in the bibliography. Similarly, the question of how to define and characterise quantum chaos in open quantum systems has attracted attention from a broad community in recent years [694–698], with many questions still open to be explored.

vi) Topological physics. The subject of topology has been discussed in these Lecture Notes in only a very tangential way; yet, this is a very important and rapidly evolving research topic. Excellent reviews exist, see for example [27, 484] where topological classification of non-Hermitian systems is discussed in depth. The topological properties of systems governed by a Lindblad dynamics are less explored and additional work is required. More information on this rapidly growing area of activities can be found in [461, 468, 473, 699] and the references therein.

vii) Adaptive protocols, feedback and active quantum matter. In Sec. 8 we discussed monitored quantum systems whose evolution is conditioned to the measurement outcomes. A novel frontier which has started to be explored is to actively use the measurement outcomes to modify or correct the subsequent evolution, thus performing adaptive dynamics and feedback [700–703]. On the one hand this could allow to characterise measurement-induced transitions at the level of the average state, a task which remains non-trivial and requires appropriate entanglement measures/witnesses [704]. In addition, this could open the way to completely new dynamical behaviour, which generalise to the quantum setting the physics explored in classical active and non-reciprocal matter [604, 705–708].

viii) Frustration and disorder. Very few works adddress the possible formation of glassy behaviour in driven-dissipative systems. Recent contributions in this direction include [114–116, 709, 710].

To conclude, we would like to mention two aspects of the entire field that may be of interest for future studies. The choice, of course, is biased by our own interests and does not represent by any means an objective priority list.

In the last chapter our discussion on monitored systems focusing on measurement induced phase transitions reflects the current status of the interests on this questions. The number of questions that may arise is nevertheless much wider and concerns the whole area of correlated/collective dynamics of quantum jumps. For example, there are connections between properties that are only visible through individual trajectories of a monitored quantum system, and properties revealed by the distribution of replicas in spin glasses [115]. Further exploration of correlated dynamics in monitored quantum systems is an open research area that may lead to unexpected results but that however requires to find strategies to monitor many-body systems and to find efficient experimental techniques for revealing entanglement phase transitions without suffering from the burden of performing a postselection, a problem that still waits for a solution.

As a second and final point, we would like to mention the interest of exploring how open many-body systems can be of relevance for quantum technologies, for example to realise quantum memories [711]. Quantum simulators are at the heart of the current discussion on the subject, promising the possibility of addressing specific problems that are out of reach for classical computers. The same may be true for open-quantum system simulators, which could be employed to study different problems.

However, this should not limit the field of possible applications to quantum technologies: other applications such as quantum thermodynamics could be envisioned [25, 712] or collective effects in sensing and metrology are also a promising direction.

## Acknowledgements

We would like to thank all the people with whom we interacted and collaborated on the topics covered in these Lecture Notes. We had many interesting and enlightening discussions with K. B. Arnardóttir, A. Biella, I. Bouchoule, M. Capone, I. Carusotto, G. Chiriacò, J. I. Cirac, C. Ciuti, A. A. Clerk, A. J. Daley, M. Dalmonte, A. Delmonte, A. De Luca, J. De Nardis, S. Diehl, J. Dubail, P. Fowler-Wright, G. Fux, D. Gerace, F. Gerbier, V. Giovannetti, S. Gopalakrishnan, L. Gotta, M. Hartman, F. Iemini, A. Imamoglu, J. Jin, C. Joshi, H. Katsura, P. Kirton, H. Landa, Z. Lenarcic, A. Lerose, I. Lesanovsky, B. L. Lev, Y. Le Gal, Z. Li, P. B. Littlewood, B. W. Lovett, P. Lucignano, M. Maghrebi, A. Marché, J. Marino, B. Marsh, F. Minganti, G. Misguich, A. Nardin, F. B. F. Nissen, G. Pagano, G. Passarelli, D. Poletti, D. Rossini, L. Rosso, A. Russomanno, O. Scarlatella, M. Seclì, K. Seetharam, S. Sharma, P. Sierant, R. D. Soares, F. Surace, M. H. Szymańska, A. Tomadin, H. E. Türeci, X. Turkeshi, M. Vanhoecke, O. Viyuela, B. Xing, H. Yoshida and P. Zoller. We are particularly grateful for helpful comments on earlier versions of these Lecture Notes from A. Biella, I. Carusotto, F. Gerbier, F. Minganti, G. Pagano, T. Tomita and O. Scarlatella.

**Funding information.** R.F. is supported by the European Research Council via the grant agreements RAVE, 101053159 and the Italian PNRR MUR project PE0000023-NQSTI (R.F.). L.M. acknowledges support from the ANR project LOQUST ANR- 23-CE47-0006-02; this work is also part of HQI (www.hqi.fr) initiative and is supported by France 2030 under the French National Research Agency grant number ANR-22-PNCQ-0002. M.S. acknowledge funding from the European Research Council (ERC) under the European Union's Horizon 2020 research and innovation program (Grant agreement No. 101002955 – CONQUER). Views and opinions expressed are those of the authors only and do not necessarily reflect those of the European Union or the European Research Council. Neither the European Union nor the granting authority can be held responsible for them.

This research was supported in part by (i) the International Centre for Theoretical Sciences (ICTS) for R.F. and J.K. participating in the program "Periodically and quasi-periodically driven complex systems" (code: ICTS/pdcs2023/6); and by (ii) the Institut Pascal of Université Paris-Saclay in the framework of the program "Open quantum many-body physics 2023" to which R.F., L.M. and M.S. participated.

## A   Derivation of Lindblad Master Equation

In this Appendix we briefly recapitulate some basic notions related to deriving the Lindblad equation. The discussion contained here is not intended to offer a complete and rigorous derivation of the Lindblad equation or a comprehensive discussion of its properties. Our main goal is to give a quick introduction to few concepts that underlie the discussion throughout the Lecture Notes. For a complete discussion, the reader is referred to the references [186, 187, 713, 714] where the topic is treated in greater detail. We will first consider a formal derivation, based on the properties of the density matrix. In the second part we briefly outline how this equation is recovered in a system and environment dynamics assuming a weak coupling approximation.

## A.1 From quantum channels to the Lindblad equation

A remarkable fact of the Lindblad equation is that its form can be derived only by requiring that during the dynamical evolution the density matrix retains all its properties (i.e. remains Hermitian, unit trace, semi-positive definite.) and that the dynamics is Markovian. Note that this universal character is lost in the case of non-Markovian dynamics for which no general dynamical equation is known (or believed to exist).

Any transformation that maps density matrices into density matrices can be expressed in the Kraus form [186, 187]

$$\hat{\rho}' = \sum_\mu \hat{K}_\mu \hat{\rho} \hat{K}_\mu^\dagger \qquad \text{with} \qquad \sum_{\mu=0} \hat{K}_\mu^\dagger \hat{K}_\mu = \mathbb{1}. \tag{214}$$

It is easy to verify that if $\hat{\rho}$ obeys all the properties of a density matrix then so will $\hat{\rho}'$. To derive the Lindblad equation, one can consider Eq. (214) in the case of an infinitesimal transformation, linking the state of the system at time $t$ to that at time $t + dt$. For such a transformation one may easily see that one of the Kraus operators, say $\hat{K}_0$, must be the identity up to corrections of the order of $dt$ while all the other Kraus operators must scale as $\sqrt{dt}$:

$$\begin{aligned} \hat{K}_0 &= \mathbb{1} + (-i\hat{H} + \hat{\mathcal{K}})dt \\ \hat{K}_\mu &= \hat{L}_\mu \sqrt{dt} \qquad \mu \neq 0 \end{aligned} \tag{215}$$

The decomposition in $\hat{K}_0$ of the operator proportional to $dt$ into its real an imaginary parts immediately suggests that $\hat{H}$ can be identified with the (effective) Hamiltonian of the system and $\hat{\mathcal{K}}$ should satisfy $\hat{\mathcal{K}} = -(1/2) \sum_{\mu \neq 0} \hat{L}_\mu^\dagger \hat{L}_\mu$. Substituting the expressions of Eq. (215) in Eq. (214) the Lindblad equation follows

$$\frac{d\hat{\rho}}{dt} = -i[\hat{H}, \hat{\rho}] + \sum_{\mu \neq 0} \left( \hat{L}_\mu \hat{\rho} \hat{L}_\mu^\dagger - \frac{1}{2} \left\{ \hat{L}_\mu^\dagger \hat{L}_\mu, \hat{\rho} \right\} \right) . \tag{216}$$

The number of allowed operators $\hat{L}_\mu$ extends up to the square of the dimension of the Hilbert space. For a many body system, this will thus scale exponentially with the system size and so is potentially very large; however, in many cases of relevance for these Lecture Notes, the jump operators are local and so the number of terms in the sum scales linearly with the system size rather than exponentially.

The operators $\hat{H}$ and $\hat{L}_\mu$ have a fundamental physical meaning: the first is the Hamiltonian and the second accounts for the action of the environment on the system. As described extensively in the Lecture Notes, the choice of the jump operators usually follows from the derivation of the reduced dynamics from global models of system and environment; however, in many situations this can be done on phenomenological grounds by considering relevant processes associated to the action of the environment on the systems or by approximate methods. In the next subsection we briefly recap a very popular and effective approach based on a weak-coupling expansion in the system-environment interaction. One should always bear in mind that the derivation from microscopic principles of appropriate dynamical equations is a highly non-trivial tasks and may require sophisticated methods.

Before concluding, let us remark that depending on the choice of the jump operators, the steady state may or may not describe an equilibrium state. If one wants a Lindblad equation that reaches the thermal equilibrium state at long times, this will generally require non-local jump operators. In this case the steady state is diagonal in the eigenstates of the Hamiltonian and the jump operators describe transitions between the eigenstates with rates obeying detailed balance. Since these eigenstates are often delocalised, the corresponding operators must also be.

## A.2  Born–Markov Master Equation

The derivation of the Lindblad dynamics for Markovian open quantum systems that we presented in the previous section leaves open important questions on how to determine the the form of the jump operators from first principles; several approaches are discussed in the Refs. [187, 713, 714]. In this Appendix we summarise a simple method—often referred to as the Bloch–Redfield approach—valid in the weak-coupling regime, that is however illuminating in understanding several features of quantum open system dynamics.

   The starting point is to assume that the system and its environment evolve globally in a unitary fashion governed by the Hamiltonian

$$\hat{H} = \hat{H}_S + \hat{H}_E + \hat{H}_{S-E} \ . \tag{217}$$

where the $\hat{H}_S$ is the Hamiltonian of the system, $\hat{H}_E$ that of the environment and $\hat{H}_{S-E}$ their interaction. The interaction can be chosen, for simplicity, as given by the product of an operator of the system $\hat{Y}_S$ with one of the environment $\hat{X}_E$, and so $\hat{H}_{S-E} = \hat{Y}_S \hat{X}_E^\dagger +$ H.c.. The evolution of the state of the "universe", $\hat{\rho}_U$, is unitary and follows $\frac{d}{dt}\hat{\rho}_U = -i[\hat{H}, \hat{\rho}_U]$. The goal is to find a dynamical equation for $\hat{\rho} = \mathrm{Tr}_E[\hat{\rho}_U]$ by tracing over the environmental degrees of freedom. The question boils down to finding (approximate) methods for the computation of the partial trace $\mathrm{Tr}_E[\hat{H}, \hat{\rho}_U]$. This trace can be easily evaluated by expanding $\hat{\rho}_U$ in powers of the interaction. By going to the interaction picture and substituting the formal solution for the density matrix back into the Liouville equation (we drop for simplicity the subscript indicating the interaction picture) one finds:

$$\frac{d}{dt}\hat{\rho} = -\int_{t_0 \to -\infty}^{t} dt' \ \mathrm{Tr}_E[\hat{H}_{S-E}(t), [\hat{H}_{S-E}(t'), \hat{\rho}_U(t')]] \ . \tag{218}$$

In this expression we dropped an irrelevant term depending on the density matrix at $t_0$. The system Hamiltonian can be always renormalized in order to make this term to disappear.

   Equation (218) has the right form to discuss the key approximations:

1. Born approximation. In general, $\hat{\rho}_U$ can be written as the factorised form $\hat{\rho}_{E,\beta} \otimes \hat{\rho}(t)$, where $\rho_\beta$ is the (assumed thermal) state of the environment, plus terms that arise due to the interaction of the system and the environment. Since the right hand side of Eq. (218) is already of second order in the interaction, parts of $\hat{\rho}_U$ that are of higher order in the coupling can be dropped. This means that one can substitute it with the product of the system and bath states, $\hat{\rho}_U(t) \simeq \hat{\rho}_{E,\beta} \otimes \hat{\rho}(t)$. Note that this Born approximation further implies that the state of the environment does not change due to the interaction with the system.

2. Markov approximation. The trace over the environment amounts to computing correlators of the form $\mathrm{Tr}_E[\hat{X}_E(t_1)\hat{X}_E(t_2)\hat{\rho}_{E,\beta}]$. If these correlations are sufficiently short-ranged in time then it is legitimate to substitute $\hat{\rho}_U(t') \to \hat{\rho}_U(t)$ in Eq. (218).

With these approximations, the equation of motion for the density matrix becomes:

$$\frac{d}{dt}\hat{\rho} = -\int_{-\infty}^{t} dt' \ \mathrm{Tr}_E[\hat{H}_{S-E}(t), [\hat{H}_{S-E}(t'), \hat{\rho}_{E,\beta} \otimes \hat{\rho}(t)]] \ \equiv \ \hat{\bar{\mathcal{L}}}_{\mathrm{BM}}[\hat{\rho}(t)] \ , \tag{219}$$

where the subscript in the superoperator indicate the Born-Markov approximation.

   There are several features that emerge already at this stage, that are worth pointing out. In the right hand side of Eq. (219) the degrees of freedom of the system and the environment "factorise". Let us examine the two of them separately, starting from the system. The double commutator appearing in Eq. (219) leads naturally to a Lindblad-like sequence

of terms. If we consider the simplest case, in which $\hat{Y}_S$ is an eigenoperator of the system Hamiltonian (where $\hat{Y}_S$ evolves monochromatically under $\hat{H}_S$, i.e. $e^{i\hat{H}_S t}\hat{Y}_S e^{-i\hat{H}_S t} = \hat{Y}_S e^{-i\epsilon t}$, or equivalently, $[\hat{Y}_S, \hat{H}_S] = \epsilon \hat{Y}_S$), then the terms that arise in the density matrix equation of motion are $\hat{Y}_S \hat{\rho} \hat{Y}_S^\dagger$, $\hat{Y}_S^\dagger \hat{Y}_S \hat{\rho}$, $\hat{\rho}\hat{Y}_S^\dagger \hat{Y}_S$ and their Hermitian conjugates. The jump operators appearing in the Lindblad equations in this special case are proportional to the operators $\hat{Y}_S$, i.e. the microscopic coupling between the system and the environment also determine how the dynamics of the reduced density matrix is affected. For example, if $\hat{Y}_S$ is a spin-$1/2$ raising/lowering operator $\sigma_i^\pm$ coupled, in each site, to a bath of bosons (photons for instance), the process accounted in Eq. (219) is the absorption/emission of a quantum of energy from/to the bath.

The perturbative character of the Born–Markov approximations appears in the fact that the only operators that appear are first order in the system-bath coupling. For example, in the spin raising case, the jump operators $\hat{\sigma}_i^\pm$ are indicating that the only allowed process is the emission or absorption of a single photon. Two- or higher-order processes can be found going beyond the second order (for example, $\hat{\sigma}_i^+ \hat{\sigma}_j^-$). The perturbation expansion in Eq. (219) should be interpreted as an expansion of the self-energy in the Dyson expansion of the many-body Green's function [715]. It includes only selected terms (in this example the single photon absorption/emission) resummed to infinite orders.

In the general case, where $\hat{Y}_S$ is not an eigenoperator of $\hat{H}_S$, there is still some extra work to do to bring Eq. (219) into a Lindblad form. In this case the time evolution can be considered as having the effect of decomposing $\hat{Y}_S$ into "eigenoperators" which individually evolve monochromatically with $\hat{H}_S$, i.e. $e^{i\hat{H}_S t}\hat{Y}_S e^{-i\hat{H}_S t} = \sum_\mu \hat{Y}_{S,\mu} e^{-i\epsilon_\mu t}$. All the details can be found, for example, in [714]. In general Eq. (219) will then take the form:

$$\frac{d}{dt}\hat{\rho} = \sum_{\mu\nu} L_{\mu\nu} \left( \hat{Y}_{S,\nu} \hat{\rho} \hat{Y}_{S,\mu}^\dagger - \frac{1}{2}\{\hat{Y}_{S,\mu}^\dagger \hat{Y}_{S,\nu}, \hat{\rho}\} \right). \tag{220}$$

This expression is not yet in Lindblad form, as the matrix $L_{\mu\nu}$ is not diagonal, and crucially may have both positive and negative eigenvalues. The Lindblad form corresponds to working in the diagonal basis, but also requiring the matrix $L$ has only positive eigenvalues, corresponding to rates of each process. There are various procedures that can be applied to ensure a positive form, such as "secularisation", which consists of neglecting all terms that are time-dependent in the interaction picture [187]. However, in a number of cases secularisation leads to a form that is of Lindblad form but generates incorrect dynamics [716, 717].

The other very important element of Eq. (219) is the "environmental part", i.e. the correlators $\text{Tr}_E[\hat{X}_E(t_1)\hat{X}_E(t_2)\hat{\rho}_{E,\beta}]$ integrated over time. This term will give the magnitude of the coupling, entering in the jump operator. Assuming a thermal bath, one may find that the different correlators (i.e., the different patterns of orders of times) are related to the familiar rates of emission and absorption obtained by Fermi's golden rule. The important aspect to highlight here is that with this method it is possible to compute how all the terms forming the density matrix evolve under the action of the external environment. With the Fermi golden rule argument alone, only the transition rates are obtained.

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
