# Peer review of "Many-Body Open Quantum Systems"

_SciPost Physics Lecture Notes_

## Round 2 · Referee Report · Anonymous (Referee 1) · 2024-12-7

Strengths

  1. An extensive and comprehensive review of a large subset of the field of open quantum many-body systems.
  2. It covers the most relevant tools and canonical models, the essential topics (from quantum phase transitions, criticality and symmetry breaking, to dynamics and connections to other fields).
  3. It is self contain and an excellent resource for onboarding newcomers (both students and early career scientists, or people from other fields with a knowledge of advanced QM).
  4. It is very up to date and written in a clear and pedagogical way.

Weaknesses

  1. Some areas, such as numerical methods (tensor networks, DMFT, stochastic equations) are described quite succinctly. These may be worse starting points for a newcomer to enter the field.

Report

The authors have developed a large, but very well organized and written lecture notes on the topic of many-body open quantum systems. They focus on a subset of the field that encompasses not only the most fundamental results and ideas, but also the tools, formalisms and canonical models that may help readers enter other areas of research. The presentation is rather exhaustive and covers a broad set of tools, sometimes superficially (e.g. numerical methods), but in general with enough key references to access all methods and concepts.

Overall I find this an extremely useful resource that not only helps the community, but that stands on its own as a wonderful text to quickly onboard beginners (i.e. students and newcomers), allowing them to critically examine topics and tools that might be more interesting for their research.

I strongly recommend publication, with the optional consideration of some scientific and presentation issues I outline below.

Requested changes

Physics comments:

On page 6, Ref. 41 does not seem to be the expected one. Perhaps the authors refer to J. Resnick, J. Garland, J. Boyd, S. Shoemaker, R. Newrock, Phys. Rev. Lett. 47 (1981) 1542 and P. Martinoli, P. Lerch, C. Leemann, H. Beck, J. Appl. Phys. 26 (1987) 1999.

In Sect. 2.2, the treatment of two-body losses in ultracold atoms (and ultracold molecules) and the emergence of strongly correlated phases is unfair to previous art. Ref. 84 is only discussed as a work engineering losses and not the physics that emerges from it: Zeno suppression of losses and emergence of a 1D TG gas.

A similar problem regards the theory, which ignores early works describing strong two-body losses as as originators of strong correlations in 1D via Zeno physics https://journals.aps.org/pra/abstract/10.1103/PhysRevA.79.023614 https://iopscience.iop.org/article/10.1088/1367-2630/11/1/013053/meta The theory in those works is applicable to molecules or atoms and, as a purely Linbladian treatment, predates the non-Hermitian papers cited in Sect. 7.2 These references would also be suitable after Eq. (36), where the dissipative Bose-Hubbard model is introduced without citations and without references to experiments where it has been realized.

In Sect. 2.3, I miss references to a canonical problem in cavities with many atoms, which is the generation of squeezing. This includes experiments without and with feedback, such as https://journals.aps.org/prl/abstract/10.1103/PhysRevLett.104.073602 and https://journals.aps.org/prl/abstract/10.1103/PhysRevLett.116.093602 and is a much simpler model that the, arguably extremely interesting, setups of Bose-Hubbard models in cavities.

On page 67, when discussing tensor networks and star topologies, I feel that there is an unnecessary link to 2020's works, when there were very exhaustive studies of renormalization strategies by Plenio and Huelga, and the development of the MPS with chain mapping algorithms that are of slight more interest. Naturally, such renormalizations have been discovered multiple times and can also be in the DMRG world (the link between TN's and DMRG seems also to be missing, even in passing), as in A. Feiguin's work on spins in electron baths in arbitrary dimensions.

Style and presentation comments:

  • The overall writing is of high quality, but there are sections where the choice of words is somewhat repetitive: e.g., at the end of page 8, the word "discuss(ion)" is used 5 times in 10 lines. Everywhere "however" is also abused and at the end of page 10 the word capture is used three times, the third one with a missing "d" at the end.

  • "tracing out the environmental degrees of freedom. [38]." An extra dot there.

  • On page 13, instead of Ref. 59 I would have expected the canonical RMP on single trapped ions and covers everything, including spontaneous emission, cooling and pumping https://journals.aps.org/rmp/pdf/10.1103/RevModPhys.75.281?casa_token=GnefutWXNywAAAAA%3AXnjSTOTlkxznRNbQLu-yX94QqHS0MMvmH6tfUmPjw3HHXeaVyiLNeQJ0mFMs0Q3cI0yKXx6lZ0T2MQ

  • At the end of page 13 and beginning of page 14, the topic of measurement-induced phase transitions is mentioned in a way that seems to make it specific to trapped ions. The authors could add a footnote that this is not specific to ions, though they are particularly well suited, due to quality and speed of measurements.

  • Because of the way it is formulated, using a different label for the stochastic variable, the paragraph around (60) seems to apply only to the second unravelling, when it is true in general: the quantum state is the mixed state that results from averaging over realizations. This paragraph is also somewhat redundant with what follows immediately afterwards, unless I am losing some subtletly.

  • On page 70, the presentation above Eq. (121) is somewhat confusing. As a student, the subindex "c" is not explained and I would be puzzled by the fact that I have been told the three operator expectation value is zero, but then I am presented with an expansion of that product. I am sure this can be fixed with one or two sentences (which are relevant, since Kubo's cited paper is behind a paywall).

  • On page 98, the text "the sum over alpha" probably means "the sum over mu".

  • On page 98, this discussion is also ambiguous. In Eq. 167, the authors assume that the eigenvectors will appear in a precise order, when the opposite phase, \rho^{>}-\rho^{<}, is essentially the same vector and eigenstate with the same eigenvalue, but different decomposition 168. Maybe this disambiguation can be referenced or anticipated.

  • Fig. 31 does not make it clear what is being plotted. At most, one can deduce a boundary computed by some means, but what this boundary is, is not clear from the figure itself.

Recommendation

Publish (surpasses expectations and criteria for this Journal; among top 10%)

---

## Round 2 · Referee Report · Anonymous (Referee 2) · 2025-1-3

Strengths

  1. A broad and contemporary overview that helps newcomers and students orient themselves in the field of open quantum systems.
  2. Covers the main theoretical as well as experimental aspects.
  3. Written in a pedagogical way.

Weaknesses

  1. While theoretical concepts are discussed in quite great detail, some more widely useful numerical techniques are explained extensively.
  2. Naturally, the numerical approaches actively used by the authors are covered at greater extent (such as DMFT), but this might give a wrong impression regarding which methods are the ones more commonly used to simulate Lindblad dynamics (such as tensor networks); using DMFT to study Lindblad is still being developed, while tensor network modeling of open quantum systems is rather well established.
  3. I find that perturbative approaches might also be mentioned as another approach to reduce the complexity of studying open systems.
  4. I am a bit surprised that authors did not include any discussion on non-Markovian dynamics, since one of the authors is an expert. I understand though, that this subject might be too vast and requires a separate review.

Report

Authors have overviewed a wide and rather complete selection of topics on open quantum systems. Their choice of section is pedagogical, as well as up to date with the current trends in the community. The overview mentions all the main experimental and theoretical approaches and directs the reader into particular papers on the given topic.

I strongly recommend the publication. Below, I give some optional suggestions, mostly about additional references and how to potentially  address the above mentioned weaknesses.

Requested changes

  • Regarding adding additional section on perturbation theory: Splitting the Liouvillian in a dominant and perturbative terms can generally reduce the complexity of studying Liouvillian dynamics, generally down to the complexity of the dominant term. A most naturally studied limit is the limit of weak openness, addressed generically in Sci. Rep. 4, 4887 (2014), PRB 97, 024302 (2018); PRE 101, 042116 (2020). Examples of usage are probably numerious, e.g., PRB 97, 134301 (2018). PRL 121 (26), 267603 (2018), so maybe hard to cite. Of course, this approach can be used also for other limiting cases and is thus quite generic. https://www.nature.com/articles/srep04887 https://journals.aps.org/prb/abstract/10.1103/PhysRevB.97.024302 https://journals.aps.org/pre/abstract/10.1103/PhysRevE.101.042116 https://journals.aps.org/prb/abstract/10.1103/PhysRevB.97.134301 https://journals.aps.org/prl/abstract/10.1103/PhysRevLett.121.267603

  • p.31, discussion after Eq.(20): A recent paper by Mori and Shiraii, PRL 125, 230604 (2020), pointed out that the Liouvillian gap does not necessarily give the slowest relaxation time to the steady state. Although a general understanding of when this happens is, to my knowledge, not yet there, it might be worth pointing out that there can be exceptions. https://journals.aps.org/prl/abstract/10.1103/PhysRevLett.125.230604

  • page 32: "As discussed in Sec. 1.3, when a system is coupled to a single environment " →was probably meant single thermal environment?

- page 34: "As noted in the introduction, in this review, we will focus on Markovian (i.e., weak coupling) models. Moreover, since we focus on driven-dissipative systems and other nonequilibrium states, we will not generically expect to reach thermal equilibrium. As such, the rest of our discussion will focus on models with local dissipation." → I disagree since any chaotic Hamiltonian that is weakly coupled to baths in bulk will relax to a density matrix that can be approximated with a thermal state, see works PRB 97, 024302 (2018), PRE 101, 042116 (2020) and as actual solutions PRL 121, 267603 (2018), PRB 97, 134301 (2018), PRL 125, 116601 (2020). In the limit of infinitesimal openness chaotic system relaxes to a thermal state. Maybe the point you want to make is that for obtaining exactly thermal state at finite openness one need to couple to a thermal bath? https://link.aps.org/doi/10.1103/PhysRevB.97.024302 https://journals.aps.org/pre/abstract/10.1103/PhysRevE.101.042116 https://link.aps.org/doi/10.1103/PhysRevLett.121.267603 https://journals.aps.org/prb/abstract/10.1103/PhysRevB.97.134301 https://journals.aps.org/prl/abstract/10.1103/PhysRevLett.125.116601

  • section 3.1.5: I think that under examples of models, it might be worth mentioning coupling to boundary Lindblads that induce a weak current in the system, since this is a rather generic way to probe the transport properties of the bulk model. Crucial references: PRL 106, 220601 (2011), PRB 99, 035143 (2019) https://link.aps.org/doi/10.1103/PhysRevLett.106.220601 https://link.aps.org/doi/10.1103/PhysRevB.99.035143

- page 47, end of Sec. 3: perhaps mention that one physical situation where the no-click limit is applicable is when dealing with condensates with an already macroscopic number of particles/excitation present, so adding/removing one is not crucial.

  • p. 48: under neural network approaches, I would add PRL 127, 230501 (2021) using neural network as a way to approximate the POVM representation of the density matrix. This seems to be a rather promising and generically applicable approach. https://journals.aps.org/prl/abstract/10.1103/PhysRevLett.127.230501

-p.51: I would possibly add additional references: PRL 126 (24), 240403, arXiv:2406.12695, Phys. Rev. Lett. 124, 160403 (2020), Phys. Rev. Lett. 110, 047201 (2013) https://journals.aps.org/prl/abstract/10.1103/PhysRevLett.126.240403 https://arxiv.org/abs/2406.12695 https://journals.aps.org/prl/abstract/10.1103/PhysRevLett.124.160403 https://journals.aps.org/prl/abstract/10.1103/PhysRevLett.110.047201

  • p.62: misprint $\Theta_{i,eff}(t)$$\Phi_{i,eff}(t)$

-p.60 In that case, I think it would be fair cite other review on driven-dissipative DMFT, e.g. Rev. Mod. Phys. 86, 779 (2014) and arxiv:2310.05201 https://journals.aps.org/rmp/abstract/10.1103/RevModPhys.86.779 https://arxiv.org/abs/2310.05201.

It might also be useful for students to know that in this framework a very common approach to opennes is to consider a quadratic bath which is integrated out and act as a new term in the effective action. The advantage of such approach is that the bath is non-Markovian and for quadratic baths the procedure is exact. For interacting bath weak coupling expansion is typically considered. Examples of such approachs are numerous and here we only suggest few : https://journals.aps.org/prl/abstract/10.1103/PhysRevLett.110.126401, https://journals.aps.org/prb/abstract/10.1103/PhysRevB.102.165136, https://journals.aps.org/prb/abstract/10.1103/PhysRevB.106.125123

- p.66: → I would call section 4.5.2 Tensor network methods since the density matrix can also be represented as a matrix-product operator, therefore MPS is a bit misleading. → I would mention that using tensor network approaches is very well suited to model open systems (better than for closed systems) since the operator entanglement entropy is naturally bounded by the coupling to the environment. That is the fundamental reason why tensor networks is one of the most popular approaches in the field. While this is common wisedom, it has been addressed in Quantum 4, 318 (2020). https://quantum-journal.org/papers/q-2020-09-11-318/ → one of the most generally used approaches to model open systems with tensor networks is vectorization of density matrix and using the usual methods for states, e.g., TEBD. This possibility is not mentioned at all, even thought it is generically more efficient than sampling over trajectories for state where bond dimension is not bounded by the dissipator. I would strongly suggest introducing it. → Alternative is to latter is also do DMRG-like optimization for the steady state, see PRL 114, 220601 (2015). https://journals.aps.org/prl/abstract/10.1103/PhysRevLett.114.220601

  • p.69: → Figure 17: I think that this schematics is not very informative. For example, the sketch in Hartmann&Carleo [279] is more informative because it makes it more explicit that some hidden neurons connect bra degrees of freedom, some ket DOF, and some connect both (making it a mixed state) → An alternative to RBM representation is to use a neural network to represent the POVM distribution corresponding to the density matrix, PRL 127 (23), 230501. In this case, one can play with the dept and architecture of the NN to achieve better expressivity. Currently, the neural network architecture doesn't ensure the physical properties of the state. Some other ansätze tried to mend for that, e.g., arXiv:2206.13488 https://journals.aps.org/prl/abstract/10.1103/PhysRevLett.127.230501 https://arxiv.org/abs/2206.13488

  • p.91: maybe add a reference to Science 383 (6689), 1332-1337, where they implemented system qubits coupled to ancillary ones with a superconducting circuit for the purpose of dissipative cooling. https://www.science.org/doi/abs/10.1126/science.adh9932

  • p. 114, Sec. 7.1.1 would be fair to cite Marko Žnidarič J. Stat. Mech. (2010) L05002 because was before ref 551 and has discussed diffusion due to noise in boundary driven setup https://iopscience.iop.org/article/10.1088/1742-5468/2010/05/L05002

- p. 117, Eq.(192) t_H → J for consistency with the text

-p. 125 In addition to [615, 616, 626], optionally mention realization with trapped ions PRR 3, 033142 (2021), quantum computers arXiv:2406.17033 and quenches in Kitaev chain PRL 129 220602 (2022) https://journals.aps.org/prresearch/abstract/10.1103/PhysRevResearch.3.033142 https://arxiv.org/abs/2406.17033 https://journals.aps.org/prl/abstract/10.1103/PhysRevLett.129.220602

Recommendation

Publish (easily meets expectations and criteria for this Journal; among top 50%)

---

## Editorial Decision

resubmitted